# Association between bisphosphonate use and COVID-19 related outcomes

Jeffrey Thompson[1†], Yidi Wang[2†], Tobias Dreischulte[3], Olga Barreiro[2], Rodrigo J Gonzalez[2], Pavel Hanč[2], Colette Matysiak[2], Harold R Neely[2], Marietta Rottenkolber[3], Thomas Haskell[1], Stefan Endres[4], Ulrich H von Andrian[2*]

[1]Cerner Enviza, Malvern, United States; [2]Dept. of Immunology, Harvard Medical School, Boston, United States; [3]Institute of General Practice and Family Medicine, University Hospital of Ludwig Maximilians-University Munich, Munich, Germany; [4]Center of Integrated Protein Science Munich and Division of Clinical Pharmacology, University Hospital, LMU Munich, Germany, Munich, Germany

*For correspondence:
uva@hms.harvard.edu

†These authors contributed equally to this work

## Abstract

**Background:** Although there are several efficacious vaccines against COVID-19, vaccination rates in many regions around the world remain insufficient to prevent continued high disease burden and emergence of viral variants. Repurposing of existing therapeutics that prevent or mitigate severe COVID-19 could help to address these challenges. The objective of this study was to determine whether prior use of bisphosphonates is associated with reduced incidence and/or severity of COVID-19.

**Methods:** A retrospective cohort study utilizing payer-complete health insurance claims data from 8,239,790 patients with continuous medical and prescription insurance January 1, 2019 to June 30, 2020 was performed. The primary exposure of interest was use of any bisphosphonate from January 1, 2019 to February 29, 2020. Bisphosphonate users were identified as patients having at least one bisphosphonate claim during this period, who were then 1:1 propensity score-matched to bisphosphonate non-users by age, gender, insurance type, primary-care-provider visit in 2019, and comorbidity burden. Main outcomes of interest included: (a) any testing for SARS-CoV-2 infection; (b) COVID-19 diagnosis; and (c) hospitalization with a COVID-19 diagnosis between March 1, 2020 and June 30, 2020. Multiple sensitivity analyses were also performed to assess core study outcomes amongst more restrictive matches between BP users/non-users, as well as assessing the relationship between BP-use and other respiratory infections (pneumonia, acute bronchitis) both during the same study period as well as before the COVID outbreak.

**Results:** A total of 7,906,603 patients for whom continuous medical and prescription insurance information was available were selected. A total of 450,366 bisphosphonate users were identified and 1:1 propensity score-matched to bisphosphonate non-users. Bisphosphonate users had lower odds ratios (OR) of testing for SARS-CoV-2 infection (OR = 0.22; 95%CI:0.21–0.23; p<0.001), COVID-19 diagnosis (OR = 0.23; 95%CI:0.22–0.24; p<0.001), and COVID-19-related hospitalization (OR = 0.26; 95%CI:0.24–0.29; p<0.001). Sensitivity analyses yielded results consistent with the primary analysis. Bisphosphonate-use was also associated with decreased odds of acute bronchitis (OR = 0.23; 95%CI:0.22–0.23; p<0.001) or pneumonia (OR = 0.32; 95%CI:0.31–0.34; p<0.001) in 2019, suggesting that bisphosphonates may protect against respiratory infections by a variety of pathogens, including but not limited to SARS-CoV-2.

**Conclusions:** Prior bisphosphonate-use was associated with dramatically reduced odds of SARS-CoV-2 testing, COVID-19 diagnosis, and COVID-19-related hospitalizations. Prospective clinical trials will be required to establish a causal role for bisphosphonate-use in COVID-19-related outcomes.

**Funding:** This study was supported by NIH grants, AR068383 and AI155865, a grant from MassCPR (to UHvA) and a CRI Irvington postdoctoral fellowship, CRI2453 (to PH).

## Editor's evaluation

Using health insurance claims data, this valuable paper reports on a retrospective propensity score matched cohort study that was performed to quantify associations between bisphosphonate (BP) use and COVID-19-related outcomes (COVID-19 diagnosis, testing, and COVID-19 hospitalization). The evidence is solid showing that in primary and sensitivity analyses, BP use was consistently associated with lower odds for COVID-19, testing, and COVID-19 hospitalization. The study is of interest to a broad readership (clinicians, public health physicians, pharmacologists and epidemiologists).

## Introduction

Throughout the COVID-19 pandemic, massive global efforts to repurpose existing drugs as potential therapeutic options for COVID-19 have been undertaken. Drug repurposing, whereby a drug already proven to be safe and effective in humans for another approved clinical indication is evaluated for novel clinical use, may allow for faster identification and deployment of therapeutic agents compared to traditional drug discovery pipelines. Using in silico and in vitro analyses, a growing list of drugs have been suggested to be potentially efficacious in treating COVID-19 by either direct or indirect antiviral actions (*Sultana et al., 2020*). Another potentially beneficial class of drugs may be agents that boost or modulate anti-viral immune responses to SARS-CoV-2 infection to reduce clinical symptoms and/or mitigate disease progression. Regardless of the mechanism of action, ultimately, randomized prospective clinical studies are needed to test the safety and efficacy of each candidate in treating or preventing COVID-19. Observational studies can help prioritize candidates for prospective clinical testing, by examining associations between the use of a candidate drug and the incidence or severity of disease in users compared to a matched group of non-users. Drugs with strong observational evidence for potential effectiveness against COVID-19 may then be considered for prospective trials (*Sultana et al., 2020*).

Here, we have investigated bisphosphonates (BPs), a class of small-molecule drugs that inhibit bone resorption by osteoclasts (*Roelofs et al., 2010b*). BPs are widely prescribed as either oral or intravenous formulations to treat osteoporosis, Paget disease, and malignancy-induced hypercalcemia. Additionally, BPs are used as adjuvant therapy for breast cancer (*Dhesy-Thind et al., 2017*). BPs are subdivided into two classes, nitrogen-containing (amino-BPs) and nitrogen-free BPs (non-amino-BPs; *Russell et al., 2008*). Both accumulate in bone but have distinct molecular mechanisms by which they kill osteoclasts to prevent bone resorption (*Roelofs et al., 2010b*).

Aside from depleting osteoclasts, clinical and experimental studies indicate that BPs exert a plethora of immunomodulatory effects, providing a rationale for exploring BPs as potential repurposed drug candidates for COVID-19 (*Brufsky et al., 2020*). Indeed, amino-BPs regulate the activation, expansion, and/or function of a major subset of human γδT cells (*Poccia et al., 2006*; *Hewitt et al., 2005*; *Tu et al., 2011*) as well as neutrophils (*Favot et al., 2013*), monocytes (*Roelofs et al., 2010a*), and macrophages (*Rogers and Holen, 2011*; *Wolf et al., 2006*); they can modulate the antigen-presentation capacity of dendritic cells (*Xia et al., 2018*); and in animal studies, both amino-BPs and non-amino-BPs exerted potent adjuvant-like activity to boost antibody and T cells responses to viral antigens (*Tonti et al., 2013*). Furthermore, observational studies have reported decreased in-hospital mortality for patients in the ICU (*Lee et al., 2016*), and reduced incidence of pneumoniae and pneumonia-related mortality in patients treated with amino-BPs versus controls (*Sing et al., 2020*). These immunological and clinical effects of BPs combine with several other characteristics that make BPs well-suited as repurposed drug candidates in the context of a pandemic: they are globally accessible as generics, affordable, straightforward to administer, and have known safety profiles in adult (*Suresh et al., 2014*) and paediatric populations (*Sbrocchi et al., 2010*; *George et al., 2015*).

In light of these considerations, we have analysed a database of health insurance claims in the U.S. to determine if prior BP-use is associated with a differential incidence and/or severity of COVID-19-related outcomes. Specifically, we assessed the relationship between use of BPs and COVID-19-related hospitalizations and COVID-19 diagnosis, as well as testing for SARS-CoV-2 infection (as a proxy for severe COVID-19 symptoms given the restricted access to testing during the initial surge). Outcomes were measured from March 1, 2020 to June 30, 2020, a period that roughly coincided with the first wave of COVID-19 in the U.S. and predated the advent of potential outcome modifiers, such as vaccines or other effective treatment options.

**eLife digest** The COVID-19 pandemic challenged the world to rapidly develop strategies to combat the virus responsible for the disease. While several effective vaccines and new drugs have since become available, these therapies are not always easy to access and take time to generate and distribute. To address these challenges, researchers have tried to find ways to repurpose existing medications that are already commonly used and known to be safe.

One potential candidate are bisphosphonates, a family of drugs used to reduce bone loss in patients with osteoporosis. Bisphosphonates have been shown to boost the immune response to viral infections, and it has been observed that patients prescribed these drugs are less likely to develop or die from pneumonia. But whether bisphosphonates are effective against COVID-19 had not been fully explored.

To investigate, Thompson, Wang et al. analyzed insurance claims data from about 8 million patients between January 2019 and June 2020, including around 450,000 individuals that had filled a prescription for bisphosphonates. Patients prescribed bisphosphonates were then compared to non-users that were similar in terms of their gender, age, the type of health insurance they had, their access to healthcare, and other health comorbidities.

The study revealed that bisphosphonate users were around three to five times less likely to be tested for, diagnosed with, or hospitalized for COVID-19 during the first four months of the pandemic. They were also less commonly diagnosed with other respiratory infections in 2019, like bronchitis or pneumonia.

Although the results suggest that bisphosphonates provide some protection against COVID-19, they cannot directly prove it. Verifying that bisphosphonates can treat or prevent COVID-19 and/or other respiratory infections requires more studies that follow patients in real-time rather than studying previously collected data.

If such studies confirm the link, bisphosphonates could be a helpful tool to protect against COVID-19 or other virus outbreaks. The drugs are widely available, safe, and affordable, and therefore may provide an alternative for patients who cannot access other medications or vaccines.

## Methods

### Study design

A retrospective cohort study was performed using health insurance claims data from January 1, 2019 to June 30, 2020 (study period) in order to assess the relationship between use of BPs and three COVID-19-related outcomes: (a) testing for SARS-CoV-2 infection; (b) COVID-19 diagnosis; and (c) hospitalization with a COVID-19 diagnosis, whereby COVID-19-related hospitalization was deemed the primary endpoint and COVID-19 diagnosis and testing were secondary endpoints. Primary and secondary endpoints were assessed during the observation period of March 1, 2020 to June 30, 2020, roughly corresponding to the first nation-wide surge of COVID-19 in the U.S. (*Figure 1A*). In the primary analysis, the risk of COVID-19-related outcomes was assessed among BP users compared to a matched sample of BP non-users with similar demographic and clinical characteristics.

### Data source

Data used for this study included closed medical (inpatient and outpatient) and outpatient-pharmacy-dispensed claims between January 1, 2019 and June 30, 2020, from the Komodo Health payer-complete dataset (https://www.komodohealth.com). This dataset is derived from over 150 private insurers in the U.S. and includes patients with commercial, individual, state exchange-purchased, Medicare Advantage, and Medicaid managed-care insurance coverage. The dataset also provides information on insurance eligibility periods. Closed claims within this dataset represent those that had undergone insurance adjudication. In total, the Komodo Health payer-complete dataset includes health insurance claims data from over 140 million individuals in the U.S. from 2015 to 2020.

### Cohort definition

All patients were required to have continuous medical and prescription insurance eligibility during the entire study period. Patients with missing information for age, gender, insurance type, or state/region were excluded.

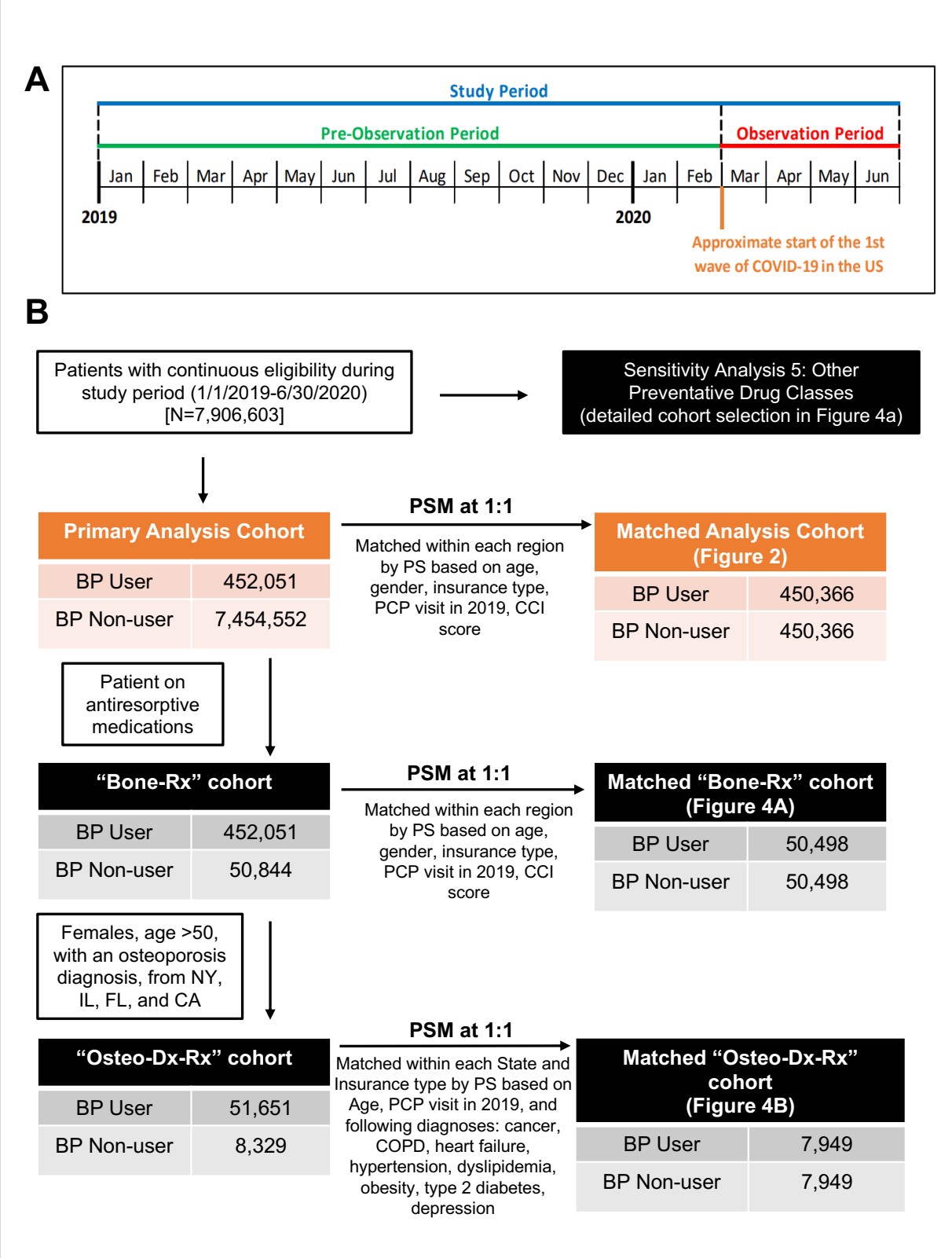

**Figure 1.** Study periods, cohort selection, and analyses of BP use on COVID-19-related outcomes. (**A**) Schematic overview of the study timeline. (**B**) Schematic flow diagram illustrating the identification of the study population and matched control populations for primary analysis and sensitivity analyses cohorts. *BP: bisphosphonate; CA: California; CCI: Charlson comorbidity index; CI: confidence interval; COPD: chronic obstructive pulmonary disease; FL: Florida; IL: Illinois; NY: New York; OR: odds ratio; PCP: primary care physician; PS: propensity score; PSM: propensity score match.*

## Exposures of interest

The primary exposure of interest was the use of any amino- or non-amino BP medication. Exposure to BPs and all other medications of interest were assessed over a 14-month pre-observation period preceding the COVID-19 pandemic in the U.S. This long duration was chosen because of the extended bioavailability of BPs, which accumulate in bone where they are retained and slowly released for up to several years (*Cremers et al., 2019*). Patients were classified as BP users if they had any claim at any time during the pre-observation period for one of the following: alendronate, alendronic acid, etidronate, ibandronate, ibandronic acid, pamidronate, risedronate, and zoledronic acid (full details in **Appendix 1**).

## Timing of BP dose

The effect of timing and formulation of BPs on COVID-19-related outcomes was more closely examined by varying the window between BP exposure and outcome measurement. The primary analysis BP user cohort, along with their propensity-score matched (see below for cohort matching) BP non-user cohort, were stratified as follows: two cohorts were used as the reference comparator with known BP-exposure during all or most of the pre-observation and the entire observation period, specifically (i) BP users who took oral alendronic acid (dosed daily or weekly) throughout the pre-observation period (i.e. at least one claim or drug-on-hand in each quarter in 2019 and in Jan/Feb. 2020) that also had a days-supply extending past June 30, 2020, and (ii) users of infusion zoledronic acid (dosed annually) with a claim in Q3 or Q4 2019; two cohorts with BP-exposure only during the pre-observation period, namely (iii) users of alendronic acid occurring during the first six months of 2019 with days-covered ending prior to June 30, 2019 and no other BP claims thereafter, and (iv) users of zoledronic acid in January or February 2019 with no other BP claims during the remainder of the study period; and, two cohorts with short-term BP exposure, specifically new users of (v) alendronic acid or (vi) zoledronic acid in February 2020, with no prior BP claims during the pre-observation period.

## Covariates

As covariates, we considered factors that may influence either the use of BPs or potential modulators of primary or secondary study endpoints. These included: age; gender; insurance type (commercial, dual, Medicaid, Medicare); having had any primary care physician (PCP) visit in 2019; and comorbidity burden. The variable 'PCP visit in 2019' was used to control for prior healthcare-use behaviour and was assigned based on any physician office claim from January 1, 2019 to December 31, 2019 with one of the following provider types: family practice, general practice, geriatric medicine, internal medicine, and preventive medicine. Comorbidity score assignment was calculated following the Charlson Comorbidity Index (CCI) methodology (*Quan et al., 2005*), and was based on diagnosis codes present on any medical claim (inpatient or outpatient) occurring during the pre-observation period. The assigned CCI score was used as the comorbidity covariate for the primary cohort propensity score matching, but to better control for differences in comorbidity burden when assessing outcomes, all regression analyses involving the primary analysis cohort included the following individual comorbidity covariates in lieu of the aggregate CCI score: osteoporosis, cancer, chronic obstructive pulmonary disease (COPD), depression, dyslipidaemia, hypertension, obesity, type 2 diabetes, cardiovascular disease overall, sickle cell anemia, stroke, dementia, HIV/AIDS, chronic kidney disease/end-stage renal disease (CKD/ESRD), and liver disease (**Appendix 1**).

## Cohort matching

For the primary analysis, BP users were propensity-score (PS) matched to BP non-users via a PS calculated using multiple variables, including age, gender, insurance type, CCI, and any PCP visit in 2019, to yield comparable populations by demographics and clinical characteristics (*Figure 1B*). To account for the differential geographic spread of COVID-19 across the U.S. during the observation period, matching was performed within each geographic region separately (Northeast, Midwest, South, West) and then combined. In addition to this within-region stratified match, a cohort build was also performed after restricting to patients from New York (NY) state only, since this state was the site of the largest outbreak in the initial COVID-19 surge in the U.S. All matching algorithms used a greedy-match propensity score technique (*Parsons, 2001*) to match BP users to non-users with a maximum permitted propensity-score difference of 0.015.

## Definition of endpoints

Primary and secondary endpoints were assigned using inpatient and outpatient medical claims that occurred during the four-month observation period. The primary endpoint, COVID-19-related hospitalization, was assigned based on the presence of an International Classification of Diseases, Tenth Revision (ICD-10) code on any inpatient medical service claim indicating test-confirmed 2019 Novel Coronavirus (2019-nCoV) acute respiratory disease, specifically U07.1. The first secondary endpoint, SARS-CoV-2 testing, was assigned using Current Procedural Terminology (CPT) codes indicating a test for active infection, specifically 87635, 87636, and 87637. The second secondary endpoint, COVID-19-related diagnosis, was assigned based on any medical service claim with the ICD-10 diagnosis code U07.1.

## Statistical analysis

Unadjusted analyses assessing the association between BP-use and COVID-19-related outcomes were performed for the primary analysis cohort using chi-square tests for categorical variables and calculation of the crude unadjusted odds ratio (OR) in the matched cohort groups overall, when stratified by region and in NY state alone, and when further stratified by age group and gender. Chi-square tests for categorical variables and t-tests for continuous variables were also performed to assess differences in demographic and clinical characteristics of BP users compared to BP non-users both pre-match and post-match to assess the success of the propensity-score match.

Multivariate logistic regression analyses, modelled separately to determine the adjusted OR for each COVID-19-related primary and secondary outcome while adjusting for demographic and clinical characteristics, were performed on the matched primary analysis cohort with all regions combined, when stratified by region, and in NY state alone. The primary exposure of interest was BP-use (yes/no) during the pre-observation period. Additional demographic/clinical characteristics also included as regression model covariates were: age group, gender, region (for all regions-combined analyses), insurance type, PCP visit in 2019, and the following comorbid conditions: osteoporosis, cancer, COPD, depression, dyslipidaemia, hypertension, obesity, type 2 diabetes, cardiovascular disease overall, sickle cell anaemia, stroke, dementia, HIV/AIDS, CKD/ESRD, and liver disease. Demographic characteristics used in the matching procedure were also included in the final outcome regressions to control for the impact of those characteristics on outcomes modelled.

All tests were two-tailed, and p-values of less than 0.05 were considered significant. All analyses were performed using SAS 9.4 (Cary, NC).

## Sensitivity analyses

Multiple sensitivity analyses were performed to assess the reliability of the primary analysis results and/or to address potential unmeasured confounding (full details in **Appendix 1**).

1. The first sensitivity analysis addressed potential confounding by indication (i.e. the possibility of the indication for BP use rather than BP use itself being responsible for differences in outcomes among BP users and non-users) by restricting the control group to an active comparator cohort of patients who had used non-BP anti-resorptive bone medications during the pre-observation period. Users of non-BP anti-resorptive bone medications, the smaller patient population, were then 1:1 matched to BP users, providing a sample where all patients had used bone health medications during the pre-observation period ('*Bone-Rx*' cohort) (*Figure 1B*). Cohort matching and regression modelling were performed following the same methodology employed for the primary analysis.
2. The second sensitivity analysis further addressed potential baseline differences between users of BPs and users of non-BP anti-resorptive bone medications in terms of indication for treatment and risk of SARS-CoV-2 exposure. To homogenise indication for treatment, we restricted the 'Bone-Rx' cohort to females aged older than 50 years with an osteoporosis diagnosis (ICD-10: M80.x, M81.x, M82.x), which is the main (but not the only) indication for use of anti-resorptive bone medications. In order to homogenise risk of COVID-19 exposure, we additionally (a) restricted both groups to residents of New York, Illinois, Florida, and California (four states with a high incidence of COVID-19 cases during the observation period, with each representing a geographic region) (*CDC, 2021a*), and (b) matched within each state by insurance-type strata (i.e. BP non-users matched to BP users with Medicaid coverage residing in New York) to control for differences in socioeconomic characteristics. Non-BP anti-resorptive bone medication users were then matched to BP users by age, PCP visit in 2019, and the following select comorbid conditions that include those thought to impact COVID-19 severity: cancer, COPD, depression,

dyslipidaemia, heart failure, hypertension, obesity, and type 2 diabetes (*Rosenthal et al., 2020*). In addition to assessing COVID-19-related outcomes, the matched cohorts that resulted from this analysis, older female patients from New York, Illinois, Florida, or California with a diagnosis of osteoporosis who were users of BP or non-BP anti-resorptive medications ('*Osteo-Dx-Rx*' cohort), were used for the third and fourth sensitivity analyses (see below).

3. The third sensitivity analysis assessed the relationship between BP-use and exploratory positive control outcomes (anticipated to be impacted by the immunomodulatory pharmacological mechanism of BPs) occurring in 2019. For this analysis, the primary, '*Bone-Rx*', and *Osteo-Dx-Rx*" cohorts were restricted to BP users who had any BP claim during the first half of 2019 and their previously-assigned BP non-user matched pair to assess the relationship between BP-use and medical services for other respiratory infectious diseases (acute bronchitis, pneumonia).

4. The fourth sensitivity analysis addressed potential bias due to the 'healthy adherer' effect, whereby users of a preventive drug may have better disease outcomes due to their healthier behaviours rather than due to drug treatment itself (*Ladova et al., 2014*). Two strategies were employed to validate the findings from our primary analysis while controlling for the potential impact of healthy adherer effect-associated bias. First, we tested whether effects observed with exposure to BPs were similarly observed with exposure to other preventive drugs, namely statins, antihypertensives, antidiabetics, and antidepressants. Second, we assessed whether the association between BP-use and COVID-19-related outcomes was maintained among the matched user/non-user populations of these other preventive drugs, i.e. BP users were compared to BP non-users within, for example, the statin user population and separately within the matched statin non-user population.

## Results

### Study population

A total of 8,239,790 patients met the inclusion criterion of continuous medical and prescription insurance eligibility over the full study period, of which 333,107 were excluded due to missing demographic information, resulting in a total eligible sample of 7,906,603 patients (*Figure 1B*). Of this full population, 452,051 (5.7%) and 7,454,552 (94.3%) patients were classified as BP users and BP non-users, respectively. Within BP users, more than 99% were prescribed an amino-BP, with oral alendronic acid (75.4%), zoledronic acid infusion (11.5%), and oral ibandronic acid (8.4%) as the most prevalent formulations (*Table 1*).

Prior to propensity-score matching, there were significant differences between BP users and non-users across all demographic and clinical characteristics. BP users were older (age >60: 82.7% vs 27.7%; p<0.001), predominantly female (91.0% vs 57.2%; p<0.001), with a higher comorbidity burden (mean CCI 0.95 vs 0.60; p<0.001), with a larger proportion of patients residing in the Western U.S. (21.1% vs 15.4%; p<0.001), covered by Medicare (43.3% vs 13.7%; p<0.001), and having visited a PCP in 2019 (63.8% versus 44.7%; p<0.001). Propensity-score matching yielded 450,366 BP users and 450,366 BP non-users with no significant differences across all characteristics used in matching (*Table 2*). Differences did exist, however, in the distribution of individual comorbid condition indicators that were used as covariates in the regression analysis, with the BP non-user cohort having a higher proportion of patients with COPD (10.2% vs 8.5%; p<0.001), cardiovascular disease (25.1% vs 18.7%; p<0.001), dyslipidemia (36.9% vs 34.6%; p<0.001), hypertension (46.4% vs 38.8%; p<0.001), obesity (10.3% vs 6.7%; p<0.001), and type 2 diabetes (22.9% vs 18.2%; p<0.001). Over 98% of all BP

**Table 1.** Most recent bisphosphonate claim among all users.

| Drug (route) | N | % |
| --- | --- | --- |
| Alendronate / alendronic acid (oral) | 340,810 | 75.4% |
| Etidronate (oral) | 14 | 0.0% |
| Ibandronate / ibandronic acid (oral) | 37,988 | 8.4% |
| Ibandronic acid (injection/infusion) | 1169 | 0.3% |
| Pamidronate (injection/infusion) | 1121 | 0.2% |
| Risedronate (oral) | 18,991 | 4.2% |
| Zoledronic acid (injection/infusion) | 51,958 | 11.5% |

**Table 2.** Primary analysis cohort (all regions), patient characteristics pre/post match.

| | All Observations Unmatched | | | | | | | All Observations Matched | | | | | | |
| | All | | BP Non-users | | BP Users | | p-value | All | | BP Non-users | | BP Users | | p-value |
| | N | % | N | % | N | % | | N | % | N | % | N | % | |
| All Patients | 7,906,603 | 100.00% | 7,454,552 | 94.30% | 452,051 | 5.70% | | 900,732 | 100.00% | 450,366 | 50.00% | 450,366 | 50.00% | |
| **Demographics** | | | | | | | | | | | | | | |
| **Age** | | | | | | | | | | | | | | |
| ≤20 | 1,840,050 | 23.30% | 1,838,922 | 24.70% | 1,128 | 0.20% | <0.001 | 2,253 | 0.30% | 1,125 | 0.20% | 1,128 | 0.30% | 1 |
| 21-40 | 1,446,999 | 18.30% | 1,443,908 | 19.40% | 3,091 | 0.70% | | 6,195 | 0.70% | 3,104 | 0.70% | 3,091 | 0.70% | |
| 41-50 | 925,309 | 11.70% | 916,758 | 12.30% | 8,551 | 1.90% | | 17,096 | 1.90% | 8,545 | 1.90% | 8,551 | 1.90% | |
| 51-60 | 1,250,190 | 15.80% | 1,184,469 | 15.90% | 65,721 | 14.50% | | 131,445 | 14.60% | 65,724 | 14.60% | 65,721 | 14.60% | |
| 61-70 | 1,181,261 | 14.90% | 1,024,383 | 13.70% | 156,878 | 34.70% | | 313,822 | 34.80% | 156,944 | 34.80% | 156,878 | 34.80% | |
| 71-80 | 783,775 | 9.90% | 642,050 | 8.60% | 141,725 | 31.40% | | 280,803 | 31.20% | 140,366 | 31.20% | 140,437 | 31.20% | |
| ≥81 | 479,019 | 6.10% | 404,062 | 5.40% | 74,957 | 16.60% | | 149,118 | 16.60% | 74,558 | 16.60% | 74,560 | 16.60% | |
| **Gender** | | | | | | | | | | | | | | |
| Female | 4,670,960 | 59.10% | 4,263,524 | 57.20% | 407,436 | 90.10% | <0.001 | 811,497 | 90.10% | 405,746 | 90.10% | 405,751 | 90.10% | 0.99 |
| Male | 3,235,643 | 40.90% | 3,191,028 | 42.80% | 44,615 | 9.90% | | 89,235 | 9.90% | 44,620 | 9.90% | 44,615 | 9.90% | |
| **Region** | | | | | | | | | | | | | | |
| Midwest | 1,467,802 | 18.60% | 1,391,835 | 18.70% | 75,967 | 16.80% | <0.001 | 151,802 | 16.90% | 75,901 | 16.90% | 75,901 | 16.90% | 1 |
| Northeast | 2,152,560 | 27.20% | 2,032,832 | 27.30% | 119,728 | 26.50% | | 238,988 | 26.50% | 119,494 | 26.50% | 119,494 | 26.50% | |
| South | 3,042,604 | 38.50% | 2,881,718 | 38.70% | 160,886 | 35.60% | | 319,408 | 35.50% | 159,704 | 35.50% | 159,704 | 35.50% | |
| West | 1,243,637 | 15.70% | 1,148,167 | 15.40% | 95,470 | 21.10% | | 190,534 | 21.20% | 95,267 | 21.20% | 95,267 | 21.20% | |
| **Insurance** | | | | | | | | | | | | | | |
| Commercial | 3,938,603 | 49.80% | 3,791,545 | 50.90% | 147,058 | 32.50% | <0.001 | 294,070 | 32.60% | 147,012 | 32.60% | 147,058 | 32.70% | 1 |
| Dual | 156,497 | 2.00% | 125,090 | 1.70% | 31,407 | 6.90% | | 59,936 | 6.70% | 29,980 | 6.70% | 29,956 | 6.70% | |
| Medicaid | 2,594,500 | 32.80% | 2,517,020 | 33.80% | 77,480 | 17.10% | | 154,519 | 17.20% | 77,272 | 17.20% | 77,247 | 17.20% | |
| Medicare | 1,217,003 | 15.40% | 1,020,897 | 13.70% | 196,106 | 43.40% | | 392,207 | 43.50% | 196,102 | 43.50% | 196,105 | 43.50% | |
| **PCP Visit 2019** | | | | | | | | | | | | | | |
| No | 4,283,697 | 54.20% | 4,119,831 | 55.30% | 163,866 | 36.20% | <0.001 | 327,383 | 36.30% | 163,659 | 36.30% | 163,724 | 36.40% | 0.89 |
| Yes | 3,622,906 | 45.80% | 3,334,721 | 44.70% | 288,185 | 63.80% | | 573,349 | 63.70% | 286,707 | 63.70% | 286,642 | 63.60% | |

*Table 2 continued on next page*

*Table 2 continued*

### Clinical Characteristics

| Characteristic | All Observations Unmatched | | | | | | | All Observations Matched | | | | | | |
|---|---|---|---|---|---|---|---|---|---|---|---|---|---|---|
| | mean / N | SD / % | mean / N | SD / % | mean / N | SD / % | p-value | mean / N | SD / % | mean / N | SD / % | mean / N | SD / % | p-value |
| CCI | 0.62 | 1.38 | 0.6 | 1.35 | 0.95 | 1.76 | <0.001 | 0.95 | 1.76 | 0.95 | 1.76 | 0.95 | 1.76 | 0.7 |

### Regression Comorbidity Covariates

| Characteristic | All Observations Unmatched | | | | | | | All Observations Matched | | | | | | |
|---|---|---|---|---|---|---|---|---|---|---|---|---|---|---|
| | N | % | N | % | N | % | p-value | N | % | N | % | N | % | p-value |
| Osteoporosis | 267,020 | 3.40% | 135,231 | 1.80% | 131,789 | 29.20% | <0.001 | 163,814 | 18.20% | 32,390 | 7.20% | 131,424 | 29.20% | <0.001 |
| Cancer | 419,083 | 5.30% | 366,786 | 4.90% | 52,297 | 11.60% | <0.001 | 94,148 | 10.50% | 41,861 | 9.30% | 52,287 | 11.60% | <0.001 |
| CKD/ESRD | 361,451 | 4.60% | 328,633 | 4.40% | 32,818 | 7.30% | <0.001 | 68,999 | 7.70% | 36,182 | 8.00% | 32,817 | 7.30% | <0.001 |
| COPD | 466,094 | 5.90% | 427,850 | 5.70% | 38,244 | 8.50% | <0.001 | 84,234 | 9.40% | 45,990 | 10.20% | 38,244 | 8.50% | <0.001 |
| CVD | 1,084,031 | 13.70% | 999,526 | 13.40% | 84,505 | 18.70% | <0.001 | 197,243 | 21.90% | 112,933 | 25.10% | 84,310 | 18.70% | <0.001 |
| Dementia | 125,811 | 1.60% | 113,778 | 1.50% | 12,033 | 2.70% | <0.001 | 24,921 | 2.80% | 12,889 | 2.90% | 12,032 | 2.70% | <0.001 |
| Depression | 571,303 | 7.20% | 531,355 | 7.10% | 39,948 | 8.80% | <0.001 | 86,280 | 9.60% | 46,431 | 10.30% | 39,849 | 8.80% | <0.001 |
| Dyslipidemia | 1,532,254 | 19.40% | 1,375,920 | 18.50% | 156,334 | 34.60% | <0.001 | 322,125 | 35.80% | 166,360 | 36.90% | 155,765 | 34.60% | <0.001 |
| HIV/AIDS | 33,229 | 0.40% | 31,711 | 0.40% | 1518 | 0.30% | <0.001 | 2897 | 0.30% | 1379 | 0.30% | 1,518 | 0.30% | 0.01 |
| Hypertension | 1,899,063 | 24.00% | 1,723,519 | 23.10% | 175,544 | 38.80% | <0.001 | 384,059 | 42.60% | 209,184 | 46.40% | 174,875 | 38.80% | <0.001 |
| Liver Disease | 251,331 | 3.20% | 231,664 | 3.10% | 19,667 | 4.40% | <0.001 | 38,697 | 4.30% | 19,031 | 4.20% | 19,666 | 4.40% | 0.001 |
| Obesity | 638,506 | 8.10% | 608,083 | 8.20% | 30,423 | 6.70% | <0.001 | 76,844 | 8.50% | 46,498 | 10.30% | 30,346 | 6.70% | <0.001 |
| Sickle Cell Anemia | 10,499 | 0.10% | 10,292 | 0.10% | 207 | 0.00% | <0.001 | 422 | 0.00% | 215 | 0.00% | 207 | 0.00% | 0.7 |
| Stroke | 104,859 | 1.30% | 97,001 | 1.30% | 7,858 | 1.70% | <0.001 | 19,395 | 2.20% | 11,569 | 2.60% | 7,826 | 1.70% | <0.001 |
| Type 2 Diabetes | 978,239 | 12.40% | 895,983 | 12.00% | 82,256 | 18.20% | <0.001 | 184,978 | 20.50% | 103,031 | 22.90% | 81,947 | 18.20% | <0.001 |

user/non-user matches for the primary analysis cohort were completed with differences in matched propensity scores <0.000001 (overall mean difference of 0.000004, max difference of 0.0147).

Similar profiles in pre-match *versus* post-match characteristics were seen when patients were stratified by region or restricted to NY-state (*Appendix 2—tables 1–3*, *Appendix 2—table 4*, *Appendix 2—table 5*). Demographic distributions, including differences between BP user *versus* BP non-user characteristics pre-match *versus* post-match characteristics were seens pre- and post-matching for all sensitivity analysis cohorts are detailed in **Appendix 2**.

## BP use and COVID-19-related outcomes

Among the full matched cohort, BP users had significantly lower rates and unadjusted (crude) odds of testing (1.2% vs 5.1%; OR = 0.22; 95%CI:0.21–0.22; p<0.001), diagnosis (0.7% vs 2.9%; OR = 0.22; 95%CI:0.21–0.23; p<0.001), and hospitalization (0.2% vs 0.7%; OR = 0.24; 95%CI:0.22–0.26; p<0.001) as compared to BP non-users (*Figure 2* and *Appendix 3—figure 1*). Consistent findings were seen when sub-stratifying the full matched cohort by age, gender, age*gender, within grouped regions, by individual region, and in NY-state alone (*Appendix 2—tables 6–11*).

Multivariate regression analyses yielded similar results for all outcomes while additionally controlling for patient demographic and comorbidity characteristics. In the full matched cohort, BP users had lower adjusted odds of testing (OR = 0.22; 95%CI:0.21–0.23; p<0.001), diagnosis (OR = 0.23; 95%CI:0.22–0.24; p<0.001), and hospitalizations (OR = 0.26; 95%CI:0.24–0.29; p<0.001). These findings were robust when comparing BP users with BP non-users when stratified by geographic region or NY-state alone.

## Timing of last BP exposure and COVID-19-related outcomes

The above results demonstrate that any BP exposure during the 14-months pre-observation period is associated with a marked reduction in each of the three COVID-19-related outcomes. To further investigate the relationship between COVID-19-related outcomes and the timing of BP exposure, we focused on the two most commonly prescribed BPs, alendronic acid (oral formulation dosed daily or weekly) and zoledronic acid (infusion dosed annually). For each BP type, COVID-19-related outcomes were assessed among users: (i-ii) with exposure or days covered (based on prescription frequency) during the pre-observation period and throughout the observation period; (iii-iv) with exposure or days covered ending prior to the observation period; and (v-vi) newly initiating therapy prior to the observation period (*Figure 3A*). Furthermore, all subgroups of BP users had decreased odds of COVID-19-related outcomes (*Figure 3B*) except for the odds of hospitalization among zoledronic acid users who were last dosed in January/February of 2019 (OR = 0.52; 95%CI:0.20–1.40; p=0.20) or newly initiated in February of 2020 (OR = 0.49; 95%CI:0.13–1.88; p=0.30).

## Sensitivity analysis 1: COVID-19-related outcomes among all users of anti-resorptive medications ('Bone-Rx' cohort)

The first sensitivity analysis was performed to address potential confounding by indication. To validate our primary findings in more comparable cohorts, analysis was restricted to comparing BP users to patients using non-BP anti-resorptive bone medications during the pre-observation period. Compared to non-BP users of anti-resorptive medications, BP users had decreased odds of testing (OR = 0.31; 95%CI:0.28–0.33; p<0.001), diagnosis (OR = 0.35; 95%CI:0.31–0.38; p<0.001), and hospitalization (OR = 0.45; 95%CI:0.36–0.56; p<0.001) (*Figure 4A* and *Appendix 3—figure 2*). Furthermore, these findings were robust when assessed separately across every geographic region as well as NY state for all outcomes except hospitalizations when restricted to the Western U.S. (p=0.08; *Appendix 2—table 12*).

## Sensitivity analysis 2: COVID-19-related outcomes among users of anti-resorptive medications with a diagnosis of osteoporosis ('Osteo-Dx-Rx' cohort)

The second sensitivity analysis was performed to address the fact that, even after restricting the comparator cohort to users of anti-resorptive medications, differences may still exist between patient cohorts that could affect COVID-19-related outcomes, including different indications for anti-resorptive medication use and other uncontrolled patient characteristics. To address this, the association between BP use and COVID-19 related outcomes were examined in a cohort restricted to

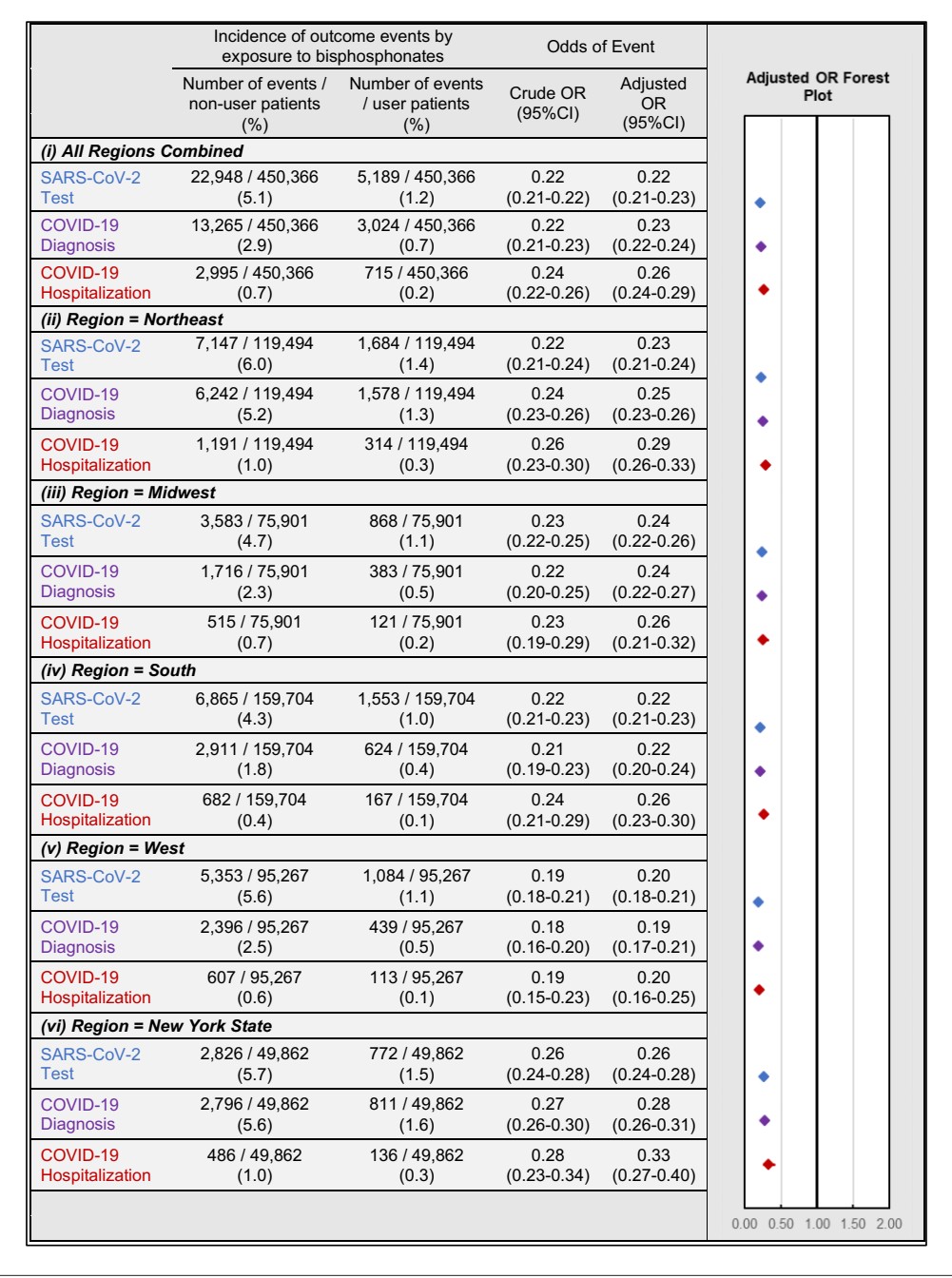

**Figure 2.** Association of BP use and COVID-19-related outcomes incidence (left) and regression-adjusted results for odds (right) of SARS-CoV-2 testing (blue), COVID-19 diagnosis (purple), and COVID-19-related hospitalizations (red) of BP users compared with BP non-users in the all-regions combined primary analysis cohort (i) and when stratified by region/state into: Northeast (ii), Midwest (iii), South (iv), West (v), and New York state (vi). For details see *Figure 2—source data 1*.

The online version of this article includes the following source data for figure 2:

**Source data 1.** COVID-19-related outcomes in the primary analysis cohort.

female patients over 50 years old, with a diagnosis of osteoporosis, using either a BP or a non-BP anti-resorptive bone medication, matched within insurance-type as a proxy for socioeconomic status, and selected from four states (NY, IL, FL, CA) with high incidences of COVID-19 cases during the observation period (*CDC, 2021a*; 'Osteo-Dx-Rx' cohort). In agreement with the results reported above, the

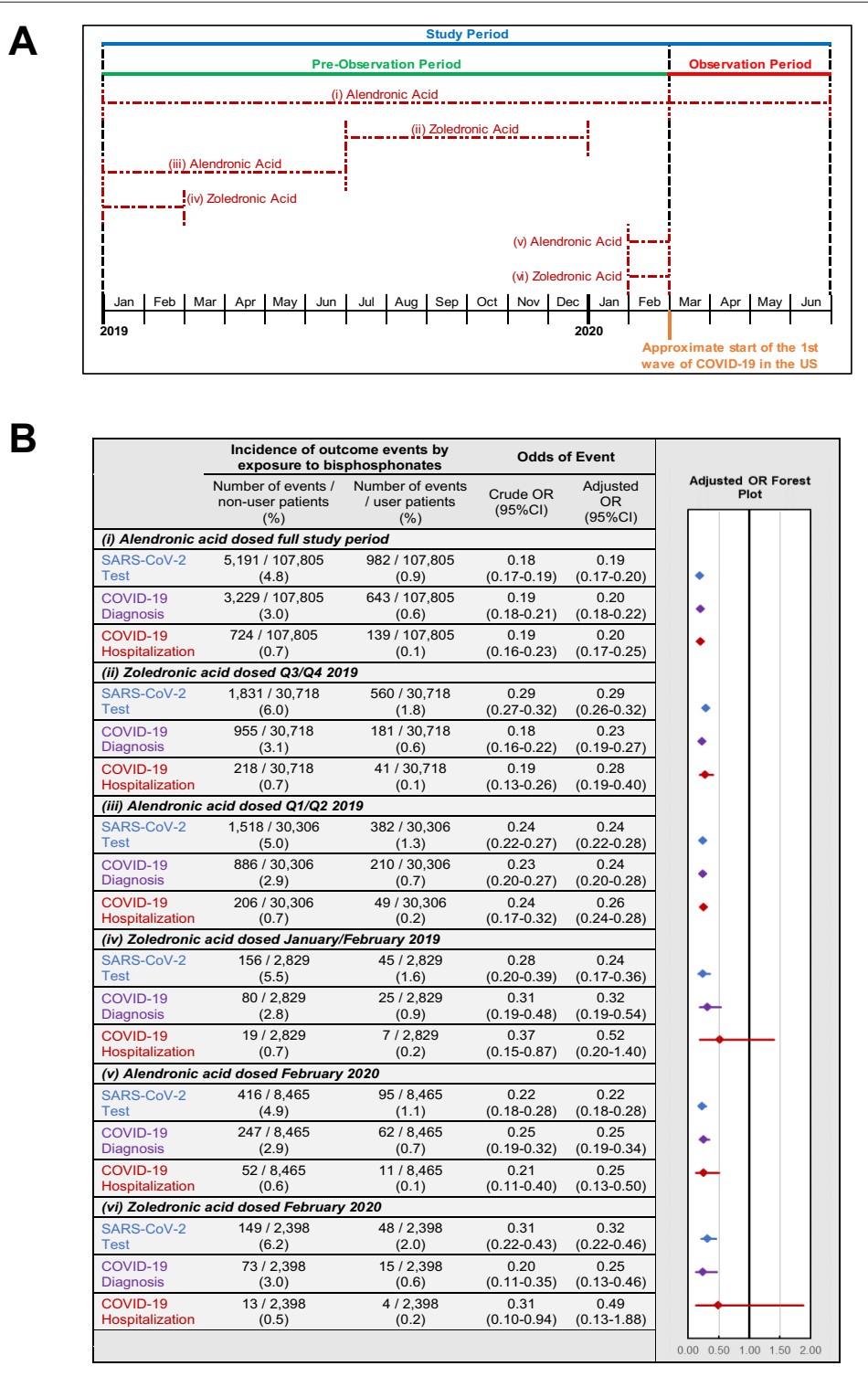

**Figure 3.** Timing of BP use and COVID-19-related outcomes. (A) Schematic of BP user sub-stratification by timing of exposure to alendronic acid or zoledronic acid prior to outcome assessment. Broken lines represent periods of active BP dosing. For zoledronic acid users, days covered was considered to extend 1 year past the dosing period based on dosing guidelines. (B) Incidence (left) and regression-adjusted results (right) for odds of SARS-CoV-2 testing, COVID-19 diagnosis, and COVID-19-related hospitalizations of BP users compared with BP non-users in pre-specified subgroups. For further details see *Figure 3—source data 1*. CI: confidence interval; OR: odds ratio.

The online version of this article includes the following source data for figure 3:

**Source data 1.** Primary analysis cohort by timing of BP dosing, COVID-19-related outcomes.

## A ("Bone-Rx" Cohort)

| | Incidence of outcome events by exposure to bisphosphonates | | Odds of Event | | Adjusted OR Forest Plot |
|---|---|---|---|---|---|
| | Number of events / non-user patients (%) | Number of events / user patients (%) | Crude OR (95%CI) | Adjusted OR (95%CI) | |
| **All Regions Combined** | | | | | |
| SARS-CoV-2 Test | 2,438 / 50,498 (4.8) | 760 / 50,498 (1.5) | 0.30 (0.28-0.33) | 0.31 (0.28-0.33) | |
| COVID-19 Diagnosis | 1,307 / 50,498 (2.6) | 461 / 50,498 (0.9) | 0.35 (0.31-0.39) | 0.35 (0.31-0.38) | |
| COVID-19 Hospitalization | 276 / 50,498 (0.5) | 123 / 50,498 (0.2) | 0.44 (0.36-0.55) | 0.45 (0.36-0.56) | |

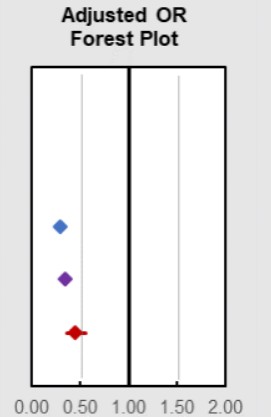

## B "Osteo-Dx-Rx" Cohort

| | Incidence of outcome events by exposure to bisphosphonates | | Odds of Event | | Adjusted OR Forest Plot |
|---|---|---|---|---|---|
| | Number of events / non-user patients (%) | Number of events / user patients (%) | Crude OR (95%CI) | Adjusted OR (95%CI) | |
| **All Regions Combined** | | | | | |
| SARS-CoV-2 Test | 395 / 7,949 (5.0) | 115 / 7,949 (1.4) | 0.28 (0.23-0.35) | 0.28 (0.23-0.35) | |
| COVID-19 Diagnosis | 300 / 7,949 (3.8) | 121 / 7,949 (1.5) | 0.40 (0.32-0.49) | 0.40 (0.32-0.49) | |
| COVID-19 Hospitalization | 47 / 7,949 (0.6) | 21 / 7,949 (0.3) | 0.45 (0.27-0.75) | 0.45 (0.26-0.75) | |

**Figure 4.** COVID-19-related outcomes among the Bone-RX and Osteo-Dx-Rx restricted cohorts. Incidence and forest plots summarizing regression-adjusted odds ratios of SARS-CoV-2 testing (blue), COVID-19 diagnosis (purple), and COVID-19-related hospitalizations (red) in the (**A**) '*Bone-Rx*' (see also *Figure 4—source data 1*) and (**B**) '*Osteo-Dx-Rx*' sensitivity analysis cohorts (see also *Figure 4—source data 2*).

The online version of this article includes the following source data for figure 4:

**Source data 1.** Source data for *Figure 4A*: Bone-Rx cohort COVID-19-related outcomes.

**Source data 2.** Source data for *Figure 4B*: Osteo-Dx-Rx cohort COVID-19-related outcomes.

decrease in odds of COVID-19-related outcomes in BP users remained robust for testing (OR = 0.28; 95%CI:0.23–0.35; p<0.001), diagnosis (OR = 0.40; 95%CI:0.32–0.49; p<0.001), and hospitalizations (OR = 0.45; 95%CI:0.26–0.75; p=0.003) (*Figure 4B*).

### Sensitivity analysis 3: Association of BP-use with exploratory positive control outcomes

The third sensitivity analysis was performed to assess if there is an association between BP-use and incidence of other respiratory infections, which has been previously reported (*Sing et al., 2020*). Medical services for acute bronchitis or pneumonia were measured during the second half of 2019,

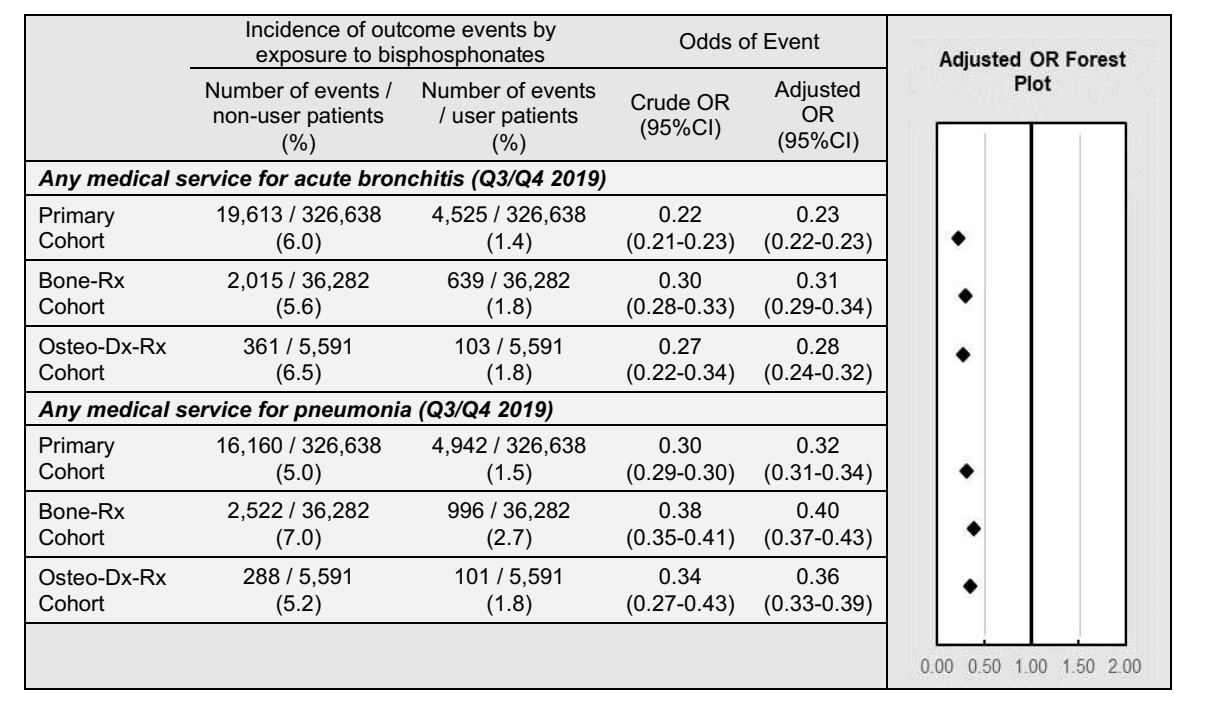

**Figure 5.** Exploratory outcomes among BP users versus BP non-users. Incidence and adjusted odds ratios of other respiratory infections, in the primary, '*Bone-Rx*', and '*Osteo-Dx-Rx*' cohorts. For details, see *Figure 5—source data 1*. CI: confidence interval; OR: odds ratio.

The online version of this article includes the following source data for figure 5:

**Source data 1.** Positive control outcomes by primary, bone-Rx, and osteo-Dx-Rx cohorts.

prior to the advent of COVID-19, in the primary, '*Bone-Rx*', and '*Osteo-Dx-Rx*' cohorts. Regression modelling found that, among all cohort variations modelled, BP users had a decreased odds of any medical service related to acute bronchitis (point estimates of ORs ranged from 0.23 to 0.28) and pneumonia (point estimates of ORs ranged from 0.32 to 0.36) (*Figure 5*).

## Sensitivity analysis 4: Association of other preventive drugs with COVID-19-related outcomes

A potential pitfall in the interpretation of apparent effects of preventive medications on health outcomes is the so-called healthy adherer effect, whereby patients may have better outcomes due to their overall healthier behaviours and not due to active drug treatment itself (*Ladova et al., 2014*). To address this possibility of unmeasured confounding, a final sensitivity analysis was performed to evaluate the association between control exposures (i.e. use of other preventive medications such as statins, antihypertensives, antidiabetics, and antidepressants) and COVID-19-related outcomes (*Figure 6A*). In comparison to BPs, the impact of other preventive drug classes on COVID-19-related outcomes was much weaker overall (*Figure 6B–E*) and varied between geographic regions in terms of magnitude or direction (*Appendix 2—tables 13–16*). Furthermore, when assessing the impact of BP-use within matched user/non-user preventive drug cohorts (e.g. BP users compared to BP non-users among the matched statin user and statin non-user populations), we found BP-use to be consistently associated with lower odds of testing (point estimates of ORs ranged from 0.21 to 0.27), diagnosis (point estimates of ORs ranged from 0.22 to 0.30), and hospitalizations (point estimates of ORs ranged from 0.25 to 0.33) across all stratified preventive user/non-user cohorts (*Figure 6B–E*).

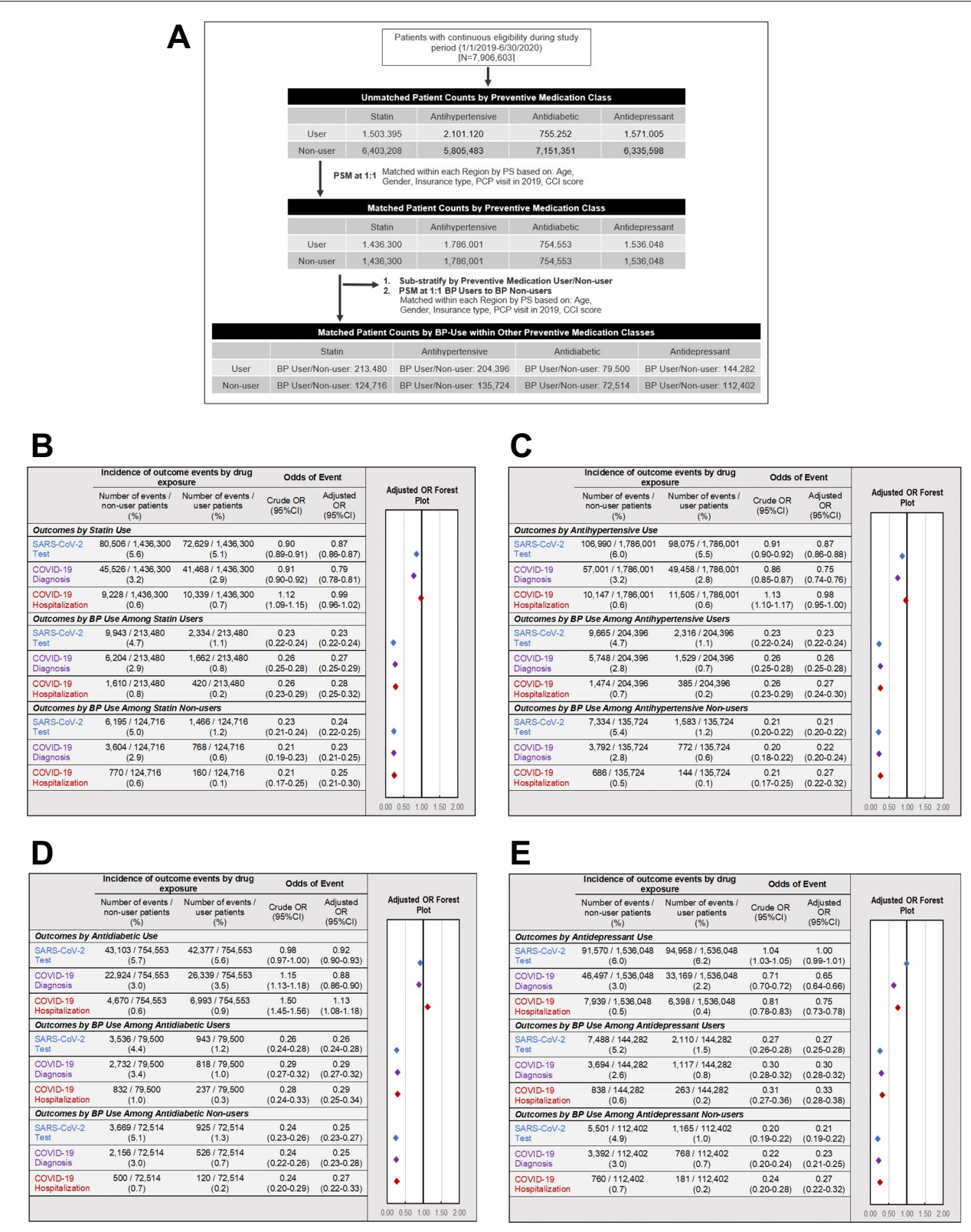

**Figure 6.** Association of other preventive drugs with COVID-19-related outcomes. (A). Schematic illustrating the identification of study populations and matched controls for each drug class. (**B–E**) Incidence and adjusted odds ratios of SARS-CoV-2 testing (blue), COVID-19 diagnosis (purple), and COVID-19-related hospitalizations (red) in users and non-users of (**B**) statins (see also *Figure 6—source data 1*), (**C**) antihypertensive medications (see also *Figure 6—source data 2*), (**D**) non-insulin antidiabetic medications (see also *Figure 6—source data 3*), and (**E**) antidepressant medications

*Figure 6 continued on next page*

*Figure 6 continued*

(see also *Figure 6—source data 4*). For each class of preventive medications, further analysis was performed comparing BP users and BP non-users within matched cohorts of medication users (middle) and medication non-users (bottom). *BP: bisphosphonate; CCI: Charlson comorbidity index; CI: confidence interval; COPD: chronic obstructive pulmonary disease; OR: odds ratio; PCP: primary care physician; PS: propensity score; PSM: propensity score match.*

The online version of this article includes the following source data for figure 6:

**Source data 1.** Source data for *Figure 6B*: COVID-19-related outcomes by statin use overall & sub-stratified by BP use.

**Source data 2.** Source data for *Figure 6C*: COVID-19-related outcomes by antihypertensive use overall & sub-stratified by BP use.

**Source data 3.** Source data for *Figure 6D*: COVID-19-related outcomes by antidiabetic use overall & sub-stratified by BP use.

**Source data 4.** Source data for *Figure 6E*: COVID-19-related outcomes by antidepressant use overall & sub-stratified by BP use.

## Discussion

This study examined the association between recent exposure to BPs and subsequent COVID-19-related outcomes during the initial outbreak of the COVID-19 pandemic in the U.S. Our findings demonstrate that amino-BP users experienced a three- to five-fold reduced incidence of SARS-CoV-2 testing, COVID-19 diagnosis, and COVID-19-related hospitalization during this period. This dramatic difference in outcomes was consistently observed when comparing BP users to BP non-users in a propensity score-matched general population, when comparing to users of other anti-resorptive bone medications, when further restricting the latter cohort to female osteoporosis patients matched by comorbidities within state of residence and by insurance type, and when comparing BP users to BP non-users stratified by use of other preventive medications. Therefore, although there are confounding-related limitations inherent within retrospective studies, the consistency and strength of our observed associations when using various methods to control for unmeasured confounding support the contention that further prospective research should be performed to determine the true magnitude of the potential immunomodulatory effects of BP use.

Our findings are consistent with previous observational studies, prior to the advent of COVID-19, that had reported associations between BP use and reduced incidence of pneumonia and pneumonia-related mortality (*Sing et al., 2020*; *Colón-Emeric et al., 2010*; *Reid et al., 2021*). Accordingly, we observed in our population that BP use was associated with decreased odds of medical services for acute bronchitis and pneumonia during the second half of 2019. Taken together, these findings suggest that BPs may play a protective role in respiratory tract infections from a variety of causes, including SARS-CoV-2.

Other recent retrospective studies have explored, to some extent, associations of anti-resorptive medication use and COVID-19-related outcomes, albeit in much smaller patient populations than were analysed here. One study found no differences in the COVID-19-related risk of hospitalization (70.7% vs 72.7%, p = 0.16) and mortality (11.9% vs 12.8%, p = 0.386) among 1,997 female patients diagnosed with COVID-19 who received anti-osteoporosis medication as compared to propensity score-matched COVID-19 patients who were not receiving such drugs (*Atmaca et al., 2022*). This study did not examine the incidence of COVID-19 among BP users, but it raises the possibility that the subset of BP users who do develop sufficient pathology to be diagnosed with COVID-19 may have a similar clinical course as BP non-users. Another retrospective cohort study in Italy examining the association of oral amino-BP use and incidence of COVID-19-related hospitalization found no difference between BP users (12.32% (95% CI, 9.61–15.04)) and BP non-users (11.55% (95% CI, 8.91–14.20)) (*Degli Esposti et al., 2021*). However, the overall incidence of COVID-19 hospitalization in the primary cohort (151/126,370 patients, or 0.12%) of this study was markedly lower than in the present analysis (3,710/900,732 patients, or 0.41%). A third study examined the influence of various anti-osteoporosis drugs, including BPs, on the cumulative incidence of COVID-19 in 2,102 patients with non-inflammatory rheumatic conditions that were compared to population estimates in the same geographic region (*Blanch-Rubió et al., 2020*). In this analysis, users of non-BP anti-resorptive medications and zoledronate, but not users of oral BPs, had a lower incidence and relative risk of COVID-19 diagnosis and hospitalization. The observations with zoledronate are consistent with the findings reported here. However, we did not detect a significant impact of non-BP anti-resorptive medications in comparison to BPs, and we found a robust association between oral BP use and lower odds of COVID-19 diagnosis and related hospitalization. The reason for these discrepancies is unclear

but could potentially reflect the large disparity in sample size between our study, which differed by more than three orders of magnitude. A fourth study, which used Israeli insurance data to perform an analysis involving two separate case-control matched cohorts to assess the risk of COVID-19 hospitalizations when stratified by recent medication use, also found that the odds COVID-19-related hospitalizations were lower among users of BPs, and ranged from an OR of 0.705 (95%CI: 0.522–0.935) to 0.567 (95%CI:0.400–0.789) (*Israel et al., 2021*).

The large size of our dataset allowed for a range of fully powered, stratified analyses to be performed to explore the robustness of our findings and to address unmeasured confounding factors and other sources of potential bias that can occur in retrospective studies using insurance claims data. Notwithstanding, a retrospective analysis of insurance claims data has inevitable limitations that should be considered. Specifically, there is the potential that key patient characteristics impacting outcomes could not be derived from claims data. For example, the interpretation of our findings depends, in part, on the assumption that BP users and non-users had a similar risk of SARS-CoV-2 infection during the observation period. However, our dataset does not allow us to restrict patient observations to those with known exposure to SARS-CoV-2. Therefore, to minimize potential differences in SARS-CoV-2 exposure between BP users and non-users in our primary study cohort, we implemented additional analytical strategies, including the sensitivity analyses, as well as matching BP users to BP non-users within geographical regions and specific states.

Despite these efforts, it is important to note that we have limited information to assess and match BP users to BP non-users by sociodemographic risk factors, such as socio-economic status and racial/ethnic minority status, that are associated with COVID-19 incidence and mortality (*Karmakar et al., 2021*; *Rogers et al., 2020*). Notably, Black/African-American and Hispanic patients have been shown to have significantly higher test positivity rates (*Kaufman et al., 2021*; *Escobar et al., 2021*; *Jacobson et al., 2021*; *Rubin-Miller, 2020*) and severity of disease at the time of testing (*Rubin-Miller, 2020*). Furthermore, Black/African American (*Azar et al., 2020*) and Hispanic patients were found to have a higher incidence of COVID-19 infection (*Escobar et al., 2021*; *CDC, 2021b*) and odds of COVID-19 related hospital admission even after adjustment for comorbidities (*Nau et al., 2021*), residence in a low-income area (*Rubin-Miller, 2020*), and insurance plan (*Azar et al., 2020*; *Price-Haywood et al., 2020*; *Muñoz-Price et al., 2020*). The greater COVID-19 burden in these groups is likely due to a combination of systemic health inequities as well as a disproportionate representation among essential workers (*Selden and Berdahl, 2020*; *US Bureau of Labor Statistics, 2019*), which could potentially increase their exposure risk to SARS-CoV-2. In addition, there are known variations in the prevalence of osteoporosis between different racial groups, which could potentially result in disproportionate frequencies of BP prescriptions (*No authors listed, 2021*). The potential confounding due to socio-economic status and differential prevalence of osteoporosis among racial/ethnic groups was addressed in our analysis of the '*Osteo-Dx-Rx*' cohort where we compared BP users to non-users after restricting to female patients with a diagnosis of osteoporosis, all using anti-resorptive bone medications, and matched by insurance type (proportion of Medicaid and dual Medicare/Medicaid users) as a proxy for social-economic status (*Figure 4B*). Nevertheless, this strategy cannot rigorously rule out a potential under-representation of groups with higher sociodemographic risk factors among BP users that could have contributed to the observed decreased odds of COVID-19 related outcomes in our primary analyses.

The potential bias introduced by a putative differential racial/ethnic group composition of BP users *versus* BP non-users is at least partially addressed by a recent study of a large Californian cohort of female BP users (*Black et al., 2020*). Compared to the racial composition of California at-large (a proxy for BP non-users) (*United States Census Bureau, 2019*), BP users were predominantly Non-Hispanic White (36.5% in California *versus* 53.3% among BP users). The proportions of Black/African-Americans and Asians among BP users in that study were similar to those in California at-large, whereas Hispanic patients represented a smaller percentage (24%) of BP users as compared to Hispanics in the state's general population (39.4%). Based on these findings and the reported differential case rates of COVID-19 infections among racial groups in California (*Reitsma et al., 2021*), we can estimate the race-adjusted incidence of COVID-19 in populations reflecting the composition of BP users and non-users (*Black et al., 2020*) to be 1.7% and 2.1%, respectively. By comparison, in our study the actual rate of COVID-19 diagnosis in the Western US was 2.5% for BP non-users *versus* 0.46% for BP users (*Figure 2*), indicating that the uneven representation of ethnic/racial groups cannot fully explain the observed differences in COVID-19 related outcomes. Moreover, we note that racial/ethnic minorities

are also under-represented among statin users (*Salami et al., 2017*), but statin-users in our primary cohort had similar odds of COVID-19 hospitalization as statin non-users (*Figure 6B*). Similarly, Black/African-Americans and Hispanics have lower utilization rates of antidepressants (*Chen and Rizzo, 2008*) and Hispanics were also reported to be undertreated with antihypertensive medications (*Gu et al., 2017*). Our analysis of COVID-19-related outcomes among users and non-users of antihypertensives showed a modest decrease in COVID-19 diagnosis and minimal association with COVID-19-related hospitalization (*Figure 6C*). By contrast, users of antidepressants had uniformly lower odds for both endpoints (*Figure 6E*), which is consistent with other recent studies (*Israel et al., 2021*; *Hoertel et al., 2021*; *Zimniak et al., 2021*). However, regardless of the class of non-BP preventive drugs analysed, concomitant BP use was consistently associated with dramatically decreased odds of COVID-19 diagnosis and hospitalization as well as testing for SARS-CoV-2 (*Figure 6B–E*).

Furthermore, specifically looking at the rate of SARS-CoV-2 testing in California (*Escobar et al., 2021*; *Jacobson et al., 2021*) or nation-wide (*Kaufman et al., 2021*), the proportions of different racial and ethnic groups among tested patients were nearly identical to estimates for the state or national population. Thus, the observed association between BP use and reduced testing for SARS-CoV-2 infection in our nation-wide cohorts is unlikely to be explained by potential differences in racial composition between BP users and non-users. It also seems unlikely that exposure to BPs reduces the actual incidence of SARS-CoV-2 infections. More likely, we propose that immune-modulatory effects of BPs may enhance the anti-viral response of BP users to SARS-CoV-2 and mitigate the development of symptoms. Milder or absent symptoms may have caused infected BP users to be less likely to seek testing. Moreover, because there was a nationwide shortage of available tests for SARS-CoV-2 during the observation period, patients needed to present with sufficiently severe disease symptoms to be eligible for testing, so fewer test-seeking BP users may have qualified. Consequently, a larger proportion of uncaptured 'silent' infections among BP users could explain why fewer diagnoses and hospitalizations were observed in this group.

The scarceness of COVID-19 tests combined with the strain on healthcare systems during the observation period could potentially have resulted in a misclassification bias whereby some patients may have been falsely diagnosed and/or hospitalized with COVID-19 without having received a confirmatory test. However, this bias should equally affect BP users and BP non-users and bias our findings towards the null. Relatedly, limited hospital capacity during the observation period could have led to rationing of inpatient hospital beds based on severity of disease and likelihood to survive (*Emanuel et al., 2020*). However, matching by age and comorbidities should produce patient populations with similar characteristics used for rationing.

A further limitation of our study is the lack of information on the result of COVID-19 tests received by patients. Therefore, as discussed above, the incidence and odds of COVID-19 testing should not be viewed as a proxy for the rate of infection, but rather reflects the incidence of patients with severe enough symptoms or exposure to warrant testing. Another potential source of confounding is the possibility that some patients in our study were classified as BP non-users due to the absence of BP exposure during the pre-observation period but may have received a BP during the observation period. The potential misclassification of BP non-users, however, would bias towards the null hypothesis, and was only seen in 1.92% of the matched BP non-user population.

An additional limitation is potential censoring of patients who died during the observation period, resulting in truncated insurance eligibility and exclusion based on the continuous insurance eligibility requirement. However, modelling the impact of censoring by using death rates observed in BP users and non-users in the first six months of 2020 and attributing all deaths as COVID-19-related did not significantly alter the decreased odds of COVID-19 diagnosis in BP users (see **Appendix 3**).

Another limitation in the current study is related to a potential 'double correction' of patient characteristics that were included in both the propensity score matching procedure as well as the outcome regression modelling, which could lead to overfitting of the regression models and an overestimation of the measured treatment effect. Covariates were included in the regression models since these characteristics could have differential impacts on the outcomes themselves, and our results show that the adjusted ORs were in fact slightly larger (showing a decreased effect size) when compared to unadjusted ORs, which show the difference in effect sizes of the matched populations alone.

Furthermore, another potential limitation in both the primary and 'Bone-Rx' cohorts is imbalanced comorbidity burden in BP user and non-user cohorts post-match. *Table 1* shows there is differential

prevalence of most co-morbid diseases despite matched cumulative CCI score between BP user and BP non-user cohorts. However, this limitation is in part addressed given (1) these covariates were controlled for during our regression analyses on study outcomes, and (2) that the key study findings were also observed in the 'Osteo-Dx-Rx' cohort, which matched based on individual comorbidities.

Additionally, limitations may be present due to misclassification bias of study outcomes due to the specific procedure/diagnostic codes used as well as the potential for residual confounding occurring for patient characteristics related to study outcomes that are unable to be operationalized in claims data, which would impact all cohort comparisons. For SARS-CoV-2 testing, procedure codes were limited to those testing for active infection, and therefore observations could be missed if they were captured via antibody testing (CPT 86318, 86328). These codes were excluded a priori due to the focus on the symptomatic COVID-19 population. Furthermore, for the COVID-19 diagnosis and hospitalization outcomes, all events were identified using the ICD-10 code for lab-confirmed COVID-19 (U07.1), and therefore events with an associated diagnosis code for suspected COVID-19 (U07.2) were not included. This was done to have a more stringent algorithm when identifying COVID-19-related events, and any impact of events identified using U07.2 is considered minimal, as previous studies of the early COVID-19 outbreak have found that U07.1 alone has a positive predictive value of 94% (*Kluberg et al., 2022*), and for this study U07.1 captured 99.2%, 99.0%, and 97.5% of all COVID-19 patient-diagnoses for the primary, 'Bone-Rx', and 'Osteo-Dx-Rx' cohorts, respectively.

Another potential limitation of this study relates to the positivity assumption, which when building comparable treatment cohorts is violated when the comparator population does not have an indication for the exposure being modelled (*Petersen et al., 2012*). This limitation is present in the primary cohort comparisons between BP users and BP non-users, as well as in the sensitivity analyses involving other preventive medications. This limitation, however, is mitigated by the fact that the outcomes in this study are related to infectious disease and are not direct clinical outcomes of known treatment benefits of BPs. The fact that the clinical benefits being assessed – the impact of BPs on COVID-related outcomes – was essentially unknown clinically at the time of the study data minimizes the impact of violation of the positivity assumption. Furthermore, our sensitivity analyses involving the 'Bone-Rx' and 'Osteo-Dx-Rx' cohorts did not suffer this potential violation, and the results from those analyses support those from the primary analysis cohort comparisons.

Moreover, we note that the propensity score-matched BP users and BP non-users in the primary analysis cohort mainly consisted of older females. According to the CDC,~75% and 95% of US women between 60–69 and 70–79 suffer from either low bone mass or osteoporosis, respectively (https://www.cdc.gov/nchs/data/databriefs/db93.pdf). Essentially all women (and 70% of men) above age 80 suffer from these conditions, which often go undiagnosed. Women aged 60 and older represent ~75% of our study population (*Table 1*). Although bone density measurements are not available for non-BP users in the matched primary cohort, there is a high probability that the incidence of osteoporosis and/or low bone mass in these patients was similar to the national average. Thus, BP therapy would have been indicated for most non-BP users in the matched primary cohort, and arguably, for these patients the positivity assumption was not violated.

One large potential bias to consider when comparing BP users to BP non-users is the healthy adherer effect, whereby adherence to drug therapy is associated with overall healthier behavior (*Dormuth et al., 2009*; *Curtis et al., 2011*). During the COVID-19 pandemic, this could have potentially resulted in differences between BP users and non-users such as, for example, adherence to mask-wearing, hand washing, or social distancing. However, if this effect accounted for the observed association between BP use and COVID-19-related outcomes, one would expect that users of other preventive medications would show similar associations. However, as discussed above, other preventive drug classes had a variable directional impact on the odds of COVID-19-related events, and sub-analyses within each drug class identified a strong association between concomitant BP use and decreased COVID-19-related events (*Figure 6B–E*). These analyses were based on the assumption that the association of unmeasured confounders with other drugs is comparable in magnitude and quality as for BPs. Taken together, these results suggest the observed association between BP use and COVID-19-related outcomes cannot solely be attributed to general behaviors associated with the healthy adherer effect.

Notably, several observational studies have reported that the use of one of our comparator preventive drug classes, statins, is associated with a lower risk of mortality in hospitalized COVID-19 patients (*Israel et al., 2021*; *Lohia et al., 2021*; *Zhang et al., 2020*). Indeed, statins are currently being tested as an adjunct

therapy for COVID-19 (NCT04380402). In our study population, statin use was associated with moderately decreased odds of SARS-CoV-2 testing and COVID-19 diagnosis, though at a much smaller magnitude than BPs, and was not consistently associated with reduced odds of COVID-19-related hospitalizations. Our analysis did not address the clinical course of hospitalized patients, so these results are not necessarily conflicting. However, we note that in our primary cohort, as many as 15.2% of statin users concomitantly used a BP. Indeed, within statin users, stratification by BP use revealed that the decreased odds of SARS-CoV-2 testing, COVID-19 diagnosis, and COVID-19-related hospitalizations remained regardless of statin use. Future studies on disease outcomes of hospitalized COVID-19 patients with antecedent use of BPs and statins alone or in combination are needed to clarify the effects of each drug class.

The differential association of amino-BPs *versus* statins with COVID-19 related outcomes is somewhat unexpected because both target the same biochemical pathway, albeit at different enzymatic steps (*Xia et al., 2018*). Statins block HMG-CoA reductase, the first and key rate-limiting enzyme in the mevalonate pathway (*Istvan and Deisenhofer, 2001*). Amino-BPs, which account for >99% of BPs prescribed in our study, inhibit a downstream enzyme in the same metabolic pathway, farnesyl pyrophosphate synthase (FPPS), which converts geranyl pyrophosphate to farnesyl pyrophosphate (*Kavanagh et al., 2006*). FPPS blockade disrupts protein prenylation and interferes with cytoskeletal rearrangement, membrane ruffling and vesicular trafficking in osteoclasts, thus preventing bone resorption (*Russell, 2007*). However, the anti-osteolytic activity of BPs per se is unlikely to account for the observed association between BP use and decreased incidence of COVID-19 and, more broadly, respiratory tract infections, because patients treated with non-BP anti-resorptive bone health medications have higher odds of respiratory infections (*Sing et al., 2020* and this study).

Another consequence of mevalonate pathway inhibition by both statins and amino-BPs is arrested endosomal maturation in antigen-presenting cells resulting in enhanced antigen presentation, T cell activation and humoral immunity (*Xia et al., 2018*). In addition to this adjuvant-like effect, FPPS blockade by amino-BPs causes the intracellular accumulation of the enzyme's substrate, isopentyl diphosphate (IPP), in myeloid leukocytes, which then stimulate Vγ9Vδ2 T cells (*Wang et al., 2011*; *Nada et al., 2017*), a large population of migratory innate lymphocytes in humans that are thought to play an important role in host defense against infectious pathogens (*Ribot et al., 2021*), including SARS-CoV-1[6]. Experiments in humanized mice that were challenged with influenza viruses have shown that amino-BP-induced expansion of Vγ9Vδ2 T cells markedly improves viral control and mitigates disease severity and mortality (*Tu et al., 2011*; *Zheng et al., 2015*). However, since statins act upstream of FPPS, they are expected to inhibit IPP synthesis and, hence, have been shown to counteract the stimulatory effect of amino-BPs on Vγ9Vδ2 T cells (*Wang et al., 2011*). However, statins and amino-BPs do not always antagonize each other. In vitro, concomitant statin and amino-BP use has been shown to be synergistic in inhibition of cancer cell growth, but mainly through downstream inhibition of geranylgeranyl transferases and subsequent protein prenylation by statins (*Abdullah et al., 2017*). The fact that the observed reduction in COVID-19-related outcomes in BP users was not altered by concomitant statin use implies that the apparent protective effects of amino-BPs may not rely solely on stimulation of Vγ9Vδ2 T cells. Indeed, in mice (in which BPs are not known to stimulate γδ T cells), BPs potently boost systemic and mucosal antiviral antibody and T cell responses (*Tonti et al., 2013*). This effect was also seen with non-nitrogenous BPs, which do not antagonize FPPS (*Tonti et al., 2013*). In the present study, the number of patients who used non-nitrogenous BPs was less than 20, and therefore too small to determine any impact on COVID-19-related outcomes. Nevertheless, in aggregate, these clinical and pre-clinical findings raise the possibility that BPs may exert (at least some) immuno-stimulatory effects by engaging an as yet unidentified additional pathway, regardless of their nitrogen content.

Irrespective of the precise molecular mechanism of action, BPs have been reported to exert a plethora of effects on additional immune cell populations in humans, including NK cells (*Sarhan et al., 2017*) and regulatory T cells (*Liu et al., 2016*). Moreover, studies of patients treated with amino-BPs found impaired chemotaxis and generation of reactive oxygen species by neutrophils (*Kuiper et al., 2012*; *Chadwick et al., 2020*), a population of inflammatory cells whose dysregulated recruitment and activation are strongly implicated in the pathogenesis of severe COVID-19 (*Meizlish et al., 2021*; *Reusch et al., 2021*). Thus, BPs may provide therapeutic benefits during infections with SARS-CoV-2 through modulation of both innate and adaptive immune responses. However, further studies to directly test these pleiotropic immuno-modulatory effects of BPs and to assess their relative contribution to the host response to SARS-CoV-2 infection are needed.

We conclude that, despite several caveats discussed above, the association between BP use and decreased odds of COVID-19-related endpoints was robust in analyses comparing BP users to BP non-users. Large differences were detected regardless of age, sex or geographic location that remained robust when using multiple approaches to address unmeasured confounding and/or potential sources of bias. These retrospective findings strongly suggest that BPs should be considered for prophylactic use in individuals at risk of SARS-CoV-2 infection. However, additional well-controlled prospective clinical studies will be needed to rigorously assess whether the observed reduction in COVID-19-related outcomes is directly caused by BPs and remains true in patient populations not commonly prescribed BPs.

A number of BPs are globally available as relatively affordable generics that are generally well tolerated and could be prescribed for off-label use. Rare, but severe adverse events that have been linked to BP use include osteonecrosis of the jaw (*Migliorati et al., 2006*) and atypical femur fractures (*Saita et al., 2015*), which are both associated with long-term BP therapy. In this context, it is important to consider the relationship between the timing of BP exposure and COVID-19-related outcomes. Remarkably, BP users of alendronic acid whose prescription ended more than eight months prior to the observation period, as well as users who initiated alendronic acid therapy immediately preceding the observation period, had similarly decreased odds of COVID-19-related outcomes (*Figure 3B*). A likely explanation for the observed long-term protection after transient BP use may be the well-documented retention of BPs in bone resulting in half-lives of several years (*Cremers et al., 2019*). Small amounts of stored BPs are continuously released, especially in regions of high bone turnover, which may result in persistent exposure of immune cells either systemically or preferentially in bone marrow, a site of active immune cell trafficking (*Mazo et al., 2005*; *Zhao et al., 2012*) where antiviral immune responses can be initiated in response to respiratory infection (*Hermesh et al., 2010*). Thus, BP use at the time of infection may not be necessary for protection against COVID-19. Rather, our results suggest that prophylactic BP therapy may be sufficient to achieve a potentially rapid and sustained immune modulation resulting in profound mitigation of the incidence and/or severity of infections by SARS-CoV-2.

## Acknowledgements

The authors acknowledge Ziqi Chen, Paris Pallis, and Flora Tierney for helpful discussions on the interpretation of study results. We are grateful to Komodo Health who provided all data used in this analysis at no cost, and we thank Vicki Guan and Ben Cohen from Komodo Health for facilitating this research. Special thanks to Kantar Health (now Cerner Enviza) who provided the support needed to complete this study with no associated financial requirements. This study was supported by NIH grants AR068383 and AI155865 (to UHvA) and a CRI Irvington postdoctoral fellowship CRI2453 (to PH).

## Additional information

### Competing interests

Jeffrey Thompson, Thomas Haskell: are full time employees of Cerner Health and have received support for attending ISPOR 2022 from Cerner Enviza (previously Kantar Health). The authors have no other competing interests to declare. Tobias Dreischulte: received payments/honoraria from Techniker Krankenkasse (public insurance fund) for editing a report on COVID-19 treatments, and research grants from BMBF (German federal ministry for research) and from the Innovationsfond (German federal research fund for health services research). The author has no other competing interests to declare. Stefan Endres: received grants from BMBF (German Federal Ministry for Research) and Bio-M (Munich Cluster Organisation). The author received royalties/licenses from TCR2, Cambridge, MA, USA and Carina Biotech Ltd, Mawson Lakes, Australia. The author received honoraria for chairing the Scientific committee at Else Kröner Fresenius Foundation (non-profit), acting as scientific advisor for the Paul-Martini-Foundation (non-profit) and textbook editor and author for Elsevier. The author received payment for expert testimony from CMS Hasche Sigle, Law firm and Gilde Healthcare, Utrecht, Netherlands (private equity investor). The author holds stock options at TCR2, Cambridge, MA, USA. Patents have been issued for Bispecific antibody molecules with antigentransfected T cells and their use in medicine, and PD1-CD28 fusions proteins and their use in medicine. Patents are pending for CXCR6

transduced T cells for targeted tumor therapy, Improving adoptive cellular therapy, CCR8 transduced T cells for targeted tumor therapy and CSF1R-targeted immunotherapies. The author has no other competing interests to declare. Ulrich H von Andrian: received the following grants unrelated to this project; HMS-AbbVie Alliance, Program Area 1; Project 1: 'Host-virus interaction dynamics in nasal mucosa and associated lymphoid tissues', Gates Foundation, OPP1155348 'Mucosal Vaccine Consortium' and Moderna-HMS ARTiMIS Alliance. Ulrich H von Andrian was granted the following patents unrelated to this project; US Patents #9539210, 8932595, 8277812, 8906381, 8343497 licensed to Selecta Biosciences, and US Patent #11111472 licensed to SQZ. The author is a paid consultant of AbbVie, Avenge Bio, Beam Therapeutics, Bluesphere Bio, FL72, DNAlite, Gate Biosciences, Gentibio, Intergalactic, intrECate Biotherapeutics, Interon, Institute for Protein Innovation, Mallinckrodt Pharmaceuticals, Moderna, Monopteros Biotherapeutics, Morphic Therapeutics, Rubius, Selecta and SQZ. The author holds stock/stock options at Avenge Bio, Beam, Bluesphere, FL72, IntrECate, Interon, Moderna, Monopteros, Morphic, Rubius, Selecta and SQZ. The author received payment/honoraria for a Keynote Lecture at 'Applied Pharmaceutical Nanotechnology 2019', Cambridge, MA (organized by Pfizer), Nov. 2019 and Mallinckrodt Mini-Symposium, Oct. 2019. The author received support as a speaker at the following conferences: Ethics in Medicine Seminar, San Servolo Italy, May 2022; Keystone Symposium 'B and T cell Memory'; Keystone Symposium 'Stromal Cells in Immunity and Disease', Feb. 2020; and HIV Prevention Workshop, South Africa, Nov. 2019. The author is an inventor on the following pending patents: Ziegler et al. 'Methods and composition for modulating immune response and immune homeostasis', Docket # BROD-4830US; Thiriot et al. 'Modulating phenotype and function of high endothelial venules' Provisional docket # 00742-304001, von Andrian and Thiriot. 'Microvessel endothelial cells and uses thereof' Provisional docket #HRVY 026-001. The author holds a leadership/fiduciary role on the Monopteros Biotherapeutics Board of Directors, intrECate Biotherapeutics Board of Directors and Councilor of the American Association of Immunologists. The author has no other competing interests to declare. The other authors declare that no competing interests exist.

## Funding

| Funder | Grant reference number | Author |
|---|---|---|
| National Institute of Allergy and Infectious Diseases | AI155865 | Ulrich H von Andrian |
| National Institute of Arthritis and Musculoskeletal and Skin Diseases | AR068383 | Ulrich H von Andrian |
| MassCPR | Evergrande COVID-19 Response Fund Award | Ulrich H von Andrian |
| Cancer Research Institute | CRI2453 | Pavel Hanč |

The funders had no role in study design, data collection and interpretation, or the decision to submit the work for publication.

## Author contributions

Jeffrey Thompson, Conceptualization, Data curation, Formal analysis, Validation, Investigation, Visualization, Methodology, Writing – original draft, Writing – review and editing; Yidi Wang, Conceptualization, Formal analysis, Investigation, Visualization, Methodology, Writing – original draft, Writing – review and editing; Tobias Dreischulte, Olga Barreiro, Rodrigo J Gonzalez, Colette Matysiak, Harold R Neely, Stefan Endres, Conceptualization, Investigation, Methodology, Writing – review and editing; Pavel Hanč, Conceptualization, Methodology, Writing – review and editing; Marietta Rottenkolber, Conceptualization, Investigation, Methodology; Thomas Haskell, Resources, Data curation, Formal analysis, Supervision, Funding acquisition, Investigation, Project administration; Ulrich H von Andrian, Conceptualization, Formal analysis, Supervision, Funding acquisition, Investigation, Methodology, Writing – original draft, Project administration, Writing – review and editing

## Author ORCIDs

Stefan Endres (ID) https://orcid.org/0000-0002-4703-537X
Ulrich H von Andrian (ID) https://orcid.org/0000-0003-4231-2283

### Ethics

The study protocol was reviewed by Pearl IRB (Indianapolis, IN) and was determined to be Exempt according to FDA 21 CFR 56.104 and 45CFR46.104(b)(4): (4) Secondary Research Uses of Data or Specimens on 02/08/2021.Protocol #21-ACUT-101.

### Decision letter and Author response

Decision letter https://doi.org/10.7554/eLife.10.7554/eLife.79548.sa1
Author response https://doi.org/10.7554/eLife.10.7554/eLife.79548.sa2

### Data availability

Excel spreadsheets of source data are provided as supplemental information for figures 1C, 2B, 3A-D, and 4B-E.The administrative claims data used in this study cannot be made publicly available as it as it is a business product of Komodo Health, who contracts with insurers to develop the combined de-identified dataset under agreements that no patient-level data is permitted outside of the Komodo Health analytics environment. All analyses for this current study were performed in the Komodo Health analytics environment.An interested researcher may contact the corresponding author listed in this article by electronic mail at the address listed, who can then further connect them to a researcher at the company who is familiar with the study. The data was analyzed using Microsoft Excel software.

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

# Appendix 1

## Study Methods

Section 1: Variable Assignment

Outcomes

The following details the identification algorithms and associated codes that were used to identify outcomes of interest, including COVID-19-related as well as the exploratory outcomes that were assessed during sensitivity analyses.

### Primary outcomes

SARS-CoV-2 testing

- Any medical services claim with a procedure code indicating polymerase chain reaction (PCR) testing for active SARS-CoV-2 infection 3/1/2020-6/30/2020
- Identified using HCPCS codes: 87635, 87636, 87637

COVID-19 diagnosis

- Any medical services claim with a diagnosis code indicating COVID-19 3/1/2020-6/30/2020
- Identified using ICD-10 code U07.1x

COVID-19-related hospitalization

- Any medical services claim occurring in an inpatient setting with a diagnosis code indicating COVID-19 3/1/2020-6/30/2020
- Identified using ICD-10 code U07.1x

### Exploratory outcomes (study observation period)

Acute cholecystitis-related service

- Any medical services claim occurring in an emergency room/inpatient setting with a diagnosis indicating acute cholecystitis 3/1/2020-6/30/2020
- Identified using ICD-10 codes K81.0x

Acute pancreatitis-related service

- Any medical services claim occurring in an emergency room/inpatient setting with a diagnosis indicating acute pancreatitis 3/1/2020-6/30/2020
- Identified using ICD-10 codes K85.x

### Exploratory outcomes (2019)

Acute cholecystitis-related service

- Any medical services claim occurring in an emergency room/inpatient setting with a diagnosis indicating acute cholecystitis 7/1/2019-12/31/2019
- Identified using ICD-10 codes K81.0x

Acute pancreatitis-related service

- Any medical services claim occurring in an emergency room/inpatient setting with a diagnosis indicating acute pancreatitis 7/1/2019-12/31/2019
- Identified using ICD-10 codes K85.x

Acute bronchitis-related service

- Any medical services claim with a diagnosis indicating acute bronchitis 7/1/2019-12/31/2019
- Identified using ICD-10 codes J20.x-J21.x

Acute pneumonia-related service

- Any medical services claim with a diagnosis indicating acute bronchitis 7/1/2019-12/31/2019
- Identified using ICD-10 codes J13.x-J18.x

### Osteonecrosis

Osteonecrosis

- Any medical services claim with a diagnosis indicating drug-induced osteonecrosis 1/1/2019-6/30/2020
- Identified using ICD-10 codes M87.1x

## Drug-exposure assignment

The following details the identification algorithms and associated inputs used for drug-exposure classification of study subjects into users/non-users of bisphosphonates, non-bisphosphonates osteoporosis medications, statins, antihypertensives, non-insulin antidiabetics, and antidepressants.

Bisphosphonates
- Any outpatient prescription or in-office dispensing 1/1/2019-2/29/2020
- Drugs included: alendronate, alendronic acid, etidronate, ibandronate, ibandronic acid, pamidronate, risedronate, and zoledronic acid

Non-BP anti-resorptive bone health medications
- Any outpatient prescription or in-office dispensing 1/1/2019-2/29/2020
- Drugs included: denosumab, calcitonin, raloxifene, romosozumab-aqqg, teriparatide, abaloparatide, or bazedoxifene

Statins
- Any outpatient prescription 1/1/2019-2/29/2020
- Drugs included: pravastatin, rosuvastatin, fluvastatin, atorvastatin, pitavastatin, or simvastatin

Antihypertensives
- Any non-ophthalmic, non-injection, outpatient prescription claim for a beta-blocker, calcium channel blocker, or renin angiotensin system antagonist 1/1/2019-2/29/2020
- Drugs included: acebutolol, atenolol, betaxolol, bisoprolol, carvedilol, labetalol, metoprolol, nadolol, nebivolol, penbutolol, pindolol, propranolol, timolol, amlodipine, diltiazem, felodipine, isradipine, nicardipine, nifedipine, nisoldipine, verapamil, aliskiren, azilsartan, benazepril, candesartan, captopril, enalapril, eprosartan, fosinopril, irbesartan, lisinopril, losartan, moexipril, olmesartan, perindopril, quinapril, ramipril, sacubitril, telmisartan, trandolapril, valsartan

Antidiabetics
- Any outpatient prescription claim for a non-insulin antidiabetic medication 1/1/2019-2/29/2020
- Drugs included: metformin, chlorpropamide, glimepiride, glipizide, glyburide, tolazamide, tolbutamide, pioglitazone, rosiglitazone, alogliptin, linagliptin, saxagliptin, sitagliptin, albiglutide, dulaglutide, exenatide, liraglutide, lixisenatide, semaglutide, nateglinide, repaglinide, canagliflozin, dapagliflozin, empagliflozin, ertugliflozin

Antidepressants
- Any outpatient prescription claim for a selective serotonin reuptake inhibitor, norepinephrine-dopamine reuptake inhibitor, serotonin-norepinephrine reuptake inhibitor, tricyclic, tetracyclic, modified cyclic, or MAO inhibitor medication 1/1/2019-2/29/2020
- Drugs included: amoxapine, bupropion, citalopram, clomipramine, desipramine, desvenlafaxine, doxepin, duloxetine, escitalopram, esketamine, fluoxetine, fluvoxamine, imipramine, isocarboxazid, levomilnacipran, maprotiline, mirtazapine, nefazodone, nortriptyline, paroxetine, phenelzine, protriptyline, selegiline, sertraline, tranylcypromine, trazodone, trimipramine, venlafaxine, vilazodone, vortioxetine

## Charlson comorbidity condition assignment

The following ICD-10 codes were used to assign the CCI condition-specific indicators that are used to calculate the overall CCI score. The time period used for identification of condition-specific indicators was the entire pre-observation period (1/1/2019-2/29/2020).

Myocardial infarction
- ICD-10 codes: I21.x, I22.x, I25.2

Congestive heart failure
- ICD-10 codes: I09.9, I11.0, I13.0, I13.2, I25.5, I42.0, I42.5 - I42.9, I43.x, I50.x, P29.0

Peripheral vascular disease
- ICD-10 codes: I70.x, I71.x, I73.8, I73.9, I77.1, I79.0, I79.2, K55.1, K55.8, K55.9, Z95.8, Z95.9

Cerebrovascular disease
- ICD-10 codes: G45.x, G46.x, H34.0, I60.x-I69.x

Dementia
- ICD-10 codes: F00.x - F03.x, F05.1, G30.x, G31.1

Chronic pulmonary disease
- ICD-10 codes: I27.8, I27.9, J40.x - J47.x, J60.x - J67.x, J68.4, J70.1, J70.3

Rheumatologic disease
- ICD-10 codes: M05.x, M06.x, M31.5, M32.x - M34.x, M35.1, M35.3, M36.0

Peptic ulcer disease
- ICD-10 codes: K25.x-K28.x

Mild liver disease
- ICD-10 codes: B18.x, K70.0 - K70.3, K70.9, K71.3 - K71.5, K71.7, K73.x, K74.x, K76.0, K76.2 - K76.4, K76.8, K76.9, Z94.4

Diabetes without chronic complications
- ICD-10 codes: E10.0, E10.1, E10.6, E10.8, E10.9, E11.0, E11.1, E11.6, E11.8, E11.9, E12.0, E12.1, E12.6, E12.8, E12.9, E13.0, E13.1, E13.6, E13.8, E13.9, E14.0, E14.1, E14.6, E14.8, E14.9

Diabetes with chronic complications
- ICD-10 codes: E10.2 - E10.5, E10.7, E11.2 - E11.5, E11.7, E12.2 - E12.5, E12.7, E13.2 - E13.5, E13.7, E14.2 - E14.5, E14.7

Hemiplegia or paraplegia
- ICD-10 codes: G04.1, G11.4, G80.1, G80.2, G81.x, G82.x, G83.0 - G83.4, G83.9

Renal disease
- ICD-10 codes: I12.0, I13.1, N03.2 - N03.7, N05.2 - N05.7, N18.x, N19.x, N25.0, Z49.0 - Z49.2, Z94.0, Z99.2

Any tumor, leukemia, or lymphoma
- ICD-10 codes: C00.x - C26.x, C30.x - C34.x, C37.x - C41.x, C43.x, C45.x - C58.x, C60.x - C76.x, C81.x - C85.x, C88.x, C90.x - C97.x

Moderate or severe liver disease
- ICD-10 codes: I85.0, I85.9, I86.4, I98.2, K70.4, K71.1, K72.1, K72.9, K76.5, K76.6, K76.7

Metastatic solid tumor
- ICD-10 codes: C77.x - C80.x

AIDS/HIV
- ICD-10 codes: B20.x - B22.x, B24.x

## Additional condition covariate assignment

The following details the ICD-10 diagnosis codes that were used to identify comorbid conditions. For all condition indicators classification was based on all medical claims occurring during the pre-observation period (1/1/2019-2/29/2020).

Osteoporosis: M80.x, M81.x, M82.x
- Cardiovascular disease overall: I3x.x-I4x.x, I20.x-I28.x, I50.x-I52.x
- Cancer: C0x.x - C9x.x
- Chronic kidney disease (CKD)/ end-stage renal disease (ESRD): I12.0, I13.1, N03.2 - N03.7, N05.2 - N05.7, N18.x, N19.x, N25.0, Z49.0 - Z49.2, Z94.0, Z99.2
- Chronic obstructive pulmonary disease (COPD): J43.x, J44.x
- Dementia: F00.x - F03.x, F05.1, G30.x, G31.1
- Depression: F32.x, F33.x
- Dyslipidemia: E78.x
- Heart failure: I50.x, I11.0xx, I13.0xx, I13.2xx
- HIV/AIDS: B20.x - B22.x, B24.x
- Hypertension: I10.x, I12.x, I11.9xx, I13.1xx
- Liver disease: B18.x, K70.0 - K70.3, K70.9, K71.3 - K71.5, K71.7, K73.x, K74.x, K76.0, K76.2 - K76.4, K76.8, K76.9, Z94.4, I85.0, I85.9, I86.4, I98.2, K70.4, K71.1, K72.1, K72.9, K76.5, K76.6, K76.7
- Obesity: E66.x
- Sickle cell disease: D57.x
- Stroke: I63.x
- Type 2 diabetes: E11.x

## Section 2: Sensitivity analyses methodologies

## Sensitivity Analysis (1): COVID-19-related outcomes in "*Bone-Rx*" cohort
## Overview and rationale

- The first sensitivity analysis was performed to validate the robustness of the primary findings by limiting all BP non-users to those who had used non-BP anti-resorptive bone health

medications during the pre-observation period, thus yielding a more comparable comparator cohort that was also receiving bone health medication therapy.

- The use of an active-comparator cohort was done to reduce the impact of unmeasured confounding that may have occurred in the primary analysis due to the use of the derived Charlson Comorbidity Index composite score as the only comorbidity matching covariate. Restriction of the patient population to users of any non-BP anti-resorptive bone health medication prior to propensity-score matching improves the probability of having drug user/non-user matches with more similar clinical characteristics.
- This sensitivity analysis, further, also acted to increase the robustness and reliability of the matched user/non-user outcome comparisons since non-BP anti-resorptive bone health medication users represented the smaller portion of the total bone health medication-user population ("*Bone-Rx*" cohort) and therefore were matched to their best BP-user pair.

### Analysis cohort definition(s)

- Continuous medical and prescription insurance coverage 1/1/2019-6/30/2020
- Patients with ≥1 claim for any anti-resorptive bone health medication 1/1/2019-2/29/2020

### Exposures of interest

- Patients were assigned into the BP user cohort if they had any claim 1/1/2019-2/29/2020 for one of the following: alendronate, alendronic acid, etidronate, ibandronate, ibandronic acid, pamidronate, risedronate, and zoledronic acid.
- Patients were assigned into the non-BP any anti-resorptive bone health medication user cohort if: (1) they had any claim 1/1/2019-2/29/2020 for one of the following: denosumab, calcitonin, raloxifene, romosozumab-aqqg, teriparatide, abaloparatide, or bazedoxifene; and (2) they had no BP claims 1/1/2019-2/29/2020.

### Outcomes

- SARS-CoV-2 testing, COVID-19 diagnosis, and COVID-19-related hospitalizations

### Cohort matching

- Non-BP anti-resorptive bone health medication users were matched to BP users based on age, gender, insurance type, any PCP visit in 2019, and comorbidity score. Matching was performed within each region separately (northeast, midwest, south, west) and then combined as well as in NY-state alone.

### Statistical analyses

- Same as was performed for the primary analysis cohort.

## Sensitivity Analysis (2): COVID-19-related outcomes in "*Osteo-Dx-Rx*" cohort
### Overview and rationale

- The second sensitivity analysis was performed to further assess the robustness of the primary analysis findings by performing a highly restricted comparator cohort matching that included patients diagnosed and treated for osteoporosis ("*Osteo-Dx-Rx*" cohort).
- The relationship between COVID-19-related outcomes and BP-exposure was modelled after restricting anti-resorptive bone health medication users to those most likely to use BPs and matching BP non-users to BP users based on the presence of comorbid diagnoses within insurance type in four states with early COVID-19 spread representing each to further reduce confounding related to differences in demographic/clinical characteristics amongst BP users/non-users, confounding due to socioeconomic status (insurance type as proxy), and confounding due to differences in COVID-19-exposure risk based on geography.

### Analysis cohort definition(s)

- Continuous medical and prescription insurance coverage 1/1/2019-6/30/2020
- Patients with ≥1 claim for any osteoporosis medication 1/1/2019-2/29/2020 who also met the following criteria: (i) female; (ii) age 51 or older; (iii) identified as residing in New York, Illinois,

Florida, or California; and (iv) had ≥1 medical claim indicating a diagnosis of osteoporosis 1/1/2019-2/29/2020

## Exposures of interest

- Patients were assigned into the BP user cohort if they had any claim 1/1/2019-2/29/2020 for one of the following: alendronate, alendronic acid, etidronate, ibandronate, ibandronic acid, pamidronate, risedronate, and zoledronic acid.
- Patients were assigned into the non-BP anti-resorptive bone health medication user cohort if: (1) they had any claim 1/1/2019-2/29/2020 for one of the following: denosumab, calcitonin, raloxifene, romosozumab-aqqg, teriparatide, abaloparatide, or bazedoxifene; and (2) they had no BP claims 1/1/2019-2/29/2020.

## Outcomes

- SARS-CoV-2 testing, COVID-19 diagnosis, and COVID-19-related hospitalizations

## Cohort matching

- Non- anti-resorptive bone health medication users were matched to BP users based on age, PCP visit in 2019, and the presence of the following comorbid conditions (assigned using ICD-10 codes on claims occurring 1/1/2019-2/29/2020): cancer, chronic obstructive pulmonary disease, depression, dyslipidaemia, heart failure, hypertension, obesity, and type 2 diabetes.
- Matching was performed within each state when stratified by insurance type (commercial, dual, Medicaid, Medicare).

## Statistical analyses

Multivariate logistic regression analyses, modelled separately for each COVID-19-related outcome of interest, were performed on the unmatched and matched samples after combining all patient observations. In addition to the key exposure variable (indicating BP user versus non-BP user), the regression model also included demographic/clinical covariate for age group, region, insurance type, PCP visit in 2019, and the following comorbid conditions: osteoporosis, cancer, chronic obstructive pulmonary disease, depression, dyslipidaemia, hypertension, obesity, type 2 diabetes, cardiovascular disease overall, sickle cell anemia, stroke, dementia, HIV/AIDS, chronic kidney disease/end-stage renal disease, and liver disease.

## Sensitivity Analysis (3): Association of BP-use with exploratory negative control outcomes

### Overview and rationale

- The third sensitivity analysis was performed to assess the relationship between BP-use and outcomes not anticipated to be impacted by the pharmacological mechanism of BPs.
- This was performed by modelling the relationship between BP-exposure and other outcomes occurring (1) during the study observation, and (2) during the second half of 2019 among BP users with claims during the first half of 2019 and their previously-assigned BP non-user matched pair, in the primary, "*Bone-Rx*", and "*Osteo-Dx-Rx*" cohorts.
- Outcomes modelled included any acute cholecystitis-related or acute pancreatitis-related inpatient/emergency-room (ER) service, used as exploratory outcomes not predicted to be modulated by BP exposure to assess the validity of the core COVID-19-related outcomes.

### Analysis cohort definition(s)

- Patients who were included in the primary analysis cohort for assessment of (1) outcomes occurring during the study observation period; for (2) outcomes assessed during the second half of 2019 the cohort was restricted to among BP users with claims during the first half of 2019 and their previously-assigned BP non-user matched pair.
- Patients who met all eligibility criteria to be included in the '*Bone-Rx*' cohort for assessment of (1) outcomes occurring during the study observation period; for (2) outcomes assessed during

the second half of 2019 the cohort was restricted to among BP users with claims during the first half of 2019 and their previously-assigned BP non-user matched pair.
- Patients who met all eligibility criteria to be included in the 'Osteo-Dx-Rx' cohort for assessment of (1) outcomes occurring during the study observation period; for (2) outcomes assessed during the second half of 2019 the cohort was restricted to among BP users with claims during the first half of 2019 and their previously-assigned BP non-user matched pair.

### Exposures of interest

- For the primary analysis cohort, the BP user / BP non-user assignment was the same as used in the core analyses.
- For the "*Bone-Rx*" and "*Osteo-Dx-Rx*" cohorts, assignment was the same as used in those analyses stratifying medication users into BP users and non-BP medication users.

### Outcomes

- Any medical claim from an ER/inpatient setting with a diagnosis indicating acute cholecystitis (ICD-10 code K81.0x) occurring 3/1/2020-6/30/2020 (observation period)
- Any medical claim from an ER/inpatient setting with a diagnosis indicating acute pancreatitis (ICD-10 code K85.x) occurring 3/1/2020-6/30/2020 (observation period)
- Any medical claim from an ER/inpatient setting with a diagnosis indicating acute cholecystitis (ICD-10 code K81.0x) occurring 7/1/2019-12/31/2019 (2019)
- Any medical claim from an ER/inpatient setting with a diagnosis indicating acute pancreatitis (ICD-10 code K85.x) occurring 7/1/2019-12/31/2019 (2019)

### Cohort matching
NA; all cohorts previously matched.

### Statistical analyses
Multivariate logistic regression analyses were performed using the same methodologies employed when assessing COVID-19 outcomes that were cohort-build-specific (i.e. followed previous approach detailed for each respective cohort build) to assess the odds of acute cholecystitis or acute pancreatitis.

## Sensitivity Analysis (4): Association of BP-use with exploratory positive control outcomes in 2019
### Overview and rationale

- The fourth sensitivity analysis was performed to assess the relationship between BP-use and select outcomes occurring in 2019 to validate the theorized BP mechanism of action.
- This was performed by modelling the relationship between BP-exposure in the first half of 2019 and other outcomes occurring during the second half of 2019 in the primary, "*Bone-Rx*", and "*Osteo-Dx-Rx*" cohorts, specifically medical services for other infectious respiratory conditions (acute bronchitis, pneumonia), used to assess the validity of the relationship between BP-use and decreased respiratory infections.

### Analysis cohort definition(s)
- The following criteria were applied to all three cohort build variations (primary analysis cohort, "*Bone-Rx*" cohort, "*Osteo-Dx-Rx*" cohort): (i) BP users were restricted to those with any BP claim 1/1/2019-6/30/2019, and the remaining previously-classified BP-user patients with their first BP-claim date occurring on/after 7/1/2019 were excluded; (ii) BP non-users were restricted to their BP-user matched-pair previously assigned.

### Exposures of interest

- In all cohort build variations, the previously-classified BP user cohorts were restricted to those with any BP-claim 1/1/2019-6/30/2019; all other previously-classified BP users were excluded.

### Outcomes

- Any medical claim with a diagnosis indicating acute bronchitis (ICD-10 code J20.x-J21.x) occurring 7/1/2019-12/31/2019
- Any medical claim with a diagnosis indicating pneumonia (ICD-10 code J13.x-J18.x) occurring 7/1/2019-12/31/2019

### Cohort matching

- NA; all cohorts previously matched.

### Statistical analyses

- Multivariate logistic regression analyses were performed using the same methodologies employed when assessing COVID-19-related outcomes that were cohort-build-specific (i.e. followed previous approach detailed for each respective cohort build) to assess the odds of acute bronchitis, or pneumonia.

## Sensitivity Analysis (5): Association between use of other drug classes and COVID-19-related outcomes

### Overview and rationale

- The fifth sensitivity analysis was performed to assess whether the observed protective effect of BPs may be associated with general healthier behaviours in patients using any medication rather than specifically BP use. To assess this unmeasured confounding due to the healthy adherer effect, which is a type of potential bias where patients may have better outcomes due to their heathier behaviours and not better outcomes related to active drug treatment itself, the first sensitivity analysis evaluated the association between use of other preventive medications (statin, antihypertensive, antidiabetic, antidepressant) and COVID-19-related outcomes were evaluated.
- This was performed following the same techniques used in the primary cohort matching and analyses but when assigned drug exposure cohorts based on the use of statin, antihypertensive, antidiabetic, or antidepressant medications. The consistency of methods was done to permit direct comparison on the association between drug-use and COVID-19-related outcomes to assess whether the healthy adherer effect alone accounts for the decrease in the odds of COVID-19 outcomes when comparing BP users to non-users in the primary analysis. Evidence to support the contention that the HAE is a significant source of unmeasured confounding would necessitate that other drug classes display a similar statistically significant trend and/or magnitude when comparing drug users to non-users. Variability in directional impact, magnitude, and/or statistical significance would, conversely, suggest that the healthy adherer effect itself does not account for the differences seen when comparing BP users to BP non-users.
- This sensitivity analysis, additionally, also employed a unique nested-matching technique wherein BP users were matched to BP non-users within the other-medication-class matched populations when stratified into the already matched but mutually exclusive user/non-user cohorts. This was performed to: (1) assess whether the decreased odds of COVID-19-realted outcomes in BP users compared to BP non-users was robust, even amongst cohorts displaying an increase in the odds of COVID-19-related outcomes; and (2) to assess whether the magnitude of decrease in odds of COVID-19-related outcomes amongst BP users compared to BP non-users seen in the primary analysis is impacted by use of other medication classes, including some that have also been identified as being associated with a reduced incidence and/or severity of COVID-19-related outcomes.

### Analysis cohort definition(s)

- Continuous medical and prescription insurance coverage 1/1/2019-6/30/2020 (*all*)
- Patients with any claim for another drug class of interest (statin, antihypertensive, antidiabetic, antidepressant) medication 1/1/2019-2/29/2020 were classified users
- Among the propensity-score matched drug user/non-user cohorts, a further stratification and propensity-score matching based on BP use 1/1/2019-2/29/2020 to yield the following:

(i) drug user/BP user matched to drug user/BP non-user, (ii) drug non-user/BP user matched to drug non-user/BP non-user.

## Exposures of interest

- Patients were assigned into the statin user cohort if they had any claim 1/1/2019-2/29/2020 for one of the following: pravastatin, rosuvastatin, fluvastatin, atorvastatin, pitavastatin, or simvastatin
- Patients were assigned into the antihypertensive user cohort if they had any non-ophthalmic, non-injection claim 1/1/2019-2/29/2020 for a beta blocker, calcium channel blocker, or renin-angiotensin system antagonist medication.
- Patients were assigned into the antidiabetic user cohort if they had any claim 1/1/2019-2/29/2020 for one of the following non-insulin medications: metformin, chlorpropamide, glimepiride, glipizide, glyburide, tolazamide, tolbutamide, pioglitazone, rosiglitazone, alogliptin, linagliptin, saxagliptin, sitagliptin, albiglutide, dulaglutide, exenatide, liraglutide, lixisenatide, semaglutide, nateglinide, repaglinide, canagliflozin, dapagliflozin, empagliflozin, ertugliflozin
- Patients were assigned into the antidepressant user cohort if they had any claim 1/1/2019-2/29/2020 for one of the following: amoxapine, bupropion, citalopram, clomipramine, desipramine, desvenlafaxine, doxepin, duloxetine, escitalopram, esketamine, fluoxetine, fluvoxamine, imipramine, isocarboxazid, levomilnacipran, maprotiline, mirtazapine, nefazodone, nortriptyline, paroxetine, phenelzine, protriptyline, selegiline, sertraline, tranylcypromine, trazodone, trimipramine, venlafaxine, vilazodone, vortioxetine

## Outcomes

- SARS-CoV-2 testing, COVID-19 diagnosis, and COVID-19-related hospitalizations

## Cohort matching

- For the larger drug-class analyses, matching was performed following the same methods used in the primary analysis: users were matched to non-users based on age, gender, insurance type, any PCP visit in 2019, and comorbidity score. Matching was performed within each region separately (northeast, midwest, south, west) and then combined, as well as in NY-state alone.
- Following this matching procedure, a nested BP user to BP non-user propensity score match was then performed on the aforementioned matched populations (i.e. within the separate and already matched statin user and statin non-user populations). Matching was performed using the same list of demographic/clinical characteristics, and was also performed within each region separately (northeast, midwest, south, west) and then combined as well as in NY-state alone.

## Statistical analyses

- Same as was performed for the primary analysis cohort.

## Appendix 2

### Additional study results; cohort characteristics pre/post match

Primary analysis study population

#### Northeast region

A total of 2,152,560 patients identified as residing in the northeast were included in the unmatched primary analysis cohort comparisons, of which 119,728 (5.6%) and 2,032,832 (94.4%) were classified as BP users and BP non-users, respectively (*Appendix 2—table 1*). Prior to propensity-score matching, there were significant differences across all demographic and clinical characteristics. Compared to BP non-users, BP users were older (97.5% age ≥51 vs 49.8%; $P<0.001$), predominantly female (90.5% vs 57.4%; $P<0.001$), with higher comorbidity burden (mean CCI = 0.93 versus 0.65; $P<0.001$), insured by Medicare (46.5% vs 18.0%; $P<0.001$), and have had a primary-care physician (PCP) visit in 2019 (58.3% vs 42.8%; $P<0.001$). Propensity-score matching yielded 119,494 BP users and 119,494 BP non-users with no significant differences across examined characteristics. A total of 234 BP users from the northeast region in the unmatched primary analysis cohort were not assigned an applicable BP non-user pair during the matching procedure and were excluded from the matched BP user population.

#### Midwest region

A total of 1,467,802 patients identified as residing in the midwest were included in the unmatched primary analysis cohort comparisons, of which 75,967 (5.2%) and 1,391,835 (94.8%) were classified as BP users and BP non-users, respectively (*Appendix 2—table 2*). Prior to propensity-score matching, there were significant differences across all demographic and clinical characteristics. Compared to BP non-users, BP users were older (96.6% age ≥51 vs 44.0%; $P<0.001$), predominantly female (90.3% vs 57.1%; $P<0.001$), with higher comorbidity burden (mean CCI = 0.99 versus 0.56; $P<0.001$), insured by Medicare (43.6% vs 14.5%; $P<0.001$), and have had a primary-care physician (PCP) visit in 2019 (62.2% vs 51.0%; $P<0.001$). Propensity-score matching yielded 75,901 BP users and 75,901 BP non-users with no significant differences across examined characteristics. A total of 66 BP users from the midwest region in the unmatched primary analysis cohort were not assigned an applicable BP non-user pair during the matching procedure and were excluded from the matched BP user population.

#### South region

A total of 3,042,604 patients identified as residing in the south were included in the unmatched primary analysis cohort comparisons, of which 160,886 (5.3%) and 2,881,718 (94.7%) were classified as BP users and BP non-users, respectively (*Appendix 2—table 3*). Prior to propensity-score matching, there were significant differences across all demographic and clinical characteristics. Compared to BP non-users, BP users were older (96.8% age ≥51 vs 39.2%; $P<0.001$), predominantly female (90.6% vs 57.4%; $P<0.001$), with higher comorbidity burden (mean CCI = 0.86 versus 0.55; $P<0.001$), insured by Medicare (41.0% vs 11.3%; $P<0.001$), and have had a primary-care physician (PCP) visit in 2019 (66.1% vs 49.2%; $P<0.001$). Propensity-score matching yielded 159,704 BP users and 159,704 BP non-users with no significant differences across examined characteristics. A total of 1,182 BP users from the south region in the unmatched primary analysis cohort were not assigned an applicable BP non-user pair during the matching procedure and were excluded from the matched BP user population.

#### West region

A total of 1,243,637 patients identified as residing in the west were included in the unmatched primary analysis cohort comparisons, of which 95,470 (7.7%) and 1,148,167 (92.3%) were classified as BP users and BP non-users, respectively (*Appendix 2—table 4*). Prior to propensity-score matching, there were significant differences across all demographic and clinical characteristics. Compared to BP non-users, BP users were older (97.8% age ≥51 vs 43.5%; $P<0.001$), predominantly female (88.7% vs 56.4%; $P<0.001$), with higher comorbidity burden (mean CCI = 1.08 versus 0.66; $P<0.001$), insured by Medicare (43.5% vs 11.0%; $P<0.001$), and have had a primary-care physician (PCP) visit in 2019 (67.7% vs 45.3%; $P<0.001$). Propensity-score matching yielded 95,267 BP users and 95,267 BP non-users with no significant differences across examined characteristics. A total of 203 BP users from the west region in the unmatched primary analysis cohort were not assigned an applicable BP non-user pair during the matching procedure and were excluded from the matched BP user population.

### New York State

A total of 968,296 patients identified as residing in New York state were included in the unmatched primary analysis NY-state restricted cohort, of which 50,035 (5.2%) and 918,261 (94.8%) were classified as BP users and BP non-users, respectively (*Appendix 2—table 5*). Prior to propensity-score matching, there were significant differences across all demographic and clinical characteristics. Compared to BP non-users, BP users were older (98.1% age ≥51 vs 50.7%; *P*<0.001), predominantly female (90.9% vs 57.5%; *P*<0.001), with higher comorbidity burden (mean CCI = 0.95 versus 0.63; *P*<0.001), insured by Medicare (57.7% vs 19.5%; *P*<0.001), and have had a primary-care physician (PCP) visit in 2019 (62.7% vs 45.3%; *P*<0. 001). Propensity-score matching yielded 49,862 BP users and 49,862 BP non-users with no significant differences across examined characteristics. A total of 173 BP users from the unmatched New York state primary analysis cohort were not assigned an applicable BP non-user pair during the matching procedure and were excluded from the matched BP user population.

## Bone-Rx analysis study population

### All observations (all regions combined)

A total of 502,895 patients were included in the unmatched "*Bone-Rx*" analysis cohort comparisons, of which 452,051 (89.9%) and 50,844 (10.1%) were classified as BP users and BP non-users, respectively (*Appendix 2—table 17*). Prior to propensity-score matching, there were significant differences across all demographic and clinical characteristics. Compared to BP non-users, BP users were younger (47.9% age ≥71 vs 55.2%; *P*<0.001), predominantly female (90.1% vs 87.2%; *P*<0.001), with a lower comorbidity burden (mean CCI = 0.95 vs 1.99; *P*<0.001), with a larger proportion of patients residing in the west (21.1% versus 15.8%; *P*<0.001), a lower proportion covered by Medicare (43.4% vs 47.5%; *P*<0.001), and a lower proportion have had a primary-care physician (PCP) visit in 2019 (63.8% vs 64.3%; *P*=0.009). Propensity-score matching yielded 50,498 BP users and 50,498 BP non-users with no significant differences across examined characteristics. A total of 346 BP non-users from the unmatched "*Bone-Rx*" analysis cohort were not assigned an applicable BP user pair during the matching procedure and were excluded from the matched BP non-user population.

### Northeast region

A total of 135,867 patients identified as residing in the northeast were included in the unmatched "*Bone-Rx*" analysis cohort comparisons, of which 119,728 (88.1%) and 16,139 (11.9%) were classified as BP users and BP non-users, respectively (*Appendix 2—table 18*). Prior to propensity-score matching based on BP-use, there were significant differences across all demographic and clinical characteristics except for any PCP visit in 2019 (*P*=0.95). Compared to BP non-users, BP users were younger (48.1% age ≥71 vs 54.8%; *P*<0.001), predominantly female (90.5% vs 87.5%; *P*<0.001), with a lower comorbidity burden (mean CCI = 0.93 vs 1.97; *P*<0.001), and a lower proportion insured by Medicare (46.5% vs 54.0%; *P*<0.001). Propensity-score matching yielded 15,993 BP users and 15,993 BP non-users with no significant differences across examined characteristics. A total of 146 BP non-users from the northeast region in the unmatched "*Bone-Rx*" analysis cohort were not assigned an applicable BP user pair during the matching procedure and were excluded from the matched BP non-user population.

### Midwest region

A total of 85,391 patients identified as residing in the midwest were included in the unmatched "*Bone-Rx*" analysis cohort comparisons, of which 75,967 (89.0%) and 9,424 (11.0%) were classified as BP users and BP non-users, respectively (*Appendix 2—table 19*). Prior to propensity-score matching, there were significant differences across all demographic and clinical characteristics. Compared to BP non-users, BP users were younger (43.0% age ≥71 vs 54.1%; *P*<0.001), predominantly female (90.3% versus 86.1%; *P*<0.001), with a lower comorbidity burden (mean CCI = 0.99 versus 2.12; *P*<0.001), had a lower proportion insured by Medicare (43.6% versus 51.9%; *P*<0.001), with a lower proportion having a primary-care physician (PCP) visit in 2019 (62.2% vs 64.7%; *P*<0.001). Propensity-score matching yielded 9,360 BP users and 9,360 BP non-users with no significant differences across examined characteristics. A total of 64 BP non-users from the midwest region in the unmatched "*Bone-Rx*" analysis cohort were not assigned an applicable BP user pair during the matching procedure and were excluded from the matched BP non-user population.

## South region

A total of 178,118 patients identified as residing in the south were included in the unmatched "*Bone-Rx*" analysis cohort comparisons, of which 160,886 (90.3%) and 17,232 (9.7%) were classified as BP users and BP non-users, respectively (*Appendix 2—table 20*). Prior to propensity-score matching, there were significant differences across all demographic and clinical characteristics except for any PCP visit in 2019 (*P*=0.45). Compared to BP non-users, BP users were younger (46.6% age ≥71 vs 53.3%; *P*<0.001), predominantly female (90.6% vs 88.1%; *P*<0.001), with a lower comorbidity burden (mean CCI = 0.86 vs 1.86; *P*<0.001), and a lower proportion insured by Medicare (41.0% vs 44.0%; *P*<0.001). Propensity-score matching yielded 17,140 BP users and 17,140 BP non-users with no significant differences across examined characteristics. A total of 92 BP non-users from the south region in the unmatched "*Bone-Rx*" analysis cohort were not assigned an applicable BP user pair during the matching procedure and were excluded from the matched BP non-user population.

## West region

A total of 103,519 patients identified as residing in the west were included in the unmatched "*Bone-Rx*" analysis cohort comparisons, of which 95,470 (92.2%) and 8,049 (7.8%) were classified as BP users and BP non-users, respectively (*Appendix 2—table 21*). Prior to propensity-score matching, there were significant differences across all demographic and clinical characteristics. Compared to BP non-users, BP users were younger (54.1% age ≥71 vs 61.6%; *P*<0.001), predominantly female (88.7% vs 86.2%; *P*<0.001), with a lower comorbidity burden (mean CCI = 1.08 versus 2.17; *P*<0.001), insured by Medicare (43.5% vs 36.9%; *P*<0.001), with a lower proportion having a primary-care physician (PCP) visit in 2019 (67.7% vs 71.6%; *P*<0.001). Propensity-score matching yielded 8,005 BP users and 8,005 BP non-users with no significant differences across examined characteristics. A total of 44 BP non-users from the west region in the unmatched "*Bone-Rx*" analysis cohort were not assigned an applicable BP user pair during the matching procedure and were excluded from the matched BP non-user population.

## New York State

A total of 57,397 patients identified as residing in New York state were included in the unmatched "*Bone-Rx*" analysis NY-state restricted cohort, of which 50,035 (87.2%) and 7,362 (12.8%) were classified as BP users and BP non-users, respectively (*Appendix 2—table 22*). Prior to propensity-score matching, there were significant differences across all demographic and clinical characteristics except for any PCP visit in 2019 (*P*=0.35). Compared to BP non-users, BP users were younger (53.2% age ≥11 vs 54.5%; *P*<0.001), predominantly female (90.9% vs 89.5%; *P*<0.001), with a lower comorbidity burden (mean CCI = 0.95 vs 1.81; *P*<0.001), and a higher proportion insured by Medicaid (18.3% vs 13.8%; *P*<0.001). Propensity-score matching yielded 7,254 BP users and 7,254 BP non-users with no significant differences across examined characteristics. A total of 108 BP non-users from the unmatched New York state "*Bone-Rx*" analysis cohort were not assigned an applicable BP user pair during the matching procedure and were excluded from the matched BP non-user population.

## Osteo-Dx-Rx analysis study population

A total of 60,043 female patients age ≥51 with a diagnosis of osteoporosis who resided in New York (NY), Illinois (IL), Florida (FL), or California (CA) were included in the unmatched "*Osteo-Dx-Rx*" analysis cohort comparison, of which 51,651 (86.0%) and 8,392 (14.0%) were classified as BP users and BP non-users, respectively (*Appendix 2—table 23*). Prior to propensity-score matching, which was performed within each state by insurance type, there were significant differences across all demographic and clinical characteristics except the proportion of patients with a diagnosis of dyslipidemia (*P*=0.08). Compared to BP non-users, BP users were younger (18.8% age ≥81 vs 26.0%; *P*<0.001), with a larger proportion of patients residing in CA (42.5% vs 30.5%; *P*<0.001), insured by Medicaid (23.1% versus 21.3%; *P*<0.001), have had a primary-care physician (PCP) visit in 2019 (77.4% vs 71.1%; *P*<0.001), had a higher proportion with a diagnosis of obesity (11.2% vs 9.6%; *P*<0.001), and had a lower proportion diagnosed with the following: cancer (11.8% vs 19.4%; *P*<0.001), COPD (10.1% vs 16.2%; *P*<0.001), heart failure (6.1% vs 10.7%; *P*<0.001), hypertension (58.0% vs 60.9%; *P*<0.001), type 2 diabetes (25.6% vs 26.9%; *P*<0.01), and depression (13.9% vs 15.2%; *P*<0.001). Propensity-score matching yielded 7,949 BP users and 7,949 BP non-users with no significant differences across examined characteristics. A total of 443 BP non-users from the unmatched "*Osteo-Dx-Rx*" analysis cohort were not assigned an applicable BP user pair during the matching procedure and were excluded from the matched BP non-user population.

## Statin user/non-user analysis

### Statin-use comparison: All observations (all regions combined)

A total of 7,906,603 patients were included in the unmatched analysis cohort comparison of statin-use, of which 1,503,395 (19.0%) and 6,403,208 (81.0%) were classified as statin users and statin non-users, respectively (***Appendix 2—table 24***). Prior to propensity-score matching, there were significant differences across all demographic and clinical characteristics. Compared to statin non-users, statin users were older (87.9% age ≥51 vs 37.1%; *P*<0.001), with a higher proportion of males (41.1% vs 40.9%; *P*<0.001), from the northeast (29.7% versus 26.6%; *P*<0.001), with higher comorbidity burden (mean CCI = 1.15 vs 0.49; *P*<0.001), insured by Medicare (32.7% vs 11.3%; *P*<0.001), and have had a primary-care physician (PCP) visit in 2019 (66.1% vs 44.1%; *P*<0.001). Propensity-score matching yielded 1,436,300 statin users and 1,436,300 statin non-users with no significant differences across age group, region, insurance type, and having had any PCP visit in 2019. The final matched population did, however, display statistically significant differences between statin users and statin non-users for gender (58.7% vs 58.4% male; *P*<0.001) and mean CCI (1.11 vs 1.12; *P*<0.001). These differences, however, are small in magnitude, and were statistically significant due to the underlying statistical power associated with the large sample size. A total of 67,095 statin users from the unmatched analysis cohort were not assigned an applicable statin non-user pair during the matching procedure and were excluded from the matched statin user population.

### Statin-use comparison: New York State

A total of 968,296 patients identified as residing in New York state were included in the unmatched analysis cohort comparison of statin-use, of which 206,301 (21.3%) and 761,995 (78.7%) were classified as statin users and statin non-users, respectively (***Appendix 2—table 25***). Prior to propensity-score matching, there were significant differences across all demographic and clinical characteristics. Compared to statin non-users, statin users were older (90.3% age ≥51 vs 43.1%; *P*<0.001), with a higher proportion of males (42.0% vs 40.4%; *P*<0.001), with higher comorbidity burden (mean CCI = 0.94=0.94=0.94 1.17 vs 0.51; *P*<0.001), insured by Medicare (47.4% versus 14.5%; *P*<0.001), and have had a primary-care physician (PCP) visit in 2019 (64.0% vs 41.3%; *P*<0.001). Propensity-score matching yielded 185,536 statin users and 185,536 statin non-users with no significant differences across age group, gender, insurance type, and having had any PCP visit in 2019. The final matched population did, however, display statistically significant differences between statin users and statin non-users for mean CCI (1.06 vs 1.08; *P*<0.001). This difference, however, is small in magnitude, and was statistically significant due to the underlying statistical power associated with the large sample size. A total of 20,765 statin users from the unmatched analysis cohort were not assigned an applicable statin non-user pair during the matching procedure and were excluded from the matched statin user population.

### BP-use comparison within statin users: All regions combined

Of the 1,436,300 statin users from the statin user/non-user propensity-score matching analysis, a total of 217,981 (15.2%) and 1,218,319 (84.8%) were classified as BP users and BP non-users, respectively (***Appendix 2—table 26***). Prior to propensity-score matching based on BP-use, there were significant differences across all demographic and clinical characteristics except for any PCP visit in 2019 (*P*=0.27). Compared to BP non-users, BP users were older (98.9% age ≥51 vs 85.3%; *P*<0.001), with a higher proportion of females (90.1% vs 53.1%; *P*<0.001), from the west (21.7% vs 14.0%; *P*<0.001), with lower comorbidity burden (mean CCI = 0.95 vs 1.13; *P*<0.001), and insured by Medicare (50.8% vs 29.7%; *P*<0.001). Propensity-score matching yielded 213,480 BP users and 213,480 BP non-users with no significant differences across examined characteristics. A total of 4,501 BP users were not assigned an applicable BP non-user pair during the matching procedure and were excluded from the matched BP user population.

### BP-use comparison within statin users: New York State

Of the 185,536 statin users from the statin user/non-user propensity-score matching analysis on patients residing in New York state, a total of 23,863 (12.9%) and 161,673 (87.1%) were classified as BP users and BP non-users, respectively (***Appendix 2—table 27***). Prior to propensity-score matching based on BP-use, there were significant differences across all demographic and clinical characteristics except for any PCP visit in 2019 (*P*=0.33). Compared to BP non-users, BP users were older (99.3% age ≥51 vs 87.7%; *P*<0.001), with a higher proportion of females (91.2% vs 53.3%; *P*<0.001), with lower comorbidity burden (mean CCI = 0.92 versus 1.08; *P*<0.001), and insured by Medicare (66.4% vs 41.9%; *P*<0.001). Propensity-score matching yielded 23,736 BP users and 23,736 BP non-users

with no significant differences across examined characteristics. A total of 127 BP users were not assigned an applicable BP non-user pair during the matching procedure and were excluded from the matched BP user population.

## BP-use comparison within statin non-users: All regions combined

Of the 1,436,300 statin non-users from the statin user/non-user propensity-score matching analysis, a total of 124,843 (8.7%) and 1,311,457 (91.3%) were classified as BP users and BP non-users, respectively (*Appendix 2—table 28*). Prior to propensity-score matching based on BP-use, there were significant differences across all demographic and clinical characteristics. Compared to BP non-users, BP users were older (98.7% age ≥51 vs 86.3%; *P*<0.001), with a higher proportion of females (89.6% vs 55.5%; *P*<0.001), from the west (21.4% vs 14.6%; *P*<0.001), with lower comorbidity burden (mean CCI = 1.02 versus 1.13; *P*<0.001), insured by Medicare (45.8% vs 31.7%; *P*<0.001), and have had a primary-care physician (PCP) visit in 2019 (71.7% vs 63.9%; *P*<0.001). Propensity-score matching yielded 124,716 BP users and 124,716 BP non-users with no significant differences across examined characteristics. A total of 127 BP users were not assigned an applicable BP non-user pair during the matching procedure and were excluded from the matched BP user population.

## BP-use comparison within statin non-users: New York State

Of the 185,536 statin non-users from the statin user/non-user propensity-score matching analysis on patients residing in New York state, a total of 14,546 (7.8%) and 170,990 (92.2%) were classified as BP users and BP non-users, respectively (*Appendix 2—table 29*). Prior to propensity-score matching based on BP-use, there were significant differences across all demographic and clinical characteristics. Compared to BP non-users, BP users were older (99.2% age ≥51 vs 88.4%; *P*<0.001), with a higher proportion of females (90.6% vs 55.1%; *P*<0.001), with lower comorbidity burden (mean CCI = 0.95 vs 1.09; *P*<0.001), insured by Medicare (59.7% versus 43.7%; *P*<0.001), and have had a primary-care physician (PCP) visit in 2019 (70.5% vs 59.4%; *P*<0.001). Propensity-score matching yielded 14,521 BP users and 14,521 BP non-users with no significant differences across examined characteristics. A total of 25 BP users were not assigned an applicable BP non-user pair during the matching procedure and were excluded from the matched BP user population.

## Antihypertensive user/non-user analysis

### Antihypertensive-use comparison: All observations (all regions combined)

A total of 7,906,603 patients were included in the unmatched analysis cohort comparison of antihypertensive-use, of which 2,101,120 (26.6%) and 5,805,483 (73.4%) were classified as antihypertensive users and antihypertensive non-users, respectively (*Appendix 2—table 30*). Prior to propensity-score matching, there were significant differences across all demographic and clinical characteristics. Compared to antihypertensive non-users, antihypertensive users were older (80.8% age ≥51 vs 34.4%; *P*<0.001), with a higher proportion of females (60.4% vs 58.6%; *P*<0.001), from the northeast (27.8% vs 27.0%; *P*<0.001), with higher comorbidity burden (mean CCI = 1.13 vs 0.43; *P*<0.001), insured by Medicare (29.5% vs 10.3%; *P*<0.001), and have had a primary-care physician (PCP) visit in 2019 (64.2% vs 39.2%; *P*<0.001). Propensity-score matching yielded 1,786,001 antihypertensive users and 1,786,001 antihypertensive non-users with no significant differences across age group, gender, region, insurance type, and having had any PCP visit in 2019. The final matched population did, however, display statistically significant difference between antihypertensive users and antihypertensive non-users for mean CCI (1.64 vs 1.66; *P*<0.05). This difference, however, is small in magnitude, and was statistically significant due to the underlying statistical power associated with the large sample size. A total of 315,119 antihypertensive users from the unmatched analysis cohort were not assigned an applicable antihypertensive non-user pair during the matching procedure and were excluded from the matched antihypertensive user population.

### Antihypertensive-use comparison: New York State

A total of 968,296 patients identified as residing in New York state were included in the unmatched analysis cohort comparison of antihypertensive-use, of which 258,652 (26.7%) and 709,644 (73.3%) were classified as antihypertensive users and antihypertensive non-users, respectively (*Appendix 2—table 31*). Prior to propensity-score matching, there were significant differences across all demographic and clinical characteristics. Compared to antihypertensive non-users, antihypertensive users were older (86.6% age ≥51 vs 40.9%; *P*<0.001), with a higher proportion of females (59.4% vs 59.2%; *P*=0.02), with higher comorbidity burden (mean CCI = 1.17 vs 0.46; *P*<0.001), insured by Medicare (45.9% vs 12.6%; *P*<0.001), and have had a primary-care physician (PCP) visit in 2019

(62.4% vs 40.3%; *P*<0.001). Propensity-score matching yielded 203,624 antihypertensive users and 203,624 antihypertensive non-users with no significant differences across examined characteristics. A total of 55,028 antihypertensive users from the unmatched analysis cohort were not assigned an applicable antihypertensive non-user pair during the matching procedure and were excluded from the matched antihypertensive user population.

## BP-use comparison within antihypertensive users: All regions combined

Of the 1,786,001 antihypertensive users from the antihypertensive user/non-user propensity-score matching analysis, a total of 206,613 (11.6%) and 1,579,388 (88.4%) were classified as BP users and BP non-users, respectively (*Appendix 2—table 32*). Prior to propensity-score matching based on BP-use, there were significant differences across all demographic and clinical characteristics. Compared to BP non-users, BP users were older (98.2% age ≥51 vs 75.2%; *P*<0.001), with a higher proportion of females (89.7% vs 56.6%; *P*<0.001), from the west (22.0% vs 14.3%; *P*<0.001), with lower comorbidity burden (mean CCI = 0.94 versus 0.95; *P*=0.02), insured by Medicare (48.6% vs 24.4%; *P*<0.001), and have not had a primary-care physician (PCP) visit in 2019 (41.2% vs 40.1%; *P*<0.001). Propensity-score matching yielded 204,396 BP users and 204,396 BP non-users with no significant differences across examined characteristics. A total of 2,217 BP users were not assigned an applicable BP non-user pair during the matching procedure and were excluded from the matched BP user population.

## BP-use comparison within antihypertensive users: New York State

Of the 203,624 antihypertensive users from the antihypertensive user/non-user propensity-score matching analysis on patients residing in New York state, a total of 21,213 (10.4%) and 182,411 (89.6%) were classified as BP users and BP non-users, respectively (*Appendix 2—table 33*). Prior to propensity-score matching based on BP-use, there were significant differences across all demographic and clinical characteristics. Compared to BP non-users, BP users were older (98.8% age ≥51 vs 81.4%; *P*<0.001), with a higher proportion of females (90.9% vs 55.5%; *P*<0.001), with lower comorbidity burden (mean CCI = 0.88 vs 0.95; *P*<0.001), insured by Medicare (64.1% vs 35.9%; *P*<0.001), and have not had a primary-care physician (PCP) visit in 2019 (53.4% vs 55.7%; *P*<0.001). Propensity-score matching yielded 21,126 BP users and 21,126 BP non-users with no significant differences across examined characteristics. A total of 87 BP users were not assigned an applicable BP non-user pair during the matching procedure and were excluded from the matched BP user population.

## BP-use comparison within antihypertensive non-users: All regions combined

Of the 1,786,001 antihypertensive non-users from the antihypertensive user/non-user propensity-score matching analysis, a total of 136,016 (7.6%) and 1,649,985 (92.4%) were classified as BP users and BP non-users, respectively (*Appendix 2—table 34*). Prior to propensity-score matching based on BP-use, there were significant differences across all demographic and clinical characteristics. Compared to BP non-users, BP users were older (97.7% age ≥51 vs 76.3%; *P*<0.001), with a higher proportion of females (90.5% vs 58.0%; *P*<0.001), from the west (20.3% vs 14.8%; *P*<0.001), with lower comorbidity burden (mean CCI = 0.88 versus 0.96; *P*<0.001), insured by Medicare (40.7% vs 26.0%; *P*<0.001), and have had a primary-care physician (PCP) visit in 2019 (68.0% vs 59.0%; *P*<0.001). Propensity-score matching yielded 135,724 BP users and 135,724 BP non-users with no significant differences across examined characteristics. A total of 292 BP users were not assigned an applicable BP non-user pair during the matching procedure and were excluded from the matched BP user population.

## BP-use comparison within antihypertensive non-users: New York State

Of the 203,624 antihypertensive non-users from the antihypertensive user/non-user propensity-score matching analysis on patients residing in New York state, a total of 14,051 (6.9%) and 189,573 (93.1%) were classified as BP users and BP non-users, respectively (*Appendix 2—table 35*). Prior to propensity-score matching based on BP-use, there were significant differences across all demographic and clinical characteristics. Compared to BP non-users, BP users were older (98.7% age ≥51 vs 82.1%; *P*<0.001), with a higher proportion of females (91.3% vs 56.8%; *P*<0.001), with lower comorbidity burden (mean CCI = 0.81 vs 0.96; *P*<0.001), insured by Medicare (54.9% vs 37.7%; *P*<0.001), and have had a primary-care physician (PCP) visit in 2019 (66.3% vs 54.7%; *P*<0.001). Propensity-score matching yielded 13,983 BP users and 13,983 BP non-users with no significant differences across examined characteristics. A total of 68 BP users were not assigned an

applicable BP non-user pair during the matching procedure and were excluded from the matched BP user population.

## Antidiabetic user/non-user analysis

### Antidiabetic-use cComparison: All observations (all regions combined)

A total of 7,906,603 patients were included in the unmatched analysis cohort comparison of antidiabetic-use, of which 755,252 (9.6%) and 7,151,351 (90.4%) were classified as antidiabetic users and antidiabetic non-users, respectively (*Appendix 2—table 36*). Prior to propensity-score matching, there were significant differences across all demographic and clinical characteristics. Compared to antidiabetic non-users, antidiabetic users were older (79.4% age ≥51 vs 43.3%; $P<0.001$), with a higher proportion of females (60.8% vs 58.9%; $P<0.001$), from the northeast (28.8% vs 27.1%; $P<0.001$), with higher comorbidity burden (mean CCI = 1.25 vs 0.55; $P<0.001$), insured by Medicare (26.2% vs 14.2%; $P<0.001$), and have had a primary-care physician (PCP) visit in 2019 (66.5% vs 43.6%; $P<0.001$). Propensity-score matching yielded 754,553 antidiabetic users and 754,553 antidiabetic non-users with no significant differences across examined characteristics. A total of 699 antidiabetic users from the unmatched analysis cohort were not assigned an applicable antidiabetic non-user pair during the matching procedure and were excluded from the matched antidiabetic user population.

### Antidiabetic-use comparison: New York State

A total of 968,296 patients identified as residing in New York state were included in the unmatched analysis cohort comparison of antidiabetic-use, of which 105,117 (10.9%) and 863,179 (89.1%) were classified as antidiabetic users and antidiabetic non-users, respectively (*Appendix 2—table 37*). Prior to propensity-score matching, there were significant differences across all demographic and clinical characteristics. Compared to antidiabetic non-users, antidiabetic users were older (83.8% age ≥51 vs 49.4%; $P<0.001$), with a higher proportion of males (42.2% vs 40.6%; $P<0.001$), with higher comorbidity burden (mean CCI = 1.34 vs 0.56; $P<0.001$), insured by Medicare (40.5% vs 19.2%; $P<0.001$), and have had a primary-care physician (PCP) visit in 2019 (64.6% vs 43.9%; $P<0.001$). Propensity-score matching yielded 104,691 antidiabetic users and 104,691 antidiabetic non-users with no significant differences across examined characteristics. A total of 426 antidiabetic users from the unmatched analysis cohort were not assigned an applicable antidiabetic non-user pair during the matching procedure and were excluded from the matched antidiabetic user population.

### BP-use comparison within antidiabetic users: All regions combined

Of the 754,553 antidiabetic users from the antidiabetic user/non-user propensity-score matching analysis, a total of 80,529 (10.7%) and 674,024 (89.3%) were classified as BP users and BP non-users, respectively (*Appendix 2—table 38*). Prior to propensity-score matching based on BP-use, there were significant differences across all demographic and clinical characteristics. Compared to BP non-users, BP users were older (98.2% age ≥51 vs 75.2%; $P<0.001$), with a higher proportion of females (98.5% vs 77.1%; $P<0.001$), from the west (22.2% versus 14.2%; $P<0.001$), with a higher comorbidity burden (mean CCI = 1.32 versus 1.23; $P<0.001$), insured by Medicare (45.2% vs 24.0%; $P<0.001$), and have had a primary-care physician (PCP) visit in 2019 (69.5% vs 66.1%; $P<0.001$). Propensity-score matching yielded 79,500 BP users and 79,500 BP non-users with no significant differences across examined characteristics. A total of 1,029 BP users were not assigned an applicable BP non-user pair during the matching procedure and were excluded from the matched BP user population.

### BP-use comparison within antidiabetic users: New York State

Of the 104,691 antidiabetic users from the antidiabetic user/non-user propensity-score matching analysis on patients residing in New York state, a total of 9,529 (9.1%) and 95,162 (90.9%) were classified as BP users and BP non-users, respectively (*Appendix 2—table 39*). Prior to propensity-score matching based on BP-use, there were significant differences across all demographic and clinical characteristics. Compared to BP non-users, BP users were older (99.1% age ≥51 vs 82.2%; $P<0.001$), with a higher proportion of females (90.1% vs 54.5%; $P<0.001$), with a higher comorbidity burden (mean CCI = 1.46 vs 1.31; $P<0.001$), insured by Medicare (64.6% vs 38.2%; $P<0.001$), and have had a primary-care physician (PCP) visit in 2019 (66.3% vs 64.4%; $P<0.001$). Propensity-score matching yielded 9,456 BP users and 9,456 BP non-users with no significant differences across

examined characteristics. A total of 73 BP users were not assigned an applicable BP non-user pair during the matching procedure and were excluded from the matched BP user population.

### BP-use comparison within antidiabetic non-users: All regions combined

Of the 754,553 antidiabetic non-users from the antidiabetic user/non-user propensity-score matching analysis, a total of 73,173 (9.7%) and 681,380 (90.3%) were classified as BP users and BP non-users, respectively (*Appendix 2—table 40*). Prior to propensity-score matching based on BP-use, there were significant differences across all demographic characteristics, but no difference was seen in mean CCI (1.24 vs 1.24; *P*=0.92). Compared to BP non-users, BP users were older (98.0% age ≥51 vs 77.3%; *P*<0.001), with a higher proportion of females (88.9% vs 57.7%; *P*<0.001), from the west (20.1% vs 14.5%; *P*<0.001), insured by Medicare (40.0% vs 24.8%; *P*<0.001), and have had a primary-care physician (PCP) visit in 2019 (74.1% vs 65.7%; *P*<0.001). Propensity-score matching yielded 72,514 BP users and 72,514 BP non-users with no significant differences across examined characteristics. A total of 659 BP users were not assigned an applicable BP non-user pair during the matching procedure and were excluded from the matched BP user population.

### BP-use comparison within antidiabetic non-users: New York State

Of the 104,691 antidiabetic non-users from the antidiabetic user/non-user propensity-score matching analysis on patients residing in New York state, a total of 9,275 (8.9%) and 95,416 (91.1%) were classified as BP users and BP non-users, respectively (*Appendix 2—table 41*). Prior to propensity-score matching based on BP-use, there were significant differences across all demographic and clinical characteristics. Compared to BP non-users, BP users were older (99.0% age ≥51 vs 82.2%; *P*<0.001), with a higher proportion of females (89.2% vs 54.7%; *P*<0.001), with a higher comorbidity burden (mean CCI = 1.37 vs 1.32; *P*<0.01), insured by Medicare (57.7% vs 38.9%; *P*<0.001), and have had a primary-care physician (PCP) visit in 2019 (72.5% vs 63.8%; *P*<0.001). Propensity-score matching yielded 13,983 BP users and 13,983 BP non-users with no significant differences across examined characteristics. A total of 131 BP users were not assigned an applicable BP non-user pair during the matching procedure and were excluded from the matched BP user population.

## Antidepressant user/non-user analysis

### Antidepressant-use comparison: All observations (all regions combined)

A total of 7,906,603 patients were included in the unmatched analysis cohort comparison of antidepressant-use, of which 1,571,005 (19.9%) and 6,335,598 (80.1%) were classified as antidepressant users and antidepressant non-users, respectively (*Appendix 2—table 42*). Prior to propensity-score matching, there were significant differences across all demographic and clinical characteristics. Compared to antidepressant non-users, antidepressant users were older (58.6% age ≥51 vs 43.8%; *P*<0.001), with a higher proportion of females (72.8% vs 55.7%; *P*<0.001), from the midwest (22.1% vs 17.7%; *P*<0.001), with higher comorbidity burden (mean CCI = 0.90 vs 0.55; *P*<0.001), insured by Medicare (18.5% vs 14.6%; *P*<0.001), and have had a primary-care physician (PCP) visit in 2019 (61.1% versus 42.0%; *P*<0.001). Propensity-score matching yielded 1,536,048 antidepressant users and 1,536,048 antidepressant non-users with no significant differences across examined characteristics. A total of 34,957 antidepressant users from the unmatched analysis cohort were not assigned an applicable antidepressant non-user pair during the matching procedure and were excluded from the matched antidepressant user population.

### Antidepressant-use comparison: New York State

A total of 968,296 patients identified as residing in New York state were included in the unmatched analysis cohort comparison of antidepressant-use, of which 136,081 (14.1%) and 832,215 (85.9%) were classified as antidepressant users and antidepressant non-users, respectively (*Appendix 2—table 43*). Prior to propensity-score matching, there were significant differences across all demographic and clinical characteristics. Compared to antidepressant non-users, antidepressant users were older (66.3% age ≥51 vs 51.0%; *P*<0.001), with a higher proportion of females (71.2% vs 57.3%; *P*<0.001), with higher comorbidity burden (mean CCI = 0.98 vs 0.59; *P*<0.001), insured by Medicare (32.2% vs 19.8%; *P*<0.001), and have had a primary-care physician (PCP) visit in 2019 (60.7% vs 43.8%; *P*<0.001). Propensity-score matching yielded 135,516 antidepressant users and 135,516 antidepressant non-users with no significant differences across examined characteristics.

A total of 565 antidepressant users from the unmatched analysis cohort were not assigned an applicable antidepressant non-user pair during the matching procedure and were excluded from the matched antidepressant user population.

## BP-use comparison within antidepressant users: All regions combined

Of the 1,536,048 antidepressant users from the antidepressant user/non-user propensity-score matching analysis, a total of 145,109 (9.4%) and 1,390,939 (90.6%) were classified as BP users and BP non-users, respectively (*Appendix 2—table 44*). Prior to propensity-score matching based on BP-use, there were significant differences across all demographic and clinical characteristics. Compared to BP non-users, BP users were older (96.7% age ≥51 vs 54.4%; *P*<0.001), with a higher proportion of females (91.9% vs 70.2%; *P*<0.001), from the west (19.6% versus 13.9%; *P*<0.001), with a higher comorbidity burden (mean CCI = 1.09 versus 0.84; *P*<0.001), insured by Medicare (42.4% vs 16.2%; *P*<0.001), and have had a primary-care physician (PCP) visit in 2019 (64.6% vs 60.2%; *P*<0.001). Propensity-score matching yielded 144,282 BP users and 144,282 BP non-users with no significant differences across examined characteristics. A total of 827 BP users were not assigned an applicable BP non-user pair during the matching procedure and were excluded from the matched BP user population.

## BP-use comparison within antidepressant users: New York State

Of the 135,516 antidepressant users from the antidepressant user/non-user propensity-score matching analysis on patients residing in New York state, a total of 12,950 (9.6%) and 122,566 (90.4%) were classified as BP users and BP non-users, respectively (*Appendix 2—table 45*). Prior to propensity-score matching based on BP-use, there were significant differences across all demographic and clinical characteristics. Compared to BP non-users, BP users were older (97.8% age ≥51 vs 63.0%; *P*<0.001), with a higher proportion of females (92.6% vs 68.9%; *P*<0.001), with a higher comorbidity burden (mean CCI = 1.13 vs 0.95; *P*<0.001), insured by Medicare (60.8% vs 29.1%; *P*<0.001), and have had a primary-care physician (PCP) visit in 2019 (65.3% vs 60.1%; *P*<0.001). Propensity-score matching yielded 12,859 BP users and 12,859 BP non-users with no significant differences across examined characteristics. A total of 91 BP users were not assigned an applicable BP non-user pair during the matching procedure and were excluded from the matched BP user population.

## BP-use comparison within antidepressant non-users: All regions combined

Of the 1,536,048 antidepressant non-users from the antidepressant user/non-user propensity-score matching analysis, a total of 113,110 (7.4%) and 1,422,938 (92.6%) were classified as BP users and BP non-users, respectively (*Appendix 2—table 46*). Prior to propensity-score matching based on BP-use, there were significant differences across all demographic characteristics. Compared to BP non-users, BP users were older (97.1% age ≥51 vs 55.4%; *P*<0.001), with a higher proportion of females (93.2% vs 70.6%; *P*<0.001), from the west (20.0% versus 14.0%; *P*<0.001), with a higher comorbidity burden (mean CCI = 1.06 versus 0.85; *P*<0.001), insured by Medicare (40.4% vs 17.0%; *P*<0.001), and have had a primary-care physician (PCP) visit in 2019 (71.2% vs 59.8%; *P*<0.001). Propensity-score matching yielded 112,402 BP users and 112,402 BP non-users with no significant differences across examined characteristics. A total of 708 BP users were not assigned an applicable BP non-user pair during the matching procedure and were excluded from the matched BP user population.

## BP-use comparison within antidepressant non-users: New York State

Of the 135,516 antidepressant non-users from the antidepressant user/non-user propensity-score matching analysis on patients residing in New York state, a total of 10,174 (7.5%) and 125,342 (92.5%) were classified as BP users and BP non-users, respectively (*Appendix 2—table 47*). Prior to propensity-score matching based on BP-use, there were significant differences across all demographic and clinical characteristics. Compared to BP non-users, BP users were older (98.4% age ≥51 vs 63.7%; *P*<0.001), with a higher proportion of females (93.6% vs 69.4%; *P*<0.001), with a higher comorbidity burden (mean CCI = 1.13 vs 0.95; *P*<0.01), insured by Medicare (60.0% vs 29.9%; *P*<0.001), and have had a primary-care physician (PCP) visit in 2019 (71.7% vs 59.7%; *P*<0.001). Propensity-score matching yielded 10,091 BP users and 10,091 BP non-users with no significant differences across examined characteristics. A total of 83 BP users were not assigned an applicable BP non-user pair during the matching procedure and were excluded from the matched BP user population.

**Appendix 2—table 1.** Primary Analysis Cohort (Region=Northeast), Patient Characteristics Pre/Post Match.

| | Region=Northeast Unmatched | | | | | | | Region=Northeast Matched | | | | | | |
| --- | --- | --- | --- | --- | --- | --- | --- | --- | --- | --- | --- | --- | --- | --- |
| | All | | BP Non-users | | BP Users | | p-value | All | | BP Non-users | | BP Users | | p-value |
| | N | % | N | % | N | % | | N | % | N | % | N | % | |
| All Patients | 2,152,560 | 100.00% | 2,032,832 | 94.40% | 119,728 | 5.60% | | 238,988 | 100.00% | 119,494 | 50.00% | 119,494 | 50.00% | |
| **Age** | | | | | | | | | | | | | | |
| ≤20 | 363,637 | 16.90% | 363,401 | 17.90% | 236 | 0.20% | <0.001 | 474 | 0.20% | 238 | 0.20% | 236 | 0.20% | 1 |
| 21-40 | 397,377 | 18.50% | 396,613 | 19.50% | 764 | 0.60% | | 1,528 | 0.60% | 764 | 0.60% | 764 | 0.60% | |
| 41-50 | 261,570 | 12.20% | 259,528 | 12.80% | 2,042 | 1.70% | | 4,084 | 1.70% | 2,042 | 1.70% | 2,042 | 1.70% | |
| 51-60 | 372,238 | 17.30% | 354,228 | 17.40% | 18,010 | 15.00% | | 36,020 | 15.10% | 18,010 | 15.10% | 18,010 | 15.10% | |
| 61-70 | 354,331 | 16.50% | 313,237 | 15.40% | 41,094 | 34.30% | | 82,233 | 34.40% | 41,139 | 34.40% | 41,094 | 34.40% | |
| 71-80 | 252,712 | 11.70% | 215,151 | 10.60% | 37,561 | 31.40% | | 74,831 | 31.30% | 37,393 | 31.30% | 37,438 | 31.30% | |
| ≥81 | 150,695 | 7.00% | 130,674 | 6.40% | 20,021 | 16.70% | | 39,818 | 16.70% | 19,908 | 16.70% | 19,910 | 16.70% | |
| **Gender** | | | | | | | | | | | | | | |
| Female | 1,275,611 | 59.30% | 1,167,241 | 57.40% | 108,370 | 90.50% | <0.001 | 216,273 | 90.50% | 108,137 | 90.50% | 108,136 | 90.50% | 0.99 |
| Male | 876,949 | 40.70% | 865,591 | 42.60% | 11,358 | 9.50% | | 22,715 | 9.50% | 11,357 | 9.50% | 11,358 | 9.50% | |
| **Insurance** | | | | | | | | | | | | | | |
| Commercial | 1,050,795 | 48.80% | 1,017,502 | 50.10% | 33,293 | 27.80% | <0.001 | 66,552 | 27.80% | 33,259 | 27.80% | 33,293 | 27.90% | 0.99 |
| Dual | 47,773 | 2.20% | 40,168 | 2.00% | 7,605 | 6.40% | | 15,114 | 6.30% | 7,576 | 6.30% | 7,538 | 6.30% | |
| Medicaid | 631,863 | 29.40% | 608,649 | 29.90% | 23,214 | 19.40% | | 46,094 | 19.30% | 23,047 | 19.30% | 23,047 | 19.30% | |
| Medicare | 422,129 | 19.60% | 366,513 | 18.00% | 55,616 | 46.50% | | 111,228 | 46.50% | 55,612 | 46.50% | 55,616 | 46.50% | |
| **PCP Visit 2019** | | | | | | | | | | | | | | |
| No | 1,212,394 | 56.30% | 1,162,527 | 57.20% | 49,867 | 41.70% | <0.001 | 99,741 | 41.70% | 49,874 | 41.70% | 49,867 | 41.70% | 0.98 |
| Yes | 940,166 | 43.70% | 870,305 | 42.80% | 69,861 | 58.30% | | 139,247 | 58.30% | 69,620 | 58.30% | 69,627 | 58.30% | |
| **Continuous Outcomes** | | | | | | | | | | | | | | |
| | mean | SD | mean | SD | mean | SD | p-value | mean | SD | mean | SD | mean | SD | p-value |
| CCI | 0.67 | 1.42 | 0.65 | 1.4 | 0.93 | 1.71 | <0.001 | 0.93 | 1.71 | 0.93 | 1.71 | 0.93 | 1.71 | 0.96 |

BP: bisphosphonate; CCI: Charlson Comorbidity Index; PCP: primary care physician; SD: standard deviation.

**Appendix 2—table 2.** Primary Analysis Cohort (Region=Midwest), Patient Characteristics Pre/Post Match.

| | Region=Midwest Unmatched | | | | | | | Region=Midwest Matched | | | | | | |
| | All | | BP Non-users | | BP Users | | p-value | All | | BP Non-users | | BP Users | | p-value |
| | N | % | N | % | N | % | | N | % | N | % | N | % | |
|---|---|---|---|---|---|---|---|---|---|---|---|---|---|---|
| All Patients | 1,467,802 | 100.0% | 1,391,835 | 94.8% | 75,967 | 5.2% | | 151,802 | 100.0% | 75,901 | 50.0% | 75,901 | 50.0% | |
| **Age** | | | | | | | | | | | | | | |
| ≤20 | 310,027 | 21.1% | 309,759 | 22.3% | 268 | 0.4% | <0.001 | 537 | 0.4% | 269 | 0.4% | 268 | 0.4% | 1.00 |
| 21-40 | 287,236 | 19.6% | 286,643 | 20.6% | 593 | 0.8% | | 1,188 | 0.8% | 595 | 0.8% | 593 | 0.8% | |
| 41-50 | 185,240 | 12.6% | 183,556 | 13.2% | 1,684 | 2.2% | | 3,367 | 2.2% | 1,683 | 2.2% | 1,684 | 2.2% | |
| 51-60 | 246,230 | 16.8% | 233,992 | 16.8% | 12,238 | 16.1% | | 24,478 | 16.1% | 12,240 | 16.1% | 12,238 | 16.1% | |
| 61-70 | 224,668 | 15.3% | 196,172 | 14.1% | 28,496 | 37.5% | | 56,991 | 37.5% | 28,495 | 37.5% | 28,496 | 37.5% | |
| 71-80 | 130,563 | 8.9% | 109,442 | 7.9% | 21,121 | 27.8% | | 42,153 | 27.8% | 21,075 | 27.8% | 21,078 | 27.8% | |
| ≥81 | 83,838 | 5.7% | 72,271 | 5.2% | 11,567 | 15.2% | | 23,088 | 15.2% | 11,544 | 15.2% | 11,544 | 15.2% | |
| **Gender** | | | | | | | | | | | | | | |
| Female | 863,156 | 58.8% | 794,578 | 57.1% | 68,578 | 90.3% | <0.001 | 137,028 | 90.3% | 68,516 | 90.3% | 68,512 | 90.3% | 0.97 |
| Male | 604,646 | 41.2% | 597,257 | 42.9% | 7,389 | 9.7% | | 14,774 | 9.7% | 7,385 | 9.7% | 7,389 | 9.7% | |
| **Insurance** | | | | | | | | | | | | | | |
| Commercial | 885,651 | 60.3% | 854,518 | 61.4% | 31,133 | 41.0% | <0.001 | 62,243 | 41.0% | 31,110 | 41.0% | 31,133 | 41.0% | 1.00 |
| Dual | 28,190 | 1.9% | 24,584 | 1.8% | 3,606 | 4.7% | | 7,211 | 4.8% | 3,605 | 4.7% | 3,606 | 4.8% | |
| Medicaid | 318,596 | 21.7% | 310,473 | 22.3% | 8,123 | 10.7% | | 16,136 | 10.6% | 8,079 | 10.6% | 8,057 | 10.6% | |
| Medicare | 235,365 | 16.0% | 202,260 | 14.5% | 33,105 | 43.6% | | 66,212 | 43.6% | 33,107 | 43.6% | 33,105 | 43.6% | |
| **PCP Visit 2019** | | | | | | | | | | | | | | |
| No | 711,308 | 48.5% | 682,601 | 49.0% | 28,707 | 37.8% | <0.001 | 57,398 | 37.8% | 28,691 | 37.8% | 28,707 | 37.8% | 0.93 |
| Yes | 756,494 | 51.5% | 709,234 | 51.0% | 47,260 | 62.2% | | 94,404 | 62.2% | 47,210 | 62.2% | 47,194 | 62.2% | |
| **Continuous Outcomes** | | | | | | | | | | | | | | |
| | mean | SD | mean | SD | mean | SD | p-value | mean | SD | mean | SD | mean | SD | p-value |
| CCI | 0.59 | 1.37 | 0.56 | 1.34 | 0.99 | 1.86 | <0.001 | 0.99 | 1.86 | 0.99 | 1.85 | 1.00 | 1.86 | 0.77 |

BP: bisphosphonate; CCI: Charlson Comorbidity Index; PCP: primary care physician; SD: standard deviation.

**Appendix 2—table 3.** Primary Analysis Cohort (Region=South), Patient Characteristics Pre/Post Match.

| | Region=South Unmatched | | | | | | | Region=South Matched | | | | | | |
| | All | | BP Non-users | | BP Users | | p-value | All | | BP Non-users | | BP Users | | p-value |
| | N | % | N | % | N | % | | N | % | N | % | N | % | |
| All Patients | 3,042,604 | 100.0% | 2,881,718 | 94.7% | 160,886 | 5.3% | | 319,408 | 100.0% | 159,704 | 50.0% | 159,704 | 50.0% | |
| **Age** | | | | | | | | | | | | | | |
| ≤20 | 890,677 | 29.3% | 890,203 | 30.9% | 474 | 0.3% | <0.001 | 943 | 0.3% | 469 | 0.3% | 474 | 0.3% | 1.00 |
| 21-40 | 527,971 | 17.4% | 526,794 | 18.3% | 1,177 | 0.7% | | 2,364 | 0.7% | 1,187 | 0.7% | 1,177 | 0.7% | |
| 41-50 | 338,262 | 11.1% | 334,841 | 11.6% | 3,421 | 2.1% | | 6,839 | 2.1% | 3,418 | 2.1% | 3,421 | 2.1% | |
| 51-60 | 442,757 | 14.6% | 417,664 | 14.5% | 25,093 | 15.6% | | 50,186 | 15.7% | 25,093 | 15.7% | 25,093 | 15.7% | |
| 61-70 | 409,854 | 13.5% | 353,958 | 12.3% | 55,896 | 34.7% | | 111,800 | 35.0% | 55,904 | 35.0% | 55,896 | 35.0% | |
| 71-80 | 272,761 | 9.0% | 222,156 | 7.7% | 50,605 | 31.5% | | 99,223 | 31.1% | 49,605 | 31.1% | 49,618 | 31.1% | |
| ≥81 | 160,322 | 5.3% | 136,102 | 4.7% | 24,220 | 15.1% | | 48,053 | 15.0% | 24,028 | 15.0% | 24,025 | 15.0% | |
| **Gender** | | | | | | | | | | | | | | |
| Female | 1,800,166 | 59.2% | 1,654,351 | 57.4% | 145,815 | 90.6% | <0.001 | 289,263 | 90.6% | 144,630 | 90.6% | 144,633 | 90.6% | 0.99 |
| Male | 1,242,438 | 40.8% | 1,227,367 | 42.6% | 15,071 | 9.4% | | 30,145 | 9.4% | 15,074 | 9.4% | 15,071 | 9.4% | |
| **Insurance** | | | | | | | | | | | | | | |
| Commercial | 1,475,456 | 48.5% | 1,416,166 | 49.1% | 59,290 | 36.9% | <0.001 | 118,587 | 37.1% | 59,297 | 37.1% | 59,290 | 37.1% | 1.00 |
| Dual | 53,474 | 1.8% | 39,414 | 1.4% | 14,060 | 8.7% | | 25,752 | 8.1% | 12,874 | 8.1% | 12,878 | 8.1% | |
| Medicaid | 1,121,606 | 36.9% | 1,099,957 | 38.2% | 21,649 | 13.5% | | 43,299 | 13.6% | 21,650 | 13.6% | 21,649 | 13.6% | |
| Medicare | 392,068 | 12.9% | 326,181 | 11.3% | 65,887 | 41.0% | | 131,770 | 41.3% | 65,883 | 41.3% | 65,887 | 41.3% | |
| **PCP Visit 2019** | | | | | | | | | | | | | | |
| No | 1,701,040 | 55.9% | 1,646,572 | 57.1% | 54,468 | 33.9% | <0.001 | 108,601 | 34.0% | 54,275 | 34.0% | 54,326 | 34.0% | 0.85 |
| Yes | 1,341,564 | 44.1% | 1,235,146 | 42.9% | 106,418 | 66.1% | | 210,807 | 66.0% | 105,429 | 66.0% | 105,378 | 66.0% | |

**Continuous Outcomes**

| | mean | SD | mean | SD | mean | SD | p-value | mean | SD | mean | SD | mean | SD | p-value |
| --- | --- | --- | --- | --- | --- | --- | --- | --- | --- | --- | --- | --- | --- | --- |
| CCI | 0.57 | 1.31 | 0.55 | 1.28 | 0.86 | 1.70 | <0.001 | 0.86 | 1.70 | 0.86 | 1.70 | 0.86 | 1.71 | 0.84 |

BP: bisphosphonate; CCI: Charlson Comorbidity Index; PCP: primary care physician; SD: standard deviation.

**Appendix 2—table 4.** Primary Analysis Cohort (Region=West), Patient Characteristics Pre/Post Match.

| | Region=West Unmatched | | | | | | | Region=West Matched | | | | | | |
| --- | --- | --- | --- | --- | --- | --- | --- | --- | --- | --- | --- | --- | --- | --- |
| | All | | BP Non-users | | BP Users | | p-value | All | | BP Non-users | | BP Users | | p-value |
| | N | % | N | % | N | % | | N | % | N | % | N | % | |
| All Patients | 1,243,637 | 100.0% | 1,148,167 | 92.3% | 95,470 | 7.7% | | 190,534 | 100.0% | 95,267 | 50.0% | 95,267 | 50.0% | |
| **Age** | | | | | | | | | | | | | | |
| ≤20 | 275,709 | 22.2% | 275,559 | 24.0% | 150 | 0.2% | <0.001 | 299 | 0.2% | 149 | 0.2% | 150 | 0.2% | 1.00 |
| 21-40 | 234,415 | 18.8% | 233,858 | 20.4% | 557 | 0.6% | | 1,115 | 0.6% | 558 | 0.6% | 557 | 0.6% | |
| 41-50 | 140,237 | 11.3% | 138,833 | 12.1% | 1,404 | 1.5% | | 2,806 | 1.5% | 1,402 | 1.5% | 1,404 | 1.5% | |
| 51-60 | 188,965 | 15.2% | 178,585 | 15.6% | 10,380 | 10.9% | | 20,761 | 10.9% | 10,381 | 10.9% | 10,380 | 10.9% | |
| 61-70 | 192,408 | 15.5% | 161,016 | 14.0% | 31,392 | 32.9% | | 62,798 | 33.0% | 31,406 | 33.0% | 31,392 | 33.0% | |
| 71-80 | 127,739 | 10.3% | 95,301 | 8.3% | 32,438 | 34.0% | | 64,596 | 33.9% | 32,293 | 33.9% | 32,303 | 33.9% | |
| ≥81 | 84,164 | 6.8% | 65,015 | 5.7% | 19,149 | 20.1% | | 38,159 | 20.0% | 19,078 | 20.0% | 19,081 | 20.0% | |
| **Gender** | | | | | | | | | | | | | | |
| Female | 732,027 | 58.9% | 647,354 | 56.4% | 84,673 | 88.7% | <0.001 | 168,933 | 88.7% | 84,463 | 88.7% | 84,470 | 88.7% | 0.96 |
| Male | 511,610 | 41.1% | 500,813 | 43.6% | 10,797 | 11.3% | | 21,601 | 11.3% | 10,804 | 11.3% | 10,797 | 11.3% | |
| **Insurance** | | | | | | | | | | | | | | |
| Commercial | 526,701 | 42.4% | 503,359 | 43.8% | 23,342 | 24.4% | <0.001 | 46,688 | 24.5% | 23,346 | 24.5% | 23,342 | 24.5% | 1.00 |
| Dual | 27,060 | 2.2% | 20,924 | 1.8% | 6,136 | 6.4% | | 11,859 | 6.2% | 5,925 | 6.2% | 5,934 | 6.2% | |
| Medicaid | 522,435 | 42.0% | 497,941 | 43.4% | 24,494 | 25.7% | | 48,990 | 25.7% | 24,496 | 25.7% | 24,494 | 25.7% | |
| Medicare | 167,441 | 13.5% | 125,943 | 11.0% | 41,498 | 43.5% | | 82,997 | 43.6% | 41,500 | 43.6% | 41,497 | 43.6% | |
| **PCP Visit 2019** | | | | | | | | | | | | | | |
| No | 658,955 | 53.0% | 628,131 | 54.7% | 30,824 | 32.3% | <0.001 | 61,643 | 32.4% | 30,819 | 32.4% | 30,824 | 32.4% | 0.98 |
| Yes | 584,682 | 47.0% | 520,036 | 45.3% | 64,646 | 67.7% | | 128,891 | 67.6% | 64,448 | 67.6% | 64,443 | 67.6% | |
| **Continuous Outcomes** | | | | | | | | | | | | | | |
| | mean | SD | mean | SD | mean | SD | p-value | mean | SD | mean | SD | mean | SD | p-value |
| CCI | 0.69 | 1.46 | 0.66 | 1.42 | 1.08 | 1.84 | <0.001 | 1.09 | 1.83 | 1.08 | 1.83 | 1.09 | 1.84 | 0.73 |

BP: bisphosphonate; CCI: Charlson Comorbidity Index; PCP: primary care physician; SD: standard deviation.

**Appendix 2—table 5.** Primary Analysis Cohort (Region=New York State), Patient Characteristics Pre/Post Match.

| | Region=New York State Unmatched | | | | | | | Region=New York State Matched | | | | | | |
| --- | --- | --- | --- | --- | --- | --- | --- | --- | --- | --- | --- | --- | --- | --- |
| | All | | BP Non-users | | BP Users | | | All | | BP Non-users | | BP Users | | |
| | N | % | N | % | N | % | p-value | N | % | N | % | N | % | p-value |
| All Patients | 968,296 | 100.0% | 918,261 | 94.8% | 50,035 | 5.2% | | 99,724 | 100.0% | 49,862 | 50.0% | 49,862 | 50.0% | |
| **Age** | | | | | | | | | | | | | | |
| ≤20 | 133,178 | 13.8% | 133,128 | 14.5% | 50 | 0.1% | <0.001 | 102 | 0.1% | 52 | 0.1% | 50 | 0.1% | 1.00 |
| 21-40 | 192,959 | 19.9% | 192,731 | 21.0% | 228 | 0.5% | | 453 | 0.5% | 225 | 0.5% | 228 | 0.5% | |
| 41-50 | 127,794 | 13.2% | 127,139 | 13.8% | 655 | 1.3% | | 1,311 | 1.3% | 656 | 1.3% | 655 | 1.3% | |
| 51-60 | 172,444 | 17.8% | 166,080 | 18.1% | 6,364 | 12.7% | | 12,732 | 12.8% | 6,368 | 12.8% | 6,364 | 12.8% | |
| 61-70 | 159,912 | 16.5% | 143,776 | 15.7% | 16,136 | 32.2% | | 32,265 | 32.4% | 16,129 | 32.3% | 16,136 | 32.4% | |
| 71-80 | 120,117 | 12.4% | 102,655 | 11.2% | 17,462 | 34.9% | | 34,693 | 34.8% | 17,352 | 34.8% | 17,341 | 34.8% | |
| ≥81 | 61,892 | 6.4% | 52,752 | 5.7% | 9,140 | 18.3% | | 18,168 | 18.2% | 9,080 | 18.2% | 9,088 | 18.2% | |
| **Gender** | | | | | | | | | | | | | | |
| Female | 573,610 | 59.2% | 528,152 | 57.5% | 45,458 | 90.9% | <0.001 | 90,567 | 90.8% | 45,282 | 90.8% | 45,285 | 90.8% | 0.97 |
| Male | 394,686 | 40.8% | 390,109 | 42.5% | 4,577 | 9.1% | | 9,157 | 9.2% | 4,580 | 9.2% | 4,577 | 9.2% | |
| **Insurance** | | | | | | | | | | | | | | |
| Commercial | 500,918 | 51.7% | 490,503 | 53.4% | 10,415 | 20.8% | <0.001 | 20,830 | 20.9% | 10,415 | 20.9% | 10,415 | 20.9% | 1.00 |
| Dual | 6,814 | 0.7% | 5,218 | 0.6% | 1,596 | 3.2% | | 3,154 | 3.2% | 1,581 | 3.2% | 1,573 | 3.2% | |
| Medicaid | 252,366 | 26.1% | 243,191 | 26.5% | 9,175 | 18.3% | | 18,044 | 18.1% | 9,019 | 18.1% | 9,025 | 18.1% | |
| Medicare | 208,198 | 21.5% | 179,349 | 19.5% | 28,849 | 57.7% | | 57,696 | 57.9% | 28,847 | 57.9% | 28,849 | 57.9% | |
| **PCP Visit 2019** | | | | | | | | | | | | | | |
| No | 521,282 | 53.8% | 502,609 | 54.7% | 18,673 | 37.3% | <0.001 | 37,253 | 37.4% | 18,616 | 37.3% | 18,637 | 37.4% | 0.89 |
| Yes | 447,014 | 46.2% | 415,652 | 45.3% | 31,362 | 62.7% | | 62,471 | 62.6% | 31,246 | 62.7% | 31,225 | 62.6% | |

**Continuous Outcomes**

| | mean | SD | mean | SD | mean | SD | p-value | mean | SD | mean | SD | mean | SD | p-value |
| --- | --- | --- | --- | --- | --- | --- | --- | --- | --- | --- | --- | --- | --- | --- |
| CCI | 0.65 | 1.39 | 0.63 | 1.37 | 0.95 | 1.68 | <0.001 | 0.95 | 1.68 | 0.95 | 1.67 | 0.95 | 1.68 | 0.93 |

BP: bisphosphonate; CCI: Charlson Comorbidity Index; PCP: primary care physician; SD: standard deviation.

**Appendix 2—table 6.** Unadjusted COVID-19-Related Outcomes Stratified by Age, Sex, & Age by Sex; Matched Primary Analysis Cohort, All-Regions Combined.

Primary Analysis Cohort, All Regions Matched

| | All | | SARS-CoV-2 Test | | | | | | COVID-19 Diagnosis | | | | | | COVID-19 Hospitalization | | | | | |
|---|---|---|---|---|---|---|---|---|---|---|---|---|---|---|---|---|---|---|---|---|
| | N | % | N | % | OR | LL | UL | p-value | N | % | OR | LL | UL | p-value | N | % | OR | LL | UL | p-value |
| All Patients | 900,732 | 100.0% | 28,137 | 3.1% | | | | | 16,289 | 1.8% | | | | | 3,710 | 0.4% | | | | |
| BP User | 450,366 | 50.0% | 5,189 | 1.2% | 0.22 | 0.21 | 0.22 | <0.001 | 3,024 | 0.7% | 0.22 | 0.21 | 0.23 | <0.001 | 715 | 0.2% | 0.24 | 0.22 | 0.26 | <0.001 |
| BP Non-user | 450,366 | 50.0% | 22,948 | 5.1% | | | | | 13,265 | 2.9% | | | | | 2,995 | 0.7% | | | | |
| **By Age** | | | | | | | | | | | | | | | | | | | | |
| Age ≤20 | 2,253 | 100.0% | 67 | 3.0% | | | | | 14 | 0.6% | | | | | 2 | 0.1% | | | | |
| BP User | 1,128 | 50.1% | 29 | 2.6% | 0.75 | 0.46 | 1.23 | 0.26 | 2 | 0.2% | 0.16 | 0.04 | 0.74 | 0.007 | 2 | 0.2% | NA | NA | NA | NA |
| BP Non-user | 1,125 | 49.9% | 38 | 3.4% | | | | | 12 | 1.1% | | | | | 0 | 0.0% | | | | |
| Age 21-40 | 6,195 | 100.0% | 335 | 5.4% | | | | | 115 | 1.9% | | | | | 13 | 0.2% | | | | |
| BP User | 3,091 | 49.9% | 58 | 1.9% | 0.20 | 0.15 | 0.26 | <0.001 | 15 | 0.5% | 0.15 | 0.08 | 0.25 | <0.001 | 4 | 0.1% | 0.45 | 0.14 | 1.45 | 0.27 |
| BP Non-user | 3,104 | 50.1% | 277 | 8.9% | | | | | 100 | 3.2% | | | | | 9 | 0.3% | | | | |
| Age 41-50 | 17,096 | 100.0% | 894 | 5.2% | | | | | 270 | 1.6% | | | | | 54 | 0.3% | | | | |
| BP User | 8,551 | 50.0% | 188 | 2.2% | 0.25 | 0.21 | 0.29 | <0.001 | 48 | 0.6% | 0.21 | 0.15 | 0.29 | <0.001 | 14 | 0.2% | 0.35 | 0.19 | 0.64 | <0.001 |
| BP Non-user | 8,545 | 50.0% | 706 | 8.3% | | | | | 222 | 2.6% | | | | | 40 | 0.5% | | | | |
| Age 51-60 | 131,445 | 100.0% | 5,765 | 4.4% | | | | | 2,371 | 1.8% | | | | | 397 | 0.3% | | | | |
| BP User | 65,721 | 50.0% | 1,104 | 1.7% | 0.22 | 0.21 | 0.24 | <0.001 | 456 | 0.7% | 0.23 | 0.21 | 0.26 | <0.001 | 83 | 0.1% | 0.26 | 0.21 | 0.34 | <0.001 |
| BP Non-user | 65,724 | 50.0% | 4,661 | 7.1% | | | | | 1,915 | 2.9% | | | | | 314 | 0.5% | | | | |
| Age 61-70 | 313,822 | 100.0% | 10,438 | 3.3% | | | | | 5,029 | 1.6% | | | | | 1,035 | 0.3% | | | | |
| BP User | 156,878 | 50.0% | 1,843 | 1.2% | 0.21 | 0.20 | 0.22 | <0.001 | 939 | 0.6% | 0.23 | 0.21 | 0.24 | <0.001 | 173 | 0.1% | 0.20 | 0.17 | 0.24 | <0.001 |
| BP Non-user | 156,944 | 50.0% | 8,595 | 5.5% | | | | | 4,090 | 2.6% | | | | | 862 | 0.5% | | | | |
| Age 71-80 | 280,803 | 100.0% | 7,179 | 2.6% | | | | | 4,827 | 1.7% | | | | | 1,212 | 0.4% | | | | |
| BP User | 140,437 | 50.0% | 1,309 | 0.9% | 0.22 | 0.20 | 0.23 | <0.001 | 877 | 0.6% | 0.22 | 0.20 | 0.23 | <0.001 | 234 | 0.2% | 0.24 | 0.21 | 0.27 | <0.001 |
| BP Non-user | 140,366 | 50.0% | 5,870 | 4.2% | | | | | 3,950 | 2.8% | | | | | 978 | 0.7% | | | | |
| Age ≥81 | 149,118 | 100.0% | 3,459 | 2.3% | | | | | 3,663 | 2.5% | | | | | 997 | 0.7% | | | | |
| BP User | 74,560 | 50.0% | 658 | 0.9% | 0.23 | 0.21 | 0.25 | <0.001 | 687 | 0.9% | 0.22 | 0.20 | 0.24 | <0.001 | 205 | 0.3% | 0.26 | 0.22 | 0.30 | <0.001 |
| BP Non-user | 74,558 | 50.0% | 2,801 | 3.8% | | | | | 2,976 | 4.0% | | | | | 792 | 1.1% | | | | |
| **Female Patients** | 811,497 | 100.0% | 24,936 | 3.1% | | | | | 14,367 | 1.8% | | | | | 3,127 | 0.4% | | | | |
| BP User | 405,751 | 50.0% | 4,519 | 1.1% | 0.21 | | 0.24 | <0.001 | 2,667 | 0.7% | 0.22 | 0.22 | 0.23 | <0.001 | 593 | 0.1% | 0.23 | | | <0.001 |

eLife Research article

Epidemiology and Global Health | Medicine

*Appendix 2—table 6 Continued*

**Primary Analysis Cohort, All Regions Matched**

| | All | | SARS-CoV-2 Test | | | | COVID-19 Diagnosis | | | | COVID-19 Hospitalization | | | |
|---|---|---|---|---|---|---|---|---|---|---|---|---|---|---|
| | N | % | N | % | OR (LL, UL) | p-value | N | % | OR (LL, UL) | p-value | N | % | OR (LL, UL) | p-value |
| **All Patients** | 900,732 | 100.0% | 28,137 | 3.1% | | | 16,289 | 1.8% | | | 3,710 | 0.4% | | |
| BP Non-user | 405,746 | 50.0% | 20,417 | 5.0% | 0.21 (0.21, 0.22) | <0.001 | 11,700 | 2.9% | 0.22 (0.21, 0.23) | <0.001 | 2,534 | 0.6% | 0.23 (0.21, 0.25) | <0.001 |
| **By Age** | | | | | | | | | | | | | | |
| Age ≤20 | 885 | 100.0% | 26 | 2.9% | | | 7 | 0.8% | | | 1 | 0.1% | | |
| BP User | 442 | 49.9% | 11 | 2.5% | 0.73 (0.33, 1.60) | 0.43 | 1 | 0.2% | 0.17 (0.02, 1.38) | 0.12 | 1 | 0.2% | NA (NA, NA) | NA |
| BP Non-user | 443 | 50.1% | 15 | 3.4% | | | 6 | 1.4% | | | 0 | 0.0% | | |
| Age 21-40 | 3,765 | 100.0% | 218 | 5.8% | | | 64 | 1.7% | | | 9 | 0.2% | | |
| BP User | 1,879 | 49.9% | 40 | 2.1% | 0.21 (0.15, 0.30) | <0.001 | 12 | 0.6% | 0.23 (0.12, 0.43) | <0.001 | 3 | 0.2% | 0.50 (0.13, 2.01) | 0.51 |
| BP Non-user | 1,886 | 50.1% | 178 | 9.4% | | | 52 | 2.8% | | | 6 | 0.3% | | |
| Age 41-50 | 13,542 | 100.0% | 730 | 5.4% | | | 206 | 1.5% | | | 37 | 0.3% | | |
| BP User | 6,774 | 50.0% | 157 | 2.3% | 0.26 (0.21, 0.31) | <0.001 | 43 | 0.6% | 0.26 (0.18, 0.36) | <0.001 | 11 | 0.2% | 0.42 (0.21, 0.85) | 0.01 |
| BP Non-user | 6,768 | 50.0% | 573 | 8.5% | | | 163 | 2.4% | | | 26 | 0.4% | | |
| Age 51-60 | 119,205 | 100.0% | 5,200 | 4.4% | | | 2,093 | 1.8% | | | 327 | 0.3% | | |
| BP User | 59,602 | 50.0% | 973 | 1.6% | 0.22 (0.20, 0.23) | <0.001 | 399 | 0.7% | 0.23 (0.21, 0.26) | <0.001 | 64 | 0.1% | 0.24 (0.18, 0.32) | <0.001 |
| BP Non-user | 59,603 | 50.0% | 4,227 | 7.1% | | | 1,694 | 2.8% | | | 263 | 0.4% | | |
| Age 61-70 | 290,276 | 100.0% | 9,474 | 3.3% | | | 4,506 | 1.6% | | | 885 | 0.3% | | |
| BP User | 145,131 | 50.0% | 1,639 | 1.1% | 0.20 (0.19, 0.21) | <0.001 | 851 | 0.6% | 0.23 (0.21, 0.25) | <0.001 | 144 | 0.1% | 0.19 (0.16, 0.23) | <0.001 |
| BP Non-user | 145,145 | 50.0% | 7,835 | 5.4% | | | 3,655 | 2.5% | | | 741 | 0.5% | | |
| Age 71-80 | 253,094 | 100.0% | 6,304 | 2.5% | | | 4,254 | 1.7% | | | 1,026 | 0.4% | | |
| BP User | 126,559 | 50.0% | 1,140 | 0.9% | 0.21 (0.20, 0.23) | <0.001 | 769 | 0.6% | 0.22 (0.20, 0.23) | <0.001 | 193 | 0.2% | 0.23 (0.20, 0.27) | <0.001 |
| BP Non-user | 126,535 | 50.0% | 5,164 | 4.1% | | | 3,485 | 2.8% | | | 833 | 0.7% | | |
| Age ≥81 | 130,730 | 100.0% | 2,984 | 2.3% | | | 3,237 | 2.5% | | | 842 | 0.6% | | |
| BP User | 65,364 | 50.0% | 559 | 0.9% | 0.22 (0.20, 0.25) | <0.001 | 592 | 0.9% | 0.22 (0.20, 0.24) | <0.001 | 177 | 0.3% | 0.26 (0.22, 0.31) | <0.001 |
| BP Non-user | 65,366 | 50.0% | 2,425 | 3.7% | | | 2,645 | 4.0% | | | 665 | 1.0% | | |
| **Male Patients** | 89,235 | 100.0% | 3,201 | 3.6% | | | 1,922 | 2.2% | | | 583 | 0.7% | | |
| BP User | 44,615 | 50.0% | 670 | 1.5% | 0.25 (0.23, 0.28) | <0.001 | 357 | 0.8% | 0.22 (0.20, 0.25) | <0.001 | 122 | 0.3% | 0.26 (0.22, 0.32) | <0.001 |
| BP Non-user | 44,620 | 50.0% | 2,531 | 5.7% | | | 1,565 | 3.5% | | | 461 | 1.0% | | |
| **By Age** | | | | | | | | | | | | | | |

*Appendix 2—table 6 Continued*

Primary Analysis Cohort, All Regions Matched

| | All | | SARS-CoV-2 Test | | | | | | COVID-19 Diagnosis | | | | | | COVID-19 Hospitalization | | | | | |
|---|---|---|---|---|---|---|---|---|---|---|---|---|---|---|---|---|---|---|---|---|
| | | | | | | OR | | | | | | OR | | | | | | OR | | |
| | N | % | N | % | LL | UL | p-value | N | % | LL | UL | p-value | N | % | LL | UL | p-value |
| All Patients | 900,732 | 100.0% | 28,137 | 3.1% | | | | 16,289 | 1.8% | | | | 3,710 | 0.4% | | | |
| Age ≤20 | 1,368 | 100.0% | 41 | 3.0% | | | | 7 | 0.5% | | | | 1 | 0.1% | | | |
| BP User | 686 | 50.1% | 18 | 2.6% | 0.77 | | 0.42 | 1 | 0.1% | 0.16 | | 0.07 | 1 | 0.1% | NA | NA | NA |
| BP Non-user | 682 | 49.9% | 23 | 3.4% | 0.41 | 1.44 | | 6 | 0.9% | 0.02 | 1.37 | | 0 | 0.0% | NA | NA | |
| Age 21-40 | 2,430 | 100.0% | 117 | 4.8% | | | | 51 | 2.1% | | | | 4 | 0.2% | | | |
| BP User | 1,212 | 49.9% | 18 | 1.5% | 0.17 | | <0.001 | 3 | 0.2% | 0.06 | | <0.001 | 1 | 0.1% | 0.33 | 3.22 | 0.63 |
| BP Non-user | 1,218 | 50.1% | 99 | 8.1% | 0.10 | 0.28 | | 48 | 3.9% | 0.02 | 0.19 | | 3 | 0.2% | 0.03 | | |
| Age 41-50 | 3,554 | 100.0% | 164 | 4.6% | | | | 64 | 1.8% | | | | 17 | 0.5% | | | |
| BP User | 1,777 | 50.0% | 31 | 1.7% | 0.22 | | <0.001 | 5 | 0.3% | 0.08 | | <0.001 | 3 | 0.2% | 0.21 | 0.74 | 0.01 |
| BP Non-user | 1,777 | 50.0% | 133 | 7.5% | 0.15 | 0.33 | | 59 | 3.3% | 0.03 | 0.21 | | 14 | 0.8% | 0.06 | | |
| Age 51-60 | 12,240 | 100.0% | 565 | 4.6% | | | | 278 | 2.3% | | | | 70 | 0.6% | | | |
| BP User | 6,119 | 50.0% | 131 | 2.1% | 0.29 | | <0.001 | 57 | 0.9% | 0.25 | | <0.001 | 19 | 0.3% | 0.37 | 0.63 | <0.001 |
| BP Non-user | 6,121 | 50.0% | 434 | 7.1% | 0.24 | 0.35 | | 221 | 3.6% | 0.19 | 0.34 | | 51 | 0.8% | 0.22 | | |
| Age 61-70 | 23,546 | 100.0% | 964 | 4.1% | | | | 523 | 2.2% | | | | 150 | 0.6% | | | |
| BP User | 11,747 | 49.9% | 204 | 1.7% | 0.26 | | <0.001 | 88 | 0.7% | 0.20 | | <0.001 | 29 | 0.2% | 0.24 | 0.36 | <0.001 |
| BP Non-user | 11,799 | 50.1% | 760 | 6.4% | 0.22 | 0.30 | | 435 | 3.7% | 0.16 | 0.28 | | 121 | 1.0% | 0.16 | | |
| Age 71-80 | 27,709 | 100.0% | 875 | 3.2% | | | | 573 | 2.1% | | | | 186 | 0.7% | | | |
| BP User | 13,878 | 50.1% | 169 | 1.2% | 0.23 | | <0.001 | 108 | 0.8% | 0.23 | | <0.001 | 41 | 0.3% | 0.28 | 0.40 | <0.001 |
| BP Non-user | 13,831 | 49.9% | 706 | 5.1% | 0.19 | 0.27 | | 465 | 3.4% | 0.18 | 0.28 | | 145 | 1.0% | 0.20 | | |
| Age ≥81 | 18,388 | 100.0% | 475 | 2.6% | | | | 426 | 2.3% | | | | 155 | 0.8% | | | |
| BP User | 9,196 | 50.0% | 99 | 1.1% | 0.26 | | <0.001 | 95 | 1.0% | 0.28 | | <0.001 | 28 | 0.3% | 0.22 | 0.33 | <0.001 |
| BP Non-user | 9,192 | 50.0% | 376 | 4.1% | 0.20 | 0.32 | | 331 | 3.6% | 0.22 | 0.35 | | 127 | 1.4% | 0.14 | | |

BP: bisphosphonate; LL: lower 95% confidence interval level; UL: upper 95% confidence interval level; NA: not applicable; OR: odds ratio; UL: upper 95% confidence interval level.

**Appendix 2—table 7.** Unadjusted COVID-19-Related Outcomes Stratified by Age, Sex, & Age by Sex; Matched Primary Analysis Cohort, Region=Northeast.

Region=Northeast Matched

| | All | | SARS-CoV-2 Test | | | | | | COVID-19 Diagnosis | | | | | | COVID-19 Hospitalization | | | | | |
|---|---|---|---|---|---|---|---|---|---|---|---|---|---|---|---|---|---|---|---|---|
| | N | % | N | % | OR | LL | UL | p-value | N | % | OR | LL | UL | p-value | N | % | OR | LL | UL | p-value |
| **All Patients** | 238,988 | 100.0% | 8,831 | 3.7% | | | | | 7,820 | 3.3% | | | | | 1,505 | 0.6% | | | | |
| BP User | 119,494 | 50.0% | 1,684 | 1.4% | 0.22 | 0.21 | 0.24 | <0.001 | 1,578 | 1.3% | 0.24 | 0.23 | 0.26 | <0.001 | 314 | 0.3% | 0.26 | 0.23 | 0.30 | <0.001 |
| BP Non-user | 119,494 | 50.0% | 7,147 | 6.0% | | | | | 6,242 | 5.2% | | | | | 1,191 | 1.0% | | | | |
| **By Age** | | | | | | | | | | | | | | | | | | | | |
| Age ≤20 | 474 | 100.0% | 14 | 3.0% | | | | | 7 | 1.5% | | | | | 2 | 0.4% | | | | |
| BP User | 236 | 49.8% | 7 | 3.0% | 1.01 | 0.35 | 2.92 | 0.99 | 2 | 0.8% | 0.40 | 0.08 | 2.07 | 0.45 | 2 | 0.8% | NA | NA | NA | NA |
| BP Non-user | 238 | 50.2% | 7 | 2.9% | | | | | 5 | 2.1% | | | | | 0 | 0.0% | | | | |
| Age 21-40 | 1,528 | 100.0% | 93 | 6.1% | | | | | 55 | 3.6% | | | | | 5 | 0.3% | | | | |
| BP User | 764 | 50.0% | 14 | 1.8% | 0.16 | 0.09 | 0.29 | <0.001 | 7 | 0.9% | 0.14 | 0.06 | 0.31 | <0.001 | 1 | 0.1% | 0.25 | 0.03 | 2.23 | 0.37 |
| BP Non-user | 764 | 50.0% | 79 | 10.3% | | | | | 48 | 6.3% | | | | | 4 | 0.5% | | | | |
| Age 41-50 | 4,084 | 100.0% | 234 | 5.7% | | | | | 118 | 2.9% | | | | | 18 | 0.4% | | | | |
| BP User | 2,042 | 50.0% | 53 | 2.6% | 0.27 | 0.20 | 0.37 | <0.001 | 17 | 0.8% | 0.16 | 0.10 | 0.27 | <0.001 | 6 | 0.3% | 0.50 | 0.19 | 1.33 | 0.16 |
| BP Non-user | 2,042 | 50.0% | 181 | 8.9% | | | | | 101 | 4.9% | | | | | 12 | 0.6% | | | | |
| Age 51-60 | 36,020 | 100.0% | 1,863 | 5.2% | | | | | 1,190 | 3.3% | | | | | 160 | 0.4% | | | | |
| BP User | 18,010 | 50.0% | 353 | 2.0% | 0.22 | 0.19 | 0.25 | <0.001 | 237 | 1.3% | 0.24 | 0.21 | 0.28 | <0.001 | 38 | 0.2% | 0.31 | 0.22 | 0.45 | <0.001 |
| BP Non-user | 18,010 | 50.0% | 1,510 | 8.4% | | | | | 953 | 5.3% | | | | | 122 | 0.7% | | | | |
| Age 61-70 | 82,233 | 100.0% | 3,200 | 3.9% | | | | | 2,424 | 2.9% | | | | | 403 | 0.5% | | | | |
| BP User | 41,094 | 50.0% | 597 | 1.5% | 0.22 | 0.20 | 0.24 | <0.001 | 507 | 1.2% | 0.26 | 0.23 | 0.28 | <0.001 | 79 | 0.2% | 0.24 | 0.19 | 0.31 | <0.001 |
| BP Non-user | 41,139 | 50.0% | 2,603 | 6.3% | | | | | 1,917 | 4.7% | | | | | 324 | 0.8% | | | | |
| Age 71-80 | 74,831 | 100.0% | 2,266 | 3.0% | | | | | 2,306 | 3.1% | | | | | 493 | 0.7% | | | | |
| BP User | 37,438 | 50.0% | 442 | 1.2% | 0.23 | 0.21 | 0.26 | <0.001 | 475 | 1.3% | 0.25 | 0.23 | 0.28 | <0.001 | 99 | 0.3% | 0.25 | 0.20 | 0.31 | <0.001 |
| BP Non-user | 37,393 | 50.0% | 1,824 | 4.9% | | | | | 1,831 | 4.7% | | | | | 394 | 1.1% | | | | |
| Age ≥81 | 39,818 | 100.0% | 1,161 | 2.9% | | | | | 1,720 | 4.3% | | | | | 424 | 1.1% | | | | |
| BP User | 19,910 | 50.0% | 218 | 1.1% | 0.22 | 0.19 | 0.26 | <0.001 | 333 | 1.7% | 0.23 | 0.20 | 0.26 | <0.001 | 89 | 0.4% | 0.26 | 0.21 | 0.33 | <0.001 |
| BP Non-user | 19,908 | 50.0% | 943 | 4.7% | | | | | 1,387 | 7.0% | | | | | 335 | 1.7% | | | | |
| **Female Patients** | 216,273 | 100.0% | 7,897 | 3.7% | | | | | 6,941 | 3.2% | | | | | 1,263 | 0.6% | | | | |
| BP User | 108,136 | 50.0% | 1,483 | 1.4% | 0.22 | 0.21 | 0.23 | <0.001 | 1,392 | 1.3% | 0.24 | 0.23 | 0.26 | <0.001 | 255 | 0.2% | 0.25 | 0.22 | 0.29 | <0.001 |
| BP Non-user | 108,137 | 50.0% | 6,414 | 5.9% | | | | | 5,549 | 5.1% | | | | | 1,008 | 0.9% | | | | |

*Appendix 2—table 7 Continued*

Region=Northeast Matched

| | All N | All % | SARS N | SARS % | SARS OR | LL | UL | p-value | Dx N | Dx % | Dx OR | LL | UL | p-value | Hosp N | Hosp % | Hosp OR | LL | UL | p-value |
|---|---|---|---|---|---|---|---|---|---|---|---|---|---|---|---|---|---|---|---|---|
| | | | SARS-CoV-2 Test | | | | | | COVID-19 Diagnosis | | | | | | COVID-19 Hospitalization | | | | | |
| All Patients | 238,988 | 100.0% | 8,831 | 3.7% | | | | | 7,820 | 3.3% | | | | | 1,505 | 0.6% | | | | |
| **By Age** | | | | | | | | | | | | | | | | | | | | |
| Age ≤20 | 180 | 100.0% | 4 | 2.2% | 1.00 | 0.14 | 7.26 | 1.00 | 3 | 1.7% | 0.49 | 0.04 | 5.55 | 1.00 | 1 | 0.6% | NA | NA | NA | NA |
| BP User | 90 | 50.0% | 2 | 2.2% | | | | | 1 | 1.1% | | | | | 1 | 1.1% | | | | |
| BP Non-user | 90 | 50.0% | 2 | 2.2% | | | | | 2 | 2.2% | | | | | 0 | 0.0% | | | | |
| Age 21-40 | 864 | 100.0% | 59 | 6.8% | 0.19 | 0.09 | 0.37 | <0.001 | 32 | 3.7% | 0.22 | 0.09 | 0.54 | <0.001 | 4 | 0.5% | 0.33 | 0.03 | 3.22 | 0.62 |
| BP User | 431 | 49.9% | 10 | 2.3% | | | | | 6 | 1.4% | | | | | 1 | 0.2% | | | | |
| BP Non-user | 433 | 50.1% | 49 | 11.3% | | | | | 26 | 6.0% | | | | | 3 | 0.7% | | | | |
| Age 41-50 | 3,176 | 100.0% | 176 | 5.5% | 0.28 | 0.19 | 0.40 | <0.001 | 87 | 2.7% | 0.20 | 0.11 | 0.35 | <0.001 | 13 | 0.4% | 0.62 | 0.20 | 1.91 | 0.40 |
| BP User | 1,588 | 50.0% | 40 | 2.5% | | | | | 15 | 0.9% | | | | | 5 | 0.3% | | | | |
| BP Non-user | 1,588 | 50.0% | 136 | 8.6% | | | | | 72 | 4.5% | | | | | 8 | 0.5% | | | | |
| Age 51-60 | 32,612 | 100.0% | 1,690 | 5.2% | 0.21 | 0.18 | 0.24 | <0.001 | 1,048 | 3.2% | 0.24 | 0.20 | 0.27 | <0.001 | 125 | 0.4% | 0.33 | 0.22 | 0.49 | <0.001 |
| BP User | 16,306 | 50.0% | 310 | 1.9% | | | | | 206 | 1.3% | | | | | 31 | 0.2% | | | | |
| BP Non-user | 16,306 | 50.0% | 1,380 | 8.5% | | | | | 842 | 5.2% | | | | | 94 | 0.6% | | | | |
| Age 61-70 | 76,403 | 100.0% | 2,933 | 3.8% | 0.21 | 0.19 | 0.23 | <0.001 | 2,181 | 2.9% | 0.26 | 0.23 | 0.28 | <0.001 | 343 | 0.4% | 0.22 | 0.17 | 0.29 | <0.001 |
| BP User | 38,200 | 50.0% | 536 | 1.4% | | | | | 456 | 1.2% | | | | | 63 | 0.2% | | | | |
| BP Non-user | 38,203 | 50.0% | 2,397 | 6.3% | | | | | 1,725 | 4.5% | | | | | 280 | 0.7% | | | | |
| Age 71-80 | 67,857 | 100.0% | 2,021 | 3.0% | 0.23 | 0.21 | 0.26 | <0.001 | 2,063 | 3.0% | 0.24 | 0.22 | 0.27 | <0.001 | 416 | 0.6% | 0.23 | 0.18 | 0.29 | <0.001 |
| BP User | 33,930 | 50.0% | 393 | 1.2% | | | | | 413 | 1.2% | | | | | 77 | 0.2% | | | | |
| BP Non-user | 33,927 | 50.0% | 1,628 | 4.8% | | | | | 1,650 | 4.9% | | | | | 339 | 1.0% | | | | |
| Age ≥81 | 35,181 | 100.0% | 1,014 | 2.9% | 0.23 | 0.19 | 0.26 | <0.001 | 1,527 | 4.3% | 0.23 | 0.20 | 0.26 | <0.001 | 361 | 1.0% | 0.27 | 0.21 | 0.34 | <0.001 |
| BP User | 17,591 | 50.0% | 192 | 1.1% | | | | | 295 | 1.7% | | | | | 77 | 0.4% | | | | |
| BP Non-user | 17,590 | 50.0% | 822 | 4.7% | | | | | 1,232 | 7.0% | | | | | 284 | 1.6% | | | | |
| **Male Patients** | 22,715 | 100.0% | 934 | 4.1% | 0.26 | 0.22 | 0.31 | <0.001 | 879 | 3.9% | 0.26 | 0.22 | 0.30 | <0.001 | 242 | 1.1% | 0.32 | 0.24 | 0.43 | <0.001 |
| BP User | 11,358 | 50.0% | 201 | 1.8% | | | | | 186 | 1.6% | | | | | 59 | 0.5% | | | | |
| BP Non-user | 11,357 | 50.0% | 733 | 6.5% | | | | | 693 | 6.1% | | | | | 183 | 1.6% | | | | |
| **By Age** | | | | | | | | | | | | | | | | | | | | |
| Age ≤20 | 294 | 100.0% | 10 | 3.4% | | | | | 4 | 1.4% | | | | | 1 | 0.3% | | | | |

*Appendix 2—table 7 Continued*

**Region=Northeast Matched**

| | All | | SARS-CoV-2 Test | | OR | | | | COVID-19 Diagnosis | | OR | | | | COVID-19 Hospitalization | | OR | | | |
|---|---|---|---|---|---|---|---|---|---|---|---|---|---|---|---|---|---|---|---|---|
| | N | % | N | % | OR | LL | UL | p-value | N | % | OR | LL | UL | p-value | N | % | OR | LL | UL | p-value |
| All Patients | 238,988 | 100.0% | 8,831 | 3.7% | | | | | 7,820 | 3.3% | | | | | 1,505 | 0.6% | | | | |
| BP User | 146 | 49.7% | 5 | 3.4% | 1.01 | 0.29 | 3.58 | 0.98 | 1 | 0.7% | 0.33 | 0.03 | 3.24 | 0.62 | 1 | 0.7% | NA | NA | NA | NA |
| BP Non-user | 148 | 50.3% | 5 | 3.4% | | | | | 3 | 2.0% | | | | | 0 | 0.0% | | | | |
| Age 21-40 | 664 | 100.0% | 34 | 5.1% | | | | | 23 | 3.5% | | | | | 1 | 0.2% | | | | |
| BP User | 333 | 50.2% | 4 | 1.2% | 0.12 | 0.04 | 0.35 | <0.001 | 1 | 0.3% | 0.04 | 0.01 | 0.32 | <0.001 | 0 | 0.0% | NA | NA | NA | NA |
| BP Non-user | 331 | 49.8% | 30 | 9.1% | | | | | 22 | 6.6% | | | | | 1 | 0.3% | | | | |
| Age 41-50 | 908 | 100.0% | 58 | 6.4% | | | | | 31 | 3.4% | | | | | 5 | 0.6% | | | | |
| BP User | 454 | 50.0% | 13 | 2.9% | 0.27 | 0.14 | 0.50 | <0.001 | 2 | 0.4% | 0.06 | 0.02 | 0.27 | <0.001 | 1 | 0.2% | 0.25 | 0.03 | 2.23 | 0.37 |
| BP Non-user | 454 | 50.0% | 45 | 9.9% | | | | | 29 | 6.4% | | | | | 4 | 0.9% | | | | |
| Age 51-60 | 3,408 | 100.0% | 173 | 5.1% | | | | | 142 | 4.2% | | | | | 35 | 1.0% | | | | |
| BP User | 1,704 | 50.0% | 43 | 2.5% | 0.31 | 0.22 | 0.45 | <0.001 | 31 | 1.8% | 0.27 | 0.18 | 0.40 | <0.001 | 7 | 0.4% | 0.25 | 0.11 | 0.57 | <0.001 |
| BP Non-user | 1,704 | 50.0% | 130 | 7.6% | | | | | 111 | 6.5% | | | | | 28 | 1.6% | | | | |
| Age 61-70 | 5,830 | 100.0% | 267 | 4.6% | | | | | 243 | 4.2% | | | | | 60 | 1.0% | | | | |
| BP User | 2,894 | 49.6% | 61 | 2.1% | 0.29 | 0.21 | 0.38 | <0.001 | 51 | 1.8% | 0.26 | 0.19 | 0.35 | <0.001 | 16 | 0.6% | 0.37 | 0.21 | 0.65 | <0.001 |
| BP Non-user | 2,936 | 50.4% | 206 | 7.0% | | | | | 192 | 6.5% | | | | | 44 | 1.5% | | | | |
| Age 71-80 | 6,974 | 100.0% | 245 | 3.5% | | | | | 243 | 3.5% | | | | | 77 | 1.1% | | | | |
| BP User | 3,508 | 50.3% | 49 | 1.4% | 0.24 | 0.17 | 0.32 | <0.001 | 62 | 1.8% | 0.33 | 0.24 | 0.44 | <0.001 | 22 | 0.6% | 0.39 | 0.24 | 0.64 | <0.001 |
| BP Non-user | 3,466 | 49.7% | 196 | 5.7% | | | | | 181 | 5.2% | | | | | 55 | 1.6% | | | | |
| Age ≥81 | 4,637 | 100.0% | 147 | 3.2% | | | | | 193 | 4.2% | | | | | 63 | 1.4% | | | | |
| BP User | 2,319 | 50.0% | 26 | 1.1% | 0.21 | 0.13 | 0.32 | <0.001 | 38 | 1.6% | 0.23 | 0.16 | 0.33 | <0.001 | 12 | 0.5% | 0.23 | 0.12 | 0.43 | <0.001 |
| BP Non-user | 2,318 | 50.0% | 121 | 5.2% | | | | | 155 | 6.7% | | | | | 51 | 2.2% | | | | |

BP: bisphosphonate; LL: lower 95% confidence interval level; NA: not applicable; OR: odds ratio; UL: upper 95% confidence interval level.

**Appendix 2—table 8.** Unadjusted COVID-19-Related Outcomes Stratified by Age, Sex, & Age by Sex; Matched Primary Analysis Cohort, Region=Midwest.

Region=Midwest Matched

| | All | | SARS-CoV-2 Test | | OR | LL | UL | p-value | COVID-19 Diagnosis | | OR | LL | UL | p-value | COVID-19 Hospitalization | | OR | LL | UL | p-value |
|---|---|---|---|---|---|---|---|---|---|---|---|---|---|---|---|---|---|---|---|---|
| | N | % | N | % | | | | | N | % | | | | | N | % | | | | |
| **All Patients** | 151,802 | 100.0% | 4,451 | 2.9% | | | | | 2,099 | 1.4% | | | | | 636 | 0.4% | | | | |
| BP User | 75,901 | 50.0% | 868 | 1.1% | 0.23 | 0.22 | 0.25 | <0.001 | 383 | 0.5% | 0.22 | 0.20 | 0.25 | <0.001 | 121 | 0.2% | 0.23 | 0.19 | 0.29 | <0.001 |
| BP Non-user | 75,901 | 50.0% | 3,583 | 4.7% | | | | | 1,716 | 2.3% | | | | | 515 | 0.7% | | | | |
| **By Age** | | | | | | | | | | | | | | | | | | | | |
| Age ≤20 | 537 | 100.0% | 15 | 2.8% | | | | | 2 | 0.4% | | | | | 0 | 0.0% | | | | |
| BP User | 268 | 49.9% | 6 | 2.2% | | 0.66 | 1.89 | 0.44 | 0 | 0.0% | NA | NA | NA | NA | 0 | 0.0% | NA | NA | NA | NA |
| BP Non-user | 269 | 50.1% | 9 | 3.3% | | | | | 2 | 0.7% | | | | | 0 | 0.0% | | | | |
| Age 21-40 | 1,188 | 100.0% | 62 | 5.2% | | | | | 17 | 1.4% | | | | | 1 | 0.1% | | | | |
| BP User | 593 | 49.9% | 7 | 1.2% | 0.12 | 0.05 | 0.26 | <0.001 | 2 | 0.3% | 0.13 | 0.03 | 0.57 | 0.002 | 0 | 0.0% | NA | NA | NA | NA |
| BP Non-user | 595 | 50.1% | 55 | 9.2% | | | | | 15 | 2.5% | | | | | 1 | 0.2% | | | | |
| Age 41-50 | 3,367 | 100.0% | 184 | 5.5% | | | | | 46 | 1.4% | | | | | 16 | 0.5% | | | | |
| BP User | 1,684 | 50.0% | 36 | 2.1% | 0.23 | 0.16 | 0.33 | <0.001 | 10 | 0.6% | 0.27 | 0.14 | 0.55 | <0.001 | 2 | 0.1% | 0.14 | 0.03 | 0.62 | 0.002 |
| BP Non-user | 1,683 | 50.0% | 148 | 8.8% | | | | | 36 | 2.1% | | | | | 14 | 0.8% | | | | |
| Age 51-60 | 24,478 | 100.0% | 951 | 3.9% | | | | | 293 | 1.2% | | | | | 80 | 0.3% | | | | |
| BP User | 12,238 | 50.0% | 180 | 1.5% | 0.22 | 0.19 | 0.26 | <0.001 | 52 | 0.4% | 0.21 | 0.16 | 0.29 | <0.001 | 15 | 0.1% | 0.23 | 0.13 | 0.40 | <0.001 |
| BP Non-user | 12,240 | 50.0% | 771 | 6.3% | | | | | 241 | 2.0% | | | | | 65 | 0.5% | | | | |
| Age 61-70 | 56,991 | 100.0% | 1,764 | 3.1% | | | | | 671 | 1.2% | | | | | 189 | 0.3% | | | | |
| BP User | 28,496 | 50.0% | 322 | 1.1% | 0.21 | 0.19 | 0.24 | <0.001 | 123 | 0.4% | 0.22 | 0.18 | 0.27 | <0.001 | 35 | 0.1% | 0.23 | 0.16 | 0.33 | <0.001 |
| BP Non-user | 28,495 | 50.0% | 1,442 | 5.1% | | | | | 548 | 1.9% | | | | | 154 | 0.5% | | | | |
| Age 71-80 | 42,153 | 100.0% | 1,009 | 2.4% | | | | | 577 | 1.4% | | | | | 200 | 0.5% | | | | |
| BP User | 21,078 | 50.0% | 209 | 1.0% | 0.25 | 0.22 | 0.30 | <0.001 | 95 | 0.5% | 0.19 | 0.16 | 0.24 | <0.001 | 37 | 0.2% | 0.23 | 0.16 | 0.32 | <0.001 |
| BP Non-user | 21,075 | 50.0% | 800 | 3.8% | | | | | 482 | 2.3% | | | | | 163 | 0.8% | | | | |
| Age ≥81 | 23,088 | 100.0% | 466 | 2.0% | | | | | 493 | 2.1% | | | | | 150 | 0.6% | | | | |
| BP User | 11,544 | 50.0% | 108 | 0.9% | 0.30 | 0.24 | 0.37 | <0.001 | 101 | 0.9% | 0.25 | 0.20 | 0.31 | <0.001 | 32 | 0.3% | 0.27 | 0.18 | 0.40 | <0.001 |
| BP Non-user | 11,544 | 50.0% | 358 | 3.1% | | | | | 392 | 3.4% | | | | | 118 | 1.0% | | | | |
| **Female Patients** | 137,028 | 100.0% | 3,945 | 2.9% | | | | | 1,828 | 1.3% | | | | | 543 | 0.4% | | | | |
| BP User | 68,512 | 50.0% | 762 | 1.1% | 0.23 | 0.21 | 0.25 | <0.001 | 333 | 0.5% | 0.22 | 0.19 | 0.25 | <0.001 | 103 | 0.2% | 0.23 | 0.19 | 0.29 | <0.001 |
| BP Non-user | 68,516 | 50.0% | 3,183 | 4.6% | | | | | 1,495 | 2.2% | | | | | 440 | 0.6% | | | | |

*Appendix 2—table 8 Continued*

**Region=Midwest Matched**

| | All | | SARS-CoV-2 Test | | OR | | | | COVID-19 Diagnosis | | OR | | | | COVID-19 Hospitalization | | OR | | | |
|---|---|---|---|---|---|---|---|---|---|---|---|---|---|---|---|---|---|---|---|---|
| | N | % | N | % | OR | LL | UL | p-value | N | % | OR | LL | UL | p-value | N | % | OR | LL | UL | p-value |
| All Patients | 151,802 | 100.0% | 4,451 | 2.9% | | | | | 2,099 | 1.4% | | | | | 636 | 0.4% | | | | |
| **By Age** | | | | | | | | | | | | | | | | | | | | |
| Age ≤20 | 226 | 100.0% | 7 | 3.1% | | | | | 1 | 0.4% | | | | | 0 | 0.0% | | | | |
| BP User | 113 | 50.0% | 3 | 2.7% | 0.74 | 0.16 | 3.40 | 1.00 | 0 | 0.0% | NA | NA | NA | NA | 0 | 0.0% | NA | NA | NA | NA |
| BP Non-user | 113 | 50.0% | 4 | 3.5% | | | | | 1 | 0.9% | | | | | 0 | 0.0% | | | | |
| Age 21-40 | 700 | 100.0% | 34 | 4.9% | | | | | 7 | 1.0% | | | | | 0 | 0.0% | | | | |
| BP User | 349 | 49.9% | 6 | 1.7% | 0.20 | 0.08 | 0.49 | <0.001 | 1 | 0.3% | 0.17 | 0.02 | 1.38 | 0.12 | 0 | 0.0% | NA | NA | NA | NA |
| BP Non-user | 351 | 50.1% | 28 | 8.0% | | | | | 6 | 1.7% | | | | | 0 | 0.0% | | | | |
| Age 41-50 | 2,639 | 100.0% | 157 | 5.9% | | | | | 32 | 1.2% | | | | | 10 | 0.4% | | | | |
| BP User | 1,319 | 50.0% | 31 | 2.4% | 0.23 | 0.15 | 0.34 | <0.001 | 8 | 0.6% | 0.33 | 0.15 | 0.74 | 0.005 | 1 | 0.1% | 0.11 | 0.01 | 0.87 | 0.02 |
| BP Non-user | 1,320 | 50.0% | 126 | 9.5% | | | | | 24 | 1.8% | | | | | 9 | 0.7% | | | | |
| Age 51-60 | 22,101 | 100.0% | 856 | 3.9% | | | | | 260 | 1.2% | | | | | 70 | 0.3% | | | | |
| BP User | 11,050 | 50.0% | 159 | 1.4% | 0.22 | 0.18 | 0.26 | <0.001 | 47 | 0.4% | 0.22 | 0.16 | 0.30 | <0.001 | 13 | 0.1% | 0.23 | 0.12 | 0.42 | <0.001 |
| BP Non-user | 11,051 | 50.0% | 697 | 6.3% | | | | | 213 | 1.9% | | | | | 57 | 0.5% | | | | |
| Age 61-70 | 52,520 | 100.0% | 1,594 | 3.0% | | | | | 591 | 1.1% | | | | | 165 | 0.3% | | | | |
| BP User | 26,260 | 50.0% | 286 | 1.1% | 0.21 | 0.18 | 0.24 | <0.001 | 107 | 0.4% | 0.22 | 0.18 | 0.27 | <0.001 | 29 | 0.1% | 0.21 | 0.14 | 0.32 | <0.001 |
| BP Non-user | 26,260 | 50.0% | 1,308 | 5.0% | | | | | 484 | 1.8% | | | | | 136 | 0.5% | | | | |
| Age 71-80 | 38,367 | 100.0% | 877 | 2.3% | | | | | 501 | 1.3% | | | | | 172 | 0.4% | | | | |
| BP User | 19,184 | 50.0% | 180 | 0.9% | 0.25 | 0.21 | 0.30 | <0.001 | 85 | 0.4% | 0.20 | 0.16 | 0.25 | <0.001 | 33 | 0.2% | 0.24 | 0.16 | 0.35 | <0.001 |
| BP Non-user | 19,183 | 50.0% | 697 | 3.6% | | | | | 416 | 2.2% | | | | | 139 | 0.7% | | | | |
| Age ≥81 | 20,475 | 100.0% | 420 | 2.1% | | | | | 436 | 2.1% | | | | | 126 | 0.6% | | | | |
| BP User | 10,237 | 50.0% | 97 | 0.9% | 0.29 | 0.23 | 0.37 | <0.001 | 85 | 0.8% | 0.24 | 0.19 | 0.30 | <0.001 | 27 | 0.3% | 0.27 | 0.18 | 0.41 | <0.001 |
| BP Non-user | 10,238 | 50.0% | 323 | 3.2% | | | | | 351 | 3.4% | | | | | 99 | 1.0% | | | | |
| **Male Patients** | **14,774** | **100.0%** | **506** | **3.4%** | | | | | **271** | **1.8%** | | | | | **93** | **0.6%** | | | | |
| BP User | 7,389 | 50.0% | 106 | 1.4% | 0.25 | 0.20 | 0.32 | <0.001 | 50 | 0.7% | 0.22 | 0.16 | 0.30 | <0.001 | 18 | 0.2% | 0.24 | 0.14 | 0.40 | <0.001 |
| BP Non-user | 7,385 | 50.0% | 400 | 5.4% | | | | | 221 | 3.0% | | | | | 75 | 1.0% | | | | |
| **By Age** | | | | | | | | | | | | | | | | | | | | |
| Age ≤20 | 311 | 100.0% | 8 | 2.6% | | | | | 1 | 0.3% | | | | | 0 | 0.0% | | | | |

*Appendix 2—table 8 Continued*

**Region=Midwest Matched**

| | All | | SARS-CoV-2 Test | | | | | COVID-19 Diagnosis | | | | | COVID-19 Hospitalization | | | | | |
|---|---|---|---|---|---|---|---|---|---|---|---|---|---|---|---|---|---|---|---|---|
| | N | % | N | % | OR | LL | UL | p-value | N | % | OR | LL | UL | p-value | N | % | OR | LL | UL | p-value |
| All Patients | 151,802 | 100.0% | 4,451 | 2.9% | | | | | 2,099 | 1.4% | | | | | 636 | 0.4% | | | | |
| BP User | 155 | 49.8% | 3 | 1.9% | 0.60 | 0.14 | 2.54 | 0.72 | 0 | 0.0% | NA | NA | NA | NA | 0 | 0.0% | NA | NA | NA | NA |
| BP Non-user | 156 | 50.2% | 5 | 3.2% | | | | | 1 | 0.6% | | | | | 0 | 0.0% | | | | |
| Age 21-40 | 488 | 100.0% | 28 | 5.7% | | | | | 10 | 2.0% | | | | | 1 | 0.2% | | | | |
| BP User | 244 | 50.0% | 1 | 0.4% | 0.03 | 0.00 | 0.25 | <0.001 | 1 | 0.4% | 0.11 | 0.01 | 0.85 | 0.02 | 0 | 0.0% | NA | NA | NA | NA |
| BP Non-user | 244 | 50.0% | 27 | 11.1% | | | | | 9 | 3.7% | | | | | 1 | 0.4% | | | | |
| Age 41-50 | 728 | 100.0% | 27 | 3.7% | | | | | 14 | 1.9% | | | | | 6 | 0.8% | | | | |
| BP User | 365 | 50.1% | 5 | 1.4% | 0.22 | 0.08 | 0.57 | <0.001 | 2 | 0.5% | 0.16 | 0.04 | 0.73 | 0.007 | 1 | 0.3% | 0.20 | 0.02 | 1.69 | 0.12 |
| BP Non-user | 363 | 49.9% | 22 | 6.1% | | | | | 12 | 3.3% | | | | | 5 | 1.4% | | | | |
| Age 51-60 | 2,377 | 100.0% | 95 | 4.0% | | | | | 33 | 1.4% | | | | | 10 | 0.4% | | | | |
| BP User | 1,188 | 50.0% | 21 | 1.8% | 0.27 | 0.17 | 0.44 | <0.001 | 5 | 0.4% | 0.18 | 0.07 | 0.46 | <0.001 | 2 | 0.2% | 0.25 | 0.05 | 1.17 | 0.11 |
| BP Non-user | 1,189 | 50.0% | 74 | 6.2% | | | | | 28 | 2.4% | | | | | 8 | 0.7% | | | | |
| Age 61-70 | 4,471 | 100.0% | 170 | 3.8% | | | | | 80 | 1.8% | | | | | 24 | 0.5% | | | | |
| BP User | 2,236 | 50.0% | 36 | 1.6% | 0.26 | 0.18 | 0.37 | <0.001 | 16 | 0.7% | 0.24 | 0.14 | 0.42 | <0.001 | 6 | 0.3% | 0.33 | 0.13 | 0.84 | 0.01 |
| BP Non-user | 2,235 | 50.0% | 134 | 6.0% | | | | | 64 | 2.9% | | | | | 18 | 0.8% | | | | |
| Age 71-80 | 3,786 | 100.0% | 132 | 3.5% | | | | | 76 | 2.0% | | | | | 28 | 0.7% | | | | |
| BP User | 1,894 | 50.0% | 29 | 1.5% | 0.27 | 0.18 | 0.41 | <0.001 | 10 | 0.5% | 0.15 | 0.08 | 0.29 | <0.001 | 4 | 0.2% | 0.16 | 0.06 | 0.48 | <0.001 |
| BP Non-user | 1,892 | 50.0% | 103 | 5.4% | | | | | 66 | 3.5% | | | | | 24 | 1.3% | | | | |
| Age ≥81 | 2,613 | 100.0% | 46 | 1.8% | | | | | 57 | 2.2% | | | | | 24 | 0.9% | | | | |
| BP User | 1,307 | 50.0% | 11 | 0.8% | 0.31 | 0.16 | 0.61 | <0.001 | 16 | 1.2% | 0.38 | 0.21 | 0.69 | <0.001 | 5 | 0.4% | 0.26 | 0.10 | 0.70 | 0.004 |
| BP Non-user | 1,306 | 50.0% | 35 | 2.7% | | | | | 41 | 3.1% | | | | | 19 | 1.5% | | | | |

BP: bisphosphonate; LL: lower 95% confidence interval level; UL: upper 95% confidence interval level; NA: not applicable; OR: odds ratio; UL: upper 95% confidence interval level.

**Appendix 2—table 9.** Unadjusted COVID-19-Related Outcomes Stratified by Age, Sex, & Age by Sex; Matched Primary Analysis Cohort, Region=South.

Region=South Matched

| | All | | SARS-CoV-2 Test | | | | | COVID-19 Diagnosis | | | | | COVID-19 Hospitalization | | | | |
|---|---|---|---|---|---|---|---|---|---|---|---|---|---|---|---|---|---|
| | N | % | N | % | OR LL | UL | p-value | N | % | OR LL | UL | p-value | N | % | OR LL | UL | p-value |
| All Patients | 319,408 | 100.0% | 8,418 | 2.6% | | | | 3,535 | 1.1% | | | | 849 | 0.3% | | | |
| BP User | 159,704 | 50.0% | 1,553 | 1.0% | 0.22 | 0.23 | <0.001 | 624 | 0.4% | 0.21 | 0.23 | <0.001 | 167 | 0.1% | 0.24 | 0.29 | <0.001 |
| BP Non-user | 159,704 | 50.0% | 6,865 | 4.3% | | | | 2,911 | 1.8% | | | | 682 | 0.4% | | | |
| **By Age** | | | | | | | | | | | | | | | | | |
| Age ≤20 | 943 | 100.0% | 29 | 3.1% | | | | 4 | 0.4% | | | | 0 | 0.0% | | | |
| BP User | 474 | 50.3% | 15 | 3.2% | 1.06 | 2.23 | 0.87 | 0 | 0.0% | NA | NA | NA | 0 | 0.0% | NA | NA | NA |
| BP Non-user | 469 | 49.7% | 14 | 3.0% | | | | 4 | 0.9% | | | | 0 | 0.0% | | | |
| Age 21-40 | 2,364 | 100.0% | 113 | 4.8% | | | | 25 | 1.1% | | | | 4 | 0.2% | | | |
| BP User | 1,177 | 49.8% | 20 | 1.7% | 0.20 | 0.33 | <0.001 | 4 | 0.3% | 0.06 | 0.55 | <0.001 | 2 | 0.2% | 1.01 | 7.17 | 1.00 |
| BP Non-user | 1,187 | 50.2% | 93 | 7.8% | | | | 21 | 1.8% | | | | 2 | 0.2% | | | |
| Age 41-50 | 6,839 | 100.0% | 329 | 4.8% | | | | 73 | 1.1% | | | | 10 | 0.1% | | | |
| BP User | 3,421 | 50.0% | 72 | 2.1% | 0.26 | 0.34 | <0.001 | 18 | 0.5% | 0.32 | 0.55 | <0.001 | 5 | 0.1% | 1.00 | 3.45 | 0.99 |
| BP Non-user | 3,418 | 50.0% | 257 | 7.5% | | | | 55 | 1.6% | | | | 5 | 0.1% | | | |
| Age 51-60 | 50,186 | 100.0% | 1,999 | 4.0% | | | | 584 | 1.2% | | | | 103 | 0.2% | | | |
| BP User | 25,093 | 50.0% | 393 | 1.6% | 0.23 | 0.26 | <0.001 | 114 | 0.5% | 0.24 | 0.29 | <0.001 | 23 | 0.1% | 0.29 | 0.46 | <0.001 |
| BP Non-user | 25,093 | 50.0% | 1,606 | 6.4% | | | | 470 | 1.9% | | | | 80 | 0.3% | | | |
| Age 61-70 | 111,800 | 100.0% | 3,246 | 2.9% | | | | 1,106 | 1.0% | | | | 247 | 0.2% | | | |
| BP User | 55,896 | 50.0% | 583 | 1.0% | 0.21 | 0.23 | <0.001 | 191 | 0.3% | 0.21 | 0.24 | <0.001 | 38 | 0.1% | 0.18 | 0.26 | <0.001 |
| BP Non-user | 55,904 | 50.0% | 2,663 | 4.8% | | | | 915 | 1.6% | | | | 209 | 0.4% | | | |
| Age 71-80 | 99,223 | 100.0% | 1,942 | 2.0% | | | | 1,029 | 1.0% | | | | 260 | 0.3% | | | |
| BP User | 49,618 | 50.0% | 322 | 0.6% | 0.19 | 0.22 | <0.001 | 170 | 0.3% | 0.20 | 0.23 | <0.001 | 55 | 0.1% | 0.27 | 0.36 | <0.001 |
| BP Non-user | 49,605 | 50.0% | 1,620 | 3.3% | | | | 859 | 1.7% | | | | 205 | 0.4% | | | |
| Age ≥81 | 48,053 | 100.0% | 760 | 1.6% | | | | 714 | 1.5% | | | | 225 | 0.5% | | | |
| BP User | 24,025 | 50.0% | 148 | 0.6% | 0.24 | 0.28 | <0.001 | 127 | 0.5% | 0.21 | 0.26 | <0.001 | 44 | 0.2% | 0.24 | 0.34 | <0.001 |
| BP Non-user | 24,028 | 50.0% | 612 | 2.5% | | | | 587 | 2.4% | | | | 181 | 0.8% | | | |
| **Female Patients** | **289,263** | **100.0%** | **7,519** | **2.6%** | | | | **3,159** | **1.1%** | | | | **745** | **0.3%** | | | |
| BP User | 144,633 | 50.0% | 1,365 | 0.9% | 0.21 | 0.23 | <0.001 | 562 | 0.4% | 0.21 | 0.23 | <0.001 | 143 | 0.1% | 0.24 | 0.28 | <0.001 |
| BP Non-user | 144,630 | 50.0% | 6,154 | 4.3% | | | | 2,597 | 1.8% | | | | 602 | 0.4% | | | |

*Appendix 2—table 9 Continued on next page*

*Appendix 2—table 9 Continued*

**Region=South Matched**

| | All | | SARS-CoV-2 Test | | | | | | COVID-19 Diagnosis | | | | | | COVID-19 Hospitalization | | | | | |
| --- | --- | --- | --- | --- | --- | --- | --- | --- | --- | --- | --- | --- | --- | --- | --- | --- | --- | --- | --- | --- |
| | N | % | N | % | OR | LL | UL | p-value | N | % | OR | LL | UL | p-value | N | % | OR | LL | UL | p-value |
| All Patients | 319,408 | 100.0% | 8,418 | 2.6% | | | | | 3,535 | 1.1% | | | | | 849 | 0.3% | | | | |
| **By Age** | | | | | | | | | | | | | | | | | | | | |
| Age ≤20 | 372 | 100.0% | 11 | 3.0% | | | | | 3 | 0.8% | | | | | 0 | 0.0% | | | | |
| BP User | 185 | 49.7% | 6 | 3.2% | 1.22 | | | 0.75 | 0 | 0.0% | NA | | | NA | 0 | 0.0% | NA | | | NA |
| BP Non-user | 187 | 50.3% | 5 | 2.7% | | 0.37 | 4.07 | | 3 | 1.6% | | NA | NA | | 0 | 0.0% | | NA | NA | |
| Age 21-40 | 1,543 | 100.0% | 81 | 5.2% | | | | | 16 | 1.0% | | | | | 3 | 0.2% | | | | |
| BP User | 770 | 49.9% | 14 | 1.8% | 0.20 | | | <0.001 | 4 | 0.5% | 0.33 | | | 0.08 | 2 | 0.3% | 2.01 | | | 0.62 |
| BP Non-user | 773 | 50.1% | 67 | 8.7% | | 0.11 | 0.35 | | 12 | 1.6% | | 0.11 | 1.03 | | 1 | 0.1% | | 0.18 | 22.22 | |
| Age 41-50 | 5,569 | 100.0% | 273 | 4.9% | | | | | 66 | 1.2% | | | | | 9 | 0.2% | | | | |
| BP User | 2,787 | 50.0% | 65 | 2.3% | 0.30 | | | <0.001 | 18 | 0.6% | 0.37 | | | <0.001 | 5 | 0.2% | 1.25 | | | 1.00 |
| BP Non-user | 2,782 | 50.0% | 208 | 7.5% | | 0.22 | 0.39 | | 48 | 1.7% | | 0.21 | 0.64 | | 4 | 0.1% | | 0.33 | 4.65 | |
| Age 51-60 | 46,012 | 100.0% | 1,819 | 4.0% | | | | | 521 | 1.1% | | | | | 89 | 0.2% | | | | |
| BP User | 23,007 | 50.0% | 358 | 1.6% | 0.23 | | | <0.001 | 100 | 0.4% | 0.23 | | | <0.001 | 16 | 0.1% | 0.22 | | | <0.001 |
| BP Non-user | 23,005 | 50.0% | 1,461 | 6.4% | | 0.21 | 0.26 | | 421 | 1.8% | | 0.19 | 0.29 | | 73 | 0.3% | | 0.13 | 0.38 | |
| Age 61-70 | 103,825 | 100.0% | 2,948 | 2.8% | | | | | 1,007 | 1.0% | | | | | 218 | 0.2% | | | | |
| BP User | 51,910 | 50.0% | 517 | 1.0% | 0.20 | | | <0.001 | 177 | 0.3% | 0.21 | | | <0.001 | 33 | 0.1% | 0.18 | | | <0.001 |
| BP Non-user | 51,915 | 50.0% | 2,431 | 4.7% | | 0.19 | 0.23 | | 830 | 1.6% | | 0.18 | 0.25 | | 185 | 0.4% | | 0.12 | 0.26 | |
| Age 71-80 | 89,474 | 100.0% | 1,729 | 1.9% | | | | | 915 | 1.0% | | | | | 230 | 0.3% | | | | |
| BP User | 44,742 | 50.0% | 283 | 0.6% | 0.19 | | | <0.001 | 153 | 0.3% | 0.20 | | | <0.001 | 47 | 0.1% | 0.26 | | | <0.001 |
| BP Non-user | 44,732 | 50.0% | 1,446 | 3.2% | | 0.17 | 0.22 | | 762 | 1.7% | | 0.17 | 0.24 | | 183 | 0.4% | | 0.19 | 0.35 | |
| Age ≥81 | 42,468 | 100.0% | 658 | 1.5% | | | | | 631 | 1.5% | | | | | 196 | 0.5% | | | | |
| BP User | 21,232 | 50.0% | 122 | 0.6% | 0.22 | | | <0.001 | 110 | 0.5% | 0.21 | | | <0.001 | 40 | 0.2% | 0.26 | | | <0.001 |
| BP Non-user | 21,236 | 50.0% | 536 | 2.5% | | 0.18 | 0.27 | | 521 | 2.5% | | 0.17 | 0.25 | | 156 | 0.7% | | 0.18 | 0.36 | |
| **Male Patients** | **30,145** | **100.0%** | **899** | **3.0%** | | | | | **376** | **1.2%** | | | | | **104** | **0.3%** | | | | |
| BP User | 15,071 | 50.0% | 188 | 1.2% | 0.26 | | | <0.001 | 62 | 0.4% | 0.19 | | | <0.001 | 24 | 0.2% | 0.30 | | | <0.001 |
| BP Non-user | 15,074 | 50.0% | 711 | 4.7% | | 0.22 | 0.30 | | 314 | 2.1% | | 0.15 | 0.26 | | 80 | 0.5% | | 0.19 | 0.47 | |
| **By Age** | | | | | | | | | | | | | | | | | | | | |
| Age ≤20 | 571 | 100.0% | 18 | 3.2% | | | | | 1 | 0.2% | | | | | 0 | 0.0% | | | | |

*Appendix 2—table 9 Continued on next page*

*Appendix 2—table 9 Continued*

**Region=South Matched**

| | All | | SARS-CoV-2 Test | | | | | | COVID-19 Diagnosis | | | | | | COVID-19 Hospitalization | | | | | |
|---|---|---|---|---|---|---|---|---|---|---|---|---|---|---|---|---|---|---|---|---|
| | | | | | OR | | | | | | OR | | | | | | OR | | | |
| | N | % | N | % | LL | UL | p-value | N | % | LL | UL | p-value | N | % | LL | UL | p-value | | | |
| All Patients | 319,408 | 100.0% | 8,418 | 2.6% | | | | 3,535 | 1.1% | | | | 849 | 0.3% | | | | | | |
| BP User | 289 | 50.6% | 9 | 3.1% | 0.98 | 0.38 | 2.49 | 0 | 0.0% | NA | NA | NA | 0 | 0.0% | NA | NA | NA | | | |
| BP Non-user | 282 | 49.4% | 9 | 3.2% | | | 0.96 | 1 | 0.4% | | | | 0 | 0.0% | | | | | | |
| Age 21-40 | 821 | 100.0% | 32 | 3.9% | | | | 9 | 1.1% | | | | 1 | 0.1% | | | | | | |
| BP User | 407 | 49.6% | 6 | 1.5% | 0.22 | 0.09 | 0.55 | 0 | 0.0% | NA | NA | NA | 0 | 0.0% | NA | NA | NA | | | |
| BP Non-user | 414 | 50.4% | 26 | 6.3% | | | <0.001 | 9 | 2.2% | | | | 1 | 0.2% | | | | | | |
| Age 41-50 | 1,270 | 100.0% | 56 | 4.4% | | | | 7 | 0.6% | | | | 1 | 0.1% | | | | | | |
| BP User | 634 | 49.9% | 7 | 1.1% | 0.13 | 0.06 | 0.30 | 0 | 0.0% | NA | NA | NA | 0 | 0.0% | NA | NA | NA | | | |
| BP Non-user | 636 | 50.1% | 49 | 7.7% | | | <0.001 | 7 | 1.1% | | | | 1 | 0.2% | | | | | | |
| Age 51-60 | 4,174 | 100.0% | 180 | 4.3% | | | | 63 | 1.5% | | | | 14 | 0.3% | | | | | | |
| BP User | 2,086 | 50.0% | 35 | 1.7% | 0.23 | 0.16 | 0.33 | 14 | 0.7% | 0.28 | 0.15 | 0.51 | 7 | 0.3% | 1.00 | 0.35 | 2.86 | | | |
| BP Non-user | 2,088 | 50.0% | 145 | 6.9% | | | <0.001 | 49 | 2.3% | | | <0.001 | 7 | 0.3% | | | 0.99 | | | |
| Age 61-70 | 7,975 | 100.0% | 298 | 3.7% | | | | 99 | 1.2% | | | | 29 | 0.4% | | | | | | |
| BP User | 3,986 | 50.0% | 66 | 1.7% | 0.27 | 0.21 | 0.36 | 14 | 0.4% | 0.16 | 0.09 | 0.29 | 5 | 0.1% | 0.21 | 0.08 | 0.54 | | | |
| BP Non-user | 3,989 | 50.0% | 232 | 5.8% | | | <0.001 | 85 | 2.1% | | | <0.001 | 24 | 0.6% | | | <0.001 | | | |
| Age 71-80 | 9,749 | 100.0% | 213 | 2.2% | | | | 114 | 1.2% | | | | 30 | 0.3% | | | | | | |
| BP User | 4,876 | 50.0% | 39 | 0.8% | 0.22 | 0.15 | 0.31 | 17 | 0.3% | 0.17 | 0.10 | 0.29 | 8 | 0.2% | 0.36 | 0.16 | 0.81 | | | |
| BP Non-user | 4,873 | 50.0% | 174 | 3.6% | | | <0.001 | 97 | 2.0% | | | <0.001 | 22 | 0.5% | | | 0.01 | | | |
| Age ≥81 | 5,585 | 100.0% | 102 | 1.8% | | | | 83 | 1.5% | | | | 29 | 0.5% | | | | | | |
| BP User | 2,793 | 50.0% | 26 | 0.9% | 0.34 | 0.21 | 0.53 | 17 | 0.6% | 0.25 | 0.15 | 0.43 | 4 | 0.1% | 0.16 | 0.06 | 0.46 | | | |
| BP Non-user | 2,792 | 50.0% | 76 | 2.7% | | | <0.001 | 66 | 2.4% | | | <0.001 | 25 | 0.9% | | | <0.001 | | | |

BP: bisphosphonate; LL: lower 95% confidence interval level; UL: upper 95% confidence interval level; NA: not applicable; OR: odds ratio; UL: upper 95% confidence interval level.

**Appendix 2—table 10.** Unadjusted COVID-19-Related Outcomes Stratified by Age, Sex, & Age by Sex; Matched Primary Analysis Cohort, Region=West.

| Region=West Matched | All | | SARS-CoV-2 Test | | OR | | | COVID-19 Diagnosis | | OR | | | COVID-19 Hospitalization | | OR | | |
|---|---|---|---|---|---|---|---|---|---|---|---|---|---|---|---|---|---|
| | N | % | N | % | LL | UL | p-value | N | % | LL | UL | p-value | N | % | LL | UL | p-value |
| All Patients | 190,534 | 100.0% | 6,437 | 3.4% | | | | 2,835 | 1.5% | | | | 720 | 0.4% | | | |
| BP User | 95,267 | 50.0% | 1,084 | 1.1% | 0.19 | 0.21 | <0.001 | 439 | 0.5% | 0.18 | 0.20 | <0.001 | 113 | 0.1% | 0.19 | 0.23 | <0.001 |
| BP Non-user | 95,267 | 50.0% | 5,353 | 5.6% | 0.18 | | | 2,396 | 2.5% | 0.16 | | | 607 | 0.6% | 0.15 | | |
| **By Age** | | | | | | | | | | | | | | | | | |
| Age ≤20 | 299 | 100.0% | 9 | 3.0% | | | | 1 | 0.3% | | | | 0 | 0.0% | | | |
| BP User | 150 | 50.2% | 1 | 0.7% | 0.12 | 0.96 | 0.02 | 0 | 0.0% | NA | NA | NA | 0 | 0.0% | NA | NA | NA |
| BP Non-user | 149 | 49.8% | 8 | 5.4% | 0.01 | | | 1 | 0.7% | NA | | | 0 | 0.0% | NA | | |
| Age 21-40 | 1,115 | 100.0% | 67 | 6.0% | | | | 18 | 1.6% | | | | 3 | 0.3% | | | |
| BP User | 557 | 50.0% | 17 | 3.1% | 0.32 | 0.56 | <0.001 | 2 | 0.4% | 0.12 | 0.53 | 0.001 | 1 | 0.2% | 0.50 | 5.53 | 1.00 |
| BP Non-user | 558 | 50.0% | 50 | 9.0% | 0.18 | | | 16 | 2.9% | 0.03 | | | 2 | 0.4% | 0.05 | | |
| Age 41-50 | 2,806 | 100.0% | 147 | 5.2% | | | | 33 | 1.2% | | | | 10 | 0.4% | | | |
| BP User | 1,404 | 50.0% | 27 | 1.9% | 0.21 | 0.32 | <0.001 | 3 | 0.2% | 0.10 | 0.32 | <0.001 | 1 | 0.1% | 0.11 | 0.87 | 0.01 |
| BP Non-user | 1,402 | 50.0% | 120 | 8.6% | 0.14 | | | 30 | 2.1% | 0.03 | | | 9 | 0.6% | 0.01 | | |
| Age 51-60 | 20,761 | 100.0% | 952 | 4.6% | | | | 304 | 1.5% | | | | 54 | 0.3% | | | |
| BP User | 10,380 | 50.0% | 178 | 1.7% | 0.22 | 0.26 | <0.001 | 53 | 0.5% | 0.21 | 0.28 | <0.001 | 7 | 0.1% | 0.15 | 0.33 | <0.001 |
| BP Non-user | 10,381 | 50.0% | 774 | 7.5% | 0.18 | | | 251 | 2.4% | 0.15 | | | 47 | 0.5% | 0.07 | | |
| Age 61-70 | 62,798 | 100.0% | 2,228 | 3.5% | | | | 828 | 1.3% | | | | 196 | 0.3% | | | |
| BP User | 31,392 | 50.0% | 341 | 1.1% | 0.17 | 0.19 | <0.001 | 118 | 0.4% | 0.16 | 0.20 | <0.001 | 21 | 0.1% | 0.12 | 0.19 | <0.001 |
| BP Non-user | 31,406 | 50.0% | 1,887 | 6.0% | 0.15 | | | 710 | 2.3% | 0.13 | | | 175 | 0.6% | 0.08 | | |
| Age 71-80 | 64,596 | 100.0% | 1,962 | 3.0% | | | | 915 | 1.4% | | | | 259 | 0.4% | | | |
| BP User | 32,303 | 50.0% | 336 | 1.0% | 0.20 | 0.22 | <0.001 | 137 | 0.4% | 0.17 | 0.21 | <0.001 | 43 | 0.1% | 0.20 | 0.27 | <0.001 |
| BP Non-user | 32,293 | 50.0% | 1,626 | 5.0% | 0.18 | | | 778 | 2.4% | 0.14 | | | 216 | 0.7% | 0.14 | | |
| Age ≥81 | 38,159 | 100.0% | 1,072 | 2.8% | | | | 736 | 1.9% | | | | 198 | 0.5% | | | |
| BP User | 19,081 | 50.0% | 184 | 1.0% | 0.20 | 0.23 | <0.001 | 126 | 0.7% | 0.20 | 0.24 | <0.001 | 40 | 0.2% | 0.25 | 0.36 | <0.001 |
| BP Non-user | 19,078 | 50.0% | 888 | 4.7% | 0.17 | | | 610 | 3.2% | 0.17 | | | 158 | 0.8% | 0.18 | | |
| **Female Patients** | **168,933** | **100.0%** | **5,575** | **3.3%** | | | | **2,439** | **1.4%** | | | | **576** | **0.3%** | | | |
| BP User | 84,470 | 50.0% | 909 | 1.1% | 0.19 | 0.20 | <0.001 | 380 | 0.4% | 0.18 | 0.20 | <0.001 | 92 | 0.1% | 0.19 | 0.24 | <0.001 |
| BP Non-user | 84,463 | 50.0% | 4,666 | 5.5% | 0.17 | | | 2,059 | 2.4% | 0.16 | | | 484 | 0.6% | 0.15 | | |

*Appendix 2—table 10 Continued on next page*

*Appendix 2—table 10 Continued*

*Appendix 2—table 12 Continued*

Region=West Matched

| | All | | SARS-CoV-2 Test | | | | | | COVID-19 Diagnosis | | | | | | COVID-19 Hospitalization | | | | | |
| --- | --- | --- | --- | --- | --- | --- | --- | --- | --- | --- | --- | --- | --- | --- | --- | --- | --- | --- | --- | --- |
| | N | % | N | % | OR | LL | UL | p-value | N | % | OR | LL | UL | p-value | N | % | OR | LL | UL | p-value |
| All Patients | 190,534 | 100.0% | 6,437 | 3.4% | | | | | 2,835 | 1.5% | | | | | 720 | 0.4% | | | | |
| **By Age** | | | | | | | | | | | | | | | | | | | | |
| Age ≤20 | 107 | 100.0% | 4 | 3.7% | | | | | 0 | 0.0% | | | | | 0 | 0.0% | | | | |
| BP User | 54 | 50.5% | 0 | 0.0% | NA | NA | NA | NA | 0 | 0.0% | NA | NA | NA | NA | 0 | 0.0% | NA | NA | NA | NA |
| BP Non-user | 53 | 49.5% | 4 | 7.5% | | | | | 0 | 0.0% | | | | | 0 | 0.0% | | | | |
| Age 21-40 | 658 | 100.0% | 44 | 6.7% | | | | | 9 | 1.4% | | | | | 2 | 0.3% | | | | |
| BP User | 329 | 50.0% | 10 | 3.0% | 0.27 | 0.13 | 0.56 | <0.001 | 1 | 0.3% | 0.12 | 0.02 | 0.98 | 0.04 | 0 | 0.0% | NA | NA | NA | NA |
| BP Non-user | 329 | 50.0% | 34 | 10.3% | | | | | 8 | 2.4% | | | | | 2 | 0.6% | | | | |
| Age 41-50 | 2,158 | 100.0% | 124 | 5.7% | | | | | 21 | 1.0% | | | | | 5 | 0.2% | | | | |
| BP User | 1,080 | 50.0% | 21 | 1.9% | 0.19 | 0.12 | 0.30 | <0.001 | 2 | 0.2% | 0.10 | 0.02 | 0.45 | <0.001 | 0 | 0.0% | NA | NA | NA | NA |
| BP Non-user | 1,078 | 50.0% | 103 | 9.6% | | | | | 19 | 1.8% | | | | | 5 | 0.5% | | | | |
| Age 51-60 | 18,480 | 100.0% | 835 | 4.5% | | | | | 264 | 1.4% | | | | | 43 | 0.2% | | | | |
| BP User | 9,239 | 50.0% | 146 | 1.6% | 0.20 | 0.17 | 0.24 | <0.001 | 46 | 0.5% | 0.21 | 0.15 | 0.29 | <0.001 | 4 | 0.0% | 0.10 | 0.04 | 0.29 | <0.001 |
| BP Non-user | 9,241 | 50.0% | 689 | 7.5% | | | | | 218 | 2.4% | | | | | 39 | 0.4% | | | | |
| Age 61-70 | 57,528 | 100.0% | 1,999 | 3.5% | | | | | 727 | 1.3% | | | | | 159 | 0.3% | | | | |
| BP User | 28,761 | 50.0% | 300 | 1.0% | 0.17 | 0.15 | 0.19 | <0.001 | 111 | 0.4% | 0.18 | 0.14 | 0.22 | <0.001 | 19 | 0.1% | 0.14 | 0.08 | 0.22 | <0.001 |
| BP Non-user | 28,767 | 50.0% | 1,699 | 5.9% | | | | | 616 | 2.1% | | | | | 140 | 0.5% | | | | |
| Age 71-80 | 57,396 | 100.0% | 1,677 | 2.9% | | | | | 775 | 1.4% | | | | | 208 | 0.4% | | | | |
| BP User | 28,703 | 50.0% | 284 | 1.0% | 0.20 | 0.17 | 0.22 | <0.001 | 118 | 0.4% | 0.18 | 0.14 | 0.21 | <0.001 | 36 | 0.1% | 0.21 | 0.15 | 0.30 | <0.001 |
| BP Non-user | 28,693 | 50.0% | 1,393 | 4.9% | | | | | 657 | 2.3% | | | | | 172 | 0.6% | | | | |
| Age ≥81 | 32,606 | 100.0% | 892 | 2.7% | | | | | 643 | 2.0% | | | | | 159 | 0.5% | | | | |
| BP User | 16,304 | 50.0% | 148 | 0.9% | 0.19 | 0.16 | 0.23 | <0.001 | 102 | 0.6% | 0.18 | 0.15 | 0.23 | <0.001 | 33 | 0.2% | 0.26 | 0.18 | 0.38 | <0.001 |
| BP Non-user | 16,302 | 50.0% | 744 | 4.6% | | | | | 541 | 3.3% | | | | | 126 | 0.8% | | | | |
| **Male Patients** | **21,601** | **100.0%** | **862** | **4.0%** | | | | | **396** | **1.8%** | | | | | **144** | **0.7%** | | | | |
| BP User | 10,797 | 50.0% | 175 | 1.6% | 0.24 | 0.21 | 0.29 | <0.001 | 59 | 0.5% | 0.17 | 0.13 | 0.23 | <0.001 | 21 | 0.2% | 0.17 | 0.11 | 0.27 | <0.001 |
| BP Non-user | 10,804 | 50.0% | 687 | 6.4% | | | | | 337 | 3.1% | | | | | 123 | 1.1% | | | | |
| **By Age** | | | | | | | | | | | | | | | | | | | | |
| Age ≤20 | 192 | 100.0% | 5 | 2.6% | | | | | 1 | 0.5% | | | | | 0 | 0.0% | | | | |

*Appendix 2—table 10 Continued on next page*

Appendix 2—table 10 Continued

Region=West Matched

| | All | | SARS-CoV-2 Test | | OR | | | | COVID-19 Diagnosis | | OR | | | | COVID-19 Hospitalization | | OR | | | |
|---|---|---|---|---|---|---|---|---|---|---|---|---|---|---|---|---|---|---|---|---|
| | N | % | N | % | OR | LL | UL | p-value | N | % | OR | LL | UL | p-value | N | % | OR | LL | UL | p-value |
| All Patients | 190,534 | 100.0% | 6,437 | 3.4% | | | | | 2,835 | 1.5% | | | | | 720 | 0.4% | | | | |
| BP User | 96 | 50.0% | 1 | 1.0% | 0.24 | | | 0.37 | 0 | 0.0% | NA | | | NA | 0 | 0.0% | NA | | | NA |
| BP Non-user | 96 | 50.0% | 4 | 4.2% | | 0.03 | 2.21 | | 1 | 1.0% | | NA | NA | | 0 | 0.0% | | NA | NA | |
| Age 21-40 | 457 | 100.0% | 23 | 5.0% | | | | | 9 | 2.0% | | | | | 1 | 0.2% | | | | |
| BP User | 228 | 49.9% | 7 | 3.1% | 0.42 | | | 0.06 | 1 | 0.4% | 0.12 | | | 0.04 | 1 | 0.4% | NA | | | NA |
| BP Non-user | 229 | 50.1% | 16 | 7.0% | | 0.17 | 1.05 | | 8 | 3.5% | | 0.02 | 0.98 | | 0 | 0.0% | | NA | NA | |
| Age 41-50 | 648 | 100.0% | 23 | 3.5% | | | | | 12 | 1.9% | | | | | 5 | 0.8% | | | | |
| BP User | 324 | 50.0% | 6 | 1.9% | 0.34 | | | 0.02 | 1 | 0.3% | 0.09 | | | 0.006 | 1 | 0.3% | 0.25 | | | 0.37 |
| BP Non-user | 324 | 50.0% | 17 | 5.2% | | 0.13 | 0.88 | | 11 | 3.4% | | 0.01 | 0.69 | | 4 | 1.2% | | 0.03 | 2.23 | |
| Age 51-60 | 2,281 | 100.0% | 117 | 5.1% | | | | | 40 | 1.8% | | | | | 11 | 0.5% | | | | |
| BP User | 1,141 | 50.0% | 32 | 2.8% | 0.36 | | | <0.001 | 7 | 0.6% | 0.21 | | | <0.001 | 3 | 0.3% | 0.37 | | | 0.15 |
| BP Non-user | 1,140 | 50.0% | 85 | 7.5% | | 0.24 | 0.54 | | 33 | 2.9% | | 0.09 | 0.47 | | 8 | 0.7% | | 0.10 | 1.41 | |
| Age 61-70 | 5,270 | 100.0% | 229 | 4.3% | | | | | 101 | 1.9% | | | | | 37 | 0.7% | | | | |
| BP User | 2,631 | 49.9% | 41 | 1.6% | 0.21 | | | <0.001 | 7 | 0.3% | 0.07 | | | <0.001 | 2 | 0.1% | 0.06 | | | <0.001 |
| BP Non-user | 2,639 | 50.1% | 188 | 7.1% | | 0.15 | 0.29 | | 94 | 3.6% | | 0.03 | 0.16 | | 35 | 1.3% | | 0.01 | 0.24 | |
| Age 71-80 | 7,200 | 100.0% | 285 | 4.0% | | | | | 140 | 1.9% | | | | | 51 | 0.7% | | | | |
| BP User | 3,600 | 50.0% | 52 | 1.4% | 0.21 | | | <0.001 | 19 | 0.5% | 0.15 | | | <0.001 | 7 | 0.2% | 0.16 | | | <0.001 |
| BP Non-user | 3,600 | 50.0% | 233 | 6.5% | | 0.16 | 0.29 | | 121 | 3.4% | | 0.09 | 0.25 | | 44 | 1.2% | | 0.07 | 0.35 | |
| Age ≥81 | 5,553 | 100.0% | 180 | 3.2% | | | | | 93 | 1.7% | | | | | 39 | 0.7% | | | | |
| BP User | 2,777 | 50.0% | 36 | 1.3% | 0.24 | | | <0.001 | 24 | 0.9% | 0.34 | | | <0.001 | 7 | 0.3% | 0.22 | | | <0.001 |
| BP Non-user | 2,776 | 50.0% | 144 | 5.2% | | 0.17 | 0.35 | | 69 | 2.5% | | 0.21 | 0.55 | | 32 | 1.2% | | 0.10 | 0.49 | |

BP: bisphosphonate; LL: lower 95% confidence interval level; NA: not applicable; OR: odds ratio; UL: upper 95% confidence interval level.

**Appendix 2—table 11.** Unadjusted COVID-19-Related Outcomes Stratified by Age, Sex, & Age by Sex; Matched Primary Analysis Cohort, Region=New York State.

Region=New York State Matched

| | All | | SARS-CoV-2 Test | | | | | | COVID-19 Diagnosis | | | | | | COVID-19 Hospitalization | | | | | |
|---|---|---|---|---|---|---|---|---|---|---|---|---|---|---|---|---|---|---|---|---|
| | | | | | OR | | | | | | OR | | | | | | OR | | | |
| | N | % | N | % | | LL | UL | p-value | N | % | | LL | UL | p-value | N | % | | LL | UL | p-value |
| All Patients | 99,724 | 100.0% | 3,598 | 3.6% | | | | | 3,607 | 3.6% | | | | | 622 | 0.6% | | | | |
| BP User | 49,862 | 50.0% | 772 | 1.5% | 0.26 | 0.26 | 0.28 | <0.001 | 811 | 1.6% | 0.28 | 0.28 | 0.30 | <0.001 | 136 | 0.3% | 0.28 | 0.28 | 0.34 | <0.001 |
| BP Non-user | 49,862 | 50.0% | 2,826 | 5.7% | | 0.24 | 0.30 | | 2,796 | 5.6% | | 0.26 | | | 486 | 1.0% | | 0.23 | | |
| **By Age** | | | | | | | | | | | | | | | | | | | | |
| Age ≤20 | 102 | 100.0% | 4 | 3.9% | | | | | 2 | 2.0% | | | | | 1 | 1.0% | | | | |
| BP User | 50 | 49.0% | 2 | 4.0% | 1.04 | 1.04 | 7.69 | 1.00 | 1 | 2.0% | 1.04 | 1.04 | 17.11 | 1.00 | 1 | 2.0% | NA | NA | NA | NA |
| BP Non-user | 52 | 51.0% | 2 | 3.8% | | 0.14 | | | 1 | 1.9% | | 0.06 | | | 0 | 0.0% | | NA | NA | |
| Age 21-40 | 453 | 100.0% | 21 | 4.6% | | | | | 15 | 3.3% | | | | | 1 | 0.2% | | | | |
| BP User | 228 | 50.3% | 3 | 1.3% | 0.15 | 0.15 | 0.53 | <0.001 | 2 | 0.9% | 0.14 | 0.14 | 0.65 | 0.004 | 0 | 0.0% | NA | NA | NA | NA |
| BP Non-user | 225 | 49.7% | 18 | 8.0% | | 0.04 | | | 13 | 5.8% | | 0.03 | | | 1 | 0.4% | | NA | NA | |
| Age 41-50 | 1,311 | 100.0% | 77 | 5.9% | | | | | 36 | 2.7% | | | | | 4 | 0.3% | | | | |
| BP User | 655 | 50.0% | 22 | 3.4% | 0.38 | 0.38 | 0.63 | <0.001 | 8 | 1.2% | 0.28 | 0.28 | 0.61 | <0.001 | 1 | 0.2% | 0.33 | 0.33 | 3.21 | 0.62 |
| BP Non-user | 656 | 50.0% | 55 | 8.4% | | 0.23 | | | 28 | 4.3% | | 0.13 | | | 3 | 0.5% | | 0.03 | | |
| Age 51-60 | 12,732 | 100.0% | 688 | 5.4% | | | | | 527 | 4.1% | | | | | 58 | 0.5% | | | | |
| BP User | 6,364 | 50.0% | 155 | 2.4% | 0.27 | 0.27 | 0.33 | <0.001 | 118 | 1.9% | 0.28 | 0.28 | 0.34 | <0.001 | 17 | 0.3% | 0.41 | 0.41 | 0.73 | 0.002 |
| BP Non-user | 6,368 | 50.0% | 533 | 8.4% | | 0.23 | | | 409 | 6.4% | | 0.22 | | | 41 | 0.6% | | 0.23 | | |
| Age 61-70 | 32,265 | 100.0% | 1,294 | 4.0% | | | | | 1,150 | 3.6% | | | | | 141 | 0.4% | | | | |
| BP User | 16,136 | 50.0% | 277 | 1.7% | 0.26 | 0.26 | 0.30 | <0.001 | 267 | 1.7% | 0.29 | 0.29 | 0.33 | <0.001 | 27 | 0.2% | 0.24 | 0.24 | 0.36 | <0.001 |
| BP Non-user | 16,129 | 50.0% | 1,017 | 6.3% | | 0.23 | | | 883 | 5.5% | | 0.25 | | | 114 | 0.7% | | 0.15 | | |
| Age 71-80 | 34,693 | 100.0% | 957 | 2.8% | | | | | 1,196 | 3.4% | | | | | 240 | 0.7% | | | | |
| BP User | 17,341 | 50.0% | 204 | 1.2% | 0.26 | 0.26 | 0.31 | <0.001 | 257 | 1.5% | 0.26 | 0.26 | 0.30 | <0.001 | 45 | 0.3% | 0.23 | 0.23 | 0.32 | <0.001 |
| BP Non-user | 17,352 | 50.0% | 753 | 4.3% | | 0.22 | | | 939 | 5.4% | | 0.23 | | | 195 | 1.1% | | 0.17 | | |
| Age ≥81 | 18,168 | 100.0% | 557 | 3.1% | | | | | 681 | 3.7% | | | | | 177 | 1.0% | | | | |
| BP User | 9,088 | 50.0% | 109 | 1.2% | 0.23 | 0.23 | 0.29 | <0.001 | 158 | 1.7% | 0.29 | 0.29 | 0.35 | <0.001 | 44 | 0.5% | 0.33 | 0.33 | 0.46 | <0.001 |
| BP Non-user | 9,080 | 50.0% | 448 | 4.9% | | 0.19 | | | 523 | 5.8% | | 0.24 | | | 133 | 1.5% | | 0.23 | | |
| **Female Patients** | **90,567** | **100.0%** | **3,255** | **3.6%** | | | | | **3,235** | **3.6%** | | | | | **537** | **0.6%** | | | | |
| BP User | 45,285 | 50.0% | 687 | 1.5% | 0.26 | 0.26 | 0.28 | <0.001 | 726 | 1.6% | 0.28 | 0.28 | 0.30 | <0.001 | 108 | 0.2% | 0.25 | 0.25 | 0.31 | <0.001 |
| BP Non-user | 45,282 | 50.0% | 2,568 | 5.7% | | 0.24 | | | 2,509 | 5.5% | | 0.26 | | | 429 | 0.9% | | 0.20 | | |

Appendix 2—table 11 Continued

**Region=New York State Matched**

| | All | | SARS-CoV-2 Test | | | | | | COVID-19 Diagnosis | | | | | | COVID-19 Hospitalization | | | | | |
|---|---|---|---|---|---|---|---|---|---|---|---|---|---|---|---|---|---|---|---|---|
| | N | % | N | % | OR | LL | UL | p-value | N | % | OR | LL | UL | p-value | N | % | OR | LL | UL | p-value |
| All Patients | 99,724 | 100.0% | 3,598 | 3.6% | | | | | 3,607 | 3.6% | | | | | 622 | 0.6% | | | | |
| **By Age** | | | | | | | | | | | | | | | | | | | | |
| Age ≤20 | 33 | 100.0% | 0 | 0.0% | | | | | 1 | 3.0% | | | | | 1 | 3.0% | | | | |
| BP User | 16 | 48.5% | 0 | 0.0% | NA | NA | NA | NA | 1 | 6.3% | NA | NA | NA | NA | 1 | 6.3% | NA | NA | NA | NA |
| BP Non-user | 17 | 51.5% | 0 | 0.0% | | | | | 0 | 0.0% | | | | | 0 | 0.0% | | | | |
| Age 21-40 | 261 | 100.0% | 16 | 6.1% | | | | | 8 | 3.1% | | | | | 1 | 0.4% | | | | |
| BP User | 132 | 50.6% | 2 | 1.5% | 0.13 | 0.03 | 0.57 | 0.002 | 2 | 1.5% | 0.32 | 0.06 | 1.59 | 0.17 | 1 | 0.8% | | | | |
| BP Non-user | 129 | 49.4% | 14 | 10.9% | | | | | 6 | 4.7% | | | | | 0 | 0.0% | | | | |
| Age 41-50 | 1,032 | 100.0% | 58 | 5.6% | | | | | 28 | 2.7% | | | | | 3 | 0.3% | | | | |
| BP User | 516 | 50.0% | 18 | 3.5% | 0.43 | 0.24 | 0.76 | 0.003 | 7 | 1.4% | 0.32 | 0.14 | 0.77 | 0.007 | 0 | 0.0% | | | | |
| BP Non-user | 516 | 50.0% | 40 | 7.8% | | | | | 21 | 4.1% | | | | | 3 | 0.6% | | | | |
| Age 51-60 | 11,699 | 100.0% | 637 | 5.4% | | | | | 482 | 4.1% | | | | | 47 | 0.4% | | | | |
| BP User | 5,849 | 50.0% | 138 | 2.4% | 0.26 | 0.21 | 0.31 | <0.001 | 110 | 1.9% | 0.28 | 0.23 | 0.35 | <0.001 | 14 | 0.2% | 0.42 | 0.23 | 0.79 | 0.006 |
| BP Non-user | 5,850 | 50.0% | 499 | 8.5% | | | | | 372 | 6.4% | | | | | 33 | 0.6% | | | | |
| Age 61-70 | 30,115 | 100.0% | 1,204 | 4.0% | | | | | 1,070 | 3.6% | | | | | 126 | 0.4% | | | | |
| BP User | 15,060 | 50.0% | 257 | 1.7% | 0.26 | 0.22 | 0.30 | <0.001 | 248 | 1.6% | 0.29 | 0.25 | 0.33 | <0.001 | 23 | 0.2% | 0.22 | 0.14 | 0.35 | <0.001 |
| BP Non-user | 15,055 | 50.0% | 947 | 6.3% | | | | | 822 | 5.5% | | | | | 103 | 0.7% | | | | |
| Age 71-80 | 31,385 | 100.0% | 858 | 2.7% | | | | | 1,052 | 3.4% | | | | | 208 | 0.7% | | | | |
| BP User | 15,688 | 50.0% | 176 | 1.1% | 0.25 | 0.21 | 0.30 | <0.001 | 221 | 1.4% | 0.26 | 0.22 | 0.30 | <0.001 | 33 | 0.2% | 0.19 | 0.13 | 0.27 | <0.001 |
| BP Non-user | 15,697 | 50.0% | 682 | 4.3% | | | | | 831 | 5.3% | | | | | 175 | 1.1% | | | | |
| Age ≥81 | 16,042 | 100.0% | 482 | 3.0% | | | | | 594 | 3.7% | | | | | 151 | 0.9% | | | | |
| BP User | 8,024 | 50.0% | 96 | 1.2% | 0.24 | 0.19 | 0.30 | <0.001 | 137 | 1.7% | 0.29 | 0.24 | 0.35 | <0.001 | 36 | 0.4% | 0.31 | 0.21 | 0.45 | <0.001 |
| BP Non-user | 8,018 | 50.0% | 386 | 4.8% | | | | | 457 | 5.7% | | | | | 115 | 1.4% | | | | |
| **Male Patients** | 9,157 | 100.0% | 343 | 3.7% | | | | | 372 | 4.1% | | | | | 85 | 0.9% | | | | |
| BP User | 4,577 | 50.0% | 85 | 1.9% | 0.32 | 0.25 | 0.41 | <0.001 | 85 | 1.9% | 0.28 | 0.22 | 0.36 | <0.001 | 28 | 0.6% | 0.49 | 0.31 | 0.77 | 0.002 |
| BP Non-user | 4,580 | 50.0% | 258 | 5.6% | | | | | 287 | 6.3% | | | | | 57 | 1.2% | | | | |
| **By Age** | | | | | | | | | | | | | | | | | | | | |
| Age ≤20 | 69 | 100.0% | 4 | 5.8% | | | | | 1 | 1.4% | | | | | 0 | 0.0% | | | | |

Appendix 2—table 11 Continued

Region=New York State Matched

| | All | | SARS-CoV-2 Test | | | | | | COVID-19 Diagnosis | | | | | | COVID-19 Hospitalization | | | | | |
|---|---|---|---|---|---|---|---|---|---|---|---|---|---|---|---|---|---|---|---|---|
| | | | | | OR | | | | | | OR | | | | | | OR | | | |
| | N | % | N | % | | LL | UL | p-value | N | % | | LL | UL | p-value | N | % | | LL | UL | p-value |
| All Patients | 99,724 | 100.0% | 3,598 | 3.6% | | | | | 3,607 | 3.6% | | | | | 622 | 0.6% | | | | |
| BP User | 34 | 49.3% | 2 | 5.9% | 1.03 | 0.14 | 7.77 | 1.00 | 0 | 0.0% | NA | NA | NA | NA | 0 | 0.0% | NA | NA | NA | NA |
| BP Non-user | 35 | 50.7% | 2 | 5.7% | | | | | 1 | 2.9% | | | | | 0 | 0.0% | | | | |
| Age 21-40 | 192 | 100.0% | 5 | 2.6% | | | | | 7 | 3.6% | | | | | 0 | 0.0% | | | | |
| BP User | 96 | 50.0% | 1 | 1.0% | 0.24 | 0.03 | 2.21 | 0.37 | 0 | 0.0% | NA | NA | NA | NA | 0 | 0.0% | NA | NA | NA | NA |
| BP Non-user | 96 | 50.0% | 4 | 4.2% | | | | | 7 | 7.3% | | | | | 0 | 0.0% | | | | |
| Age 41-50 | 279 | 100.0% | 19 | 6.8% | | | | | 8 | 2.9% | | | | | 1 | 0.4% | | | | |
| BP User | 139 | 49.8% | 4 | 2.9% | 0.25 | 0.08 | 0.76 | 0.02 | 1 | 0.7% | 0.14 | 0.02 | 1.13 | 0.07 | 1 | 0.7% | NA | NA | NA | NA |
| BP Non-user | 140 | 50.2% | 15 | 10.7% | | | | | 7 | 5.0% | | | | | 0 | 0.0% | | | | |
| Age 51-60 | 1,033 | 100.0% | 51 | 4.9% | | | | | 45 | 4.4% | | | | | 11 | 1.1% | | | | |
| BP User | 515 | 49.9% | 17 | 3.3% | 0.49 | 0.27 | 0.88 | 0.02 | 8 | 1.6% | 0.21 | 0.09 | 0.44 | <0.001 | 3 | 0.6% | 0.37 | 0.10 | 1.42 | 0.22 |
| BP Non-user | 518 | 50.1% | 34 | 6.6% | | | | | 37 | 7.1% | | | | | 8 | 1.5% | | | | |
| Age 61-70 | 2,150 | 100.0% | 90 | 4.2% | | | | | 80 | 3.7% | | | | | 15 | 0.7% | | | | |
| BP User | 1,076 | 50.0% | 20 | 1.9% | 0.27 | 0.16 | 0.45 | <0.001 | 19 | 1.8% | 0.30 | 0.18 | 0.50 | <0.001 | 4 | 0.4% | 0.36 | 0.11 | 1.14 | 0.08 |
| BP Non-user | 1,074 | 50.0% | 70 | 6.5% | | | | | 61 | 5.7% | | | | | 11 | 1.0% | | | | |
| Age 71-80 | 3,308 | 100.0% | 99 | 3.0% | | | | | 144 | 4.4% | | | | | 32 | 1.0% | | | | |
| BP User | 1,653 | 50.0% | 28 | 1.7% | 0.38 | 0.25 | 0.60 | <0.001 | 36 | 2.2% | 0.32 | 0.22 | 0.47 | <0.001 | 12 | 0.7% | 0.60 | 0.29 | 1.23 | 0.16 |
| BP Non-user | 1,655 | 50.0% | 71 | 4.3% | | | | | 108 | 6.5% | | | | | 20 | 1.2% | | | | |
| Age ≥81 | 2,126 | 100.0% | 75 | 3.5% | | | | | 87 | 4.1% | | | | | 26 | 1.2% | | | | |
| BP User | 1,064 | 50.0% | 13 | 1.2% | 0.20 | 0.11 | 0.37 | <0.001 | 21 | 2.0% | 0.30 | 0.18 | 0.50 | <0.001 | 8 | 0.8% | 0.44 | 0.19 | 1.02 | 0.05 |
| BP Non-user | 1,062 | 50.0% | 62 | 5.8% | | | | | 66 | 6.2% | | | | | 18 | 1.7% | | | | |

BP: bisphosphonate; LL: lower 95% confidence interval level; NA: not applicable; OR: odds ratio; UL: upper 95% confidence interval level.

**Appendix 2—table 12.** Unadjusted COVID-19-Related Outcomes Stratified by Age, Sex, & Age by Sex; Matched Primary Analysis Cohort, Region=New York State.

| | | SARS-CoV-2 Test | | | | COVID-19 Diagnosis | | | | COVID-19 Hospitalization | | | |
|---|---|---|---|---|---|---|---|---|---|---|---|---|---|
| | | OR | LL | UL | p value | OR | LL | UL | p value | OR | LL | UL | p value |
| All | Unadjusted | 0.22 | 0.21 | 0.22 | <0.001 | 0.22 | 0.21 | 0.23 | <0.001 | 0.24 | 0.22 | 0.26 | <0.001 |
| | Adjusted | 0.22 | 0.21 | 0.23 | <0.001 | 0.23 | 0.22 | 0.24 | <0.001 | 0.26 | 0.24 | 0.29 | <0.001 |
| Northeast | Unadjusted | 0.22 | 0.21 | 0.24 | <0.001 | 0.24 | 0.23 | 0.26 | <0.001 | 0.26 | 0.23 | 0.30 | <0.001 |
| | Adjusted | 0.23 | 0.21 | 0.24 | <0.001 | 0.25 | 0.23 | 0.26 | <0.001 | 0.29 | 0.26 | 0.33 | <0.001 |
| Midwest | Unadjusted | 0.23 | 0.22 | 0.25 | <0.001 | 0.22 | 0.20 | 0.25 | <0.001 | 0.23 | 0.19 | 0.29 | <0.001 |
| | Adjusted | 0.24 | 0.22 | 0.26 | <0.001 | 0.24 | 0.22 | 0.27 | <0.001 | 0.26 | 0.21 | 0.32 | <0.001 |
| South | Unadjusted | 0.22 | 0.21 | 0.23 | <0.001 | 0.21 | 0.19 | 0.23 | <0.001 | 0.24 | 0.21 | 0.29 | <0.001 |
| | Adjusted | 0.22 | 0.21 | 0.23 | <0.001 | 0.22 | 0.20 | 0.24 | <0.001 | 0.26 | 0.23 | 0.30 | <0.001 |
| West | Unadjusted | 0.19 | 0.18 | 0.21 | <0.001 | 0.18 | 0.16 | 0.20 | <0.001 | 0.19 | 0.15 | 0.23 | <0.001 |
| | Adjusted | 0.20 | 0.18 | 0.21 | <0.001 | 0.19 | 0.17 | 0.21 | <0.001 | 0.20 | 0.16 | 0.25 | <0.001 |
| New York | Unadjusted | 0.26 | 0.24 | 0.28 | <0.001 | 0.28 | 0.26 | 0.30 | <0.001 | 0.28 | 0.23 | 0.34 | <0.001 |
| | Adjusted | 0.26 | 0.24 | 0.28 | <0.001 | 0.28 | 0.26 | 0.31 | <0.001 | 0.33 | 0.27 | 0.40 | <0.001 |

LL: lower 95% confidence interval level; OR: odds ratio; UL: upper 95% confidence interval level.

**Appendix 2—table 13.** Statin Use Sensitivity Analysis, Unadjusted/Adjusted Odds Ratio for COVID-19-Related Outcomes, Stratified by Region and New York State.

### Statin Uses versus Non-users

| | | SARS-CoV-2 Test | | | | COVID-19 Diagnosis | | | | COVID-19 Hospitalization | | | |
|---|---|---|---|---|---|---|---|---|---|---|---|---|---|
| | | OR | LL | UL | p value | OR | LL | UL | p value | OR | LL | UL | p value |
| All | Unadjusted | 0.90 | 0.89 | 0.91 | <0.001 | 0.91 | 0.90 | 0.92 | <0.001 | 1.12 | 1.09 | 1.15 | <0.001 |
| | Adjusted | 0.87 | 0.86 | 0.87 | <0.001 | 0.79 | 0.78 | 0.81 | <0.001 | 0.99 | 0.96 | 1.02 | 0.48 |
| Northeast | Unadjusted | 0.87 | 0.85 | 0.88 | <0.001 | 0.88 | 0.86 | 0.90 | <0.001 | 1.16 | 1.11 | 1.21 | <0.001 |
| | Adjusted | 0.85 | 0.84 | 0.87 | <0.001 | 0.77 | 0.75 | 0.78 | <0.001 | 1.03 | 0.98 | 1.07 | 0.22 |
| Midwest | Unadjusted | 0.97 | 0.95 | 0.99 | 0.02 | 1.10 | 1.07 | 1.14 | <0.001 | 1.27 | 1.19 | 1.36 | <0.001 |
| | Adjusted | 0.92 | 0.90 | 0.94 | <0.001 | 0.99 | 0.96 | 1.03 | 0.75 | 1.15 | 1.08 | 1.23 | <0.001 |
| South | Unadjusted | 0.90 | 0.88 | 0.91 | <0.001 | 0.90 | 0.88 | 0.93 | <0.001 | 1.00 | 0.95 | 1.06 | 0.90 |
| | Adjusted | 0.85 | 0.84 | 0.87 | <0.001 | 0.80 | 0.78 | 0.83 | <0.001 | 0.88 | 0.83 | 0.94 | <0.001 |
| West | Unadjusted | 0.88 | 0.86 | 0.90 | <0.001 | 0.83 | 0.80 | 0.86 | <0.001 | 1.02 | 0.95 | 1.10 | 0.58 |
| | Adjusted | 0.86 | 0.83 | 0.88 | <0.001 | 0.71 | 0.68 | 0.74 | <0.001 | 0.87 | 0.80 | 0.94 | <0.001 |
| New York | Unadjusted | 0.91 | 0.89 | 0.93 | <0.001 | 0.93 | 0.91 | 0.96 | <0.001 | 1.21 | 1.14 | 1.29 | <0.001 |
| | Adjusted | 0.92 | 0.90 | 0.95 | <0.001 | 0.79 | 0.77 | 0.82 | <0.001 | 1.05 | 0.98 | 1.13 | 0.15 |

### BP Users versus BP Non-users among Statin Users

| | | SARS-CoV-2 Test | | | | COVID-19 Diagnosis | | | | COVID-19 Hospitalization | | | |
|---|---|---|---|---|---|---|---|---|---|---|---|---|---|
| | | OR | LL | UL | p value | OR | LL | UL | p value | OR | LL | UL | p value |
| All | Unadjusted | 0.23 | 0.22 | 0.24 | <0.001 | 0.26 | 0.25 | 0.28 | <0.001 | 0.26 | 0.23 | 0.29 | <0.001 |
| | Adjusted | 0.23 | 0.22 | 0.24 | <0.001 | 0.27 | 0.25 | 0.29 | <0.001 | 0.28 | 0.25 | 0.32 | <0.001 |
| Northeast | Unadjusted | 0.25 | 0.23 | 0.27 | <0.001 | 0.29 | 0.27 | 0.31 | <0.001 | 0.28 | 0.24 | 0.34 | <0.001 |
| | Adjusted | 0.25 | 0.23 | 0.27 | <0.001 | 0.29 | 0.27 | 0.32 | <0.001 | 0.32 | 0.26 | 0.38 | <0.001 |
| Midwest | Unadjusted | 0.24 | 0.22 | 0.27 | <0.001 | 0.22 | 0.19 | 0.25 | <0.001 | 0.21 | 0.16 | 0.27 | <0.001 |
| | Adjusted | 0.25 | 0.23 | 0.29 | <0.001 | 0.23 | 0.22 | 0.25 | <0.001 | 0.22 | 0.17 | 0.30 | <0.001 |
| South | Unadjusted | 0.22 | 0.21 | 0.24 | <0.001 | 0.26 | 0.23 | 0.29 | <0.001 | 0.26 | 0.21 | 0.33 | <0.001 |
| | Adjusted | 0.22 | 0.20 | 0.24 | <0.001 | 0.27 | 0.24 | 0.31 | <0.001 | 0.28 | 0.22 | 0.36 | <0.001 |
| West | Unadjusted | 0.20 | 0.18 | 0.22 | <0.001 | 0.22 | 0.19 | 0.25 | <0.001 | 0.25 | 0.20 | 0.33 | <0.001 |
| | Adjusted | 0.20 | 0.18 | 0.22 | <0.001 | 0.23 | 0.20 | 0.27 | <0.001 | 0.28 | 0.21 | 0.36 | <0.001 |
| New York | Unadjusted | 0.27 | 0.24 | 0.30 | <0.001 | 0.31 | 0.28 | 0.35 | <0.001 | 0.30 | 0.23 | 0.39 | <0.001 |
| | Adjusted | 0.28 | 0.25 | 0.32 | <0.001 | 0.31 | 0.28 | 0.35 | <0.001 | 0.33 | 0.25 | 0.44 | <0.001 |

*Appendix 2—table 13 Continued on next page*

Appendix 2—table 13 Continued

| | | SARS-CoV-2 Test | | | | COVID-19 Diagnosis | | | | COVID-19 Hospitalization | | | |
|---|---|---|---|---|---|---|---|---|---|---|---|---|---|
| | | BP Users versus BP Non-users among Statin Non-users | | | | | | | | | | | |
| | | OR | LL | UL | p value | OR | LL | UL | p value | OR | LL | UL | p value |
| All | Unadjusted | 0.23 | 0.21 | 0.24 | <0.001 | 0.21 | 0.19 | 0.23 | <0.001 | 0.21 | 0.17 | 0.25 | <0.001 |
| | Adjusted | 0.24 | 0.22 | 0.25 | <0.001 | 0.23 | 0.21 | 0.25 | <0.001 | 0.25 | 0.21 | 0.30 | <0.001 |
| Northeast | Unadjusted | 0.25 | 0.22 | 0.27 | <0.001 | 0.22 | 0.20 | 0.25 | <0.001 | 0.24 | 0.19 | 0.31 | <0.001 |
| | Adjusted | 0.26 | 0.23 | 0.29 | <0.001 | 0.25 | 0.22 | 0.28 | <0.001 | 0.29 | 0.22 | 0.37 | <0.001 |
| Midwest | Unadjusted | 0.24 | 0.21 | 0.28 | <0.001 | 0.22 | 0.18 | 0.27 | <0.001 | 0.21 | 0.14 | 0.31 | <0.001 |
| | Adjusted | 0.24 | 0.20 | 0.28 | <0.001 | 0.25 | 0.20 | 0.32 | <0.001 | 0.26 | 0.17 | 0.39 | <0.001 |
| South | Unadjusted | 0.23 | 0.21 | 0.25 | <0.001 | 0.19 | 0.15 | 0.22 | <0.001 | 0.18 | 0.12 | 0.27 | <0.001 |
| | Adjusted | 0.24 | 0.21 | 0.27 | <0.001 | 0.21 | 0.17 | 0.25 | <0.001 | 0.22 | 0.15 | 0.33 | <0.001 |
| West | Unadjusted | 0.19 | 0.17 | 0.22 | <0.001 | 0.18 | 0.15 | 0.22 | <0.001 | 0.16 | 0.11 | 0.25 | <0.001 |
| | Adjusted | 0.20 | 0.17 | 0.23 | <0.001 | 0.19 | 0.18 | 0.21 | <0.001 | 0.18 | 0.11 | 0.29 | <0.001 |
| New York | Unadjusted | 0.26 | 0.23 | 0.30 | <0.001 | 0.26 | 0.22 | 0.30 | <0.001 | 0.27 | 0.19 | 0.39 | <0.001 |
| | Adjusted | 0.26 | 0.22 | 0.31 | <0.001 | 0.25 | 0.21 | 0.30 | <0.001 | 0.35 | 0.23 | 0.52 | <0.001 |

LL: lower 95% confidence interval level; OR: odds ratio; UL: upper 95% confidence interval level.

**Appendix 2—table 14.** Antihypertensive Use Sensitivity Analysis, Unadjusted/Adjusted Odds Ratio for COVID-19-Related Outcomes, Stratified by Region and New York State.

### Antihypertensive Users versus Non-users

| | | Odds of SARS-CoV-2 Test | | | | Odds of COVID-19 Diagnosis | | | | Odds of COVID-19 Hospitalization | | | |
|---|---|---|---|---|---|---|---|---|---|---|---|---|---|
| | | OR | LL | UL | p value | OR | LL | UL | p value | OR | LL | UL | p value |
| All | Unadjusted | 0.91 | 0.90 | 0.92 | <0.001 | 0.86 | 0.85 | 0.87 | <0.001 | 1.13 | 1.10 | 1.17 | <0.001 |
| | Adjusted | 0.87 | 0.86 | 0.88 | <0.001 | 0.75 | 0.74 | 0.76 | <0.001 | 0.98 | 0.95 | 1.00 | 0.10 |
| Northeast | Unadjusted | 0.86 | 0.84 | 0.87 | <0.001 | 0.83 | 0.82 | 0.85 | <0.001 | 1.20 | 1.15 | 1.25 | <0.001 |
| | Adjusted | 0.82 | 0.81 | 0.83 | <0.001 | 0.72 | 0.71 | 0.73 | <0.001 | 1.04 | 0.99 | 1.08 | 0.10 |
| Midwest | Unadjusted | 1.00 | 0.98 | 1.02 | 0.98 | 1.06 | 1.03 | 1.10 | <0.001 | 1.28 | 1.20 | 1.36 | <0.001 |
| | Adjusted | 0.94 | 0.91 | 0.96 | <0.001 | 0.94 | 0.90 | 0.97 | <0.001 | 1.11 | 1.04 | 1.19 | 0.002 |
| South | Unadjusted | 0.93 | 0.92 | 0.94 | <0.001 | 0.88 | 0.86 | 0.90 | <0.001 | 1.02 | 0.96 | 1.07 | 0.58 |
| | Adjusted | 0.88 | 0.87 | 0.89 | <0.001 | 0.78 | 0.76 | 0.80 | <0.001 | 0.89 | 0.84 | 0.94 | <0.001 |
| West | Unadjusted | 0.90 | 0.88 | 0.92 | <0.001 | 0.75 | 0.73 | 0.78 | <0.001 | 0.99 | 0.92 | 1.06 | 0.83 |
| | Adjusted | 0.87 | 0.85 | 0.89 | <0.001 | 0.65 | 0.62 | 0.67 | <0.001 | 0.84 | 0.78 | 0.90 | <0.001 |
| New York | Unadjusted | 0.92 | 0.90 | 0.94 | <0.001 | 0.90 | 0.87 | 0.92 | <0.001 | 1.23 | 1.15 | 1.31 | <0.001 |
| | Adjusted | 0.90 | 0.87 | 0.92 | <0.001 | 0.75 | 0.73 | 0.77 | <0.001 | 1.01 | 0.95 | 1.09 | 0.70 |

### BP Users versus BP Non-users among Antihypertensive Users

| | | Odds of SARS-CoV-2 Test | | | | Odds of COVID-19 Diagnosis | | | | Odds of COVID-19 Hospitalization | | | |
|---|---|---|---|---|---|---|---|---|---|---|---|---|---|
| | | OR | LL | UL | p value | OR | LL | UL | p value | OR | LL | UL | p value |
| All | Unadjusted | 0.23 | 0.22 | 0.24 | <0.001 | 0.26 | 0.25 | 0.28 | <0.001 | 0.26 | 0.23 | 0.29 | <0.001 |
| | Adjusted | 0.23 | 0.22 | 0.24 | <0.001 | 0.26 | 0.25 | 0.28 | <0.001 | 0.27 | 0.24 | 0.30 | <0.001 |
| Northeast | Unadjusted | 0.24 | 0.22 | 0.26 | <0.001 | 0.28 | 0.26 | 0.31 | <0.001 | 0.27 | 0.22 | 0.32 | <0.001 |
| | Adjusted | 0.23 | 0.21 | 0.26 | <0.001 | 0.28 | 0.26 | 0.31 | <0.001 | 0.29 | 0.24 | 0.34 | <0.001 |
| Midwest | Unadjusted | 0.26 | 0.23 | 0.29 | <0.001 | 0.27 | 0.23 | 0.31 | <0.001 | 0.27 | 0.21 | 0.35 | <0.001 |
| | Adjusted | 0.27 | 0.24 | 0.30 | <0.001 | 0.28 | 0.26 | 0.30 | <0.001 | 0.27 | 0.20 | 0.35 | <0.001 |
| South | Unadjusted | 0.23 | 0.21 | 0.25 | <0.001 | 0.24 | 0.22 | 0.28 | <0.001 | 0.26 | 0.20 | 0.32 | <0.001 |
| | Adjusted | 0.23 | 0.21 | 0.25 | <0.001 | 0.24 | 0.21 | 0.28 | <0.001 | 0.25 | 0.20 | 0.32 | <0.001 |
| West | Unadjusted | 0.20 | 0.18 | 0.22 | <0.001 | 0.21 | 0.18 | 0.25 | <0.001 | 0.24 | 0.18 | 0.31 | <0.001 |
| | Adjusted | 0.20 | 0.18 | 0.22 | <0.001 | 0.22 | 0.18 | 0.25 | <0.001 | 0.24 | 0.18 | 0.33 | <0.001 |
| New York | Unadjusted | 0.26 | 0.23 | 0.29 | <0.001 | 0.30 | 0.26 | 0.33 | <0.001 | 0.29 | 0.22 | 0.38 | <0.001 |
| | Adjusted | 0.25 | 0.22 | 0.29 | <0.001 | 0.30 | 0.26 | 0.34 | <0.001 | 0.33 | 0.24 | 0.44 | <0.001 |

*Appendix 2—table 14 Continued*

| | | Odds of SARS-CoV-2 Test | | | | Odds of COVID-19 Diagnosis | | | | Odds of COVID-19 Hospitalization | | | |
| --- | --- | --- | --- | --- | --- | --- | --- | --- | --- | --- | --- | --- | --- |
| | | BP Users versus BP Non-users among Antihypertensive Non-users | | | | | | | | | | | |
| | | OR | LL | UL | p value | OR | LL | UL | p value | OR | LL | UL | p value |
| All | Unadjusted | 0.21 | 0.20 | 0.22 | <0.001 | 0.20 | 0.18 | 0.22 | <0.001 | 0.21 | 0.17 | 0.25 | <0.001 |
| | Adjusted | 0.21 | 0.20 | 0.22 | <0.001 | 0.22 | 0.20 | 0.24 | <0.001 | 0.27 | 0.22 | 0.32 | <0.001 |
| Northeast | Unadjusted | 0.21 | 0.19 | 0.23 | <0.001 | 0.22 | 0.19 | 0.24 | <0.001 | 0.23 | 0.18 | 0.31 | <0.001 |
| | Adjusted | 0.22 | 0.20 | 0.25 | <0.001 | 0.25 | 0.22 | 0.28 | <0.001 | 0.30 | 0.22 | 0.40 | <0.001 |
| Midwest | Unadjusted | 0.22 | 0.19 | 0.25 | <0.001 | 0.16 | 0.12 | 0.20 | <0.001 | 0.20 | 0.13 | 0.31 | <0.001 |
| | Adjusted | 0.21 | 0.18 | 0.25 | <0.001 | 0.18 | 0.14 | 0.23 | <0.001 | 0.26 | 0.16 | 0.42 | <0.001 |
| South | Unadjusted | 0.20 | 0.18 | 0.22 | <0.001 | 0.19 | 0.16 | 0.22 | <0.001 | 0.22 | 0.15 | 0.32 | <0.001 |
| | Adjusted | 0.20 | 0.18 | 0.22 | <0.001 | 0.21 | 0.17 | 0.25 | <0.001 | 0.28 | 0.19 | 0.41 | <0.001 |
| West | Unadjusted | 0.19 | 0.17 | 0.22 | <0.001 | 0.18 | 0.15 | 0.22 | <0.001 | 0.15 | 0.09 | 0.24 | <0.001 |
| | Adjusted | 0.20 | 0.17 | 0.22 | <0.001 | 0.20 | 0.16 | 0.25 | <0.001 | 0.19 | 0.11 | 0.32 | <0.001 |
| New York | Unadjusted | 0.26 | 0.23 | 0.31 | <0.001 | 0.25 | 0.21 | 0.29 | <0.001 | 0.23 | 0.15 | 0.36 | <0.001 |
| | Adjusted | 0.27 | 0.23 | 0.32 | <0.001 | 0.26 | 0.22 | 0.31 | <0.001 | 0.26 | 0.16 | 0.43 | <0.001 |

LL: lower 95% confidence interval level; OR: odds ratio; UL: upper 95% confidence interval level.

**Appendix 2—table 15.** Antidiabetic Use Sensitivity Analysis, Unadjusted/Adjusted Odds Ratio for COVID-19-Related Outcomes, Stratified by Region and New York State.

**Antidiabetic Users versus Non-users**

| | | Odds of SARS-CoV-2 Test | | | | Odds of COVID-19 Diagnosis | | | | Odds of COVID-19 Hospitalization | | | |
|---|---|---|---|---|---|---|---|---|---|---|---|---|---|
| | | OR | LL | UL | p value | OR | LL | UL | p value | OR | LL | UL | p value |
| All | Unadjusted | 0.98 | 0.97 | 0.99 | 0.01 | 1.15 | 1.13 | 1.18 | <0.001 | 1.50 | 1.45 | 1.56 | <0.001 |
| | Adjusted | 0.92 | 0.90 | 0.93 | <0.001 | 0.88 | 0.86 | 0.90 | <0.001 | 1.13 | 1.08 | 1.18 | <0.001 |
| Northeast | Unadjusted | 1.00 | 0.98 | 1.02 | 0.92 | 1.11 | 1.09 | 1.14 | <0.001 | 1.55 | 1.47 | 1.64 | <0.001 |
| | Adjusted | 0.94 | 0.92 | 0.97 | <0.001 | 0.84 | 0.81 | 0.86 | <0.001 | 1.18 | 1.11 | 1.27 | <0.001 |
| Midwest | Unadjusted | 1.04 | 1.01 | 1.08 | 0.01 | 1.39 | 1.33 | 1.46 | <0.001 | 1.61 | 1.47 | 1.76 | <0.001 |
| | Adjusted | 0.95 | 0.91 | 0.99 | 0.01 | 1.11 | 1.04 | 1.17 | <0.001 | 1.30 | 1.17 | 1.44 | <0.001 |
| South | Unadjusted | 0.97 | 0.95 | 0.99 | 0.01 | 1.16 | 1.12 | 1.21 | <0.001 | 1.39 | 1.29 | 1.50 | <0.001 |
| | Adjusted | 0.90 | 0.88 | 0.93 | <0.001 | 0.91 | 0.87 | 0.95 | <0.001 | 1.04 | 0.95 | 1.14 | 0.40 |
| West | Unadjusted | 0.91 | 0.88 | 0.94 | <0.001 | 1.07 | 1.01 | 1.12 | 0.01 | 1.43 | 1.30 | 1.58 | <0.001 |
| | Adjusted | 0.86 | 0.82 | 0.89 | <0.001 | 0.80 | 0.75 | 0.85 | <0.001 | 0.97 | 0.86 | 1.09 | 0.60 |
| New York | Unadjusted | 1.06 | 1.03 | 1.10 | <0.001 | 1.15 | 1.11 | 1.19 | <0.001 | 1.59 | 1.46 | 1.72 | <0.001 |
| | Adjusted | 1.06 | 1.02 | 1.10 | 0.007 | 0.87 | 0.83 | 0.90 | <0.001 | 1.18 | 1.07 | 1.30 | 0.001 |

**BP Users versus BP Non-users among Antidiabetic Users**

| | | Odds of SARS-CoV-2 Test | | | | Odds of COVID-19 Diagnosis | | | | Odds of COVID-19 Hospitalization | | | |
|---|---|---|---|---|---|---|---|---|---|---|---|---|---|
| | | OR | LL | UL | p value | OR | LL | UL | p value | OR | LL | UL | p value |
| All | Unadjusted | 0.26 | 0.24 | 0.28 | <0.001 | 0.29 | 0.27 | 0.32 | <0.001 | 0.28 | 0.24 | 0.33 | <0.001 |
| | Adjusted | 0.26 | 0.24 | 0.28 | <0.001 | 0.29 | 0.27 | 0.32 | <0.001 | 0.29 | 0.25 | 0.34 | <0.001 |
| Northeast | Unadjusted | 0.28 | 0.24 | 0.32 | <0.001 | 0.32 | 0.28 | 0.35 | <0.001 | 0.29 | 0.23 | 0.36 | <0.001 |
| | Adjusted | 0.28 | 0.24 | 0.32 | <0.001 | 0.31 | 0.27 | 0.35 | <0.001 | 0.30 | 0.24 | 0.39 | <0.001 |
| Midwest | Unadjusted | 0.27 | 0.22 | 0.33 | <0.001 | 0.30 | 0.24 | 0.38 | <0.001 | 0.28 | 0.19 | 0.41 | <0.001 |
| | Adjusted | 0.27 | 0.22 | 0.34 | <0.001 | 0.32 | 0.26 | 0.41 | <0.001 | 0.29 | 0.19 | 0.42 | <0.001 |
| South | Unadjusted | 0.29 | 0.26 | 0.33 | <0.001 | 0.31 | 0.26 | 0.36 | <0.001 | 0.35 | 0.26 | 0.47 | <0.001 |
| | Adjusted | 0.30 | 0.26 | 0.34 | <0.001 | 0.30 | 0.25 | 0.36 | <0.001 | 0.36 | 0.26 | 0.48 | <0.001 |
| West | Unadjusted | 0.19 | 0.16 | 0.22 | <0.001 | 0.20 | 0.17 | 0.25 | <0.001 | 0.21 | 0.15 | 0.30 | <0.001 |
| | Adjusted | 0.19 | 0.16 | 0.23 | <0.001 | 0.21 | 0.17 | 0.26 | <0.001 | 0.22 | 0.15 | 0.31 | <0.001 |
| New York | Unadjusted | 0.33 | 0.27 | 0.40 | <0.001 | 0.34 | 0.29 | 0.39 | <0.001 | 0.35 | 0.26 | 0.49 | <0.001 |
| | Adjusted | 0.32 | 0.26 | 0.40 | <0.001 | 0.32 | 0.28 | 0.36 | <0.001 | 0.40 | 0.28 | 0.56 | <0.001 |

*Appendix 2—table 15 Continued on next page*

Appendix 2—table 15 Continued

| | | Odds of SARS-CoV-2 Test | | | | Odds of COVID-19 Diagnosis | | | | Odds of COVID-19 Hospitalization | | | |
|---|---|---|---|---|---|---|---|---|---|---|---|---|---|
| | | BP Users versus BP Non-users among Antidiabetic Non-users | | | | | | | | | | | |
| | | OR | LL | UL | p value | OR | LL | UL | p value | OR | LL | UL | p value |
| All | Unadjusted | 0.24 | 0.23 | 0.26 | <0.001 | 0.24 | 0.22 | 0.26 | <0.001 | 0.24 | 0.20 | 0.29 | <0.001 |
| | Adjusted | 0.25 | 0.23 | 0.27 | <0.001 | 0.25 | 0.23 | 0.28 | <0.001 | 0.27 | 0.22 | 0.33 | <0.001 |
| Northeast | Unadjusted | 0.24 | 0.22 | 0.28 | <0.001 | 0.26 | 0.22 | 0.29 | <0.001 | 0.25 | 0.19 | 0.34 | <0.001 |
| | Adjusted | 0.25 | 0.22 | 0.29 | <0.001 | 0.27 | 0.24 | 0.32 | <0.001 | 0.28 | 0.20 | 0.39 | <0.001 |
| Midwest | Unadjusted | 0.27 | 0.22 | 0.32 | <0.001 | 0.22 | 0.17 | 0.30 | <0.001 | 0.26 | 0.16 | 0.42 | <0.001 |
| | Adjusted | 0.28 | 0.24 | 0.31 | <0.001 | 0.23 | 0.17 | 0.31 | <0.001 | 0.26 | 0.16 | 0.45 | <0.001 |
| South | Unadjusted | 0.24 | 0.21 | 0.27 | <0.001 | 0.25 | 0.20 | 0.30 | <0.001 | 0.29 | 0.20 | 0.43 | <0.001 |
| | Adjusted | 0.24 | 0.21 | 0.27 | <0.001 | 0.24 | 0.21 | 0.28 | <0.001 | 0.33 | 0.22 | 0.49 | <0.001 |
| West | Unadjusted | 0.23 | 0.20 | 0.27 | <0.001 | 0.18 | 0.14 | 0.24 | <0.001 | 0.13 | 0.07 | 0.23 | <0.001 |
| | Adjusted | 0.23 | 0.20 | 0.28 | <0.001 | 0.20 | 0.15 | 0.26 | <0.001 | 0.15 | 0.08 | 0.28 | <0.001 |
| New York | Unadjusted | 0.30 | 0.25 | 0.37 | <0.001 | 0.30 | 0.25 | 0.36 | <0.001 | 0.22 | 0.14 | 0.36 | <0.001 |
| | Adjusted | 0.30 | 0.25 | 0.37 | <0.001 | 0.31 | 0.25 | 0.37 | <0.001 | 0.24 | 0.14 | 0.41 | <0.001 |

LL: lower 95% confidence interval level; OR: odds ratio; UL: upper 95% confidence interval level.

**Appendix 2—table 16.** Antidepressant Use Sensitivity Analysis, Unadjusted/Adjusted Odds Ratio for COVID-19-Related Outcomes, Stratified by Region and New York State.

**Antidepressant Users versus Non-users**

| | | Odds of SARS-CoV-2 Test | | | | Odds of COVID-19 Diagnosis | | | | Odds of COVID-19 Hospitalization | | | |
|---|---|---|---|---|---|---|---|---|---|---|---|---|---|
| | | OR | LL | UL | p value | OR | LL | UL | p value | OR | LL | UL | p value |
| All | Unadjusted | 1.04 | 1.03 | 1.05 | <0.001 | 0.71 | 0.70 | 0.72 | <0.001 | 0.81 | 0.78 | 0.83 | <0.001 |
| | Adjusted | 1.00 | 0.99 | 1.01 | 0.61 | 0.65 | 0.64 | 0.66 | <0.001 | 0.75 | 0.73 | 0.78 | <0.001 |
| Northeast | Unadjusted | 1.01 | 0.99 | 1.02 | 0.54 | 0.71 | 0.69 | 0.72 | <0.001 | 0.84 | 0.80 | 0.88 | <0.001 |
| | Adjusted | 0.97 | 0.95 | 0.99 | 0.001 | 0.65 | 0.63 | 0.66 | <0.001 | 0.77 | 0.73 | 0.82 | <0.001 |
| Midwest | Unadjusted | 1.10 | 1.08 | 1.12 | <0.001 | 0.75 | 0.72 | 0.78 | <0.001 | 0.84 | 0.78 | 0.90 | <0.001 |
| | Adjusted | 1.05 | 1.03 | 1.07 | <0.001 | 0.69 | 0.66 | 0.71 | <0.001 | 0.78 | 0.73 | 0.84 | <0.001 |
| South | Unadjusted | 1.04 | 1.02 | 1.05 | <0.001 | 0.68 | 0.66 | 0.70 | <0.001 | 0.74 | 0.70 | 0.79 | <0.001 |
| | Adjusted | 0.99 | 0.98 | 1.01 | 0.49 | 0.64 | 0.62 | 0.66 | <0.001 | 0.72 | 0.68 | 0.77 | <0.001 |
| West | Unadjusted | 1.04 | 1.02 | 1.06 | 0.00 | 0.70 | 0.67 | 0.73 | <0.001 | 0.77 | 0.70 | 0.84 | <0.001 |
| | Adjusted | 0.99 | 0.97 | 1.02 | 0.46 | 0.64 | 0.61 | 0.66 | <0.001 | 0.70 | 0.64 | 0.77 | <0.001 |
| New York | Unadjusted | 1.00 | 0.97 | 1.03 | 0.86 | 0.77 | 0.74 | 0.80 | <0.001 | 0.83 | 0.76 | 0.91 | <0.001 |
| | Adjusted | 0.98 | 0.95 | 1.01 | 0.27 | 0.72 | 0.70 | 0.75 | <0.001 | 0.77 | 0.70 | 0.85 | <0.001 |

**BP Users versus BP Non-users among Antidepressant Users**

| | | Odds of SARS-CoV-2 Test | | | | Odds of COVID-19 Diagnosis | | | | Odds of COVID-19 Hospitalization | | | |
|---|---|---|---|---|---|---|---|---|---|---|---|---|---|
| | | OR | LL | UL | p value | OR | LL | UL | p value | OR | LL | UL | p value |
| All | Unadjusted | 0.27 | 0.26 | 0.28 | <0.001 | 0.30 | 0.28 | 0.32 | <0.001 | 0.31 | 0.27 | 0.36 | <0.001 |
| | Adjusted | 0.27 | 0.25 | 0.28 | <0.001 | 0.30 | 0.28 | 0.32 | <0.001 | 0.33 | 0.28 | 0.38 | <0.001 |
| Northeast | Unadjusted | 0.28 | 0.26 | 0.31 | <0.001 | 0.33 | 0.30 | 0.37 | <0.001 | 0.36 | 0.29 | 0.45 | <0.001 |
| | Adjusted | 0.28 | 0.25 | 0.30 | <0.001 | 0.32 | 0.29 | 0.36 | <0.001 | 0.37 | 0.29 | 0.47 | <0.001 |
| Midwest | Unadjusted | 0.30 | 0.27 | 0.34 | <0.001 | 0.26 | 0.22 | 0.31 | <0.001 | 0.25 | 0.18 | 0.34 | <0.001 |
| | Adjusted | 0.30 | 0.26 | 0.34 | <0.001 | 0.27 | 0.22 | 0.33 | <0.001 | 0.26 | 0.18 | 0.36 | <0.001 |
| South | Unadjusted | 0.26 | 0.24 | 0.29 | <0.001 | 0.27 | 0.23 | 0.31 | <0.001 | 0.32 | 0.24 | 0.41 | <0.001 |
| | Adjusted | 0.26 | 0.24 | 0.28 | <0.001 | 0.27 | 0.23 | 0.32 | <0.001 | 0.32 | 0.24 | 0.43 | <0.001 |
| West | Unadjusted | 0.25 | 0.22 | 0.28 | <0.001 | 0.27 | 0.22 | 0.32 | <0.001 | 0.29 | 0.20 | 0.41 | <0.001 |
| | Adjusted | 0.24 | 0.21 | 0.27 | <0.001 | 0.29 | 0.28 | 0.30 | <0.001 | 0.33 | 0.23 | 0.48 | <0.001 |
| New York | Unadjusted | 0.30 | 0.26 | 0.34 | <0.001 | 0.33 | 0.28 | 0.38 | <0.001 | 0.24 | 0.16 | 0.36 | <0.001 |
| | Adjusted | 0.30 | 0.25 | 0.34 | <0.001 | 0.31 | 0.27 | 0.37 | <0.001 | 0.25 | 0.16 | 0.39 | <0.001 |

*Appendix 2—table 16 Continued on next page*

*Appendix 2—table 16 Continued*

| | | Odds of SARS-CoV-2 Test | | | | Odds of COVID-19 Diagnosis | | | | Odds of COVID-19 Hospitalization | | | |
|---|---|---|---|---|---|---|---|---|---|---|---|---|---|
| | | BP Users versus BP Non-users among Antidepressant Non-users | | | | | | | | | | | |
| | | OR | LL | UL | p value | OR | LL | UL | p value | OR | LL | UL | p value |
| All | Unadjusted | 0.20 | 0.19 | 0.22 | <0.001 | 0.22 | 0.20 | 0.24 | <0.001 | 0.24 | 0.20 | 0.28 | <0.001 |
| | Adjusted | 0.21 | 0.19 | 0.22 | <0.001 | 0.23 | 0.21 | 0.25 | <0.001 | 0.27 | 0.22 | 0.32 | <0.001 |
| Northeast | Unadjusted | 0.21 | 0.19 | 0.24 | <0.001 | 0.23 | 0.20 | 0.26 | <0.001 | 0.25 | 0.19 | 0.32 | <0.001 |
| | Adjusted | 0.22 | 0.19 | 0.25 | <0.001 | 0.24 | 0.22 | 0.25 | <0.001 | 0.29 | 0.22 | 0.39 | <0.001 |
| Midwest | Unadjusted | 0.22 | 0.19 | 0.26 | <0.001 | 0.23 | 0.18 | 0.28 | <0.001 | 0.28 | 0.19 | 0.39 | <0.001 |
| | Adjusted | 0.21 | 0.18 | 0.25 | <0.001 | 0.26 | 0.24 | 0.27 | <0.001 | 0.32 | 0.22 | 0.47 | <0.001 |
| South | Unadjusted | 0.20 | 0.18 | 0.22 | <0.001 | 0.21 | 0.18 | 0.25 | <0.001 | 0.21 | 0.15 | 0.30 | <0.001 |
| | Adjusted | 0.20 | 0.18 | 0.23 | <0.001 | 0.23 | 0.19 | 0.27 | <0.001 | 0.22 | 0.16 | 0.32 | <0.001 |
| West | Unadjusted | 0.18 | 0.16 | 0.21 | <0.001 | 0.20 | 0.16 | 0.25 | <0.001 | 0.20 | 0.13 | 0.30 | <0.001 |
| | Adjusted | 0.19 | 0.16 | 0.22 | <0.001 | 0.20 | 0.20 | 0.21 | <0.001 | 0.22 | 0.14 | 0.35 | <0.001 |
| New York | Unadjusted | 0.26 | 0.22 | 0.32 | <0.001 | 0.27 | 0.23 | 0.32 | <0.001 | 0.29 | 0.19 | 0.43 | <0.001 |
| | Adjusted | 0.26 | 0.23 | 0.30 | <0.001 | 0.26 | 0.22 | 0.32 | <0.001 | 0.35 | 0.22 | 0.54 | <0.001 |

LL: lower 95% confidence interval level; OR: odds ratio; UL: upper 95% confidence interval level.

**Appendix 2—table 17.** "Bone-Rx" Cohort (All Regions), Patient Characteristics Pre/Post Match.

| | "Bone-Rx" Cohort / All Observations Unmatched | | | | | | | "Bone-Rx" Cohort / All Observations Matched | | | | | | |
|---|---|---|---|---|---|---|---|---|---|---|---|---|---|---|
| | All | | BP Non-user | | BP User | | p-value | All | | BP Non-user | | BP User | | p-value |
| | N | % | N | % | N | % | | N | % | N | % | N | % | |
| All Patients | 502,895 | 100.0% | 50,844 | 10.1% | 452,051 | 89.9% | | 100,996 | 100.0% | 50,498 | 50.0% | 50,498 | 50.0% | |
| **Age** | | | | | | | | | | | | | | |
| ≤20 | 1,164 | 0.2% | 36 | 0.1% | 1,128 | 0.2% | <0.001 | 67 | 0.1% | 36 | 0.1% | 31 | 0.1% | 0.97 |
| 21-40 | 3,501 | 0.7% | 410 | 0.8% | 3,091 | 0.7% | | 790 | 0.8% | 403 | 0.8% | 387 | 0.8% | |
| 41-50 | 9,631 | 1.9% | 1,080 | 2.1% | 8,551 | 1.9% | | 2,107 | 2.1% | 1,069 | 2.1% | 1,038 | 2.1% | |
| 51-60 | 72,139 | 14.3% | 6,418 | 12.6% | 65,721 | 14.5% | | 12,777 | 12.7% | 6,395 | 12.7% | 6,382 | 12.6% | |
| 61-70 | 171,687 | 34.1% | 14,809 | 29.1% | 156,878 | 34.7% | | 29,509 | 29.2% | 14,751 | 29.2% | 14,758 | 29.2% | |
| 71-80 | 157,877 | 31.4% | 16,152 | 31.8% | 141,725 | 31.4% | | 32,129 | 31.8% | 16,055 | 31.8% | 16,074 | 31.8% | |
| ≥81 | 86,896 | 17.3% | 11,939 | 23.5% | 74,957 | 16.6% | | 23,617 | 23.4% | 11,789 | 23.3% | 11,828 | 23.4% | |
| **Gender** | | | | | | | | | | | | | | |
| Female | 451,790 | 89.8% | 44,354 | 87.2% | 407,436 | 90.1% | <0.001 | 88,552 | 87.7% | 44,235 | 87.6% | 44,317 | 87.8% | 0.43 |
| Male | 51,105 | 10.2% | 6,490 | 12.8% | 44,615 | 9.9% | | 12,444 | 12.3% | 6,263 | 12.4% | 6,181 | 12.2% | |
| **Region** | | | | | | | | | | | | | | |
| Midwest | 85,391 | 17.0% | 9,424 | 18.5% | 75,967 | 16.8% | <0.001 | 18,720 | 18.5% | 9,360 | 18.5% | 9,360 | 18.5% | 1.00 |
| Northeast | 135,867 | 27.0% | 16,139 | 31.7% | 119,728 | 26.5% | | 31,986 | 31.7% | 15,993 | 31.7% | 15,993 | 31.7% | |
| South | 178,118 | 35.4% | 17,232 | 33.9% | 160,886 | 35.6% | | 34,280 | 33.9% | 17,140 | 33.9% | 17,140 | 33.9% | |
| West | 103,519 | 20.6% | 8,049 | 15.8% | 95,470 | 21.1% | | 16,010 | 15.9% | 8,005 | 15.9% | 8,005 | 15.9% | |
| **Insurance** | | | | | | | | | | | | | | |
| Commercial | 164,150 | 32.6% | 17,092 | 33.6% | 147,058 | 32.5% | <0.001 | 33,977 | 33.6% | 16,963 | 33.6% | 17,014 | 33.7% | 0.91 |
| Dual | 33,969 | 6.8% | 2,562 | 5.0% | 31,407 | 6.9% | | 5,056 | 5.0% | 2,547 | 5.0% | 2,509 | 5.0% | |
| Medicaid | 84,514 | 16.8% | 7,034 | 13.8% | 77,480 | 17.1% | | 13,925 | 13.8% | 6,986 | 13.8% | 6,939 | 13.7% | |
| Medicare | 220,262 | 43.8% | 24,156 | 47.5% | 196,106 | 43.4% | | 48,038 | 47.6% | 24,002 | 47.5% | 24,036 | 47.6% | |
| **PCP Visit 2019** | | | | | | | | | | | | | | |
| No | 181,996 | 36.2% | 18,130 | 35.7% | 163,866 | 36.2% | 0.009 | 35,943 | 35.6% | 17,979 | 35.6% | 17,964 | 35.6% | 0.92 |
| Yes | 320,899 | 63.8% | 32,714 | 64.3% | 288,185 | 63.8% | | 65,053 | 64.4% | 32,519 | 64.4% | 32,534 | 64.4% | |
| **Continuous Outcomes** | mean | SD | mean | SD | mean | SD | p-value | mean | SD | mean | SD | mean | SD | p-value |
| CCI | 1.05 | 1.91 | 1.99 | 2.71 | 0.95 | 1.76 | <0.001 | 1.93 | 2.59 | 1.93 | 2.60 | 1.92 | 2.59 | 0.76 |

BP: bisphosphonate; CCI: Charlson Comorbidity Index; PCP: primary care physician; SD: standard deviation.

**Appendix 2—table 18.** "Bone-Rx" Cohort (Region=Northeast), Patient Characteristics Pre/Post Match.

| | "Bone-Rx" Cohort / Region=Northeast Unmatched | | | | | | | "Bone-Rx" Cohort / Region=Northeast Matched | | | | | | |
| | All | | BP Non-user | | BP User | | p-value | All | | BP Non-user | | BP User | | p-value |
| | N | % | N | % | N | % | | N | % | N | % | N | % | |
| **All Patients** | 135,867 | 100.0% | 16,139 | 11.9% | 119,728 | 88.1% | | 31,986 | 100.0% | 15,993 | 50.0% | 15,993 | 50.0% | |
| **Age** | | | | | | | | | | | | | | |
| ≤20 | 245 | 0.2% | ≤10 | 0.1% | 236 | 0.2% | <0.001 | 15 | 0.0% | ≤10 | 0.1% | ≤10 | 0.0% | 0.99 |
| 21-40 | 891 | 0.7% | 127 | 0.8% | 764 | 0.6% | | 250 | 0.8% | 124 | 0.8% | 126 | 0.8% | |
| 41-50 | 2,340 | 1.7% | 298 | 1.8% | 2,042 | 1.7% | | 570 | 1.8% | 290 | 1.8% | 280 | 1.8% | |
| 51-60 | 20,069 | 14.8% | 2,059 | 12.8% | 18,010 | 15.0% | | 4,088 | 12.8% | 2,049 | 12.8% | 2,039 | 12.7% | |
| 61-70 | 45,896 | 33.8% | 4,802 | 29.8% | 41,094 | 34.3% | | 9,526 | 29.8% | 4,767 | 29.8% | 4,759 | 29.8% | |
| 71-80 | 42,828 | 31.5% | 5,267 | 32.6% | 37,561 | 31.4% | | 10,465 | 32.7% | 5,226 | 32.7% | 5,239 | 32.8% | |
| ≥81 | 23,598 | 17.4% | 3,577 | 22.2% | 20,021 | 16.7% | | 7,072 | 22.1% | 3,528 | 22.1% | 3,544 | 22.2% | |
| **Gender** | | | | | | | | | | | | | | |
| Female | 122,485 | 90.2% | 14,115 | 87.5% | 108,370 | 90.5% | <0.001 | 28,157 | 88.0% | 14,062 | 87.9% | 14,095 | 88.1% | 0.57 |
| Male | 13,382 | 9.8% | 2,024 | 12.5% | 11,358 | 9.5% | | 3,829 | 12.0% | 1,931 | 12.1% | 1,898 | 11.9% | |
| **Insurance** | | | | | | | | | | | | | | |
| Commercial | 37,810 | 27.8% | 4,517 | 28.0% | 33,293 | 27.8% | <0.001 | 8,927 | 27.9% | 4,459 | 27.9% | 4,468 | 27.9% | 0.99 |
| Dual | 8,434 | 6.2% | 829 | 5.1% | 7,605 | 6.4% | | 1,637 | 5.1% | 824 | 5.2% | 813 | 5.1% | |
| Medicaid | 25,296 | 18.6% | 2,082 | 12.9% | 23,214 | 19.4% | | 4,122 | 12.9% | 2,067 | 12.9% | 2,055 | 12.8% | |
| Medicare | 64,327 | 47.3% | 8,711 | 54.0% | 55,616 | 46.5% | | 17,300 | 54.1% | 8,643 | 54.0% | 8,657 | 54.1% | |
| **PCP Visit 2019** | | | | | | | | | | | | | | |
| No | 56,593 | 41.7% | 6,726 | 41.7% | 49,867 | 41.7% | 0.95 | 13,307 | 41.6% | 6,654 | 41.6% | 6,653 | 41.6% | 0.99 |
| Yes | 79,274 | 58.3% | 9,413 | 58.3% | 69,861 | 58.3% | | 18,679 | 58.4% | 9,339 | 58.4% | 9,340 | 58.4% | |
| **Continuous Outcomes** | | | | | | | | | | | | | | |
| | mean | SD | mean | SD | mean | SD | p-value | mean | SD | mean | SD | mean | SD | p-value |
| CCI | 1.06 | 1.89 | 1.97 | 2.70 | 0.93 | 1.71 | <0.001 | 1.89 | 2.57 | 1.89 | 2.58 | 1.89 | 2.57 | 0.91 |

BP: bisphosphonate; CCI: Charlson Comorbidity Index; PCP: primary care physician; SD: standard deviation.

**Appendix 2—table 19.** "Bone-Rx" Cohort (Region=Midwest), Patient Characteristics Pre/Post Match.

| | "Bone-Rx" Cohort / Region=Midwest Unmatched | | | | | | | | "Bone-Rx" Cohort / Region=Midwest Matched | | | | | | | |
| | All | | BP Non-user | | BP User | | p-value | All | | BP Non-user | | BP User | | p-value |
| | N | % | N | % | N | % | | N | % | N | % | N | % | |
| All Patients | 85,391 | 100.0% | 9,424 | 11.0% | 75,967 | 89.0% | | 18,720 | 100.0% | 9,360 | 50.0% | 9,360 | 50.0% | |
| **Age** | | | | | | | | | | | | | | |
| ≤20 | 274 | 0.3% | ≤10 | 0.1% | 268 | 0.4% | <0.001 | 13 | 0.1% | ≤10 | 0.1% | ≤10 | 0.1% | 1.00 |
| 21-40 | 672 | 0.8% | 79 | 0.8% | 593 | 0.8% | | 154 | 0.8% | 78 | 0.8% | 76 | 0.8% | |
| 41-50 | 1,886 | 2.2% | 202 | 2.1% | 1,684 | 2.2% | | 389 | 2.1% | 200 | 2.1% | 189 | 2.0% | |
| 51-60 | 13,522 | 15.8% | 1,284 | 13.6% | 12,238 | 16.1% | | 2,559 | 13.7% | 1,280 | 13.7% | 1,279 | 13.7% | |
| 61-70 | 31,256 | 36.6% | 2,760 | 29.3% | 28,496 | 37.5% | | 5,512 | 29.4% | 2,754 | 29.4% | 2,758 | 29.5% | |
| 71-80 | 23,887 | 28.0% | 2,766 | 29.4% | 21,121 | 27.8% | | 5,492 | 29.3% | 2,748 | 29.4% | 2,744 | 29.3% | |
| ≥81 | 13,894 | 16.3% | 2,327 | 24.7% | 11,567 | 15.2% | | 4,601 | 24.6% | 2,294 | 24.5% | 2,307 | 24.6% | |
| **Gender** | | | | | | | | | | | | | | |
| Female | 76,696 | 89.8% | 8,118 | 86.1% | 68,578 | 90.3% | <0.001 | 16,223 | 86.7% | 8,102 | 86.6% | 8,121 | 86.8% | 0.68 |
| Male | 8,695 | 10.2% | 1,306 | 13.9% | 7,389 | 9.7% | | 2,497 | 13.3% | 1,258 | 13.4% | 1,239 | 13.2% | |
| **Insurance** | | | | | | | | | | | | | | |
| Commercial | 34,494 | 40.4% | 3,361 | 35.7% | 31,133 | 41.0% | <0.001 | 6,699 | 35.8% | 3,345 | 35.7% | 3,354 | 35.8% | 0.96 |
| Dual | 4,042 | 4.7% | 436 | 4.6% | 3,606 | 4.7% | | 852 | 4.6% | 429 | 4.6% | 423 | 4.5% | |
| Medicaid | 8,856 | 10.4% | 733 | 7.8% | 8,123 | 10.7% | | 1,441 | 7.7% | 729 | 7.8% | 712 | 7.6% | |
| Medicare | 37,999 | 44.5% | 4,894 | 51.9% | 33,105 | 43.6% | | 9,728 | 52.0% | 4,857 | 51.9% | 4,871 | 52.0% | |
| **PCP Visit 2019** | | | | | | | | | | | | | | |
| No | 32,037 | 37.5% | 3,330 | 35.3% | 28,707 | 37.8% | <0.001 | 6,628 | 35.4% | 3,312 | 35.4% | 3,316 | 35.4% | 0.95 |
| Yes | 53,354 | 62.5% | 6,094 | 64.7% | 47,260 | 62.2% | | 12,092 | 64.6% | 6,048 | 64.6% | 6,044 | 64.6% | |
| **Continuous Outcomes** | | | | | | | | | | | | | | |
| | mean | SD | mean | SD | mean | SD | p-value | mean | SD | mean | SD | mean | SD | p-value |
| CCI | 1.12 | 2.02 | 2.12 | 2.83 | 0.99 | 1.86 | <0.001 | 2.05 | 2.72 | 2.06 | 2.72 | 2.05 | 2.72 | 0.91 |

BP: bisphosphonate; CCI: Charlson Comorbidity Index; PCP: primary care physician; SD: standard deviation.

**Appendix 2—table 20.** *"Bone-Rx"* Cohort (Region=South), Patient Characteristics Pre/Post Match.

| | "Bone-Rx" Cohort / Region=South Unmatched | | | | | | | "Bone-Rx" Cohort / Region=South Matched | | | | | | |
| --- | --- | --- | --- | --- | --- | --- | --- | --- | --- | --- | --- | --- | --- | --- |
| | All | | BP Non-user | | BP User | | p-value | All | | BP Non-user | | BP User | | p-value |
| | N | % | N | % | N | % | | N | % | N | % | N | % | |
| **All Patients** | 178,118 | 100.0% | 17,232 | 9.7% | 160,886 | 90.3% | | 34,280 | 100.0% | 17,140 | 50.0% | 17,140 | 50.0% | 1.00 |
| **Age** | | | | | | | <0.001 | | | | | | | |
| ≤20 | 490 | 0.3% | 16 | 0.1% | 474 | 0.3% | | 31 | 0.1% | 16 | 0.1% | 15 | 0.1% | |
| 21-40 | 1,313 | 0.7% | 136 | 0.8% | 1,177 | 0.7% | | 262 | 0.8% | 134 | 0.8% | 128 | 0.7% | |
| 41-50 | 3,866 | 2.2% | 445 | 2.6% | 3,421 | 2.1% | | 884 | 2.6% | 444 | 2.6% | 440 | 2.6% | |
| 51-60 | 27,389 | 15.4% | 2,296 | 13.3% | 25,093 | 15.6% | | 4,574 | 13.3% | 2,290 | 13.4% | 2,284 | 13.3% | |
| 61-70 | 61,038 | 34.3% | 5,142 | 29.8% | 55,896 | 34.7% | | 10,271 | 30.0% | 5,129 | 29.9% | 5,142 | 30.0% | |
| 71-80 | 56,126 | 31.5% | 5,521 | 32.0% | 50,605 | 31.5% | | 10,990 | 32.1% | 5,493 | 32.0% | 5,497 | 32.1% | |
| ≥81 | 27,896 | 15.7% | 3,676 | 21.3% | 24,220 | 15.1% | | 7,268 | 21.2% | 3,634 | 21.2% | 3,634 | 21.2% | |
| **Gender** | | | | | | | <0.001 | | | | | | | |
| Female | 160,994 | 90.4% | 15,179 | 88.1% | 145,815 | 90.6% | | 30,322 | 88.5% | 15,149 | 88.4% | 15,173 | 88.5% | 0.69 |
| Male | 17,124 | 9.6% | 2,053 | 11.9% | 15,071 | 9.4% | | 3,958 | 11.5% | 1,991 | 11.6% | 1,967 | 11.5% | |
| **Insurance** | | | | | | | <0.001 | | | | | | | |
| Commercial | 66,332 | 37.2% | 7,042 | 40.9% | 59,290 | 36.9% | | 14,052 | 41.0% | 7,007 | 40.9% | 7,045 | 41.1% | 0.95 |
| Dual | 14,829 | 8.3% | 769 | 4.5% | 14,060 | 8.7% | | 1,523 | 4.4% | 769 | 4.5% | 754 | 4.4% | |
| Medicaid | 23,492 | 13.2% | 1,843 | 10.7% | 21,649 | 13.5% | | 3,639 | 10.6% | 1,829 | 10.7% | 1,810 | 10.6% | |
| Medicare | 73,465 | 41.2% | 7,578 | 44.0% | 65,887 | 41.0% | | 15,066 | 43.9% | 7,535 | 44.0% | 7,531 | 43.9% | |
| **PCP Visit 2019** | | | | | | | 0.454 | | | | | | | |
| No | 60,253 | 33.8% | 5,785 | 33.6% | 54,468 | 33.9% | | 11,462 | 33.4% | 5,736 | 33.5% | 5,726 | 33.4% | 0.91 |
| Yes | 117,865 | 66.2% | 11,447 | 66.4% | 106,418 | 66.1% | | 22,818 | 66.6% | 11,404 | 66.5% | 11,414 | 66.6% | |

| **Continuous Outcomes** | mean | SD | mean | SD | mean | SD | p-value | mean | SD | mean | SD | mean | SD | p-value |
| --- | --- | --- | --- | --- | --- | --- | --- | --- | --- | --- | --- | --- | --- | --- |
| CCI | 0.95 | 1.84 | 1.86 | 2.65 | 0.86 | 1.70 | <0.001 | 1.80 | 2.54 | 1.80 | 2.54 | 1.79 | 2.53 | 0.78 |

BP: bisphosphonate; CCI: Charlson Comorbidity Index; PCP: primary care physician; SD: standard deviation.

**Appendix 2—table 21.** "Bone-Rx" Cohort (Region=West), Patient Characteristics Pre/Post Match.

| | "Bone-Rx" Cohort / Region=West Unmatched | | | | | | | "Bone-Rx" Cohort / Region=West Matched | | | | | | |
|---|---|---|---|---|---|---|---|---|---|---|---|---|---|---|
| | All | | BP Non-user | | BP User | | p-value | All | | BP Non-user | | BP User | | p-value |
| | N | % | N | % | N | % | | N | % | N | % | N | % | |
| All Patients | 103,519 | 100.0% | 8,049 | 7.8% | 95,470 | 92.2% | | 16,010 | 100.0% | 8,005 | 50.0% | 8,005 | 50.0% | |
| **Age** | | | | | | | | | | | | | | |
| ≤20 | 155 | 0.1% | ≤10 | 0.1% | 150 | 0.2% | <0.001 | ≤10 | 0.0% | ≤10 | 0.1% | ≤10 | 0.0% | 0.96 |
| 21-40 | 625 | 0.6% | 68 | 0.8% | 557 | 0.6% | | 124 | 0.8% | 67 | 0.8% | 57 | 0.7% | |
| 41-50 | 1,539 | 1.5% | 135 | 1.7% | 1,404 | 1.5% | | 264 | 1.6% | 135 | 1.7% | 129 | 1.6% | |
| 51-60 | 11,159 | 10.8% | 779 | 9.7% | 10,380 | 10.9% | | 1,556 | 9.7% | 776 | 9.7% | 780 | 9.7% | |
| 61-70 | 33,497 | 32.4% | 2,105 | 26.2% | 31,392 | 32.9% | | 4,200 | 26.2% | 2,101 | 26.2% | 2,099 | 26.2% | |
| 71-80 | 35,036 | 33.8% | 2,598 | 32.3% | 32,438 | 34.0% | | 5,182 | 32.4% | 2,588 | 32.3% | 2,594 | 32.4% | |
| ≥81 | 21,508 | 20.8% | 2,359 | 29.3% | 19,149 | 20.1% | | 4,676 | 29.2% | 2,333 | 29.1% | 2,343 | 29.3% | |
| **Gender** | | | | | | | | | | | | | | |
| Female | 91,615 | 88.5% | 6,942 | 86.2% | 84,673 | 88.7% | <0.001 | 13,850 | 86.5% | 6,922 | 86.5% | 6,928 | 86.5% | 0.89 |
| Male | 11,904 | 11.5% | 1,107 | 13.8% | 10,797 | 11.3% | | 2,160 | 13.5% | 1,083 | 13.5% | 1,077 | 13.5% | |
| **Insurance** | | | | | | | | | | | | | | |
| Commercial | 25,514 | 24.6% | 2,172 | 27.0% | 23,342 | 24.4% | <0.001 | 4,299 | 26.9% | 2,152 | 26.9% | 2,147 | 26.8% | 1.00 |
| Dual | 6,664 | 6.4% | 528 | 6.6% | 6,136 | 6.4% | | 1,044 | 6.5% | 525 | 6.6% | 519 | 6.5% | |
| Medicaid | 26,870 | 26.0% | 2,376 | 29.5% | 24,494 | 25.7% | | 4,723 | 29.5% | 2,361 | 29.5% | 2,362 | 29.5% | |
| Medicare | 44,471 | 43.0% | 2,973 | 36.9% | 41,498 | 43.5% | | 5,944 | 37.1% | 2,967 | 37.1% | 2,977 | 37.2% | |
| **PCP Visit 2019** | | | | | | | | | | | | | | |
| No | 33,113 | 32.0% | 2,289 | 28.4% | 30,824 | 32.3% | <0.001 | 4,546 | 28.4% | 2,277 | 28.4% | 2,269 | 28.3% | 0.89 |
| Yes | 70,406 | 68.0% | 5,760 | 71.6% | 64,646 | 67.7% | | 11,464 | 71.6% | 5,728 | 71.6% | 5,736 | 71.7% | |
| **Continuous Outcomes** | | | | | | | | | | | | | | |
| | mean | SD | mean | SD | mean | SD | p-value | mean | SD | mean | SD | mean | SD | p-value |
| CCI | 1.17 | 1.94 | 2.17 | 2.67 | 1.08 | 1.84 | <0.001 | 2.12 | 2.59 | 2.12 | 2.59 | 2.12 | 2.59 | 0.93 |

BP: bisphosphonate; CCI: Charlson Comorbidity Index; PCP: primary care physician; SD: standard deviation.

**Appendix 2—table 22.** *"Bone-Rx"* Cohort (Region=New York State), Patient Characteristics Pre/Post Match.

| | "Bone-Rx" Cohort / Region=New York State Unmatched | | | | | | | "Bone-Rx" Cohort / Region=New York State Matched | | | | | | |
| --- | --- | --- | --- | --- | --- | --- | --- | --- | --- | --- | --- | --- | --- | --- |
| | All | | BP Non-user | | BP User | | p-value | All | | BP Non-user | | BP User | | p-value |
| | N | % | N | % | N | % | | N | % | N | % | N | % | |
| **All Patients** | 57,397 | 100.0% | 7,362 | 12.8% | 50,035 | 87.2% | | 14,508 | 100.0% | 7,254 | 50.0% | 7,254 | 50.0% | |
| **Age** | | | | | | | <0.001 | | | | | | | 0.96 |
| ≤20 | 56 | 0.1% | ≤10 | 0.1% | 50 | 0.1% | | 11 | 0.1% | ≤10 | 0.1% | ≤10 | 0.1% | |
| 21-40 | 272 | 0.5% | 44 | 0.6% | 228 | 0.5% | | 76 | 0.5% | 42 | 0.6% | 34 | 0.5% | |
| 41-50 | 775 | 1.4% | 120 | 1.6% | 655 | 1.3% | | 207 | 1.4% | 107 | 1.5% | 100 | 1.4% | |
| 51-60 | 7,249 | 12.6% | 885 | 12.0% | 6,364 | 12.7% | | 1,744 | 12.0% | 871 | 12.0% | 873 | 12.0% | |
| 61-70 | 18,433 | 32.1% | 2,297 | 31.2% | 16,136 | 32.2% | | 4,540 | 31.3% | 2,264 | 31.2% | 2,276 | 31.4% | |
| 71-80 | 19,944 | 34.7% | 2,482 | 33.7% | 17,462 | 34.9% | | 4,934 | 34.0% | 2,455 | 33.8% | 2,479 | 34.2% | |
| ≥81 | 10,668 | 18.6% | 1,528 | 20.8% | 9,140 | 18.3% | | 2,996 | 20.7% | 1,509 | 20.8% | 1,487 | 20.5% | |
| **Gender** | | | | | | | <.001 | | | | | | | 0.13 |
| Female | 52,047 | 90.7% | 6,589 | 89.5% | 45,458 | 90.9% | | 13,106 | 90.3% | 6,526 | 90.0% | 6,580 | 90.7% | |
| Male | 5,350 | 9.3% | 773 | 10.5% | 4,577 | 9.1% | | 1,402 | 9.7% | 728 | 10.0% | 674 | 9.3% | |
| **Insurance** | | | | | | | <0.001 | | | | | | | 1.00 |
| Commercial | 12,309 | 21.4% | 1,894 | 25.7% | 10,415 | 20.8% | | 3,706 | 25.5% | 1,850 | 25.5% | 1,856 | 25.6% | |
| Dual | 1,750 | 3.0% | 154 | 2.1% | 1,596 | 3.2% | | 307 | 2.1% | 153 | 2.1% | 154 | 2.1% | |
| Medicaid | 10,191 | 17.8% | 1,016 | 13.8% | 9,175 | 18.3% | | 1,968 | 13.6% | 987 | 13.6% | 981 | 13.5% | |
| Medicare | 33,147 | 57.8% | 4,298 | 58.4% | 28,849 | 57.7% | | 8,527 | 58.8% | 4,264 | 58.8% | 4,263 | 58.8% | |
| **PCP Visit 2019** | | | | | | | 0.35 | | | | | | | 0.73 |
| No | 21,462 | 37.4% | 2,789 | 37.9% | 18,673 | 37.3% | | 5,468 | 37.7% | 2,744 | 37.8% | 2,724 | 37.6% | |
| Yes | 35,935 | 62.6% | 4,573 | 62.1% | 31,362 | 62.7% | | 9,040 | 62.3% | 4,510 | 62.2% | 4,530 | 62.4% | |
| **Continuous Outcomes** | mean | SD | mean | SD | mean | SD | p-value | mean | SD | mean | SD | mean | SD | p-value |
| CCI | 1.06 | 1.84 | 1.81 | 2.56 | 0.95 | 1.68 | <0.001 | 1.69 | 2.35 | 1.69 | 2.36 | 1.69 | 2.35 | 0.98 |

BP: bisphosphonate; CCI: Charlson Comorbidity Index; PCP: primary care physician; SD: standard deviation.

**Appendix 2—table 23.** "Osteo-Dx-Rx" Cohort, Patient Characteristics Pre/Post Match.

| | "Osteo-Dx-Rx" Cohort / All Observations Unmatched | | | | | | | "Osteo-Dx-Rx" Cohort / All Observations Matched | | | | | | |
| --- | --- | --- | --- | --- | --- | --- | --- | --- | --- | --- | --- | --- | --- | --- |
| | All | | BP Non-user | | BP User | | p-value | All | | BP Non-user | | BP User | | p-value |
| | N | % | N | % | N | % | | N | % | N | % | N | % | |
| **All Patients** | 60,043 | 100.0% | 8,392 | 14.0% | 51,651 | 86.0% | | 15,898 | 100.0% | 7,949 | 50.0% | 7,949 | 50.0% | |
| **Age** | | | | | | | | | | | | | | |
| 51-60 | 6,443 | 10.7% | 753 | 9.0% | 5,690 | 11.0% | <0.001 | 1,430 | 9.0% | 723 | 9.1% | 707 | 8.9% | 0.95 |
| 61-70 | 20,187 | 33.6% | 2,492 | 29.7% | 17,695 | 34.3% | | 4,821 | 30.3% | 2,397 | 30.2% | 2,424 | 30.5% | |
| 71-80 | 21,545 | 35.9% | 2,964 | 35.3% | 18,581 | 36.0% | | 5,677 | 35.7% | 2,841 | 35.7% | 2,836 | 35.7% | |
| ≥81 | 11,868 | 19.8% | 2,183 | 26.0% | 9,685 | 18.8% | | 3,970 | 25.0% | 1,988 | 25.0% | 1,982 | 24.9% | |
| **State** | | | | | | | | | | | | | | |
| CA | 24,489 | 40.8% | 2,558 | 30.5% | 21,931 | 42.5% | <0.001 | 4,886 | 30.7% | 2,443 | 30.7% | 2,443 | 30.7% | 1.00 |
| FL | 11,904 | 19.8% | 1,767 | 21.1% | 10,137 | 19.6% | | 3,256 | 20.5% | 1,628 | 20.5% | 1,628 | 20.5% | |
| IL | 4,447 | 7.4% | 678 | 8.1% | 3,769 | 7.3% | | 1,168 | 7.3% | 584 | 7.3% | 584 | 7.3% | |
| NY | 19,203 | 32.0% | 3,389 | 40.4% | 15,814 | 30.6% | | 6,588 | 41.4% | 3,294 | 41.4% | 3,294 | 41.4% | |
| **Insurance** | | | | | | | | | | | | | | |
| Commercial | 12,990 | 21.6% | 2,048 | 24.4% | 10,942 | 21.2% | <0.001 | 3,736 | 23.5% | 1,868 | 23.5% | 1,868 | 23.5% | 1.00 |
| Dual | 3,652 | 6.1% | 313 | 3.7% | 3,339 | 6.5% | | 554 | 3.5% | 277 | 3.5% | 277 | 3.5% | |
| Medicaid | 13,698 | 22.8% | 1,785 | 21.3% | 11,913 | 23.1% | | 3,392 | 21.3% | 1,696 | 21.3% | 1,696 | 21.3% | |
| Medicare | 29,703 | 49.5% | 4,246 | 50.6% | 25,457 | 49.3% | | 8,216 | 51.7% | 4,108 | 51.7% | 4,108 | 51.7% | |
| **PCP Visit 2019** | | | | | | | | | | | | | | |
| No | 14,089 | 23.5% | 2,427 | 28.9% | 11,662 | 22.6% | <0.001 | 4,487 | 28.2% | 2,243 | 28.2% | 2,244 | 28.2% | 0.99 |
| Yes | 45,954 | 76.5% | 5,965 | 71.1% | 39,989 | 77.4% | | 11,411 | 71.8% | 5,706 | 71.8% | 5,705 | 71.8% | |
| **Cancer Dx** | | | | | | | | | | | | | | |
| No | 52,301 | 87.1% | 6,765 | 80.6% | 45,536 | 88.2% | <0.001 | 13,116 | 82.5% | 6,548 | 82.4% | 6,568 | 82.6% | 0.68 |
| Yes | 7,742 | 12.9% | 1,627 | 19.4% | 6,115 | 11.8% | | 2,782 | 17.5% | 1,401 | 17.6% | 1,381 | 17.4% | |
| **COPD Dx** | | | | | | | | | | | | | | |
| No | 53,446 | 89.0% | 7,035 | 83.8% | 46,411 | 89.9% | <0.001 | 13,705 | 86.2% | 6,834 | 86.0% | 6,871 | 86.4% | 0.39 |
| Yes | 6,597 | 11.0% | 1,357 | 16.2% | 5,240 | 10.1% | | 2,193 | 13.8% | 1,115 | 14.0% | 1,078 | 13.6% | |
| **Heart Failure Dx** | | | | | | | | | | | | | | |
| No | 56,005 | 93.3% | 7,492 | 89.3% | 48,513 | 93.9% | <0.001 | 14,475 | 91.0% | 7,218 | 90.8% | 7,257 | 91.3% | 0.28 |
| Yes | 4,038 | 6.7% | 900 | 10.7% | 3,138 | 6.1% | | 1,423 | 9.0% | 731 | 9.2% | 692 | 8.7% | |

*Appendix 2—table 23 Continued on next page*

Appendix 2—table 23 Continued

| | "Osteo-Dx-Rx" Cohort / All Observations Unmatched | | | | | | | | "Osteo-Dx-Rx" Cohort / All Observations Matched | | | | | | | |
| | All | | BP Non-user | | BP User | | p-value | | All | | BP Non-user | | BP User | | p-value |
| | N | % | N | % | N | % | | N | % | N | % | N | % | |
| **Hypertension Dx** | | | | | | | | | | | | | | |
| No | 24,966 | 41.6% | 3,281 | 39.1% | 21,685 | 42.0% | <0.001 | 6,268 | 39.4% | 3,137 | 39.5% | 3,131 | 39.4% | 0.92 |
| Yes | 35,077 | 58.4% | 5,111 | 60.9% | 29,966 | 58.0% | | 9,630 | 60.6% | 4,812 | 60.5% | 4,818 | 60.6% | |
| **Dyslipidemia Dx** | | | | | | | | | | | | | | |
| No | 24,095 | 40.1% | 3,295 | 39.3% | 20,800 | 40.3% | 0.08 | 6,187 | 38.9% | 3,101 | 39.0% | 3,086 | 38.8% | 0.81 |
| Yes | 35,948 | 59.9% | 5,097 | 60.7% | 30,851 | 59.7% | | 9,711 | 61.1% | 4,848 | 61.0% | 4,863 | 61.2% | |
| **Obesity Dx** | | | | | | | | | | | | | | |
| No | 53,453 | 89.0% | 7,583 | 90.4% | 45,870 | 88.8% | <0.001 | 14,468 | 91.0% | 7,217 | 90.8% | 7,251 | 91.2% | 0.35 |
| Yes | 6,590 | 11.0% | 809 | 9.6% | 5,781 | 11.2% | | 1,430 | 9.0% | 732 | 9.2% | 698 | 8.8% | |
| **Type 2 Diabetes Dx** | | | | | | | | | | | | | | |
| No | 44,565 | 74.2% | 6,132 | 73.1% | 38,433 | 74.4% | 0.009 | 11,759 | 74.0% | 5,859 | 73.7% | 5,900 | 74.2% | 0.46 |
| Yes | 15,478 | 25.8% | 2,260 | 26.9% | 13,218 | 25.6% | | 4,139 | 26.0% | 2,090 | 26.3% | 2,049 | 25.8% | |
| **Depression Dx** | | | | | | | | | | | | | | |
| No | 51,609 | 86.0% | 7,114 | 84.8% | 44,495 | 86.1% | 0.001 | 13,697 | 86.2% | 6,844 | 86.1% | 6,853 | 86.2% | 0.84 |
| Yes | 8,434 | 14.0% | 1,278 | 15.2% | 7,156 | 13.9% | | 2,201 | 13.8% | 1,105 | 13.9% | 1,096 | 13.8% | |

BP: bisphosphonate; CCI: Charlson Comorbidity Index; CA: California; Dx: diagnosis; FL: Florida; IL: Illinois; NY: New York; PCP: primary care physician.

**Appendix 2—table 24.** Statin Cohort (All Regions), Patient Characteristics Pre/Post Match.

| | All Observations by Statin Use: Unmatched | | | | | | | All Observations by Statin Use: Matched | | | | | | |
| | All | | Statin Non-users | | Statin Users | | p-value | All | | Statin Non-users | | Statin Users | | p-value |
| | N | % | N | % | N | % | | N | % | N | % | N | % | |
|---|---|---|---|---|---|---|---|---|---|---|---|---|---|---|
| **All Patients** | 7,906,603 | 100.00% | 6,403,208 | 81.00% | 1,503,395 | 19.00% | | 2,872,600 | 100.00% | 1,436,300 | 50.00% | 1,436,300 | 50.00% | |
| **Age** | | | | | | | <0.001 | | | | | | | 0.11 |
| ≤20 | 1,840,050 | 23.30% | 1,838,665 | 28.70% | 1,385 | 0.10% | | 2,772 | 0.10% | 1,387 | 0.10% | 1,385 | 0.10% | |
| 21–40 | 1,446,999 | 18.30% | 1,402,606 | 21.90% | 44,393 | 3.00% | | 88,760 | 3.10% | 44,371 | 3.10% | 44,389 | 3.10% | |
| 41–50 | 925,309 | 11.70% | 789,385 | 12.30% | 135,924 | 9.00% | | 271,615 | 9.50% | 135,748 | 9.50% | 135,867 | 9.50% | |
| 51–60 | 1,250,190 | 15.80% | 888,510 | 13.90% | 361,680 | 24.10% | | 710,481 | 24.70% | 354,449 | 24.70% | 356,032 | 24.80% | |
| 61–70 | 1,181,261 | 14.90% | 728,702 | 11.40% | 452,559 | 30.10% | | 857,269 | 29.80% | 428,326 | 29.80% | 428,943 | 29.90% | |
| 71–80 | 783,775 | 9.90% | 452,267 | 7.10% | 331,508 | 22.10% | | 605,360 | 21.10% | 303,279 | 21.10% | 302,081 | 21.00% | |
| ≥81 | 479,019 | 6.10% | 303,073 | 4.70% | 175,946 | 11.70% | | 336,343 | 11.70% | 168,740 | 11.70% | 167,603 | 11.70% | |
| **Gender** | | | | | | | <0.001 | | | | | | | <0.001 |
| Female | 4,670,960 | 59.10% | 3,785,061 | 59.10% | 885,899 | 58.90% | | 1,682,354 | 58.60% | 839,207 | 58.40% | 843,147 | 58.70% | |
| Male | 3,235,643 | 40.90% | 2,618,147 | 40.90% | 617,496 | 41.10% | | 1,190,246 | 41.40% | 597,093 | 41.60% | 593,153 | 41.30% | |
| **Region** | | | | | | | <0.001 | | | | | | | 1 |
| Midwest | 1,467,802 | 18.60% | 1,188,569 | 18.60% | 279,233 | 18.60% | | 542,638 | 18.90% | 271,319 | 18.90% | 271,319 | 18.90% | |
| Northeast | 2,152,560 | 27.20% | 1,706,021 | 26.60% | 446,539 | 29.70% | | 847,868 | 29.50% | 423,934 | 29.50% | 423,934 | 29.50% | |
| South | 3,042,604 | 38.50% | 2,490,630 | 38.90% | 551,974 | 36.70% | | 1,046,224 | 36.40% | 523,112 | 36.40% | 523,112 | 36.40% | |
| West | 1,243,637 | 15.70% | 1,017,988 | 15.90% | 225,649 | 15.00% | | 435,870 | 15.20% | 217,935 | 15.20% | 217,935 | 15.20% | |
| **Insurance** | | | | | | | <0.001 | | | | | | | 0.34 |
| Commercial | 3,938,603 | 49.80% | 3,350,332 | 52.30% | 588,271 | 39.10% | | 1,175,472 | 40.90% | 587,847 | 40.90% | 587,625 | 40.90% | |
| Dual | 156,497 | 2.00% | 73,532 | 1.10% | 82,965 | 5.50% | | 110,207 | 3.80% | 54,851 | 3.80% | 55,356 | 3.90% | |
| Medicaid | 2,594,500 | 32.80% | 2,254,531 | 35.20% | 339,969 | 22.60% | | 641,345 | 22.30% | 320,434 | 22.30% | 320,911 | 22.30% | |
| Medicare | 1,217,003 | 15.40% | 724,813 | 11.30% | 492,190 | 32.70% | | 945,576 | 32.90% | 473,168 | 32.90% | 472,408 | 32.90% | |
| **PCP Visit 2019** | | | | | | | <0.001 | | | | | | | 0.29 |
| No | 4,283,697 | 54.20% | 3,773,784 | 58.90% | 509,913 | 33.90% | | 1,016,313 | 35.40% | 508,587 | 35.40% | 507,726 | 35.30% | |
| Yes | 3,622,906 | 45.80% | 2,629,424 | 41.10% | 993,482 | 66.10% | | 1,856,287 | 64.60% | 927,713 | 64.60% | 928,574 | 64.70% | |
| **Continuous Outcomes** | | | | | | | | | | | | | | |
| | mean | SD | mean | SD | mean | SD | p-value | mean | SD | mean | SD | mean | SD | p-value |
| CCI | 0.62 | 1.38 | 0.49 | 1.23 | 1.15 | 1.79 | <0.001 | 1.11 | 1.77 | 1.12 | 1.79 | 1.11 | 1.75 | <0.001 |

CCI: Charlson Comorbidity Index; PCP: primary care physician; SD: standard deviation.

**Appendix 2—table 25.** Statin Cohort (Region=New York State), Patient Characteristics Pre/Post Match.

| | Region=NY by Statin Use: Unmatched | | | | | | | Region=NY by Statin Use: Matched | | | | | | |
| --- | --- | --- | --- | --- | --- | --- | --- | --- | --- | --- | --- | --- | --- | --- |
| | All | | Statin Non-users | | Statin Users | | p-value | All | | Statin Non-users | | Statin Users | | p-value |
| | N | % | N | % | N | % | | N | % | N | % | N | % | |
| All Patients | 968,296 | 100.0% | 761,995 | 78.7% | 206,301 | 21.3% | | 371,072 | 100.0% | 185,536 | 50.0% | 185,536 | 50.0% | 1.00 |
| **Age** | | | | | | | <0.001 | | | | | | | 1.00 |
| ≤20 | 133,178 | 13.8% | 133,111 | 17.5% | 67 | 0.0% | | 134 | 0.0% | 67 | 0.0% | 67 | 0.0% | |
| 21-40 | 192,959 | 19.9% | 188,446 | 24.7% | 4,513 | 2.2% | | 9,019 | 2.4% | 4,508 | 2.4% | 4,511 | 2.4% | |
| 41-50 | 127,794 | 13.2% | 112,342 | 14.7% | 15,452 | 7.5% | | 30,860 | 8.3% | 15,420 | 8.3% | 15,440 | 8.3% | |
| 51-60 | 172,444 | 17.8% | 128,472 | 16.9% | 43,972 | 21.3% | | 86,136 | 23.2% | 43,068 | 23.2% | 43,068 | 23.2% | |
| 61-70 | 159,912 | 16.5% | 100,884 | 13.2% | 59,028 | 28.6% | | 106,460 | 28.7% | 53,233 | 28.7% | 53,227 | 28.7% | |
| 71-80 | 120,117 | 12.4% | 64,549 | 8.5% | 55,568 | 26.9% | | 91,337 | 24.6% | 45,675 | 24.6% | 45,662 | 24.6% | |
| ≥81 | 61,892 | 6.4% | 34,191 | 4.5% | 27,701 | 13.4% | | 47,126 | 12.7% | 23,565 | 12.7% | 23,561 | 12.7% | |
| **Gender** | | | | | | | <0.001 | | | | | | | 0.08 |
| Female | 573,610 | 59.2% | 454,050 | 59.6% | 119,560 | 58.0% | | 215,375 | 58.0% | 107,420 | 57.9% | 107,955 | 58.2% | |
| Male | 394,686 | 40.8% | 307,945 | 40.4% | 86,741 | 42.0% | | 155,697 | 42.0% | 78,116 | 42.1% | 77,581 | 41.8% | |
| **Insurance** | | | | | | | <0.001 | | | | | | | 0.57 |
| Commercial | 500,918 | 51.7% | 442,990 | 58.1% | 57,928 | 28.1% | | 116,123 | 31.3% | 58,206 | 31.4% | 57,917 | 31.2% | |
| Dual | 6,814 | 0.7% | 2,410 | 0.3% | 4,404 | 2.1% | | 4,447 | 1.2% | 2,190 | 1.2% | 2,257 | 1.2% | |
| Medicaid | 252,366 | 26.1% | 206,109 | 27.0% | 46,257 | 22.4% | | 83,550 | 22.5% | 41,703 | 22.5% | 41,847 | 22.6% | |
| Medicare | 208,198 | 21.5% | 110,486 | 14.5% | 97,712 | 47.4% | | 166,952 | 45.0% | 83,437 | 45.0% | 83,515 | 45.0% | |
| **PCP Visit 2019** | | | | | | | <0.001 | | | | | | | 0.20 |
| No | 521,282 | 53.8% | 446,929 | 58.7% | 74,353 | 36.0% | | 146,967 | 39.6% | 73,675 | 39.7% | 73,292 | 39.5% | |
| Yes | 447,014 | 46.2% | 315,066 | 41.3% | 131,948 | 64.0% | | 224,105 | 60.4% | 111,861 | 60.3% | 112,244 | 60.5% | |

**Continuous Outcomes**

| | mean | SD | mean | SD | mean | SD | p-value | mean | SD | mean | SD | mean | SD | p-value |
| --- | --- | --- | --- | --- | --- | --- | --- | --- | --- | --- | --- | --- | --- | --- |
| CCI | 0.65 | 1.39 | 0.51 | 1.24 | 1.17 | 1.77 | <0.001 | 1.07 | 1.73 | 1.08 | 1.76 | 1.06 | 1.70 | <0.001 |

CCI: Charlson Comorbidity Index; PCP: primary care physician; SD: standard deviation.

**Appendix 2—table 26.** Statin User Cohort (All Regions) by BP Use, Patient Characteristics Pre/Post Match of BP Users/Non-users.

| | All Statin Users by BP: Unmatched | | | | | | | All Statin Users by BP: Matched | | | | | | |
| | All | | BP Non-user | | BP User | | p-value | All | | BP Non-user | | BP User | | p-value |
| | N | % | N | % | N | % | | N | % | N | % | N | % | |
|---|---|---|---|---|---|---|---|---|---|---|---|---|---|---|
| **All Patients** | 1,436,300 | 100.0% | 1,218,319 | 84.8% | 217,981 | 15.2% | | 426,960 | 100.0% | 213,480 | 50.0% | 213,480 | 50.0% | |
| **Age** | | | | | | | | | | | | | | |
| ≤20 | 1,385 | 0.1% | 1,365 | 0.1% | 20 | 0.0% | <0.001 | 42 | 0.0% | 22 | 0.0% | 20 | 0.0% | 1.00 |
| 21-40 | 44,389 | 3.1% | 44,042 | 3.6% | 347 | 0.2% | | 704 | 0.2% | 357 | 0.2% | 347 | 0.2% | |
| 41-50 | 135,867 | 9.5% | 133,850 | 11.0% | 2,017 | 0.9% | | 4,033 | 0.9% | 2,016 | 0.9% | 2,017 | 0.9% | |
| 51-60 | 356,032 | 24.8% | 333,325 | 27.4% | 22,707 | 10.4% | | 45,439 | 10.6% | 22,732 | 10.6% | 22,707 | 10.6% | |
| 61-70 | 428,943 | 29.9% | 356,208 | 29.2% | 72,735 | 33.4% | | 144,861 | 33.9% | 72,341 | 33.9% | 72,520 | 34.0% | |
| 71-80 | 302,081 | 21.0% | 223,651 | 18.4% | 78,430 | 36.0% | | 150,527 | 35.3% | 75,316 | 35.3% | 75,211 | 35.2% | |
| ≥81 | 167,603 | 11.7% | 125,878 | 10.3% | 41,725 | 19.1% | | 81,354 | 19.1% | 40,696 | 19.1% | 40,658 | 19.0% | |
| **Gender** | | | | | | | | | | | | | | |
| Female | 843,147 | 58.7% | 646,846 | 53.1% | 196,301 | 90.1% | <0.001 | 383,586 | 89.8% | 191,786 | 89.8% | 191,800 | 89.8% | 0.94 |
| Male | 593,153 | 41.3% | 571,473 | 46.9% | 21,680 | 9.9% | | 43,374 | 10.2% | 21,694 | 10.2% | 21,680 | 10.2% | |
| **Region** | | | | | | | | | | | | | | |
| Midwest | 271,319 | 18.9% | 237,718 | 19.5% | 33,601 | 15.4% | <0.001 | 67,050 | 15.7% | 33,525 | 15.7% | 33,525 | 15.7% | 1.00 |
| Northeast | 423,934 | 29.5% | 366,936 | 30.1% | 56,998 | 26.1% | | 113,308 | 26.5% | 56,654 | 26.5% | 56,654 | 26.5% | |
| South | 523,112 | 36.4% | 442,996 | 36.4% | 80,116 | 36.8% | | 157,838 | 37.0% | 78,919 | 37.0% | 78,919 | 37.0% | |
| West | 217,935 | 15.2% | 170,669 | 14.0% | 47,266 | 21.7% | | 88,764 | 20.8% | 44,382 | 20.8% | 44,382 | 20.8% | |
| **Insurance** | | | | | | | | | | | | | | |
| Commercial | 587,625 | 40.9% | 533,843 | 43.8% | 53,782 | 24.7% | <0.001 | 107,552 | 25.2% | 53,774 | 25.2% | 53,778 | 25.2% | 1.00 |
| Dual | 55,356 | 3.9% | 42,041 | 3.5% | 13,315 | 6.1% | | 24,380 | 5.7% | 12,183 | 5.7% | 12,197 | 5.7% | |
| Medicaid | 320,911 | 22.3% | 280,799 | 23.0% | 40,112 | 18.4% | | 76,121 | 17.8% | 38,050 | 17.8% | 38,071 | 17.8% | |
| Medicare | 472,408 | 32.9% | 361,636 | 29.7% | 110,772 | 50.8% | | 218,907 | 51.3% | 109,473 | 51.3% | 109,434 | 51.3% | |
| **PCP Visit 2019** | | | | | | | | | | | | | | |
| No | 507,726 | 35.3% | 430,446 | 35.3% | 77,280 | 35.5% | 0.27 | 151,395 | 35.5% | 75,614 | 35.4% | 75,781 | 35.5% | 0.59 |
| Yes | 928,574 | 64.7% | 787,873 | 64.7% | 140,701 | 64.5% | | 275,565 | 64.5% | 137,866 | 64.6% | 137,699 | 64.5% | |

| **Continuous Outcomes** | | | | | | | | | | | | | | |
| | mean | SD | mean | SD | mean | SD | p-value | mean | SD | mean | SD | mean | SD | p-value |
|---|---|---|---|---|---|---|---|---|---|---|---|---|---|---|
| CCI | 1.11 | 1.75 | 1.13 | 1.77 | 0.95 | 1.66 | <0.001 | 0.97 | 1.66 | 0.97 | 1.66 | 0.97 | 1.67 | 0.79 |

BP: bisphosphonate; CCI: Charlson Comorbidity Index; PCP: primary care physician; SD: standard deviation.

**Appendix 2—table 27.** Statin User Cohort (Region=New York State) by BP Use, Patient Characteristics Pre/Post Match of BP Users/Non-users.

| | Region=NY Statin Users by BP: Unmatched | | | | | | | Region=NY Statin Users by BP: Matched | | | | | | |
| | All | | BP Non-user | | BP User | | p-value | All | | BP Non-user | | BP User | | p-value |
| | N | % | N | % | N | % | | N | % | N | % | N | % | |
| All Patients | 185,536 | 100.0% | 161,673 | 87.1% | 23,863 | 12.9% | | 47,472 | 100.0% | 23,736 | 50.0% | 23,736 | 50.0% | |
| **Age** | | | | | | | | | | | | | | |
| ≤20 | 67 | 0.0% | 67 | 0.0% | 0 | 0.0% | <0.001 | 52 | 0.1% | 26 | 0.1% | 26 | 0.1% | 1.00 |
| 21-40 | 4,511 | 2.4% | 4,485 | 2.8% | 26 | 0.1% | | 304 | 0.6% | 152 | 0.6% | 152 | 0.6% | |
| 41-50 | 15,440 | 8.3% | 15,288 | 9.5% | 152 | 0.6% | | 4,381 | 9.2% | 2,192 | 9.2% | 2,189 | 9.2% | |
| 51-60 | 43,068 | 23.2% | 40,879 | 25.3% | 2,189 | 9.2% | | 14,717 | 31.0% | 7,358 | 31.0% | 7,359 | 31.0% | |
| 61-70 | 53,227 | 28.7% | 45,861 | 28.4% | 7,366 | 30.9% | | 18,189 | 38.3% | 9,092 | 38.3% | 9,097 | 38.3% | |
| 71-80 | 45,662 | 24.6% | 36,474 | 22.6% | 9,188 | 38.5% | | 9,829 | 20.7% | 4,916 | 20.7% | 4,913 | 20.7% | |
| ≥81 | 23,561 | 12.7% | 18,619 | 11.5% | 4,942 | 20.7% | | 0 | 0.0% | | 0.0% | | 0.0% | |
| **Gender** | | | | | | | | | | | | | | |
| Female | 107,955 | 58.2% | 86,194 | 53.3% | 21,761 | 91.2% | <0.001 | 43,265 | 91.1% | 21,631 | 91.1% | 21,634 | 91.1% | 0.96 |
| Male | 77,581 | 41.8% | 75,479 | 46.7% | 2,102 | 8.8% | | 4,207 | 8.9% | 2,105 | 8.9% | 2,102 | 8.9% | |
| **Insurance** | | | | | | | | | | | | | | |
| Commercial | 57,917 | 31.2% | 54,411 | 33.7% | 3,506 | 14.7% | <0.001 | 7,008 | 14.8% | 3,502 | 14.8% | 3,506 | 14.8% | 1.00 |
| Dual | 2,257 | 1.2% | 1,664 | 1.0% | 593 | 2.5% | | 1,128 | 2.4% | 564 | 2.4% | 564 | 2.4% | |
| Medicaid | 41,847 | 22.6% | 37,926 | 23.5% | 3,921 | 16.4% | | 7,644 | 16.1% | 3,821 | 16.1% | 3,823 | 16.1% | |
| Medicare | 83,515 | 45.0% | 67,672 | 41.9% | 15,843 | 66.4% | | 31,692 | 66.8% | 15,849 | 66.8% | 15,843 | 66.7% | |
| **PCP Visit 2019** | | | | | | | | | | | | | | |
| No | 73,292 | 39.5% | 63,797 | 39.5% | 9,495 | 39.8% | 0.33 | 18,870 | 39.7% | 9,434 | 39.7% | 9,436 | 39.8% | 0.99 |
| Yes | 112,244 | 60.5% | 97,876 | 60.5% | 14,368 | 60.2% | | 28,602 | 60.3% | 14,302 | 60.3% | 14,300 | 60.2% | |
| **Continuous Outcomes** | | | | | | | | | | | | | | |
| | mean | SD | mean | SD | mean | SD | p-value | mean | SD | mean | SD | mean | SD | p-value |
| CCI | 1.06 | 1.70 | 1.08 | 1.71 | 0.92 | 1.59 | <0.001 | 0.92 | 1.58 | 0.92 | 1.57 | 0.93 | 1.59 | 0.64 |

BP: bisphosphonate; CCI: Charlson Comorbidity Index; PCP: primary care physician; SD: standard deviation.

**Appendix 2—table 28.** Statin Non-user Cohort (All Regions) by BP Use, Patient Characteristics Pre/Post Match of BP Users/Non-users.

### All Statin Non-users by BP Use: Unmatched

| | All N | All % | BP Non-users N | BP Non-users % | BP Users N | BP Users % | p-value |
|---|---|---|---|---|---|---|---|
| **All Patients** | 1,436,300 | 100.0% | 1,311,457 | 91.3% | 124,843 | 8.7% | |
| **Age** | | | | | | | |
| ≤20 | 1,387 | 0.1% | 1,383 | 0.1% | 4 | 0.0% | <0.001 |
| 21-40 | 44,371 | 3.1% | 44,170 | 3.4% | 201 | 0.2% | |
| 41-50 | 135,748 | 9.5% | 134,305 | 10.2% | 1,443 | 1.2% | |
| 51-60 | 354,449 | 24.7% | 336,779 | 25.7% | 17,670 | 14.2% | |
| 61-70 | 428,326 | 29.8% | 381,936 | 29.1% | 46,390 | 37.2% | |
| 71-80 | 303,279 | 21.1% | 264,157 | 20.1% | 39,122 | 31.3% | |
| ≥81 | 168,740 | 11.7% | 148,727 | 11.3% | 20,013 | 16.0% | |
| **Gender** | | | | | | | |
| Female | 839,207 | 58.4% | 727,324 | 55.5% | 111,883 | 89.6% | <0.001 |
| Male | 597,093 | 41.6% | 584,133 | 44.5% | 12,960 | 10.4% | |
| **Region** | | | | | | | |
| Midwest | 271,319 | 18.9% | 249,383 | 19.0% | 21,936 | 17.6% | <0.001 |
| Northeast | 423,934 | 29.5% | 390,134 | 29.7% | 33,800 | 27.1% | |
| South | 523,112 | 36.4% | 480,680 | 36.7% | 42,432 | 34.0% | |
| West | 217,935 | 15.2% | 191,260 | 14.6% | 26,675 | 21.4% | |
| **Insurance** | | | | | | | |
| Commercial | 587,847 | 40.9% | 552,487 | 42.1% | 35,360 | 28.3% | <0.001 |
| Dual | 54,851 | 3.8% | 46,371 | 3.5% | 8,480 | 6.8% | |
| Medicaid | 320,434 | 22.3% | 296,591 | 22.6% | 23,843 | 19.1% | |
| Medicare | 473,168 | 32.9% | 416,008 | 31.7% | 57,160 | 45.8% | |
| **PCP Visit 2019** | | | | | | | |
| No | 508,587 | 35.4% | 473,241 | 36.1% | 35,346 | 28.3% | <0.001 |
| Yes | 927,713 | 64.6% | 838,216 | 63.9% | 89,497 | 71.7% | |

| Continuous Outcomes | All mean | All SD | BP Non-users mean | BP Non-users SD | BP Users mean | BP Users SD | p-value |
|---|---|---|---|---|---|---|---|
| CCI | 1.12 | 1.79 | 1.13 | 1.79 | 1.02 | 1.86 | <0.001 |

### All Statin Non-users by BP: Matched

| | All N | All % | BP Non-users N | BP Non-users % | BP Users N | BP Users % | p-value |
|---|---|---|---|---|---|---|---|
| **All Patients** | 249,432 | 100.0% | 124,716 | 50.0% | 124,716 | 50.0% | |
| **Age** | | | | | | | |
| ≤20 | 6 | 0.0% | 2 | 0.0% | 4 | 0.0% | 0.99 |
| 21-40 | 413 | 0.2% | 212 | 0.2% | 201 | 0.2% | |
| 41-50 | 2,880 | 1.2% | 1,437 | 1.2% | 1,443 | 1.2% | |
| 51-60 | 35,335 | 14.2% | 17,665 | 14.2% | 17,670 | 14.2% | |
| 61-70 | 92,791 | 37.2% | 46,401 | 37.2% | 46,390 | 37.2% | |
| 71-80 | 78,077 | 31.3% | 39,037 | 31.3% | 39,040 | 31.3% | |
| ≥81 | 39,930 | 16.0% | 19,962 | 16.0% | 19,968 | 16.0% | |
| **Gender** | | | | | | | |
| Female | 223,501 | 89.6% | 111,745 | 89.6% | 111,756 | 89.6% | 0.94 |
| Male | 25,931 | 10.4% | 12,971 | 10.4% | 12,960 | 10.4% | |
| **Region** | | | | | | | |
| Midwest | 43,870 | 17.6% | 21,935 | 17.6% | 21,935 | 17.6% | 1.00 |
| Northeast | 67,594 | 27.1% | 33,797 | 27.1% | 33,797 | 27.1% | |
| South | 84,618 | 33.9% | 42,309 | 33.9% | 42,309 | 33.9% | |
| West | 53,350 | 21.4% | 26,675 | 21.4% | 26,675 | 21.4% | |
| **Insurance** | | | | | | | |
| Commercial | 70,725 | 28.4% | 35,365 | 28.4% | 35,360 | 28.4% | 1.00 |
| Dual | 16,696 | 6.7% | 8,342 | 6.7% | 8,354 | 6.7% | |
| Medicaid | 47,674 | 19.1% | 23,832 | 19.1% | 23,842 | 19.1% | |
| Medicare | 114,337 | 45.8% | 57,177 | 45.8% | 57,160 | 45.8% | |
| **PCP Visit 2019** | | | | | | | |
| No | 70,689 | 28.3% | 35,343 | 28.3% | 35,346 | 28.3% | 0.99 |
| Yes | 178,743 | 71.7% | 89,373 | 71.7% | 89,370 | 71.7% | |

| Continuous Outcomes | All mean | All SD | BP Non-users mean | BP Non-users SD | BP Users mean | BP Users SD | p-value |
|---|---|---|---|---|---|---|---|
| CCI | 1.02 | 1.85 | 1.02 | 1.84 | 1.02 | 1.86 | 0.49 |

BP: bisphosphonate; CCI: Charlson Comorbidity Index; PCP: primary care physician; SD: standard deviation.

**Appendix 2—table 29.** Statin Non-user Cohort (Region=New York State) by BP Use, Patient Characteristics Pre/Post Match of BP Users/Non-users.

| | Region=NY Statin Non-users by BP: Unmatched | | | | | | | Region=NY Statin Non-users by BP: Matched | | | | | | |
|---|---|---|---|---|---|---|---|---|---|---|---|---|---|---|
| | All | | BP Non-users | | BP Users | | p-value | All | | BP Non-users | | BP Users | | p-value |
| | N | % | N | % | N | % | | N | % | N | % | N | % | |
| **All Patients** | 185,536 | 100.0% | 170,990 | 92.2% | 14,546 | 7.8% | | 29,042 | 100.0% | 14,521 | 50.0% | 14,521 | 50.0% | |
| **Age** | | | | | | | | | | | | | | |
| ≤20 | 67 | 0.0% | 67 | 0.0% | 0 | 0.0% | <0.001 | 0 | 0.0% | 0 | 0.0% | 0 | 0.0% | 1.00 |
| 21-40 | 4,508 | 2.4% | 4,498 | 2.6% | 10 | 0.1% | | 23 | 0.1% | 13 | 0.1% | 10 | 0.1% | |
| 41-50 | 15,420 | 8.3% | 15,314 | 9.0% | 106 | 0.7% | | 211 | 0.7% | 105 | 0.7% | 106 | 0.7% | |
| 51-60 | 43,068 | 23.2% | 41,317 | 24.2% | 1,751 | 12.0% | | 3,502 | 12.1% | 1,751 | 12.1% | 1,751 | 12.1% | |
| 61-70 | 53,233 | 28.7% | 48,148 | 28.2% | 5,085 | 35.0% | | 10,174 | 35.0% | 5,089 | 35.0% | 5,085 | 35.0% | |
| 71-80 | 45,675 | 24.6% | 40,731 | 23.8% | 4,944 | 34.0% | | 9,877 | 34.0% | 4,937 | 34.0% | 4,940 | 34.0% | |
| ≥81 | 23,565 | 12.7% | 20,915 | 12.2% | 2,650 | 18.2% | | 5,255 | 18.1% | 2,626 | 18.1% | 2,629 | 18.1% | |
| **Gender** | | | | | | | | | | | | | | |
| Female | 107,420 | 57.9% | 94,242 | 55.1% | 13,178 | 90.6% | <0.001 | 26,304 | 90.6% | 13,151 | 90.6% | 13,153 | 90.6% | 0.97 |
| Male | 78,116 | 42.1% | 76,748 | 44.9% | 1,368 | 9.4% | | 2,738 | 9.4% | 1,370 | 9.4% | 1,368 | 9.4% | |
| **Insurance** | | | | | | | | | | | | | | |
| Commercial | 58,206 | 31.4% | 56,313 | 32.9% | 1,893 | 13.0% | <0.001 | 3,785 | 13.0% | 1,892 | 13.0% | 1,893 | 13.0% | 0.96 |
| Dual | 2,190 | 1.2% | 1,754 | 1.0% | 436 | 3.0% | | 883 | 3.0% | 449 | 3.1% | 434 | 3.0% | |
| Medicaid | 41,703 | 22.5% | 38,177 | 22.3% | 3,526 | 24.2% | | 6,994 | 24.1% | 3,491 | 24.0% | 3,503 | 24.1% | |
| Medicare | 83,437 | 45.0% | 74,746 | 43.7% | 8,691 | 59.7% | | 17,380 | 59.8% | 8,689 | 59.8% | 8,691 | 59.9% | |
| **PCP Visit 2019** | | | | | | | | | | | | | | |
| No | 73,675 | 39.7% | 69,382 | 40.6% | 4,293 | 29.5% | <0.001 | 8,564 | 29.5% | 4,280 | 29.5% | 4,284 | 29.5% | 0.96 |
| Yes | 111,861 | 60.3% | 101,608 | 59.4% | 10,253 | 70.5% | | 20,478 | 70.5% | 10,241 | 70.5% | 10,237 | 70.5% | |
| **Continuous Outcomes** | | | | | | | | | | | | | | |
| | mean | SD | mean | SD | mean | SD | p-value | mean | SD | mean | SD | mean | SD | p-value |
| CCI | 1.08 | 1.76 | 1.09 | 1.76 | 0.95 | 1.75 | <0.001 | 0.95 | 1.74 | 0.95 | 1.73 | 0.95 | 1.75 | 0.82 |

BP: bisphosphonate; CCI: Charlson Comorbidity Index; PCP: primary care physician; SD: standard deviation.

**Appendix 2—table 30.** Antihypertensive Cohort (All Regions), Patient Characteristics Pre/Post Match.

| | All Observations by Antihypertensive Use: Unmatched | | | | | | | All Observations by Antihypertensive Use: Matched | | | | | | |
| --- | --- | --- | --- | --- | --- | --- | --- | --- | --- | --- | --- | --- | --- | --- |
| | All | | HTN Non-users | | HTN Users | | p-value | All | | HTN Non-users | | HTN Users | | p-value |
| | N | % | N | % | N | % | | N | % | N | % | N | % | |
| **All Patients** | 7,906,603 | 100.0% | 5,805,483 | 73.4% | 2,101,120 | 26.6% | | 3,572,002 | 100.0% | 1,786,001 | 50.0% | 1,786,001 | 50.0% | |
| **Age** | | | | | | | | | | | | | | |
| ≤20 | 1,840,050 | 23.3% | 1,823,229 | 31.4% | 16,821 | 0.8% | <0.001 | 33,574 | 0.9% | 16,785 | 0.9% | 16,789 | 0.9% | 0.44 |
| 21-40 | 1,446,999 | 18.3% | 1,299,520 | 22.4% | 147,479 | 7.0% | | 293,445 | 8.2% | 146,712 | 8.2% | 146,733 | 8.2% | |
| 41-50 | 925,309 | 11.7% | 685,931 | 11.8% | 239,378 | 11.4% | | 463,130 | 13.0% | 231,312 | 13.0% | 231,818 | 13.0% | |
| 51-60 | 1,250,190 | 15.8% | 759,987 | 13.1% | 490,203 | 23.3% | | 870,549 | 24.4% | 434,995 | 24.4% | 435,554 | 24.4% | |
| 61-70 | 1,181,261 | 14.9% | 626,235 | 10.8% | 555,026 | 26.4% | | 918,823 | 25.7% | 459,192 | 25.7% | 459,631 | 25.7% | |
| 71-80 | 783,775 | 9.9% | 381,957 | 6.6% | 401,818 | 19.1% | | 619,578 | 17.3% | 309,898 | 17.4% | 309,680 | 17.3% | |
| ≥81 | 479,019 | 6.1% | 228,624 | 3.9% | 250,395 | 11.9% | | 372,903 | 10.4% | 187,107 | 10.5% | 185,796 | 10.4% | |
| **Gender** | | | | | | | | | | | | | | |
| Female | 4,670,960 | 59.1% | 3,402,357 | 58.6% | 1,268,603 | 60.4% | <0.001 | 2,159,365 | 60.5% | 1,079,468 | 60.4% | 1,079,897 | 60.5% | 0.64 |
| Male | 3,235,643 | 40.9% | 2,403,126 | 41.4% | 832,517 | 39.6% | | 1,412,637 | 39.5% | 706,533 | 39.6% | 706,104 | 39.5% | |
| **Region** | | | | | | | | | | | | | | |
| Midwest | 1,467,802 | 18.6% | 1,065,772 | 18.4% | 402,030 | 19.1% | <0.001 | 694,206 | 19.4% | 347,103 | 19.4% | 347,103 | 19.4% | 1.00 |
| Northeast | 2,152,560 | 27.2% | 1,568,239 | 27.0% | 584,321 | 27.8% | | 997,132 | 27.9% | 498,566 | 27.9% | 498,566 | 27.9% | |
| South | 3,042,604 | 38.5% | 2,240,163 | 38.6% | 802,441 | 38.2% | | 1,338,570 | 37.5% | 669,285 | 37.5% | 669,285 | 37.5% | |
| West | 1,243,637 | 15.7% | 931,309 | 16.0% | 312,328 | 14.9% | | 542,094 | 15.2% | 271,047 | 15.2% | 271,047 | 15.2% | |
| **Insurance** | | | | | | | | | | | | | | |
| Commercial | 3,938,603 | 49.8% | 3,060,354 | 52.7% | 878,249 | 41.8% | <0.001 | 1,695,516 | 47.5% | 848,106 | 47.5% | 847,410 | 47.4% | 0.80 |
| Dual | 156,497 | 2.0% | 55,827 | 1.0% | 100,670 | 4.8% | | 93,467 | 2.6% | 46,774 | 2.6% | 46,693 | 2.6% | |
| Medicaid | 2,594,500 | 32.8% | 2,091,349 | 36.0% | 503,151 | 23.9% | | 812,737 | 22.8% | 406,012 | 22.7% | 406,725 | 22.8% | |
| Medicare | 1,217,003 | 15.4% | 597,953 | 10.3% | 619,050 | 29.5% | | 970,282 | 27.2% | 485,109 | 27.2% | 485,173 | 27.2% | |
| **PCP Visit 2019** | | | | | | | | | | | | | | |
| No | 4,283,697 | 54.2% | 3,531,914 | 60.8% | 751,783 | 35.8% | <0.001 | 1,438,005 | 40.3% | 719,756 | 40.3% | 718,249 | 40.2% | 0.10 |
| Yes | 3,622,906 | 45.8% | 2,273,569 | 39.2% | 1,349,337 | 64.2% | | 2,133,997 | 59.7% | 1,066,245 | 59.7% | 1,067,752 | 59.8% | |
| **Continuous Outcomes** | | | | | | | | | | | | | | |
| | mean | SD | mean | SD | mean | SD | p-value | mean | SD | mean | SD | mean | SD | p-value |
| CCI | 0.62 | 1.38 | 0.43 | 1.14 | 1.13 | 1.80 | <0.001 | 0.95 | 1.65 | 0.96 | 1.66 | 0.95 | 1.64 | <0.05 |

CCI: Charlson Comorbidity Index; HTN: antihypertensive; PCP: primary care physician; SD: standard deviation.

**Appendix 2—table 31.** Antihypertensive Cohort (Region=New York State), Patient Characteristics Pre/Post Match.

| | Region=NY by Antihypertensive Use: Unmatched | | | | | | | Region=NY by Antihypertensive Use: Matched | | | | | | |
|---|---|---|---|---|---|---|---|---|---|---|---|---|---|---|
| | All | | HTN Non-users | | HTN Users | | p-value | All | | HTN Non-users | | HTN Users | | p-value |
| | N | % | N | % | N | % | | N | % | N | % | N | % | |
| All Patients | 968,296 | 100.0% | 709,644 | 73.3% | 258,652 | 26.7% | | 407,248 | 100.0% | 203,624 | 50.0% | 203,624 | 50.0% | |
| **Age** | | | | | | | | | | | | | | |
| ≤20 | 133,178 | 13.8% | 132,352 | 18.7% | 826 | 0.3% | <0.001 | 1,622 | 0.4% | 811 | 0.4% | 811 | 0.4% | 1.00 |
| 21-40 | 192,959 | 19.9% | 181,447 | 25.6% | 11,512 | 4.5% | | 22,930 | 5.6% | 11,465 | 5.6% | 11,465 | 5.6% | |
| 41-50 | 127,794 | 13.2% | 105,490 | 14.9% | 22,304 | 8.6% | | 43,846 | 10.8% | 21,923 | 10.8% | 21,923 | 10.8% | |
| 51-60 | 172,444 | 17.8% | 119,643 | 16.9% | 52,801 | 20.4% | | 96,318 | 23.7% | 48,159 | 23.7% | 48,159 | 23.7% | |
| 61-70 | 159,912 | 16.5% | 92,103 | 13.0% | 67,809 | 26.2% | | 109,858 | 27.0% | 54,929 | 27.0% | 54,929 | 27.0% | |
| 71-80 | 120,117 | 12.4% | 54,076 | 7.6% | 66,041 | 25.5% | | 88,734 | 21.8% | 44,367 | 21.8% | 44,367 | 21.8% | |
| ≥81 | 61,892 | 6.4% | 24,533 | 3.5% | 37,359 | 14.4% | | 43,940 | 10.8% | 21,970 | 10.8% | 21,970 | 10.8% | |
| **Gender** | | | | | | | | | | | | | | |
| Female | 573,610 | 59.2% | 419,901 | 59.2% | 153,709 | 59.4% | 0.02 | 240,930 | 59.2% | 120,465 | 59.2% | 120,465 | 59.2% | 1.00 |
| Male | 394,686 | 40.8% | 289,743 | 40.8% | 104,943 | 40.6% | | 166,318 | 40.8% | 83,159 | 40.8% | 83,159 | 40.8% | |
| **Insurance** | | | | | | | | | | | | | | |
| Commercial | 500,918 | 51.7% | 425,181 | 59.9% | 75,737 | 29.3% | <0.001 | 150,918 | 37.1% | 75,459 | 37.1% | 75,459 | 37.1% | 1.00 |
| Dual | 6,814 | 0.7% | 1,659 | 0.2% | 5,155 | 2.0% | | 2,986 | 0.7% | 1,493 | 0.7% | 1,493 | 0.7% | |
| Medicaid | 252,366 | 26.1% | 193,207 | 27.2% | 59,159 | 22.9% | | 95,032 | 23.3% | 47,516 | 23.3% | 47,516 | 23.3% | |
| Medicare | 208,198 | 21.5% | 89,597 | 12.6% | 118,601 | 45.9% | | 158,312 | 38.9% | 79,156 | 38.9% | 79,156 | 38.9% | |
| **PCP Visit 2019** | | | | | | | | | | | | | | |
| No | 521,282 | 53.8% | 423,952 | 59.7% | 97,330 | 37.6% | <0.001 | 181,234 | 44.5% | 90,617 | 44.5% | 90,617 | 44.5% | 1.00 |
| Yes | 447,014 | 46.2% | 285,692 | 40.3% | 161,322 | 62.4% | | 226,014 | 55.5% | 113,007 | 55.5% | 113,007 | 55.5% | |
| **Continuous Outcomes** | | | | | | | | | | | | | | |
| | mean | SD | mean | SD | mean | SD | p-value | mean | SD | mean | SD | mean | SD | p-value |
| CCI | 0.65 | 1.39 | 0.46 | 1.16 | 1.17 | 1.80 | <0.001 | 0.95 | 1.60 | 0.95 | 1.60 | 0.95 | 1.60 | 1.00 |

CCI: Charlson Comorbidity Index; HTN: antihypertensive; PCP: primary care physician; SD: standard deviation.

**Appendix 2—table 32.** Antihypertensive User Cohort (All Regions) by BP Use, Patient Characteristics Pre/Post Match of BP Users/Non-users.

### All Antihypertensive Users by BP: Unmatched

| | All N | All % | BP Non-user N | BP Non-user % | BP User N | BP User % | p-value |
|---|---|---|---|---|---|---|---|
| All Patients | 1,786,001 | 100.0% | 1,579,388 | 88.4% | 206,613 | 11.6% | |
| **Age** | | | | | | | |
| ≤20 | 16,789 | 0.9% | 16,586 | 1.1% | 203 | 0.1% | <0.001 |
| 21-40 | 146,733 | 8.2% | 145,872 | 9.2% | 861 | 0.4% | |
| 41-50 | 231,818 | 13.0% | 229,150 | 14.5% | 2,668 | 1.3% | |
| 51-60 | 435,554 | 24.4% | 413,155 | 26.2% | 22,399 | 10.8% | |
| 61-70 | 459,631 | 25.7% | 390,664 | 24.7% | 68,967 | 33.4% | |
| 71-80 | 309,680 | 17.3% | 237,749 | 15.1% | 71,931 | 34.8% | |
| ≥81 | 185,796 | 10.4% | 146,212 | 9.3% | 39,584 | 19.2% | |
| **Gender** | | | | | | | |
| Female | 1,079,897 | 60.5% | 894,472 | 56.6% | 185,425 | 89.7% | <0.001 |
| Male | 706,104 | 39.5% | 684,916 | 43.4% | 21,188 | 10.3% | |
| **Region** | | | | | | | |
| Midwest | 347,103 | 19.4% | 313,523 | 19.9% | 33,580 | 16.3% | <0.001 |
| Northeast | 498,566 | 27.9% | 444,828 | 28.2% | 53,738 | 26.0% | |
| South | 669,285 | 37.5% | 595,410 | 37.7% | 73,875 | 35.8% | |
| West | 271,047 | 15.2% | 225,627 | 14.3% | 45,420 | 22.0% | |
| **Insurance** | | | | | | | |
| Commercial | 847,410 | 47.4% | 787,519 | 49.9% | 59,891 | 29.0% | <0.001 |
| Dual | 46,693 | 2.6% | 37,153 | 2.4% | 9,540 | 4.6% | |
| Medicaid | 406,725 | 22.8% | 369,893 | 23.4% | 36,832 | 17.8% | |
| Medicare | 485,173 | 27.2% | 384,823 | 24.4% | 100,350 | 48.6% | |
| **PCP Visit 2019** | | | | | | | |
| No | 718,249 | 40.2% | 633,042 | 40.1% | 85,207 | 41.2% | <0.001 |
| Yes | 1,067,752 | 59.8% | 946,346 | 59.9% | 121,406 | 58.8% | |

**Continuous Outcomes**

| | All mean | All SD | BP Non-user mean | BP Non-user SD | BP User mean | BP User SD | p-value |
|---|---|---|---|---|---|---|---|
| CCI | 0.95 | 1.64 | 0.95 | 1.64 | 0.94 | 1.68 | 0.02 |

### All Antihypertensive Users by BP: Matched

| | All N | All % | BP Non-user N | BP Non-user % | BP User N | BP User % | p-value |
|---|---|---|---|---|---|---|---|
| All Patients | 408,792 | 100.0% | 204,396 | 50.0% | 204,396 | 50.0% | |
| **Age** | | | | | | | |
| ≤20 | 411 | 0.1% | 208 | 0.1% | 203 | 0.1% | 1.00 |
| 21-40 | 1,728 | 0.4% | 868 | 0.4% | 860 | 0.4% | |
| 41-50 | 5,333 | 1.3% | 2,667 | 1.3% | 2,666 | 1.3% | |
| 51-60 | 44,796 | 11.0% | 22,399 | 11.0% | 22,397 | 11.0% | |
| 61-70 | 137,730 | 33.7% | 68,862 | 33.7% | 68,868 | 33.7% | |
| 71-80 | 140,882 | 34.5% | 70,439 | 34.5% | 70,443 | 34.5% | |
| ≥81 | 77,912 | 19.1% | 38,953 | 19.1% | 38,959 | 19.1% | |
| **Gender** | | | | | | | |
| Female | 366,424 | 89.6% | 183,212 | 89.6% | 183,212 | 89.6% | 1.00 |
| Male | 42,368 | 10.4% | 21,184 | 10.4% | 21,184 | 10.4% | |
| **Region** | | | | | | | |
| Midwest | 67,058 | 16.4% | 33,529 | 16.4% | 33,529 | 16.4% | 1.00 |
| Northeast | 107,150 | 26.2% | 53,575 | 26.2% | 53,575 | 26.2% | |
| South | 146,890 | 35.9% | 73,445 | 35.9% | 73,445 | 35.9% | |
| West | 87,694 | 21.5% | 43,847 | 21.5% | 43,847 | 21.5% | |
| **Insurance** | | | | | | | |
| Commercial | 119,737 | 29.3% | 59,863 | 29.3% | 59,874 | 29.3% | 1.00 |
| Dual | 17,884 | 4.4% | 8,945 | 4.4% | 8,939 | 4.4% | |
| Medicaid | 70,769 | 17.3% | 35,387 | 17.3% | 35,382 | 17.3% | |
| Medicare | 200,402 | 49.0% | 100,201 | 49.0% | 100,201 | 49.0% | |
| **PCP Visit 2019** | | | | | | | |
| No | 168,255 | 41.2% | 84,128 | 41.2% | 84,127 | 41.2% | 1.00 |
| Yes | 240,537 | 58.8% | 120,268 | 58.8% | 120,269 | 58.8% | |

**Continuous Outcomes**

| | All mean | All SD | BP Non-user mean | BP Non-user SD | BP User mean | BP User SD | p-value |
|---|---|---|---|---|---|---|---|
| CCI | 0.95 | 1.67 | 0.95 | 1.67 | 0.95 | 1.68 | 0.68 |

BP: bisphosphonate; CCI: Charlson Comorbidity Index; PCP: primary care physician; SD: standard deviation.

**Appendix 2—table 33.** Antihypertensive User Cohort (Region=New York State) by BP Use, Patient Characteristics Pre/Post Match of BP Users/Non-users.

| | Region=NY Antihypertensive Users by BP: Unmatched | | | | | | | Region=NY Antihypertensive Users by BP: Matched | | | | | | |
| --- | --- | --- | --- | --- | --- | --- | --- | --- | --- | --- | --- | --- | --- | --- |
| | All | | BP Non-user | | BP User | | p-value | All | | BP Non-user | | BP User | | p-value |
| | N | % | N | % | N | % | | N | % | N | % | N | % | |
| All Patients | 203,624 | 100.0% | 182,411 | 89.6% | 21,213 | 10.4% | | 42,252 | 100.0% | 21,126 | 50.0% | 21,126 | 50.0% | |
| **Age** | | | | | | | | | | | | | | |
| ≤20 | 811 | 0.4% | 798 | 0.4% | 13 | 0.1% | <0.001 | 27 | 0.1% | 14 | 0.1% | 13 | 0.1% | 1.00 |
| 21-40 | 11,465 | 5.6% | 11,396 | 6.2% | 69 | 0.3% | | 137 | 0.3% | 68 | 0.3% | 69 | 0.3% | |
| 41-50 | 21,923 | 10.8% | 21,747 | 11.9% | 176 | 0.8% | | 354 | 0.8% | 178 | 0.8% | 176 | 0.8% | |
| 51-60 | 48,159 | 23.7% | 46,047 | 25.2% | 2,112 | 10.0% | | 4,218 | 10.0% | 2,108 | 10.0% | 2,110 | 10.0% | |
| 61-70 | 54,929 | 27.0% | 48,022 | 26.3% | 6,907 | 32.6% | | 13,804 | 32.7% | 6,902 | 32.7% | 6,902 | 32.7% | |
| 71-80 | 44,367 | 21.8% | 36,409 | 20.0% | 7,958 | 37.5% | | 15,777 | 37.3% | 7,886 | 37.3% | 7,891 | 37.4% | |
| ≥81 | 21,970 | 10.8% | 17,992 | 9.9% | 3,978 | 18.8% | | 7,935 | 18.8% | 3,970 | 18.8% | 3,965 | 18.8% | |
| **Gender** | | | | | | | | | | | | | | |
| Female | 120,465 | 59.2% | 101,190 | 55.5% | 19,275 | 90.9% | <0.001 | 38,380 | 90.8% | 19,190 | 90.8% | 19,190 | 90.8% | 1.00 |
| Male | 83,159 | 40.8% | 81,221 | 44.5% | 1,938 | 9.1% | | 3,872 | 9.2% | 1,936 | 9.2% | 1,936 | 9.2% | |
| **Insurance** | | | | | | | | | | | | | | |
| Commercial | 75,459 | 37.1% | 71,460 | 39.2% | 3,999 | 18.9% | <0.001 | 7,993 | 18.9% | 3,997 | 18.9% | 3,996 | 18.9% | 1.00 |
| Dual | 1,493 | 0.7% | 1,151 | 0.6% | 342 | 1.6% | | 643 | 1.5% | 322 | 1.5% | 321 | 1.5% | |
| Medicaid | 47,516 | 23.3% | 44,248 | 24.3% | 3,268 | 15.4% | | 6,414 | 15.2% | 3,207 | 15.2% | 3,207 | 15.2% | |
| Medicare | 79,156 | 38.9% | 65,552 | 35.9% | 13,604 | 64.1% | | 27,202 | 64.4% | 13,600 | 64.4% | 13,602 | 64.4% | |
| **PCP Visit 2019** | | | | | | | | | | | | | | |
| No | 90,617 | 44.5% | 80,739 | 44.3% | 9,878 | 46.6% | <0.001 | 19,672 | 46.6% | 9,837 | 46.6% | 9,835 | 46.6% | 0.98 |
| Yes | 113,007 | 55.5% | 101,672 | 55.7% | 11,335 | 53.4% | | 22,580 | 53.4% | 11,289 | 53.4% | 11,291 | 53.4% | |
| **Continuous Outcomes** | | | | | | | | | | | | | | |
| | mean | SD | mean | SD | mean | SD | p-value | mean | SD | mean | SD | mean | SD | p-value |
| CCI | 0.95 | 1.60 | 0.95 | 1.61 | 0.88 | 1.54 | <0.001 | 0.87 | 1.53 | 0.87 | 1.52 | 0.87 | 1.53 | 0.87 |

BP: bisphosphonate; CCI: Charlson Comorbidity Index; PCP: primary care physician; SD: standard deviation.

**Appendix 2—table 34.** Antihypertensive Non-user Cohort (All Regions) by BP Use, Patient Characteristics Pre/Post Match of BP Users/Non-users.

| | All Antihypertensive Non-users by BP: Unmatched | | | | | | | All Antihypertensive Non-users by BP: Matched | | | | | | |
| --- | --- | --- | --- | --- | --- | --- | --- | --- | --- | --- | --- | --- | --- | --- |
| | All | | BP Non-user | | BP User | | p-value | All | | BP Non-user | | BP User | | p-value |
| | N | % | N | % | N | % | | N | % | N | % | N | % | |
| All Patients | 1,786,001 | 100.0% | 1,649,985 | 92.4% | 136,016 | 7.6% | | 271,448 | 100.0% | 135,724 | 50.0% | 135,724 | 50.0% | 1.00 |
| **Age** | | | | | | | <0.001 | | | | | | | 1.00 |
| ≤20 | 16,785 | 0.9% | 16,767 | 1.0% | 18 | 0.0% | | 34 | 0.0% | 16 | 0.0% | 18 | 0.0% | |
| 21-40 | 146,712 | 8.2% | 146,210 | 8.9% | 502 | 0.4% | | 1,009 | 0.4% | 507 | 0.4% | 502 | 0.4% | |
| 41-50 | 231,312 | 13.0% | 228,725 | 13.9% | 2,587 | 1.9% | | 5,163 | 1.9% | 2,577 | 1.9% | 2,586 | 1.9% | |
| 51-60 | 434,995 | 24.4% | 410,636 | 24.9% | 24,359 | 17.9% | | 48,700 | 17.9% | 24,349 | 17.9% | 24,351 | 17.9% | |
| 61-70 | 459,192 | 25.7% | 404,445 | 24.5% | 54,747 | 40.3% | | 109,415 | 40.3% | 54,711 | 40.3% | 54,704 | 40.3% | |
| 71-80 | 309,898 | 17.4% | 271,617 | 16.5% | 38,281 | 28.1% | | 76,139 | 28.0% | 38,070 | 28.0% | 38,069 | 28.0% | |
| ≥81 | 187,107 | 10.5% | 171,585 | 10.4% | 15,522 | 11.4% | | 30,988 | 11.4% | 15,494 | 11.4% | 15,494 | 11.4% | |
| **Gender** | | | | | | | <0.001 | | | | | | | 0.93 |
| Female | 1,079,468 | 60.4% | 956,403 | 58.0% | 123,065 | 90.5% | | 245,537 | 90.5% | 122,762 | 90.4% | 122,775 | 90.5% | |
| Male | 706,533 | 39.6% | 693,582 | 42.0% | 12,951 | 9.5% | | 25,911 | 9.5% | 12,962 | 9.6% | 12,949 | 9.5% | |
| **Region** | | | | | | | <0.001 | | | | | | | 1.00 |
| Midwest | 347,103 | 19.4% | 321,267 | 19.5% | 25,836 | 19.0% | | 51,638 | 19.0% | 25,819 | 19.0% | 25,819 | 19.0% | |
| Northeast | 498,566 | 27.9% | 463,273 | 28.1% | 35,293 | 25.9% | | 70,544 | 26.0% | 35,272 | 26.0% | 35,272 | 26.0% | |
| South | 669,285 | 37.5% | 622,064 | 37.7% | 47,221 | 34.7% | | 93,980 | 34.6% | 46,990 | 34.6% | 46,990 | 34.6% | |
| West | 271,047 | 15.2% | 243,381 | 14.8% | 27,666 | 20.3% | | 55,286 | 20.4% | 27,643 | 20.4% | 27,643 | 20.4% | |
| **Insurance** | | | | | | | <0.001 | | | | | | | 1.00 |
| Commercial | 848,106 | 47.5% | 798,579 | 48.4% | 49,527 | 36.4% | | 99,039 | 36.5% | 49,523 | 36.5% | 49,516 | 36.5% | |
| Dual | 46,774 | 2.6% | 40,212 | 2.4% | 6,562 | 4.8% | | 12,645 | 4.7% | 6,319 | 4.7% | 6,326 | 4.7% | |
| Medicaid | 406,012 | 22.7% | 381,472 | 23.1% | 24,540 | 18.0% | | 49,025 | 18.1% | 24,516 | 18.1% | 24,509 | 18.1% | |
| Medicare | 485,109 | 27.2% | 429,722 | 26.0% | 55,387 | 40.7% | | 110,739 | 40.8% | 55,366 | 40.8% | 55,373 | 40.8% | |
| **PCP Visit 2019** | | | | | | | <0.001 | | | | | | | 1.00 |
| No | 719,756 | 40.3% | 676,255 | 41.0% | 43,501 | 32.0% | | 86,956 | 32.0% | 43,478 | 32.0% | 43,478 | 32.0% | |
| Yes | 1,066,245 | 59.7% | 973,730 | 59.0% | 92,515 | 68.0% | | 184,492 | 68.0% | 92,246 | 68.0% | 92,246 | 68.0% | |

**Continuous Outcomes**

| | mean | SD | mean | SD | mean | SD | p-value | mean | SD | mean | SD | mean | SD | p-value |
| --- | --- | --- | --- | --- | --- | --- | --- | --- | --- | --- | --- | --- | --- | --- |
| CCI | 0.96 | 1.66 | 0.96 | 1.65 | 0.88 | 1.76 | <0.001 | 0.88 | 1.75 | 0.88 | 1.74 | 0.88 | 1.75 | 0.76 |

BP: bisphosphonate; CCI: Charlson Comorbidity Index; PCP: primary care physician; SD: standard deviation.

**Appendix 2—table 35.** Antihypertensive Non-user Cohort (Region=New York State) by BP Use, Patient Characteristics Pre/Post Match of BP Users/Non-users.

| | Region=NY Antihypertensive Non-Users by BP: Unmatched | | | | | | | Region=NY Antihypertensive Non-users by BP: Matched | | | | | | |
| | All | | BP Non-user | | BP User | | p-value | All | | BP Non-user | | BP User | | p-value |
| | N | % | N | % | N | % | | N | % | N | % | N | % | |
| **All Patients** | 203,624 | 100.0% | 189,573 | 93.1% | 14,051 | 6.9% | | 27,966 | 100.0% | 13,983 | 50.0% | 13,983 | 50.0% | |
| **Age** | | | | | | | | | | | | | | |
| ≤20 | 811 | 0.4% | 810 | 0.4% | 1 | 0.0% | <0.001 | 2 | 0.0% | 1 | 0.0% | 1 | 0.0% | 1.00 |
| 21-40 | 11,465 | 5.6% | 11,451 | 6.0% | 14 | 0.1% | | 28 | 0.1% | 14 | 0.1% | 14 | 0.1% | |
| 41-50 | 21,923 | 10.8% | 21,762 | 11.5% | 161 | 1.1% | | 324 | 1.2% | 163 | 1.2% | 161 | 1.2% | |
| 51-60 | 48,159 | 23.7% | 46,035 | 24.3% | 2,124 | 15.1% | | 4,245 | 15.2% | 2,121 | 15.2% | 2,124 | 15.2% | |
| 61-70 | 54,929 | 27.0% | 49,409 | 26.1% | 5,520 | 39.3% | | 11,027 | 39.4% | 5,512 | 39.4% | 5,515 | 39.4% | |
| 71-80 | 44,367 | 21.8% | 39,789 | 21.0% | 4,578 | 32.6% | | 9,054 | 32.4% | 4,528 | 32.4% | 4,526 | 32.4% | |
| ≥81 | 21,970 | 10.8% | 20,317 | 10.7% | 1,653 | 11.8% | | 3,286 | 11.7% | 1,644 | 11.8% | 1,642 | 11.7% | |
| **Gender** | | | | | | | | | | | | | | |
| Female | 120,465 | 59.2% | 107,632 | 56.8% | 12,833 | 91.3% | <0.001 | 25,530 | 91.3% | 12,764 | 91.3% | 12,766 | 91.3% | 0.97 |
| Male | 83,159 | 40.8% | 81,941 | 43.2% | 1,218 | 8.7% | | 2,436 | 8.7% | 1,219 | 8.7% | 1,217 | 8.7% | |
| **Insurance** | | | | | | | | | | | | | | |
| Commercial | 75,459 | 37.1% | 73,115 | 38.6% | 2,344 | 16.7% | <0.001 | 4,683 | 16.7% | 2,342 | 16.7% | 2,341 | 16.7% | 1.00 |
| Dual | 1,493 | 0.7% | 1,211 | 0.6% | 282 | 2.0% | | 554 | 2.0% | 277 | 2.0% | 277 | 2.0% | |
| Medicaid | 47,516 | 23.3% | 43,809 | 23.1% | 3,707 | 26.4% | | 7,295 | 26.1% | 3,648 | 26.1% | 3,647 | 26.1% | |
| Medicare | 79,156 | 38.9% | 71,438 | 37.7% | 7,718 | 54.9% | | 15,434 | 55.2% | 7,716 | 55.2% | 7,718 | 55.2% | |
| **PCP Visit 2019** | | | | | | | | | | | | | | |
| No | 90,617 | 44.5% | 85,875 | 45.3% | 4,742 | 33.7% | <0.001 | 9,461 | 33.8% | 4,728 | 33.8% | 4,733 | 33.8% | 0.95 |
| Yes | 113,007 | 55.5% | 103,698 | 54.7% | 9,309 | 66.3% | | 18,505 | 66.2% | 9,255 | 66.2% | 9,250 | 66.2% | |
| **Continuous Outcomes** | | | | | | | | | | | | | | |
| | mean | SD | mean | SD | mean | SD | p-value | mean | SD | mean | SD | mean | SD | p-value |
| CCI | 0.95 | 1.60 | 0.96 | 1.60 | 0.81 | 1.60 | <0.001 | 0.81 | 1.59 | 0.81 | 1.58 | 0.81 | 1.59 | 0.92 |

BP: bisphosphonate; CCI: Charlson Comorbidity Index; PCP: primary care physician; SD: standard deviation.

**Appendix 2—table 36.** Antidiabetic Cohort (All Regions), Patient Characteristics Pre/Post Match.

| | All Observations by Antidiabetic Use: Unmatched | | | | | | | All Observations by Antidiabetic Use: Matched | | | | | | |
| --- | --- | --- | --- | --- | --- | --- | --- | --- | --- | --- | --- | --- | --- | --- |
| | All | | DIAB Non-users | | DIAB Users | | p-value | All | | DIAB Non-users | | DIAB Users | | p-value |
| | N | % | N | % | N | % | | N | % | N | % | N | % | |
| All Patients | 7,906,603 | 100.0% | 7,151,351 | 90.4% | 755,252 | 9.6% | | 1,509,106 | 100.0% | 754,553 | 50.0% | 754,553 | 50.0% | |
| **Age** | | | | | | | | | | | | | | |
| ≤20 | 1,840,050 | 23.3% | 1,833,838 | 25.6% | 6,212 | 0.8% | <0.001 | 12,422 | 0.8% | 6,211 | 0.8% | 6,211 | 0.8% | 1.00 |
| 21-40 | 1,446,999 | 18.3% | 1,389,243 | 19.4% | 57,756 | 7.6% | | 115,448 | 7.7% | 57,723 | 7.6% | 57,725 | 7.7% | |
| 41-50 | 925,309 | 11.7% | 833,333 | 11.7% | 91,976 | 12.2% | | 183,810 | 12.2% | 91,905 | 12.2% | 91,905 | 12.2% | |
| 51-60 | 1,250,190 | 15.8% | 1,058,878 | 14.8% | 191,312 | 25.3% | | 382,390 | 25.3% | 191,196 | 25.3% | 191,194 | 25.3% | |
| 61-70 | 1,181,261 | 14.9% | 973,670 | 13.6% | 207,591 | 27.5% | | 414,869 | 27.5% | 207,435 | 27.5% | 207,434 | 27.5% | |
| 71-80 | 783,775 | 9.9% | 645,256 | 9.0% | 138,519 | 18.3% | | 276,619 | 18.3% | 138,310 | 18.3% | 138,309 | 18.3% | |
| ≥81 | 479,019 | 6.1% | 417,133 | 5.8% | 61,886 | 8.2% | | 123,548 | 8.2% | 61,773 | 8.2% | 61,775 | 8.2% | |
| **Gender** | | | | | | | | | | | | | | |
| Female | 4,670,960 | 59.1% | 4,212,086 | 58.9% | 458,874 | 60.8% | <0.001 | 916,914 | 60.8% | 458,455 | 60.8% | 458,459 | 60.8% | 0.99 |
| Male | 3,235,643 | 40.9% | 2,939,265 | 41.1% | 296,378 | 39.2% | | 592,192 | 39.2% | 296,098 | 39.2% | 296,094 | 39.2% | |
| **Region** | | | | | | | | | | | | | | |
| Midwest | 1,467,802 | 18.6% | 1,333,631 | 18.6% | 134,171 | 17.8% | <0.001 | 268,044 | 17.8% | 134,022 | 17.8% | 134,022 | 17.8% | 1.00 |
| Northeast | 2,152,560 | 27.2% | 1,935,311 | 27.1% | 217,249 | 28.8% | | 434,080 | 28.8% | 217,040 | 28.8% | 217,040 | 28.8% | |
| South | 3,042,604 | 38.5% | 2,752,618 | 38.5% | 289,986 | 38.4% | | 579,562 | 38.4% | 289,781 | 38.4% | 289,781 | 38.4% | |
| West | 1,243,637 | 15.7% | 1,129,791 | 15.8% | 113,846 | 15.1% | | 227,420 | 15.1% | 113,710 | 15.1% | 113,710 | 15.1% | |
| **Insurance** | | | | | | | | | | | | | | |
| Commercial | 3,938,603 | 49.8% | 3,631,514 | 50.8% | 307,089 | 40.7% | <0.001 | 614,045 | 40.7% | 307,022 | 40.7% | 307,023 | 40.7% | 1.00 |
| Dual | 156,497 | 2.0% | 113,496 | 1.6% | 43,001 | 5.7% | | 85,209 | 5.6% | 42,603 | 5.6% | 42,606 | 5.6% | |
| Medicaid | 2,594,500 | 32.8% | 2,387,519 | 33.4% | 206,981 | 27.4% | | 413,743 | 27.4% | 206,875 | 27.4% | 206,868 | 27.4% | |
| Medicare | 1,217,003 | 15.4% | 1,018,822 | 14.2% | 198,181 | 26.2% | | 396,109 | 26.2% | 198,053 | 26.2% | 198,056 | 26.2% | |
| **PCP Visit 2019** | | | | | | | | | | | | | | |
| No | 4,283,697 | 54.2% | 4,030,804 | 56.4% | 252,893 | 33.5% | <0.001 | 505,500 | 33.5% | 252,752 | 33.5% | 252,748 | 33.5% | 0.99 |
| Yes | 3,622,906 | 45.8% | 3,120,547 | 43.6% | 502,359 | 66.5% | | 1,003,606 | 66.5% | 501,801 | 66.5% | 501,805 | 66.5% | |
| **Continuous Outcomes** | mean | SD | mean | SD | mean | SD | p-value | mean | SD | mean | SD | mean | SD | p-value |
| CCI | 0.62 | 1.38 | 0.55 | 1.30 | 1.25 | 1.84 | <0.001 | 1.24 | 1.82 | 1.24 | 1.82 | 1.24 | 1.82 | 0.99 |

CCI: Charlson Comorbidity Index; DIAB: antidiabetic; PCP: primary care physician; SD: standard deviation.

**Appendix 2—table 37.** Antidiabetic Cohort (Region=New York State), Patient Characteristics Pre/Post Match.

| | Region=NY by Antidiabetic Use: Unmatched | | | | | | | | Region=NY by Antidiabetic Use: Matched | | | | | | | |
| --- | --- | --- | --- | --- | --- | --- | --- | --- | --- | --- | --- | --- | --- | --- | --- | --- |
| | All | | DIAB Non-users | | DIAB Users | | p-value | | All | | DIAB Non-users | | DIAB Users | | p-value |
| | N | % | N | % | N | % | | | N | % | N | % | N | % | |
| All Patients | 968,296 | 100.0% | 863,179 | 89.1% | 105,117 | 10.9% | | | 209,382 | 100.0% | 104,691 | 50.0% | 104,691 | 50.0% | |
| **Age** | | | | | | | | | | | | | | | |
| ≤20 | 133,178 | 13.8% | 132,723 | 15.4% | 455 | 0.4% | <0.001 | | 910 | 0.4% | 455 | 0.4% | 455 | 0.4% | 1.00 |
| 21-40 | 192,959 | 19.9% | 186,785 | 21.6% | 6,174 | 5.9% | | | 12,328 | 5.9% | 6,164 | 5.9% | 6,164 | 5.9% | |
| 41-50 | 127,794 | 13.2% | 117,342 | 13.6% | 10,452 | 9.9% | | | 20,880 | 10.0% | 10,440 | 10.0% | 10,440 | 10.0% | |
| 51-60 | 172,444 | 17.8% | 148,040 | 17.2% | 24,404 | 23.2% | | | 48,735 | 23.3% | 24,369 | 23.3% | 24,366 | 23.3% | |
| 61-70 | 159,912 | 16.5% | 130,968 | 15.2% | 28,944 | 27.5% | | | 57,638 | 27.5% | 28,819 | 27.5% | 28,819 | 27.5% | |
| 71-80 | 120,117 | 12.4% | 95,621 | 11.1% | 24,496 | 23.3% | | | 48,625 | 23.2% | 24,311 | 23.2% | 24,314 | 23.2% | |
| ≥81 | 61,892 | 6.4% | 51,700 | 6.0% | 10,192 | 9.7% | | | 20,266 | 9.7% | 10,133 | 9.7% | 10,133 | 9.7% | |
| **Gender** | | | | | | | | | | | | | | | |
| Female | 573,610 | 59.2% | 512,889 | 59.4% | 60,721 | 57.8% | <0.001 | | 120,937 | 57.8% | 60,467 | 57.8% | 60,470 | 57.8% | 0.99 |
| Male | 394,686 | 40.8% | 350,290 | 40.6% | 44,396 | 42.2% | | | 88,445 | 42.2% | 44,224 | 42.2% | 44,221 | 42.2% | |
| **Insurance** | | | | | | | | | | | | | | | |
| Commercial | 500,918 | 51.7% | 468,804 | 54.3% | 32,114 | 30.6% | <0.001 | | 64,200 | 30.7% | 32,100 | 30.7% | 32,100 | 30.7% | 1.00 |
| Dual | 6,814 | 0.7% | 4,408 | 0.5% | 2,406 | 2.3% | | | 4,389 | 2.1% | 2,196 | 2.1% | 2,193 | 2.1% | |
| Medicaid | 252,366 | 26.1% | 224,334 | 26.0% | 28,032 | 26.7% | | | 55,853 | 26.7% | 27,925 | 26.7% | 27,928 | 26.7% | |
| Medicare | 208,198 | 21.5% | 165,633 | 19.2% | 42,565 | 40.5% | | | 84,940 | 40.6% | 42,470 | 40.6% | 42,470 | 40.6% | |
| **PCP Visit 2019** | | | | | | | | | | | | | | | |
| No | 521,282 | 53.8% | 484,071 | 56.1% | 37,211 | 35.4% | <0.001 | | 74,215 | 35.4% | 37,106 | 35.4% | 37,109 | 35.4% | 0.99 |
| Yes | 447,014 | 46.2% | 379,108 | 43.9% | 67,906 | 64.6% | | | 135,167 | 64.6% | 67,585 | 64.6% | 67,582 | 64.6% | |

| | Region=NY by Antidiabetic Use: Unmatched | | | | | | | | Region=NY by Antidiabetic Use: Matched | | | | | | | |
| --- | --- | --- | --- | --- | --- | --- | --- | --- | --- | --- | --- | --- | --- | --- | --- | --- |
| **Continuous Outcomes** | All | | DIAB Non-users | | DIAB Users | | p-value | | All | | DIAB Non-users | | DIAB Users | | p-value |
| | mean | SD | mean | SD | mean | SD | | | mean | SD | mean | SD | mean | SD | |
| CCI | 0.65 | 1.39 | 0.56 | 1.30 | 1.34 | 1.84 | <0.001 | | 1.32 | 1.79 | 1.32 | 1.79 | 1.32 | 1.79 | 0.98 |

CCI: Charlson Comorbidity Index; DIAB: antidiabetic; PCP: primary care physician; SD: standard deviation.

**Appendix 2—table 38.** Antidiabetic User Cohort (All Regions) by BP Use, Patient Characteristics Pre/Post Match of BP Users/Non users.

| | All Antidiabetic Users by BP: Unmatched | | | | | | | All Antidiabetic Users by BP: Matched | | | | | | |
| | All | | BP Non-user | | BP User | | p-value | All | | BP Non-user | | BP User | | p-value |
| | N | % | N | % | N | % | | N | % | N | % | N | % | |
| **All Patients** | 754,553 | 100.0% | 674,024 | 89.3% | 80,529 | 10.7% | | 159,000 | 100.0% | 79,500 | 50.0% | 79,500 | 50.0% | |
| **Age** | | | | | | | | | | | | | | |
| ≤20 | 6,211 | 0.8% | 6,169 | 0.9% | 42 | 0.1% | <0.001 | 83 | 0.1% | 41 | 0.1% | 42 | 0.1% | 1.00 |
| 21-40 | 57,725 | 7.7% | 57,535 | 8.5% | 190 | 0.2% | | 380 | 0.2% | 190 | 0.2% | 190 | 0.2% | |
| 41-50 | 91,905 | 12.2% | 90,952 | 13.5% | 953 | 1.2% | | 1,905 | 1.2% | 952 | 1.2% | 953 | 1.2% | |
| 51-60 | 191,194 | 25.3% | 182,922 | 27.1% | 8,272 | 10.3% | | 16,536 | 10.4% | 8,268 | 10.4% | 8,268 | 10.4% | |
| 61-70 | 207,434 | 27.5% | 180,895 | 26.8% | 26,539 | 33.0% | | 53,028 | 33.4% | 26,512 | 33.3% | 26,516 | 33.4% | |
| 71-80 | 138,309 | 18.3% | 107,467 | 15.9% | 30,842 | 38.3% | | 60,240 | 37.9% | 30,121 | 37.9% | 30,119 | 37.9% | |
| ≥81 | 61,775 | 8.2% | 48,084 | 7.1% | 13,691 | 17.0% | | 26,828 | 16.9% | 13,416 | 16.9% | 13,412 | 16.9% | |
| **Gender** | | | | | | | | | | | | | | |
| Female | 458,459 | 60.8% | 386,400 | 57.3% | 72,059 | 89.5% | <0.001 | 142,068 | 89.4% | 71,027 | 89.3% | 71,041 | 89.4% | 0.91 |
| Male | 296,094 | 39.2% | 287,624 | 42.7% | 8,470 | 10.5% | | 16,932 | 10.6% | 8,473 | 10.7% | 8,459 | 10.6% | |
| **Region** | | | | | | | | | | | | | | |
| Midwest | 134,022 | 17.8% | 123,909 | 18.4% | 10,113 | 12.6% | <0.001 | 20,168 | 12.7% | 10,084 | 12.7% | 10,084 | 12.7% | 1.00 |
| Northeast | 217,040 | 28.8% | 196,723 | 29.2% | 20,317 | 25.2% | | 40,446 | 25.4% | 20,223 | 25.4% | 20,223 | 25.4% | |
| South | 289,781 | 38.4% | 257,599 | 38.2% | 32,182 | 40.0% | | 63,740 | 40.1% | 31,870 | 40.1% | 31,870 | 40.1% | |
| West | 113,710 | 15.1% | 95,793 | 14.2% | 17,917 | 22.2% | | 34,646 | 21.8% | 17,323 | 21.8% | 17,323 | 21.8% | |
| **Insurance** | | | | | | | | | | | | | | |
| Commercial | 307,023 | 40.7% | 290,957 | 43.2% | 16,066 | 20.0% | <0.001 | 32,086 | 20.2% | 16,043 | 20.2% | 16,043 | 20.2% | 1.00 |
| Dual | 42,606 | 5.6% | 32,797 | 4.9% | 9,809 | 12.2% | | 18,653 | 11.7% | 9,321 | 11.7% | 9,332 | 11.7% | |
| Medicaid | 206,868 | 27.4% | 188,638 | 28.0% | 18,230 | 22.6% | | 35,513 | 22.3% | 17,759 | 22.3% | 17,754 | 22.3% | |
| Medicare | 198,056 | 26.2% | 161,632 | 24.0% | 36,424 | 45.2% | | 72,748 | 45.8% | 36,377 | 45.8% | 36,371 | 45.7% | |
| **PCP Visit 2019** | | | | | | | | | | | | | | |
| No | 252,748 | 33.5% | 228,203 | 33.9% | 24,545 | 30.5% | <0.001 | 48,374 | 30.4% | 24,184 | 30.4% | 24,190 | 30.4% | 0.97 |
| Yes | 501,805 | 66.5% | 445,821 | 66.1% | 55,984 | 69.5% | | 110,626 | 69.6% | 55,316 | 69.6% | 55,310 | 69.6% | |
| **Continuous Outcomes** | | | | | | | | | | | | | | |
| | mean | SD | mean | SD | mean | SD | p-value | mean | SD | mean | SD | mean | SD | p-value |
| CCI | 1.24 | 1.82 | 1.23 | 1.81 | 1.32 | 1.90 | <0.001 | 1.31 | 1.88 | 1.31 | 1.87 | 1.32 | 1.88 | 0.75 |

BP: bisphosphonate; CCI: Charlson Comorbidity Index; PCP: primary care physician; SD: standard deviation.

**Appendix 2—table 39.** Influences on exploratory choice including WASI scores.

| | Region=NY Antidiabetic Users by BP: Unmatched | | | | | | | Region=NY Antidiabetic Users by BP: Matched | | | | | | |
| | All | | BP Non-user | | BP User | | p-value | All | | BP Non-user | | BP User | | p-value |
| | N | % | N | % | N | % | | N | % | N | % | N | % | |
| **All Patients** | 104,691 | 100.0% | 95,162 | 90.9% | 9,529 | 9.1% | | 18,912 | 100.0% | 9,456 | 50.0% | 9,456 | 50.0% | |
| **Age** | | | | | | | | | | | | | | |
| ≤20 | 455 | 0.4% | 454 | 0.5% | 1 | 0.0% | <0.001 | 2 | 0.0% | 1 | 0.0% | 1 | 0.0% | 1.00 |
| 21-40 | 6,164 | 5.9% | 6,152 | 6.5% | 12 | 0.1% | | 25 | 0.1% | 13 | 0.1% | 12 | 0.1% | |
| 41-50 | 10,440 | 10.0% | 10,363 | 10.9% | 77 | 0.8% | | 151 | 0.8% | 75 | 0.8% | 76 | 0.8% | |
| 51-60 | 24,366 | 23.3% | 23,532 | 24.7% | 834 | 8.8% | | 1,665 | 8.8% | 831 | 8.8% | 834 | 8.8% | |
| 61-70 | 28,819 | 27.5% | 25,939 | 27.3% | 2,880 | 30.2% | | 5,741 | 30.4% | 2,870 | 30.4% | 2,871 | 30.4% | |
| 71-80 | 24,314 | 23.2% | 20,338 | 21.4% | 3,976 | 41.7% | | 7,880 | 41.7% | 3,941 | 41.7% | 3,939 | 41.7% | |
| ≥81 | 10,133 | 9.7% | 8,384 | 8.8% | 1,749 | 18.4% | | 3,448 | 18.2% | 1,725 | 18.2% | 1,723 | 18.2% | |
| **Gender** | | | | | | | | | | | | | | |
| Female | 60,470 | 57.8% | 51,884 | 54.5% | 8,586 | 90.1% | <0.001 | 17,022 | 90.0% | 8,509 | 90.0% | 8,513 | 90.0% | 0.92 |
| Male | 44,221 | 42.2% | 43,278 | 45.5% | 943 | 9.9% | | 1,890 | 10.0% | 947 | 10.0% | 943 | 10.0% | |
| **Insurance** | | | | | | | | | | | | | | |
| Commercial | 32,100 | 30.7% | 31,172 | 32.8% | 928 | 9.7% | <0.001 | 1,849 | 9.8% | 924 | 9.8% | 925 | 9.8% | 1.00 |
| Dual | 2,193 | 2.1% | 1,693 | 1.8% | 500 | 5.2% | | 978 | 5.2% | 490 | 5.2% | 488 | 5.2% | |
| Medicaid | 27,928 | 26.7% | 25,978 | 27.3% | 1,950 | 20.5% | | 3,793 | 20.1% | 1,897 | 20.1% | 1,896 | 20.1% | |
| Medicare | 42,470 | 40.6% | 36,319 | 38.2% | 6,151 | 64.6% | | 12,292 | 65.0% | 6,145 | 65.0% | 6,147 | 65.0% | |
| **PCP Visit 2019** | | | | | | | | | | | | | | |
| No | 37,109 | 35.4% | 33,894 | 35.6% | 3,215 | 33.7% | <.001 | 6,363 | 33.6% | 3,182 | 33.7% | 3,181 | 33.6% | 0.99 |
| Yes | 67,582 | 64.6% | 61,268 | 64.4% | 6,314 | 66.3% | | 12,549 | 66.4% | 6,274 | 66.3% | 6,275 | 66.4% | |
| **Continuous Outcomes** | | | | | | | | | | | | | | |
| | mean | SD | mean | SD | mean | SD | p-value | mean | SD | mean | SD | mean | SD | p-value |
| CCI | 1.32 | 1.79 | 1.31 | 1.79 | 1.46 | 1.87 | <0.001 | 1.44 | 1.83 | 1.44 | 1.82 | 1.45 | 1.84 | 0.75 |

BP: bisphosphonate; CCI: Charlson Comorbidity Index; PCP: primary care physician; SD: standard deviation.

**Appendix 2—table 40.** Antidiabetic Non-user Cohort (All Regions) by BP Use, Patient Characteristics Pre/Post Match of BP Users/Non-users.

| | All Antidiabetic Non-users by BP: Unmatched | | | | | | | All Antidiabetic Non-users by BP: Matched | | | | | | |
| --- | --- | --- | --- | --- | --- | --- | --- | --- | --- | --- | --- | --- | --- | --- |
| | All | | BP Non-user | | BP User | | p-value | All | | BP Non-user | | BP User | | p-value |
| | N | % | N | % | N | % | | N | % | N | % | N | % | |
| **All Patients** | 754,553 | 100.0% | 681,380 | 90.3% | 73,173 | 9.7% | | 145,028 | 100.0% | 72,514 | 50.0% | 72,514 | 50.0% | |
| **Age** | | | | | | | <0.001 | | | | | | | 1.00 |
| ≤20 | 6,211 | 0.8% | 6,199 | 0.9% | 12 | 0.0% | | 24 | 0.0% | 12 | 0.0% | 12 | 0.0% | |
| 21-40 | 57,723 | 7.6% | 57,497 | 8.4% | 226 | 0.3% | | 455 | 0.3% | 229 | 0.3% | 226 | 0.3% | |
| 41-50 | 91,905 | 12.2% | 90,693 | 13.3% | 1,212 | 1.7% | | 2,421 | 1.7% | 1,209 | 1.7% | 1,212 | 1.7% | |
| 51-60 | 191,196 | 25.3% | 180,332 | 26.5% | 10,864 | 14.8% | | 21,721 | 15.0% | 10,860 | 15.0% | 10,861 | 15.0% | |
| 61-70 | 207,435 | 27.5% | 180,825 | 26.5% | 26,610 | 36.4% | | 53,115 | 36.6% | 26,558 | 36.6% | 26,557 | 36.6% | |
| 71-80 | 138,310 | 18.3% | 114,018 | 16.7% | 24,292 | 33.2% | | 47,723 | 32.9% | 23,861 | 32.9% | 23,862 | 32.9% | |
| ≥81 | 61,773 | 8.2% | 51,816 | 7.6% | 9,957 | 13.6% | | 19,569 | 13.5% | 9,785 | 13.5% | 9,784 | 13.5% | |
| **Gender** | | | | | | | <0.001 | | | | | | | 0.91 |
| Female | 458,455 | 60.8% | 393,376 | 57.7% | 65,079 | 88.9% | | 128,836 | 88.8% | 64,411 | 88.8% | 64,425 | 88.8% | |
| Male | 296,098 | 39.2% | 288,004 | 42.3% | 8,094 | 11.1% | | 16,192 | 11.2% | 8,103 | 11.2% | 8,089 | 11.2% | |
| **Region** | | | | | | | <0.001 | | | | | | | 1.00 |
| Midwest | 134,022 | 17.8% | 123,283 | 18.1% | 10,739 | 14.7% | | 21,390 | 14.7% | 10,695 | 14.7% | 10,695 | 14.7% | |
| Northeast | 217,040 | 28.8% | 197,710 | 29.0% | 19,330 | 26.4% | | 38,510 | 26.6% | 19,255 | 26.6% | 19,255 | 26.6% | |
| South | 289,781 | 38.4% | 261,382 | 38.4% | 28,399 | 38.8% | | 55,812 | 38.5% | 27,906 | 38.5% | 27,906 | 38.5% | |
| West | 113,710 | 15.1% | 99,005 | 14.5% | 14,705 | 20.1% | | 29,316 | 20.2% | 14,658 | 20.2% | 14,658 | 20.2% | |
| **Insurance** | | | | | | | <0.001 | | | | | | | 1.00 |
| Commercial | 307,022 | 40.7% | 289,018 | 42.4% | 18,004 | 24.6% | | 35,983 | 24.8% | 17,988 | 24.8% | 17,995 | 24.8% | |
| Dual | 42,603 | 5.6% | 33,444 | 4.9% | 9,159 | 12.5% | | 17,221 | 11.9% | 8,611 | 11.9% | 8,610 | 11.9% | |
| Medicaid | 206,875 | 27.4% | 190,166 | 27.9% | 16,709 | 22.8% | | 33,264 | 22.9% | 16,636 | 22.9% | 16,628 | 22.9% | |
| Medicare | 198,053 | 26.2% | 168,752 | 24.8% | 29,301 | 40.0% | | 58,560 | 40.4% | 29,279 | 40.4% | 29,281 | 40.4% | |
| **PCP Visit 2019** | | | | | | | <0.001 | | | | | | | 0.97 |
| No | 252,752 | 33.5% | 233,775 | 34.3% | 18,977 | 25.9% | | 37,812 | 26.1% | 18,903 | 26.1% | 18,909 | 26.1% | |
| Yes | 501,801 | 66.5% | 447,605 | 65.7% | 54,196 | 74.1% | | 107,216 | 73.9% | 53,611 | 73.9% | 53,605 | 73.9% | |

**Continuous Outcomes**

| | All | | BP Non-user | | BP User | | p-value | All | | BP Non-user | | BP User | | p-value |
| --- | --- | --- | --- | --- | --- | --- | --- | --- | --- | --- | --- | --- | --- | --- |
| | mean | SD | mean | SD | mean | SD | | mean | SD | mean | SD | mean | SD | |
| CCI | 1.24 | 1.82 | 1.24 | 1.81 | 1.24 | 1.89 | 0.92 | 1.24 | 1.87 | 1.24 | 1.87 | 1.25 | 1.88 | 0.63 |

BP: bisphosphonate; CCI: Charlson Comorbidity Index; PCP: primary care physician; SD: standard deviation.

**Appendix 2—table 41.** Antidiabetic Non-user Cohort (Region=New York State) by BP Use, Patient Characteristics Pre/Post Match of BP Users/Non-users.

| | Region=NY Antidiabetic Non-users by BP: Unmatched | | | | | | | Region=NY Antidiabetic Non-users by BP: Matched | | | | | | |
|---|---|---|---|---|---|---|---|---|---|---|---|---|---|---|
| | All | | BP Non-user | | BP User | | p-value | All | | BP Non-user | | BP User | | p-value |
| | N | % | N | % | N | % | | N | % | N | % | N | % | |
| **All Patients** | 104,691 | 100.0% | 95,416 | 91.1% | 9,275 | 8.9% | | 18,288 | 100.0% | 9,144 | 50.0% | 9,144 | 50.0% | |
| **Age** | | | | | | | | | | | | | | |
| ≤20 | 455 | 0.4% | 455 | 0.5% | 0 | 0.0% | <0.001 | 0 | 0.0% | 0 | 0.0% | 0 | 0.0% | 1.00 |
| 21-40 | 6,164 | 5.9% | 6,146 | 6.4% | 18 | 0.2% | | 36 | 0.2% | 18 | 0.2% | 18 | 0.2% | |
| 41-50 | 10,440 | 10.0% | 10,367 | 10.9% | 73 | 0.8% | | 147 | 0.8% | 74 | 0.8% | 73 | 0.8% | |
| 51-60 | 24,369 | 23.3% | 23,304 | 24.4% | 1,065 | 11.5% | | 2,128 | 11.6% | 1,064 | 11.6% | 1,064 | 11.6% | |
| 61-70 | 28,819 | 27.5% | 25,720 | 27.0% | 3,099 | 33.4% | | 6,190 | 33.8% | 3,097 | 33.9% | 3,093 | 33.8% | |
| 71-80 | 24,311 | 23.2% | 20,826 | 21.8% | 3,485 | 37.6% | | 6,839 | 37.4% | 3,419 | 37.4% | 3,420 | 37.4% | |
| ≥81 | 10,133 | 9.7% | 8,598 | 9.0% | 1,535 | 16.5% | | 2,948 | 16.1% | 1,472 | 16.1% | 1,476 | 16.1% | |
| **Gender** | | | | | | | | | | | | | | |
| Female | 60,467 | 57.8% | 52,194 | 54.7% | 8,273 | 89.2% | <0.001 | 16,291 | 89.1% | 8,146 | 89.1% | 8,145 | 89.1% | 0.98 |
| Male | 44,224 | 42.2% | 43,222 | 45.3% | 1,002 | 10.8% | | 1,997 | 10.9% | 998 | 10.9% | 999 | 10.9% | |
| **Insurance** | | | | | | | | | | | | | | |
| Commercial | 32,100 | 30.7% | 31,095 | 32.6% | 1,005 | 10.8% | <0.001 | 2,002 | 10.9% | 1,000 | 10.9% | 1,002 | 11.0% | 1.00 |
| Dual | 2,196 | 2.1% | 1,675 | 1.8% | 521 | 5.6% | | 1,006 | 5.5% | 502 | 5.5% | 504 | 5.5% | |
| Medicaid | 27,925 | 26.7% | 25,530 | 26.8% | 2,395 | 25.8% | | 4,575 | 25.0% | 2,289 | 25.0% | 2,286 | 25.0% | |
| Medicare | 42,470 | 40.6% | 37,116 | 38.9% | 5,354 | 57.7% | | 10,705 | 58.5% | 5,353 | 58.5% | 5,352 | 58.5% | |
| **PCP Visit 2019** | | | | | | | | | | | | | | |
| No | 37,106 | 35.4% | 34,553 | 36.2% | 2,553 | 27.5% | <0.001 | 5,039 | 27.6% | 2,518 | 27.5% | 2,521 | 27.6% | 0.96 |
| Yes | 67,585 | 64.6% | 60,863 | 63.8% | 6,722 | 72.5% | | 13,249 | 72.4% | 6,626 | 72.5% | 6,623 | 72.4% | |
| **Continuous Outcomes** | | | | | | | | | | | | | | |
| | mean | SD | mean | SD | mean | SD | p-value | mean | SD | mean | SD | mean | SD | p-value |
| CCI | 1.32 | 1.79 | 1.32 | 1.79 | 1.37 | 1.81 | 0.007 | 1.37 | 1.78 | 1.36 | 1.78 | 1.37 | 1.79 | 0.92 |

BP: bisphosphonate; CCI: Charlson Comorbidity Index; PCP: primary care physician; SD: standard deviation.

**Appendix 2—table 42.** Antidepressant Cohort (All Regions), Patient Characteristics Pre/Post Match.

| | All Observations by Antidepressant Use: Unmatched | | | | | | | All Observations by Antidepressant Use: Matched | | | | | | |
| --- | --- | --- | --- | --- | --- | --- | --- | --- | --- | --- | --- | --- | --- | --- |
| | All | | DEPR Non-users | | DEPR Users | | p-value | All | | DEPR Non-users | | DEPR Users | | p-value |
| | N | % | N | % | N | % | | N | % | N | % | N | % | |
| **All Patients** | 7,906,603 | 100.0% | 6,335,598 | 80.1% | 1,571,005 | 19.9% | | 3,072,096 | 100.0% | 1,536,048 | 50.0% | 1,536,048 | 50.0% | 1.00 |
| **Age** | | | | | | | | | | | | | | |
| ≤20 | 1,840,050 | 23.3% | 1,750,435 | 27.6% | 89,615 | 5.7% | <0.001 | 179,136 | 5.8% | 89,565 | 5.8% | 89,571 | 5.8% | 1.00 |
| 21-40 | 1,446,999 | 18.3% | 1,128,316 | 17.8% | 318,683 | 20.3% | | 631,186 | 20.5% | 315,593 | 20.5% | 315,593 | 20.5% | |
| 41-50 | 925,309 | 11.7% | 683,455 | 10.8% | 241,854 | 15.4% | | 466,681 | 15.2% | 233,336 | 15.2% | 233,345 | 15.2% | |
| 51-60 | 1,250,190 | 15.8% | 899,512 | 14.2% | 350,678 | 22.3% | | 667,305 | 21.7% | 333,650 | 21.7% | 333,655 | 21.7% | |
| 61-70 | 1,181,261 | 14.9% | 879,560 | 13.9% | 301,701 | 19.2% | | 592,345 | 19.3% | 296,182 | 19.3% | 296,163 | 19.3% | |
| 71-80 | 783,775 | 9.9% | 613,922 | 9.7% | 169,853 | 10.8% | | 338,594 | 11.0% | 169,295 | 11.0% | 169,299 | 11.0% | |
| ≥81 | 479,019 | 6.1% | 380,398 | 6.0% | 98,621 | 6.3% | | 196,849 | 6.4% | 98,427 | 6.4% | 98,422 | 6.4% | |
| **Gender** | | | | | | | | | | | | | | |
| Female | 4,670,960 | 59.1% | 3,527,859 | 55.7% | 1,143,101 | 72.8% | <0.001 | 2,219,179 | 72.2% | 1,109,580 | 72.2% | 1,109,599 | 72.2% | 0.98 |
| Male | 3,235,643 | 40.9% | 2,807,739 | 44.3% | 427,904 | 27.2% | | 852,917 | 27.8% | 426,468 | 27.8% | 426,449 | 27.8% | |
| **Region** | | | | | | | | | | | | | | |
| Midwest | 1,467,802 | 18.6% | 1,120,969 | 17.7% | 346,833 | 22.1% | <0.001 | 671,016 | 21.8% | 335,508 | 21.8% | 335,508 | 21.8% | 1.00 |
| Northeast | 2,152,560 | 27.2% | 1,765,134 | 27.9% | 387,426 | 24.7% | | 766,046 | 24.9% | 383,023 | 24.9% | 383,023 | 24.9% | |
| South | 3,042,604 | 38.5% | 2,428,383 | 38.3% | 614,221 | 39.1% | | 1,192,058 | 38.8% | 596,029 | 38.8% | 596,029 | 38.8% | |
| West | 1,243,637 | 15.7% | 1,021,112 | 16.1% | 222,525 | 14.2% | | 442,976 | 14.4% | 221,488 | 14.4% | 221,488 | 14.4% | |
| **Insurance** | | | | | | | | | | | | | | |
| Commercial | 3,938,603 | 49.8% | 3,230,475 | 51.0% | 708,128 | 45.1% | <0.001 | 1,415,351 | 46.1% | 707,675 | 46.1% | 707,676 | 46.1% | 1.00 |
| Dual | 156,497 | 2.0% | 94,682 | 1.5% | 61,815 | 3.9% | | 109,676 | 3.6% | 54,836 | 3.6% | 54,840 | 3.6% | |
| Medicaid | 2,594,500 | 32.8% | 2,083,688 | 32.9% | 510,812 | 32.5% | | 972,897 | 31.7% | 486,446 | 31.7% | 486,451 | 31.7% | |
| Medicare | 1,217,003 | 15.4% | 926,753 | 14.6% | 290,250 | 18.5% | | 574,172 | 18.7% | 287,091 | 18.7% | 287,081 | 18.7% | |
| **PCP Visit 2019** | | | | | | | | | | | | | | |
| No | 4,283,697 | 54.2% | 3,672,879 | 58.0% | 610,818 | 38.9% | <0.001 | 1,210,520 | 39.4% | 605,256 | 39.4% | 605,264 | 39.4% | 0.99 |
| Yes | 3,622,906 | 45.8% | 2,662,719 | 42.0% | 960,187 | 61.1% | | 1,861,576 | 60.6% | 930,792 | 60.6% | 930,784 | 60.6% | |

**Continuous Outcomes**

| | All | | DEPR Non-users | | DEPR Users | | p-value | All | | DEPR Non-users | | DEPR Users | | p-value |
| --- | --- | --- | --- | --- | --- | --- | --- | --- | --- | --- | --- | --- | --- | --- |
| | mean | SD | mean | SD | mean | SD | | mean | SD | mean | SD | mean | SD | |
| CCI | 0.62 | 1.38 | 0.55 | 1.29 | 0.90 | 1.65 | <0.001 | 0.87 | 1.60 | 0.87 | 1.60 | 0.87 | 1.60 | 0.98 |

CCI: Charlson Comorbidity Index; DEPR: antidepressant; PCP: primary care physician; SD: standard deviation.

**Appendix 2—table 43.** Antidepressant Cohort (Region=New York State), Patient Characteristics Pre/Post Match.

| | Region=NY by Antidepressant Use: Unmatched | | | | | | | Region=NY by Antidepressant Use: Matched | | | | | | |
| --- | --- | --- | --- | --- | --- | --- | --- | --- | --- | --- | --- | --- | --- | --- |
| | All | | DEPR Non-users | | DEPR Users | | p-value | All | | DEPR Non-users | | DEPR Users | | p-value |
| | N | % | N | % | N | % | | N | % | N | % | N | % | |
| **All Patients** | 968,296 | 100.0% | 832,215 | 85.9% | 136,081 | 14.1% | | 271,032 | 100.0% | 135,516 | 50.0% | 135,516 | 50.0% | |
| **Age** | | | | | | | | | | | | | | |
| ≤20 | 133,178 | 13.8% | 128,810 | 15.5% | 4,368 | 3.2% | <0.001 | 8,728 | 3.2% | 4,365 | 3.2% | 4,363 | 3.2% | 1.00 |
| 21-40 | 192,959 | 19.9% | 170,076 | 20.4% | 22,883 | 16.8% | | 45,666 | 16.8% | 22,832 | 16.8% | 22,834 | 16.8% | |
| 41-50 | 127,794 | 13.2% | 109,184 | 13.1% | 18,610 | 13.7% | | 36,965 | 13.6% | 18,483 | 13.6% | 18,482 | 13.6% | |
| 51-60 | 172,444 | 17.8% | 142,702 | 17.1% | 29,742 | 21.9% | | 58,966 | 21.8% | 29,481 | 21.8% | 29,485 | 21.8% | |
| 61-70 | 159,912 | 16.5% | 132,317 | 15.9% | 27,595 | 20.3% | | 55,083 | 20.3% | 27,543 | 20.3% | 27,540 | 20.3% | |
| 71-80 | 120,117 | 12.4% | 99,040 | 11.9% | 21,077 | 15.5% | | 42,076 | 15.5% | 21,038 | 15.5% | 21,038 | 15.5% | |
| ≥81 | 61,892 | 6.4% | 50,086 | 6.0% | 11,806 | 8.7% | | 23,548 | 8.7% | 11,774 | 8.7% | 11,774 | 8.7% | |
| **Gender** | | | | | | | | | | | | | | |
| Female | 573,610 | 59.2% | 476,684 | 57.3% | 96,926 | 71.2% | <0.001 | 192,930 | 71.2% | 96,468 | 71.2% | 96,462 | 71.2% | 0.98 |
| Male | 394,686 | 40.8% | 355,531 | 42.7% | 39,155 | 28.8% | | 78,102 | 28.8% | 39,048 | 28.8% | 39,054 | 28.8% | |
| **Insurance** | | | | | | | | | | | | | | |
| Commercial | 500,918 | 51.7% | 449,071 | 54.0% | 51,847 | 38.1% | <0.001 | 103,658 | 38.2% | 51,829 | 38.2% | 51,829 | 38.2% | 1.00 |
| Dual | 6,814 | 0.7% | 5,072 | 0.6% | 1,742 | 1.3% | | 3,191 | 1.2% | 1,591 | 1.2% | 1,600 | 1.2% | |
| Medicaid | 252,366 | 26.1% | 213,705 | 25.7% | 38,661 | 28.4% | | 77,136 | 28.5% | 38,569 | 28.5% | 38,567 | 28.5% | |
| Medicare | 208,198 | 21.5% | 164,367 | 19.8% | 43,831 | 32.2% | | 87,047 | 32.1% | 43,527 | 32.1% | 43,520 | 32.1% | |
| **PCP Visit 2019** | | | | | | | | | | | | | | |
| No | 521,282 | 53.8% | 467,739 | 56.2% | 53,543 | 39.3% | <0.001 | 106,797 | 39.4% | 53,397 | 39.4% | 53,400 | 39.4% | 0.99 |
| Yes | 447,014 | 46.2% | 364,476 | 43.8% | 82,538 | 60.7% | | 164,235 | 60.6% | 82,119 | 60.6% | 82,116 | 60.6% | |
| **Continuous Outcomes** | | | | | | | | | | | | | | |
| | mean | SD | mean | SD | mean | SD | p-value | mean | SD | mean | SD | mean | SD | p-value |
| CCI | 0.65 | 1.39 | 0.59 | 1.32 | 0.98 | 1.71 | <0.001 | 0.96 | 1.68 | 0.96 | 1.68 | 0.96 | 1.68 | 0.99 |

CCI: Charlson Comorbidity Index; DEPR: antidepressant; PCP: primary care physician; SD: standard deviation.

**Appendix 2—table 44.** Antidepressant User Cohort (All Regions) by BP Use, Patient Characteristics Pre/Post Match of BP Users/Non-users.

| | All Antidepressant Users by BP: Unmatched | | | | | | | All Antidepressant Users by BP: Matched | | | | | | |
| | All | | BP Non-user | | BP User | | p-value | All | | BP Non-user | | BP User | | p-value |
| | N | % | N | % | N | % | | N | % | N | % | N | % | |
| **All Patients** | 1,536,048 | 100.0% | 1,390,939 | 90.6% | 145,109 | 9.4% | | 288,564 | 100.0% | 144,282 | 50.0% | 144,282 | 50.0% | |
| **Age** | | | | | | | | | | | | | | |
| ≤20 | 89,571 | 5.8% | 89,415 | 6.4% | 156 | 0.1% | <0.001 | 313 | 0.1% | 157 | 0.1% | 156 | 0.1% | 1.00 |
| 21-40 | 315,593 | 20.5% | 314,429 | 22.6% | 1,164 | 0.8% | | 2,326 | 0.8% | 1,162 | 0.8% | 1,164 | 0.8% | |
| 41-50 | 233,345 | 15.2% | 229,878 | 16.5% | 3,467 | 2.4% | | 6,933 | 2.4% | 3,467 | 2.4% | 3,466 | 2.4% | |
| 51-60 | 333,655 | 21.7% | 310,316 | 22.3% | 23,339 | 16.1% | | 46,674 | 16.2% | 23,339 | 16.2% | 23,335 | 16.2% | |
| 61-70 | 296,163 | 19.3% | 244,247 | 17.6% | 51,916 | 35.8% | | 103,798 | 36.0% | 51,905 | 36.0% | 51,893 | 36.0% | |
| 71-80 | 169,299 | 11.0% | 126,089 | 9.1% | 43,210 | 29.8% | | 85,292 | 29.6% | 42,643 | 29.6% | 42,649 | 29.6% | |
| ≥81 | 98,422 | 6.4% | 76,565 | 5.5% | 21,857 | 15.1% | | 43,228 | 15.0% | 21,609 | 15.0% | 21,619 | 15.0% | |
| **Gender** | | | | | | | | | | | | | | |
| Female | 1,109,599 | 72.2% | 976,214 | 70.2% | 133,385 | 91.9% | <0.001 | 265,123 | 91.9% | 132,553 | 91.9% | 132,570 | 91.9% | 0.91 |
| Male | 426,449 | 27.8% | 414,725 | 29.8% | 11,724 | 8.1% | | 23,441 | 8.1% | 11,729 | 8.1% | 11,712 | 8.1% | |
| **Region** | | | | | | | | | | | | | | |
| Midwest | 335,508 | 21.8% | 309,597 | 22.3% | 25,911 | 17.9% | <0.001 | 51,754 | 17.9% | 25,877 | 17.9% | 25,877 | 17.9% | 1.00 |
| Northeast | 383,023 | 24.9% | 347,944 | 25.0% | 35,079 | 24.2% | | 70,010 | 24.3% | 35,005 | 24.3% | 35,005 | 24.3% | |
| South | 596,029 | 38.8% | 540,382 | 38.9% | 55,647 | 38.3% | | 110,518 | 38.3% | 55,259 | 38.3% | 55,259 | 38.3% | |
| West | 221,488 | 14.4% | 193,016 | 13.9% | 28,472 | 19.6% | | 56,282 | 19.5% | 28,141 | 19.5% | 28,141 | 19.5% | |
| **Insurance** | | | | | | | | | | | | | | |
| Commercial | 707,676 | 46.1% | 664,625 | 47.8% | 43,051 | 29.7% | <0.001 | 86,053 | 29.8% | 43,023 | 29.8% | 43,030 | 29.8% | 1.00 |
| Dual | 54,840 | 3.6% | 43,171 | 3.1% | 11,669 | 8.0% | | 22,384 | 7.8% | 11,193 | 7.8% | 11,191 | 7.8% | |
| Medicaid | 486,451 | 31.7% | 457,656 | 32.9% | 28,795 | 19.8% | | 56,959 | 19.7% | 28,479 | 19.7% | 28,480 | 19.7% | |
| Medicare | 287,081 | 18.7% | 225,487 | 16.2% | 61,594 | 42.4% | | 123,168 | 42.7% | 61,587 | 42.7% | 61,581 | 42.7% | |
| **PCP Visit 2019** | | | | | | | | | | | | | | |
| No | 605,264 | 39.4% | 553,886 | 39.8% | 51,378 | 35.4% | <0.001 | 102,148 | 35.4% | 51,064 | 35.4% | 51,084 | 35.4% | 0.94 |
| Yes | 930,784 | 60.6% | 837,053 | 60.2% | 93,731 | 64.6% | | 186,416 | 64.6% | 93,218 | 64.6% | 93,198 | 64.6% | |

| **Continuous Outcomes** | | | | | | | | | | | | | | |
| | mean | SD | mean | SD | mean | SD | p-value | mean | SD | mean | SD | mean | SD | p-value |
| CCI | 0.87 | 1.60 | 0.84 | 1.58 | 1.09 | 1.81 | <0.001 | 1.09 | 1.79 | 1.08 | 1.78 | 1.09 | 1.79 | 0.56 |

BP: bisphosphonate; CCI: Charlson Comorbidity Index; PCP: primary care physician; SD: standard deviation.

**Appendix 2—table 45.** Antidepressant User Cohort (Region=New York State) by BP Use, Patient Characteristics Pre/Post Match of BP Users/Non-users.

| | Region=NY Antidepressant Users by BP: Unmatched | | | | | | | Region=NY Antidepressant Users by BP: Matched | | | | | | |
| --- | --- | --- | --- | --- | --- | --- | --- | --- | --- | --- | --- | --- | --- | --- |
| | All | | BP Non-user | | BP User | | p-value | All | | BP Non-user | | BP User | | p-value |
| | N | % | N | % | N | % | | N | % | N | % | N | % | |
| All Patients | 135,516 | 100.0% | 122,566 | 90.4% | 12,950 | 9.6% | | 25,718 | 100.0% | 12,859 | 50.0% | 12,859 | 50.0% | |
| **Age** | | | | | | | | | | | | | | |
| ≤20 | 4,363 | 3.2% | 4,357 | 3.6% | 6 | 0.0% | <0.001 | 12 | 0.0% | 6 | 0.0% | 6 | 0.0% | 1.00 |
| 21-40 | 22,834 | 16.8% | 22,770 | 18.6% | 64 | 0.5% | | 126 | 0.5% | 62 | 0.5% | 64 | 0.5% | |
| 41-50 | 18,482 | 13.6% | 18,263 | 14.9% | 219 | 1.7% | | 440 | 1.7% | 221 | 1.7% | 219 | 1.7% | |
| 51-60 | 29,485 | 21.8% | 27,702 | 22.6% | 1,783 | 13.8% | | 3,570 | 13.9% | 1,788 | 13.9% | 1,782 | 13.9% | |
| 61-70 | 27,540 | 20.3% | 23,385 | 19.1% | 4,155 | 32.1% | | 8,292 | 32.2% | 4,146 | 32.2% | 4,146 | 32.2% | |
| 71-80 | 21,038 | 15.5% | 16,548 | 13.5% | 4,490 | 34.7% | | 8,863 | 34.5% | 4,430 | 34.5% | 4,433 | 34.5% | |
| ≥81 | 11,774 | 8.7% | 9,541 | 7.8% | 2,233 | 17.2% | | 4,415 | 17.2% | 2,206 | 17.2% | 2,209 | 17.2% | |
| **Gender** | | | | | | | | | | | | | | |
| Female | 96,462 | 71.2% | 84,469 | 68.9% | 11,993 | 92.6% | <0.001 | 23,810 | 92.6% | 11,906 | 92.6% | 11,904 | 92.6% | 0.96 |
| Male | 39,054 | 28.8% | 38,097 | 31.1% | 957 | 7.4% | | 1,908 | 7.4% | 953 | 7.4% | 955 | 7.4% | |
| **Insurance** | | | | | | | | | | | | | | |
| Commercial | 51,829 | 38.2% | 49,332 | 40.2% | 2,497 | 19.3% | <0.001 | 4,991 | 19.4% | 2,495 | 19.4% | 2,496 | 19.4% | 1.00 |
| Dual | 1,600 | 1.2% | 1,221 | 1.0% | 379 | 2.9% | | 710 | 2.8% | 356 | 2.8% | 354 | 2.8% | |
| Medicaid | 38,567 | 28.5% | 36,366 | 29.7% | 2,201 | 17.0% | | 4,269 | 16.6% | 2,131 | 16.6% | 2,138 | 16.6% | |
| Medicare | 43,520 | 32.1% | 35,647 | 29.1% | 7,873 | 60.8% | | 15,748 | 61.2% | 7,877 | 61.3% | 7,871 | 61.2% | |
| **PCP Visit 2019** | | | | | | | | | | | | | | |
| No | 53,400 | 39.4% | 48,911 | 39.9% | 4,489 | 34.7% | <0.001 | 8,901 | 34.6% | 4,449 | 34.6% | 4,452 | 34.6% | 0.97 |
| Yes | 82,116 | 60.6% | 73,655 | 60.1% | 8,461 | 65.3% | | 16,817 | 65.4% | 8,410 | 65.4% | 8,407 | 65.4% | |
| **Continuous Outcomes** | | | | | | | | | | | | | | |
| | mean | SD | mean | SD | mean | SD | p-value | mean | SD | mean | SD | mean | SD | p-value |
| CCI | 0.96 | 1.68 | 0.95 | 1.66 | 1.13 | 1.78 | <0.001 | 1.12 | 1.76 | 1.12 | 1.75 | 1.12 | 1.77 | 0.86 |

BP: bisphosphonate; CCI: Charlson Comorbidity Index; PCP: primary care physician; SD: standard deviation.

**Appendix 2—table 46.** Antidepressant Non-user Cohort (All Regions) by BP Use, Patient Characteristics Pre/Post Match of BP Users/Non-users.

| | All Antidepressant Non-users by BP: Unmatched | | | | | | | All Antidepressant Non-users by BP: Matched | | | | | | |
|---|---|---|---|---|---|---|---|---|---|---|---|---|---|---|
| | All | | BP Non-user | | BP User | | p-value | All | | BP Non-user | | BP User | | p-value |
| | N | % | N | % | N | % | | N | % | N | % | N | % | |
| All Patients | 1,536,048 | 100.0% | 1,422,938 | 92.6% | 113,110 | 7.4% | | 224,804 | 100.0% | 112,402 | 50.0% | 112,402 | 50.0% | |
| **Age** | | | | | | | | | | | | | | |
| ≤20 | 89,565 | 5.8% | 89,486 | 6.3% | 79 | 0.1% | <0.001 | 155 | 0.1% | 76 | 0.1% | 79 | 0.1% | 1.00 |
| 21–40 | 315,593 | 20.5% | 314,815 | 22.1% | 778 | 0.7% | | 1,562 | 0.7% | 784 | 0.7% | 778 | 0.7% | |
| 41–50 | 233,336 | 15.2% | 230,961 | 16.2% | 2,375 | 2.1% | | 4,746 | 2.1% | 2,371 | 2.1% | 2,375 | 2.1% | |
| 51–60 | 333,650 | 21.7% | 314,109 | 22.1% | 19,541 | 17.3% | | 39,072 | 17.4% | 19,536 | 17.4% | 19,536 | 17.4% | |
| 61–70 | 296,182 | 19.3% | 254,286 | 17.9% | 41,896 | 37.0% | | 83,664 | 37.2% | 41,834 | 37.2% | 41,830 | 37.2% | |
| 71–80 | 169,295 | 11.0% | 136,746 | 9.6% | 32,549 | 28.8% | | 64,163 | 28.5% | 32,073 | 28.5% | 32,090 | 28.5% | |
| ≥81 | 98,427 | 6.4% | 82,535 | 5.8% | 15,892 | 14.1% | | 31,442 | 14.0% | 15,728 | 14.0% | 15,714 | 14.0% | |
| **Gender** | | | | | | | | | | | | | | |
| Female | 1,109,580 | 72.2% | 1,004,112 | 70.6% | 105,468 | 93.2% | <0.001 | 209,510 | 93.2% | 104,743 | 93.2% | 104,767 | 93.2% | 0.84 |
| Male | 426,468 | 27.8% | 418,826 | 29.4% | 7,642 | 6.8% | | 15,294 | 6.8% | 7,659 | 6.8% | 7,635 | 6.8% | |
| **Region** | | | | | | | | | | | | | | |
| Midwest | 335,508 | 21.8% | 315,179 | 22.1% | 20,329 | 18.0% | <0.001 | 40,548 | 18.0% | 20,274 | 18.0% | 20,274 | 18.0% | 1.00 |
| Northeast | 383,023 | 24.9% | 356,184 | 25.0% | 26,839 | 23.7% | | 53,590 | 23.8% | 26,795 | 23.8% | 26,795 | 23.8% | |
| South | 596,029 | 38.8% | 552,754 | 38.8% | 43,275 | 38.3% | | 85,440 | 38.0% | 42,720 | 38.0% | 42,720 | 38.0% | |
| West | 221,488 | 14.4% | 198,821 | 14.0% | 22,667 | 20.0% | | 45,226 | 20.1% | 22,613 | 20.1% | 22,613 | 20.1% | |
| **Insurance** | | | | | | | | | | | | | | |
| Commercial | 707,675 | 46.1% | 672,990 | 47.3% | 34,685 | 30.7% | <0.001 | 69,354 | 30.9% | 34,675 | 30.8% | 34,679 | 30.9% | 1.00 |
| Dual | 54,836 | 3.6% | 44,281 | 3.1% | 10,555 | 9.3% | | 19,871 | 8.8% | 9,927 | 8.8% | 9,944 | 8.8% | |
| Medicaid | 486,446 | 31.7% | 463,857 | 32.6% | 22,589 | 20.0% | | 45,057 | 20.0% | 22,537 | 20.1% | 22,520 | 20.0% | |
| Medicare | 287,091 | 18.7% | 241,810 | 17.0% | 45,281 | 40.0% | | 90,522 | 40.3% | 45,263 | 40.3% | 45,259 | 40.3% | |
| **PCP Visit 2019** | | | | | | | | | | | | | | |
| No | 605,256 | 39.4% | 572,701 | 40.2% | 32,555 | 28.8% | <0.001 | 64,959 | 28.9% | 32,483 | 28.9% | 32,476 | 28.9% | 0.97 |
| Yes | 930,792 | 60.6% | 850,237 | 59.8% | 80,555 | 71.2% | | 159,845 | 71.1% | 79,919 | 71.1% | 79,926 | 71.1% | |

**Continuous Outcomes**

| | All Antidepressant Non-users by BP: Unmatched | | | | | | | All Antidepressant Non-users by BP: Matched | | | | | | |
|---|---|---|---|---|---|---|---|---|---|---|---|---|---|---|
| | All | | BP Non-user | | BP User | | p-value | All | | BP Non-user | | BP User | | p-value |
| | mean | SD | mean | SD | mean | SD | | mean | SD | mean | SD | mean | SD | |
| CCI | 0.87 | 1.60 | 0.85 | 1.58 | 1.06 | 1.84 | <0.001 | 1.06 | 1.82 | 1.05 | 1.81 | 1.06 | 1.83 | 0.57 |

BP: bisphosphonate; CCI: Charlson Comorbidity Index; PCP: primary care physician; SD: standard deviation.

**Appendix 2—table 47.** Antidepressant Non-user Cohort (Region=New York State) by BP Use, Patient Characteristics Pre/Post Match of BP Users/Non-users.

| | Region=NY Antidepressant Non-users by BP: Unadjusted | | | | | | | Region=NY Antidepressant Non-users by BP: Matched | | | | | | |
| --- | --- | --- | --- | --- | --- | --- | --- | --- | --- | --- | --- | --- | --- | --- |
| | All | | BP Non-user | | BP User | | | All | | BP Non-user | | BP User | | |
| | N | % | N | % | N | % | p-value | N | % | N | % | N | % | p-value |
| All Patients | 135,516 | 100.0% | 125,342 | 92.5% | 10,174 | 7.5% | | 20,182 | 100.0% | 10,091 | 50.0% | 10,091 | 50.0% | |
| Age | | | | | | | | | | | | | | |
| ≤20 | 4,365 | 3.2% | 4,364 | 3.5% | 1 | 0.0% | <0.001 | 2 | 0.0% | 1 | 0.0% | 1 | 0.0% | 1.00 |
| 21-40 | 22,832 | 16.8% | 22,799 | 18.2% | 33 | 0.3% | | 66 | 0.3% | 33 | 0.3% | 33 | 0.3% | |
| 41-50 | 18,483 | 13.6% | 18,350 | 14.6% | 133 | 1.3% | | 267 | 1.3% | 134 | 1.3% | 133 | 1.3% | |
| 51-60 | 29,481 | 21.8% | 28,038 | 22.4% | 1,443 | 14.2% | | 2,879 | 14.3% | 1,440 | 14.3% | 1,439 | 14.3% | |
| 61-70 | 27,543 | 20.3% | 24,197 | 19.3% | 3,346 | 32.9% | | 6,686 | 33.1% | 3,345 | 33.1% | 3,341 | 33.1% | |
| 71-80 | 21,038 | 15.5% | 17,695 | 14.1% | 3,343 | 32.9% | | 6,589 | 32.6% | 3,294 | 32.6% | 3,295 | 32.7% | |
| ≥81 | 11,774 | 8.7% | 9,899 | 7.9% | 1,875 | 18.4% | | 3,693 | 18.3% | 1,844 | 18.3% | 1,849 | 18.3% | |
| Gender | | | | | | | | | | | | | | |
| Female | 96,468 | 71.2% | 86,945 | 69.4% | 9,523 | 93.6% | <0.001 | 18,892 | 93.6% | 9,446 | 93.6% | 9,446 | 93.6% | 1.00 |
| Male | 39,048 | 28.8% | 38,397 | 30.6% | 651 | 6.4% | | 1,290 | 6.4% | 645 | 6.4% | 645 | 6.4% | |
| Insurance | | | | | | | | | | | | | | |
| Commercial | 51,829 | 38.2% | 50,405 | 40.2% | 1,424 | 14.0% | <0.001 | 2,848 | 14.1% | 1,425 | 14.1% | 1,423 | 14.1% | 1.00 |
| Dual | 1,591 | 1.2% | 1,210 | 1.0% | 381 | 3.7% | | 690 | 3.4% | 345 | 3.4% | 345 | 3.4% | |
| Medicaid | 38,569 | 28.5% | 36,303 | 29.0% | 2,266 | 22.3% | | 4,449 | 22.0% | 2,226 | 22.1% | 2,223 | 22.0% | |
| Medicare | 43,527 | 32.1% | 37,424 | 29.9% | 6,103 | 60.0% | | 12,195 | 60.4% | 6,095 | 60.4% | 6,100 | 60.4% | |
| PCP Visit 2019 | | | | | | | | | | | | | | |
| No | 53,397 | 39.4% | 50,515 | 40.3% | 2,882 | 28.3% | <0.001 | 5,723 | 28.4% | 2,863 | 28.4% | 2,860 | 28.3% | 0.96 |
| Yes | 82,119 | 60.6% | 74,827 | 59.7% | 7,292 | 71.7% | | 14,459 | 71.6% | 7,228 | 71.6% | 7,231 | 71.7% | |
| **Continuous Outcomes** | | | | | | | | | | | | | | |
| | mean | SD | mean | SD | mean | SD | p-value | mean | SD | mean | SD | mean | SD | p-value |
| CCI | 0.96 | 1.68 | 0.95 | 1.66 | 1.13 | 1.81 | <0.001 | 1.11 | 1.77 | 1.11 | 1.76 | 1.12 | 1.78 | 0.78 |

BP: bisphosphonate; CCI: Charlson Comorbidity Index; PCP: primary care physician; SD: standard deviation.

## Appendix 3

### Post-hoc analysis on the impact of censoring due to death

#### Background

Following completion of all core study analyses, an additional post-hoc investigation was performed to assess whether censoring bias due to patient death could impact our current findings of a decrease in the odds of COVID-19 outcomes seen amongst BP users. Typically, it is very difficult to perform assessments on this type of bias due to the fact that insurance claims databases in the United States do not include this information. Some claims database providers, including Komodo Health, do have the capability to 'link' their de-identified claims data with external sources on decedent enrolees, but at the time of study initiation and data extraction there were enhanced HIPAA constraints associated with claims datasets that included COVID-identifying diagnosis/treatment codes due to the heightened risk of patient re-identification due to the then lower prevalence and high visibility associated for patients with COVID-19. Eventually the increased prevalence of COVID-19 reduced the HIPAA concerns on working with claims data that include COVID-19-identifiers, and in support of this analysis and the potentially significant public health implications of our findings, Komodo Health linked their COVID-identifiable dataset with mortality data sources that account for roughly 80–85% of available death records. In conjunction with Komodo Health, queries on this mortality-linked COVID-19-identifiable dataset were performed to determine whether bias caused by patient censoring due to death could have impacted the validity and/or reliability of our current findings

#### Methodological concerns of patient censoring due to death

The single motivating factor for initiation of this post-hoc analysis was the fact that the decrease in odds of COVID-19 outcomes among BP users in this study was found to be statistically significant, large in magnitude, and robust across almost all analysis variations performed. The exhaustive use of methodological techniques to control for unmeasured confounding and/or outside sources of bias employed in this current study were undertaken not in search of statistical significance, but in search of non-significance. This was undertaken because the consistency seen in statistical significance, in addition to the magnitude of the decrease in the odds of our outcomes of interest, are typically not seen to this degree. As such, the next logical step after exhausting all methodological techniques is to search for other sources that could induce a large-enough bias on the underlying patient population itself, such as censoring of the target study cohort, that could drastically alter the typical composition of the overall sample and thus impact the reliability and validity of outcomes measured.

The high rate of death associated with COVID-19 infection, which was even worse during the early months of the pandemic, represents such an instance where outside influences could impact the underlying data, and as such, the validity of research performed on that data. The primary concern is whether patients who have died are censored from the analytical sample due to the application of one of the most fundamental inclusion/exclusion criteria used in claims-based research, the requirement for continuous insurance eligibility over the entire study period that is needed so that healthcare resource utilization events from all subjects are captured and available in the data for analysis. If in our current sample, a larger number of BP users died after contracting COVID-19 and were censored due to insurance eligibility, and a lower number of BP non-users survived and thus met the insurance eligibility criteria, then the remaining study sample would be comprised of healthier-looking BP users and a higher number of BP non-users with COVID-19 related healthcare services.

The potential for such a censoring bias in this current study sample, and the impact of that bias on the magnitude and statistical significance of our core study findings, was assessed in this post-hoc analysis by: (1) adjusting eligibility criteria to prevent the censoring of patients that may have died during the first half of 2020; (2) replicating key exposure (BP-use, use of other non-BP bone health medications) and outcomes (COVID-19 diagnosis) in this expanded sample that aligns with the core study methods; (3) analysing the impact on study findings that would result from the retention and inclusion of deceased-patient observations in the core study sample on the odds of COVID-19 diagnosis; and (4) calculating the number of missing patient observations censored due to death that would be required to reach a statistically non-significant difference in the odds of COVID-19.

## Post-Hoc analysis

### Methods

### Cohort definition

- Continuous insurance eligibility 1/1/2019-12/31/2019; used to ensure that any censoring due to death occurs during the observation period of 1/1/2020-6/30/2020
- BP users compared to BP non-users to produce a cohort comparison similar to the primary analysis cohort
- BP users compared to users of non-BP anti-resorptive bone health medications to produce a cohort comparison similar to the "*Bone-Rx*" active comparator analysis

### Exposures of interest

- Patients were assigned into the BP user cohort if they had any claim 1/1/2019-2/29/2020 for one of the following: alendronate, alendronic acid, etidronate, ibandronate, ibandronic acid, pamidronate, risedronate, and zoledronic acid; for the cohort comparison of all osteoporosis medication users BP users were further restricted to those that had no claims for a non-BP anti-resorptive bone health medication 1/1/2019-2/29/2020.
- Patients were assigned into the non-BP anti-resorptive bone health medication user cohort if: (1) they had any claim 1/1/2019-2/29/2020 for one of the following: denosumab, calcitonin, raloxifene, romosozumab-aqqg, teriparatide, abaloparatide, or bazedoxifene; and (2) they had no BP claims

### Outcomes / endpoints

- Patients were assigned into the COVID-19 diagnosis cohort based on any medical service claim with an ICD-10 diagnosis code of U07.1 occurring 1/1/2200-6/30/2020
- Patients with a date-of-death between 1/1/2020-6/30/2020 were classified into the deceased cohort

### Statistical analysis

- Chi-square testing was used to assess whether statistically significant differences exist between BP users and BP non-users in the unadjusted odds of having any COVID-19 diagnosis during the first half of 2020 among cohorts that approximate the primary analysis and "*Bone-Rx*" study cohorts for the following:

1. Among all patient-observations with a COVID-19 diagnosis to assess the potential 'true' comparison that would occur
2. With deceased patient-observations that had a known COVID-19 diagnosis removed prior to testing to replicate findings that would occur if these observations were censored
3. When making the assumption that all patients who died during this period died due to COVID-19, and thus should be classified as having a COVID-19 diagnosis

An additional analysis was performed on the last variation modelled (assuming all patients died due to COVID-19) to determine the additional BP user patient observations that would be needed to be classified as having had a COVID-19 diagnosis to yield a similar distribution of COVID-19 diagnosis (yes/no) as was seen in the BP non-user cohort to yield an odds ratio ~1.0

Finally, the impact on odds ratio testing results comparing BP users to BP non-users was modelled based on the additional number of BP users needed to be classified as having been diagnosed with COVID-19 to reach statistical non-significance

### Results

### Patient count distribution

Among the full sample a decreased rate of COVID-19 among BP users compared to BP non-users was seen in both the full sample population (1.2% vs 4.7%) as well as when restricted to users of non-BP anti-resorptive bone health medications (1.2% versus 4.3%) (*Appendix 3—table 1*)

### Unadjusted Chi-square comparison inclusive of deceased patients

The decrease in the odds of any COVID-19 diagnosis amongst BP users compared to BP non-users was found to be robust in both the full (OR = 0.24) and "Bone-Rx" (OR = 0.35) comparisons when including deceased patients with a known COVID-19 diagnosis (*Appendix 3—table 2*)

### Unadjusted Chi-square comparison with deceased patients removed

The decrease in the odds of any COVID-19 diagnosis amongst BP users compared to BP non-users was found to be robust in both the full (OR = 0.23) and "Bone-Rx" (OR = 0.26) comparisons when removing deceased patients with a known COVID-19 diagnosis (*Appendix 3—table 3*)

### Unadjusted Chi-square comparison assuming all deceased patients had COVID-19

- The decrease in the odds of any COVID-19 diagnosis amongst BP users compared to BP non-users was found to be robust in both the full (OR = 0.39) and "*Bone-Rx*" (OR = 0.29) comparisons when assuming that all deceased patients had a COVID-19 diagnosis (*Appendix 3—table 4*)
- Among this final analysis that assumes all deceased patients had a diagnosis of COVID-19, the percentage of BP non-users with an assumed COVID-19 diagnosis was 5.5% and 7.2% for the full and OPRX comparisons, respectively.
- These proportions were then used to estimate the number of additional BP users with a COVID-19 diagnosis that would be needed to have the same distribution and thus an odds ratio ~1.0 (*Appendix 3—table 5*)
- It would require an additional 22,235 (37,095-14,860) BP-user patient observations from the full cohort comparison to be classified as having a COVID-19 diagnosis to have an equivalent odds of being diagnosed with COVID-19 as was seen among the BP non-user cohort
- It would require an additional 32,598 (46,637-14,039) BP-user patient observations from the "*Bone-Rx*" cohort comparison to be classified as having a COVID-19 diagnosis to have an equivalent odds of being diagnosed with COVID-19 as was seen among the BP non-user cohort
- In the full (all observations) comparison, the minimum number of additional BP users classified as having a COVID-19 diagnosis needed to reach statistical non-significance for the calculated unadjusted odds ratio was 21,860 (*Appendix 3—figure 1*)
- In the "*Bone-Rx*" comparison, the minimum number of additional BP users classified as having a COVID-19 diagnosis needed to reach statistical non-significance for the calculated unadjusted odds ratio was 31,360 (*Appendix 3—figure 2*)

**Appendix 3—table 1.** Patient Count Distribution Inclusive of Deceased Enrolees.

| | All Observations | | All Bone Health Rx Users ("Bone-Rx") | |
| --- | --- | --- | --- | --- |
| | BP Users | BP Non-users | BP Users | BP Non-users |
| Total (N) | 672,913 | 10,978,373 | 645,118 | 75,195 |
| Deceased (N) [*any reason*] | 7,364 | 101,282 | 6,922 | 2,450 |
| COVID-19 Dx (N) | 7,927 | 519,387 | 7,527 | 3,201 |
| COVID-19 Dx (%) | 1.2% | 4.7% | 1.2% | 4.3% |
| COVID-19 Dx & Deceased (N) | 431 | 15,470 | 410 | 215 |
| COVID-19 Dx & Deceased (%) | 5.4% | 3.0% | 5.4% | 6.7% |

Dx: diagnosis.

**Appendix 3—table 2.** Unadjusted Chi-Square Comparison Inclusive of Deceased Patients.

| | All Observations (with deceased) | | "Bone-Rx" Observations (with deceased) | |
|---|---|---|---|---|
| | COVID-19 Dx | No COVID-19 Dx | COVID-19 Dx | No COVID-19 Dx |
| BP users | 7,927 | 664,986 | 7,527 | 637,591 |
| BP Non-users | 519,387 | 10,458,986 | 2,450 | 71,994 |
| Odds Ratio | 0.24 | | Odds Ratio | 0.35 |
| 95 % CI: | 0.2347 to 0.2455 | | 95 % CI: | 0.3312 to 0.3633 |
| p-value | P < 0.0001 | | p-value | P < 0.0001 |

BP: bisphosphonate; CI: confidence interval; Dx: diagnosis.

**Appendix 3—table 3.** Unadjusted Chi-Square Comparison with Deceased Patients Removed.

| | All Observations (without deceased) | | "Bone-Rx" Observations (without deceased) | |
|---|---|---|---|---|
| | COVID-19 Dx | No COVID-19 Dx | COVID-19 Dx | No COVID-19 Dx |
| BP users | 7,496 | 657,622 | 7,117 | 630,669 |
| BP Non-users | 503,917 | 10,357,704 | 2,986 | 69,544 |
| Odds Ratio | 0.23 | | Odds Ratio | 0.26 |
| 95 % CI: | 0.2290–0.2397 | | 95 % CI: | 0.2516–0.2745 |
| p-value | P<0.0001 | | p-value | P<0.0001 |

BP: bisphosphonate; CI: confidence interval; Dx: diagnosis.

**Appendix 3—table 4.** Unadjusted Chi-Square Comparison Assuming all Deceased Patients had COVID-19.

| | All Observations (assume deceased = COVID-19) | | "Bone-Rx" Observations (assume deceased = COVID-19) | |
|---|---|---|---|---|
| | COVID-19 Dx | No COVID-19 Dx | COVID-19 Dx | No COVID-19 Dx |
| BP users | 14,860 | 658,053 | 14,039 | 631,079 |
| BP Non-users | 605,199 | 10,373,174 | 5,436 | 69,759 |
| Odds Ratio | 0.39 | | Odds Ratio | 0.29 |
| 95 % CI: | 0.3807–0.3935 | | 95 % CI: | 0.2764–0.2948 |
| p-value | P<0.0001 | | p-value | P<0.0001 |

BP: bisphosphonate; CI: confidence interval; Dx: diagnosis.

**Appendix 3—table 5.** Unadjusted Chi-Square Comparison to Yield Odds Ratio = 1.00 (no difference).

| | All Observations (assume deceased = COVID-19) | | "Bone-Rx" Observations (assume deceased = COVID-19) | |
|---|---|---|---|---|
| | COVID-19 Dx | No COVID-19 Dx | COVID-19 Dx | No COVID-19 Dx |
| BP users | 37,095 | 635,818 | 46,637 | 598,481 |
| BP Non-users | 605,199 | 10,373,174 | 5,436 | 69,759 |
| Odds Ratio | 1.00 | | Odds Ratio | 1.00 |
| 95 % CI: | 0.9893–1.0108 | | 95 % CI: | 0.9713–1.0296 |
| p-value | P=0.9987 | | p-value | P=0.9999 |

BP: bisphosphonate; CI: confidence interval; Dx: diagnosis.

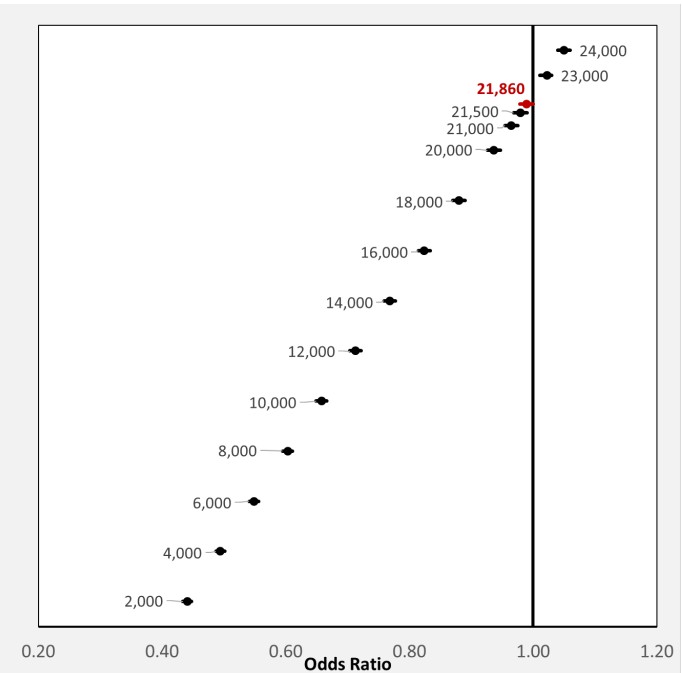

**Appendix 3—figure 1.** Full cohort: dds ratio by additional number of BP users classified as having COVID-19 diagnosis.

Forest plot of the change in the crude odds ratio (OR) of BP users having a COVID-19 diagnosis as a factor of the additional number of BP users needed to be classified as having a COVID-19 diagnosis to reach statistical non-significance for all observations.

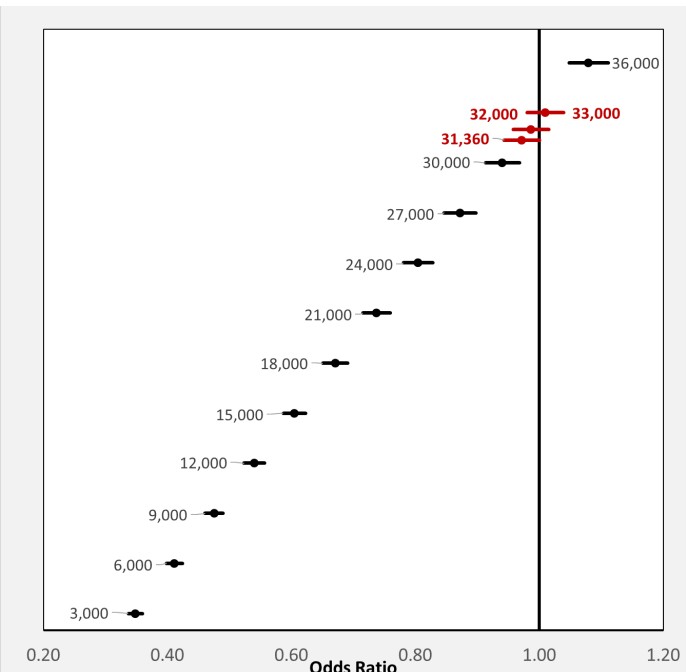

**Appendix 3—figure 2.** Bone-Rx cohort: odds ratio by additional number of BP users classified as having COVID-19 diagnosis. Forest plot of the change in the crude odds ratio (OR) of BP users having a COVID-19 diagnosis as a factor of the additional number of BP users needed to be classified as having a COVID-19 diagnosis to reach statistical non-significance when comparing BP users to users of non-BP anti-resorptive bone medication

