## [Editor Report]

Using health insurance claims data, this valuable paper reports on a retrospective propensity score matched cohort study that was performed to quantify associations between bisphosphonate (BP) use and COVID-19-related outcomes (COVID-19 diagnosis, testing, and COVID-19 hospitalization). The evidence is solid showing that in primary and sensitivity analyses, BP use was consistently associated with lower odds for COVID-19, testing, and COVID-19 hospitalization. The study is of interest to a broad readership (clinicians, public health physicians, pharmacologists and epidemiologists).

---

## [Decision Letter]

**Decision letter after peer review:**

Thank you for submitting your article "Association between Bisphosphonate use and COVID-19 related outcomes: a retrospective cohort study" for consideration by *eLife*. Your article has been reviewed by 2 peer reviewers, including Marc J Bonten as Reviewing Editor and Reviewer #1, and the evaluation has been overseen by Jos van der Meer as the Senior Editor. The following individual involved in the review of your submission has agreed to reveal their identity: Henri van Werkhoven (Reviewer #2).

Essential revisions:

*Reviewer #2 (Recommendations for the authors):*

Please consider the following comments:

Abstract:

– Methods: outcome of interest: to me it is not unambiguous whether (a) testing for SARS-CoV-2 infection means any testing or testing positive.

– Methods: I recommend to move "propensity score-matched to bisphosphonate non-users by age, gender, insurance type, primary-care-provider visit in 2019, and comorbidity burden." to the methods part.

– Also please explain what is 'comorbidity burden'? (But see comment in Results section on using CCI in the PS.)

– Pneumonia and bronchitis are not mentioned in abstract methods. The reason for this analysis should be explained or the results not reported in the abstract.

Results:

– The C-statistic / ROC-AUC of the propensity model should be reported.

– Line 495: I suggest to remove the word 'strikingly'. In general: to be objective in the Results section; in this case: it is not so striking IMHO. It is clear that this subgroup is quite small compared to other subgroups (hence the wide confidence interval) and it may well be a matter of chance that this subgroup has a bit of a higher point estimate.

– There is a lot of repetition of methods in the Results section of sensitivity analyses. Redundancy should be avoided.

Discussion:

– Line 893-9: this is also in line with the observation of similar ORs for the overall and severe COVID-19 outcomes in the current study. Although this is less evident in the sensitivity analyses (but with wider confidence intervals).

– The discussion is quite lengthy and could be shortened. E.g. line 923-33 repeats which sensitivity analyses were performed.

– Line 1059 "one would expect that BP users would have distinct odds for outcomes not predicted to be modulated by BPs" → I do not understand this sentence.

---

## [Author Response]

Essential revisions:Reviewer #2 (Recommendations for the authors):Please consider the following comments:Abstract:– Methods: outcome of interest: to me it is not unambiguous whether (a) testing for SARS-CoV-2 infection means any testing or testing positive.

We agree. We have modified the methods section in the abstract by adding “any” testing for SARSCoV-2 infection (line 59).

– Methods: I recommend to move "propensity score-matched to bisphosphonate non-users by age, gender, insurance type, primary-care-provider visit in 2019, and comorbidity burden." to the methods part.

This passage has been moved as suggested.

– Also please explain what is 'comorbidity burden'? (But see comment in Results section on using CCI in the PS.)

Comorbidity burden is defined as the overall health state of each patient that we are trying to control for via the use of the CCI (for core matches) or via the larger comorbidity CV list used in sensitivity analysis 2.

– Pneumonia and bronchitis are not mentioned in abstract methods. The reason for this analysis should be explained or the results not reported in the abstract.

We have added the following to the abstract methods (line 61-64):

“Multiple sensitivity analyses were also performed to assess core study outcomes amongst more restrictive matches between BP users/nonusers, as well as assessing the relationship between BP-use and other respiratory infections (pneumonia, acute bronchitis) both during the same study period as well as before the COVID outbreak.”

Results:– The C-statistic / ROC-AUC of the propensity model should be reported.

See our reply above regarding this issue.

– Line 495: I suggest to remove the word 'strikingly'. In general: to be objective in the Results section; in this case: it is not so striking IMHO. It is clear that this subgroup is quite small compared to other subgroups (hence the wide confidence interval) and it may well be a matter of chance that this subgroup has a bit of a higher point estimate.

We have removed the word ‘strikingly’.

– There is a lot of repetition of methods in the Results section of sensitivity analyses. Redundancy should be avoided.

We acknowledge that there is some redundancy between the Methods and Results sections, but feel it necessary to include sufficient methodological details to aid readers who are not completely familiar with HEOR work due to the high degree of complexity related to the sensitivity analyses.

Discussion:– Line 893-9: this is also in line with the observation of similar ORs for the overall and severe COVID-19 outcomes in the current study. Although this is less evident in the sensitivity analyses (but with wider confidence intervals).

We agree.

– The discussion is quite lengthy and could be shortened. E.g. line 923-33 repeats which sensitivity analyses were performed.

We recognize the length of the Discussion and have condensed this section by deleting repetitive statments where possible.

– Line 1059 "one would expect that BP users would have distinct odds for outcomes not predicted to be modulated by BPs" → I do not understand this sentence.

This passage was addressing our analysis of negative control outcomes, which we have removed in this revised version of our paper. Therefore, this sentence was deleted.